# ImageNet-RIB Benchmark: Large Pre-Training Datasets Don't Always Guarantee Robustness after Fine-Tuning

## Abstract

Highly performant large-scale pre-trained models promise to also provide a valuable foundation for learning specialized tasks, by fine-tuning the model to the desired task. By starting from a good general-purpose model, the goal is to achieve both specialization in the target task and maintain robustness. To assess the robustness of models to out-of-distribution samples after fine-tuning on downstream datasets, we introduce a new robust fine-tuning benchmark, ImageNet-RIB (Robustness Inheritance Benchmark). The benchmark consists of a set of related but distinct specialized (downstream) tasks; pre-trained models are fine-tuned on one task in the set and their robustness is assessed on the rest, iterating across all tasks for fine-tuning and assessment. We find that the continual learning methods, EWC and LwF maintain robustness after fine-tuning though fine-tuning generally does reduce performance on generalization to related downstream tasks across models. Not surprisingly, models pre-trained on large and rich datasets exhibit higher initial robustness across datasets and suffer more pronounced degradation during fine-tuning. The distance between the pre-training and downstream datasets, measured by optimal transport, predicts this performance degradation on the pre-training dataset. However, counterintuitively, model robustness after fine-tuning on related downstream tasks is the worst when the pre-training dataset is the richest and the most diverse. This suggests that starting with the strongest foundation model is not necessarily the best approach for performance on specialist tasks. The benchmark thus offers key insights for developing more resilient fine-tuning strategies and building robust machine learning models[1].

## 1 Introduction

Deep learning has progressed towards utilizing larger datasets (Lin et al., 2014; Russakovsky et al., 2015; Schuhmann et al., 2022) and deeper architectures (Dosovitskiy et al., 2021; He et al., 2016; Jiang et al., 2023). Consequently, starting with a model pre-trained on a large-scale dataset and fine-tuning it for specific downstream tasks has become standard in machine learning to achieve better performance than training from scratch. While this approach capitalizes on the extensive knowledge embedded in pre-trained models, it often results in a significant loss of that knowledge due to catastrophic forgetting (French, 1999; Robins, 1995). To mitigate this issue, methods only training a part of the pre-trained model such as linear probing, low-rank adaptation (Hu et al., 2021), and visual prompt (Bahng et al., 2022) have been proposed. However, these methods typically underperform compared to fine-tuning the entire model on the downstream task.

Fine-tuning on the downstream task also negatively impacts a model's robustness to out-of-distribution (OOD) samples as the model is optimized for a narrower distribution (Figure 1). This issue has been extensively studied using various OOD datasets, typically beginning with an ImageNet pre-trained model and evaluating it on OOD datasets that exhibit natural distribution shifts, such as changes in viewpoints (Barbu et al., 2019), time (Recht et al., 2019), styles (Hendrycks et al., 2021a; Wang et al., 2019), or synthetic data based on the original dataset (Hendrycks & Dietterich, 2019; Salvador & Oberman, 2022).

---

[1] https://jd730.github.io/projects/ImageNet-RIB

Figure 1: **Conceptual diagram of the impact of fine-tuning on pre-trained models on out-of-distribution (OOD) generalization.** A model $f_{\text{pre}}$ pre-trained on the dataset $D_{\text{pre}}$ (purple solid circle) can generalize to certain out-of-distribution data (purple dashed circle). The dashed gray line represents a volume ($\mathbb{D}$) containing inter-related OOD datasets (dark gray ellipsoids). Fine-tuning on one of these datasets, referred to as the downstream dataset ($D_{\text{down}}$, blue) shifts $f_{\text{pre}}$ to $f_{\text{down}}$, making it more specialized to $D_{\text{down}}$ (blue dashed ellipsoid). This specialization improves performance on $D_{\text{down}}$ and possibly others within the inter-related OOD set (green solid ellipsoids), but may also lead to degradation on some OOD datasets (red solid ellipsoids).

Taori et al. (2020) proposed a benchmark that evaluates pre-trained models fine-tuned on ImageNet-1K across multiple realistic ImageNet-based OOD datasets. It is widely used to measure model robustness changes after fine-tuning (Kumar et al., 2022; Wortsman et al., 2022a;b) However, this benchmark only uses one downstream task (ImageNet-1K), and certain pre-training datasets may include parts of ImageNet as they are often uncurated (Schuhmann et al., 2022). This limitation motivates the need for a broader and more comprehensive evaluation of robustness across multiple OOD datasets.

In this paper, we introduce ImageNet-RIB (Robustness Inheritance Benchmark), a new benchmark designed to assess the robustness of fine-tuned models across diverse downstream and evaluation OOD dataset pairs related to ImageNet. For each experiment, we fine-tune a pre-trained model on one OOD dataset (as a downstream dataset) and evaluate its performance on the remaining OOD datasets. This process is repeated across all available datasets to thoroughly assess how well the model retains robustness after fine-tuning. To achieve this, we employ a variety of fine-tuning strategies, including vanilla fine-tuning, linear probing (fine-tuning the last layer only), LoRA (Hu et al., 2021), regularization-based continual learning methods (Li & Hoiem, 2017; Zenke et al., 2017), and robust fine-tuning methods (Kumar et al., 2022; Wortsman et al., 2022a;b). We also investigate the relationship between dataset distance metrics and robust fine-tuning outcomes, allowing us to estimate the robustness changes prior to training.

Our experimental results show that models pre-trained on larger and more diverse datasets demonstrate superior robustness on the OOD datasets and accuracy on ImageNet-1K. However, fine-tuning causes performance drops on ImageNet-1K, which we find is aligned with the distance between the pre-training dataset and the downstream dataset, as measured by Optimal Transport Dataset Distance (OTDD) (Alvarez-Melis & Fusi, 2020). The combination of Model Soup (Wortsman et al., 2022a) with regularization-based continual learning methods achieves the best performance in the benchmark, while linear probing performs the best when using LAION-2B pre-trained models. Furthermore, our findings indicate that continual learning methods not only mitigate catastrophic forgetting related to the pre-training dataset but also enhance robustness when compared to standard fine-tuning. This improvement is attributed to leveraging the distributional properties of both pre-training and downstream datasets. Interestingly, pre-training on LAION-2B, despite its size and diversity, does not always yield the best results when fine-tuned on downstream tasks, suggesting that starting with large, rich datasets may not always be the optimal approach for preserving robustness.

In summary, the contributions of this paper are four-fold:

- We propose ImageNet-RIB, a new benchmark leveraging multiple ImageNet-based OOD datasets to quantify the robustness of fine-tuned models in comparison to pre-trained models.
- We find that the performance drop on the pre-training dataset during fine-tuning can be predicted by the distance between pre-training and downstream datasets.

- We empirically demonstrate that regularization-based continual learning methods improve robustness by leveraging both the pre-training and downstream dataset distributions and this improvement is amplified when combined with robust fine-tuning methods.

- The absolute performance of the models pre-trained on richer datasets is worse on the downstream tasks, suggesting that starting with rich foundation models may not always be the best approach.

## 2 RELATED WORK

### 2.1 ROBUSTNESS IN MACHINE LEARNING

Robustness in machine learning refers to a model's ability to maintain performance under various perturbations, such as noise, corruption, and domain shifts. Robustness is typically evaluated on synthetic datasets derived from original data (Hendrycks & Dietterich, 2019; Salvador & Oberman, 2022) or real-world datasets featuring distribution shifts, such as different viewpoints (Barbu et al., 2019), styles (Hendrycks et al., 2021a; Wang et al., 2019), or temporal changes (Recht et al., 2019). To develop more robust models, data augmentation techniques have been widely explored including style transfer (Geirhos et al., 2019), perturbation-based image-to-image networks (Hendrycks et al., 2021a), and adversarial logit pairing (Kannan et al., 2018). Robust-fine-tuning usually aims to maintain the robustness of the pre-trained model to OOD datasets during fine-tuning. Taori et al. (2020) address the limitations of previous robustness evaluations that used synthetic datasets by proposing a new evaluation protocol that utilizes realistic datasets; ImageNet-V2, ImageNet-A, ImageNet-R, ImageNet-Sketch, and ObjectNet after fine-tuning on ImageNet. This benchmark is widely used with vision and language models such as CLIP (Radford et al., 2021). Shi et al. (2023) extend this to joint training on two dataset; ImageNet-1K with CIFAR-10 (Krizhevsky et al., 2009) or YFCC (Thomee et al., 2016). To solve this problem, Wortsman et al. (2022a) demonstrate that averaging the parameters of multiple trained models improves both in-distribution and OOD performance. WiSE-FT (Wortsman et al., 2022b) further shows that linearly interpolating the weights of pre-trained CLIP and ImageNet fine-tuned CLIP improves robustness, although it requires tuning the interpolation ratio for optimal performance. Goyal et al. (2023) show that contrastive learning using text encoder in fine-tuning improves robustness. Kumar et al. (2022) propose a two-stage method (LP-FT) that first applies linear probing followed by fine-tuning the entire network. Recently concurrent work (Ramanujan et al., 2024) analyzes the effect of pre-training datasets on robust fine-tuning in the WILDS (Koh et al., 2021) dataset, showing that having more data is beneficial, while greater diversity per class is not. Unlike existing benchmarks (Shi et al., 2023; Taori et al., 2020), which only fine-tune on ImageNet or two datasets simultaneously from unknown or uncurated pre-training datasets, our benchmark provides diverse downstream datasets for a comprehensive understanding of robust fine-tuning.

### 2.2 SINGLE DOMAIN GENERALIZATION

Single-domain generalization refers to the task where only one source domain is available during training, and the model is evaluated on multiple unseen target domains (Qiao et al., 2020). While the high-level concept is similar to the existing robust fine-tuning benchmark (Taori et al., 2020), the objectives differ. Robust fine-tuning focuses on maintaining or improving a model's robustness to OOD datasets during fine-tuning, whereas single-domain generalization aims to achieve generalization to unseen OOD datasets, often through meta-learning-based data augmentation (Chen et al., 2023; Qiao et al., 2020) or adaptive batch normalization (Fan et al., 2021). Recently, Fan et al. (2021) applied single-domain generalization to the PACS dataset (Li et al., 2017), using one domain as the training set and the remaining domains as test sets. This setup resembles our ImageNet-RIB benchmark in that each dataset is used for training while the others are used for testing. However, the goals of the two benchmarks differ: our robust fine-tuning benchmark aims to mitigate robustness degradation during fine-tuning, while single-domain generalization benchmarks focus on improving generalizability from a single source domain.

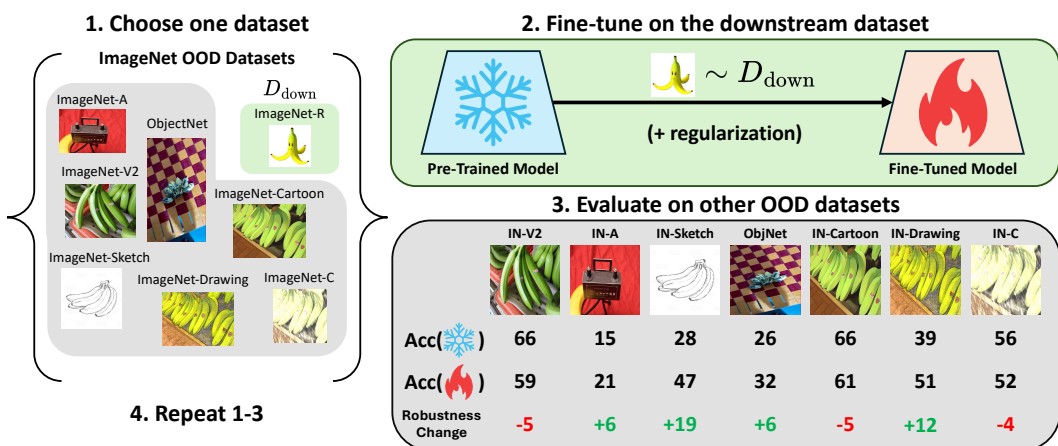

Figure 2: **Illustration of the ImageNet-RIB benchmark.** (1) The process begins by selecting one dataset from the set of ImageNet OOD datasets as the downstream dataset $D_{\text{down}}$ (2) The pre-trained model is fine-tuned on $D_{\text{down}}$, then (3) evaluated on the remaining OOD datasets to assess robustness changes compared to the pre-trained model. (4) This process is repeated across all OOD datasets as downstream tasks, ensuring a detailed evaluation of fine-tuning's impact on robustness.

### 2.3 CONTINUAL LEARNING

Continual learning aims to enable models to learn new tasks without forgetting previously learned knowledge. Existing approaches can be broadly categorized into three types: regularization-based methods, replay-based methods, and architecture-based methods. Regularization-based methods (Cheung et al., 2019; Kirkpatrick et al., 2017; Li & Hoiem, 2017; Zenke et al., 2017) use additional loss terms to limit changes to the model's parameters, ensuring that previously learned knowledge is retained. For instance, Kirkpatrick et al. (2017) employ the Fisher information matrix to determine the importance of each parameter, helping to preserve critical weights from earlier tasks. Li & Hoiem (2017) use knowledge distillation to transfer outputs from a model trained on past tasks to guide learning new tasks. Replay-based methods (Robins, 1995) mitigate catastrophic forgetting by creating a replay buffer that contains a small subset of previous task data or synthetic data (Van de Ven et al., 2020) and a model is trained on the buffer along with a new task. Techniques such as reservoir sampling, reinforcement learning (Rebuffi et al., 2017), and gradient-based selection (Aljundi et al., 2019) help efficiently manage memory and select important data. Architecture-based methods modify the model's structure to accommodate new tasks. These methods dynamically grow the network as needed. For example, Rusu et al. (2016), Yan et al. (2021), and Wang et al. (2022) introduce new model components for each task and use distillation to integrate them with the previous model. In our work, we focus on regularization-based continual learning methods to ensure a fair comparison with other fine-tuning approaches.

## 3 IMAGENET ROBUSTNESS INHERITANCE BENCHMARK (IMAGENET-RIB)

We propose the ImageNet-RIB (Robustness Inheritance Benchmark), a novel benchmark designed to measure robustness using existing ImageNet-related out-of-distribution (OOD) datasets as both downstream and evaluation datasets. ImageNet-RIB fine-tunes pre-trained models on a variety of downstream datasets, then evaluates robustness to other OOD datasets in the benchmark (Figure 2), offering a more comprehensive understanding of robustness fine-tuning.

### 3.1 BENCHMARK PROTOCOL AND ROBUSTNESS METRIC

**Protocol** Figure 2 and Algorithm 1 in Appendix illustrate the protocol of our benchmark. Given a set of out-of-distribution (OOD) datasets $\mathbb{D} = \{D_1, D_2, ..., D_n\}$, a model pre-trained on the dataset $D_{\text{pre}}$ is fine-tuned on the downstream dataset $D_{\text{down}} \sim \mathbb{D}$. After fine-tuning, both the pre-trained

model and the fine-tuned model are evaluated on the remaining datasets in $\mathbb{D} \setminus D_{\text{down}}$. This process is repeated by selecting each dataset in $\mathbb{D}$ as the downstream dataset.

**Metric**    We define the robustness improvement score ($RI$) as the average relative robustness (Taori et al., 2020). Specifically, RI measures the accuracy difference between fine-tuned and pre-trained models on OOD datasets. Formally, robustness improvement ($RI$) after fine-tuning on $D_i(= D_{\text{down}})$ is defined as:

$$RI_i = \frac{1}{n-1} \sum_{j=1, j \neq i}^{n} A_i^{(j)} - A_{\text{pre}}^{(j)}, \tag{1}$$

where $A_{\text{pre}}^{(j)}$ and $A_i^{(j)}$ denote the average accuracies of pre-trained and fine-tuned models on $D_j$, respectively. In addition to relative robustness, effective robustness (Taori et al., 2020) is an alternative metric commonly used to evaluate OOD performance. Effective robustness measures how much the accuracy of a model deviates from an expected baseline, typically using a reference in-distribution dataset (*e.g.*, ImageNet-1K). While effective robustness is insightful, we use relative robustness in this benchmark to facilitate direct comparisons between different fine-tuning methods and initial pre-training datasets. We summarize the overall robustness improvement across all datasets as the mean robustness improvement ($mRI$).

## 3.2 DATASET SUITES

We leverage all existing ImageNet OOD datasets designed for measuring robustness to distribution shifts: ImageNet-V2 (Recht et al., 2019), ImageNet-A (Hendrycks et al., 2021b), ImageNet-Drawing (Salvador & Oberman, 2022), ImageNet-Cartoon (Salvador & Oberman, 2022), and ImageNet-Sketch (Wang et al., 2019), ObjectNet (Barbu et al., 2019), and ImageNet-C (Hendrycks & Dietterich, 2019). Although ObjectNet and ImageNet-C were originally designed solely for evaluating the OOD performance of ImageNet pre-trained models, with restrictions on their use for training, we extend their application in this benchmark by fine-tuning models on these datasets and evaluating their robustness on other OOD datasets. For detailed descriptions of each dataset, please refer to Appendix A.1.

## 4 EXPERIMENTS

We use the RIB benchmark to assess the robustness of different pre-trained models to fine-tune on a set of related downstream tasks. The goal is to assess which methods of fine-tuning do best across multiple pre-training datasets.

## 4.1 EXPERIMENTAL DETAILS

**Pre-Trained Models**    We use several architectures of Vision Transformer (ViT) (Dosovitskiy et al., 2021) and ResNet (He et al., 2016). The models are pre-trained on ImageNet-1K (Russakovsky et al., 2015), or ImageNet-21K (Ridnik et al., 2021) and then fine-tuned on ImageNet-1K. The standard data augmentation and regularization technique for ViT, AugReg (Steiner et al., 2022) can also be used for training on ImageNet-1K or ImageNet-21K. Alternatively, some models are pre-trained on LAION-2B (Schuhmann et al., 2022) or OpenAI CLIP (Radford et al., 2021), followed by fine-tuning on ImageNet-1K. In other words, all pre-trained models are trained on ImageNet-1K to directly leverage its classifier before conducting experiments. For simplicity, we refer to them by the names of the first pre-training datasets (*e.g.*, ImageNet-21K, LAION-2B).

In the main paper, we focus on ImageNet-1K with AugReg pre-trained ViT-B/16 and experiments using other pre-trained models are reported in Appendix D along with ImageNet-1K with SAM (Chen et al., 2022) and ImageNet-21K-P (Ridnik et al., 2021). We employ the timm (Wightman, 2019) and torchvision (maintainers & contributors, 2016) for acquiring model weights and implementation. Please refer to Appendix A.2 for more details.

**Methods**    We employ standard fine-tuning methods, regularization-based continual learning methods, and robust fine-tuning methods for measuring performance on the proposed benchmark. The fine-tuning methods we evaluate include vanilla fine-tuning (FT), Linear Probing, LoRA (Hu et al.,

Table 1: The average accuracy of various pre-trained ViT-B/16 on each OOD dataset. LAION-2B pre-trained model generally has the best performance.

| $D_{\text{pre}}$ | ImageNet-1K | Realistic OOD | | | | | Synthetic OOD | | |
|---|---|---|---|---|---|---|---|---|---|
| | | IN-V2 | IN-A | IN-R | IN-Sketch | ObjNet | IN-Cartoon | IN-Drawing | IN-C |
| IN-1K + AugReg | 79.2 | 66.4 | 15.0 | 38.0 | 28.0 | 25.7 | 66.2 | 39.1 | 56.0 |
| IN-21K | 81.8 | 71.4 | 32.0 | 47.3 | 35.8 | 33.1 | 69.4 | 44.1 | 58.3 |
| IN-21K + AugReg | 84.5 | 74.0 | 43.2 | 56.8 | 43.2 | 39.1 | 75.1 | 54.9 | **66.5** |
| OpenAI | 85.3 | **75.7** | **47.3** | 65.9 | 50.9 | **50.7** | 76.3 | 55.7 | 62.6 |
| LAION-2B | **85.5** | 75.6 | 41.5 | **68.8** | **55.4** | 42.3 | **78.2** | **58.4** | 63.0 |

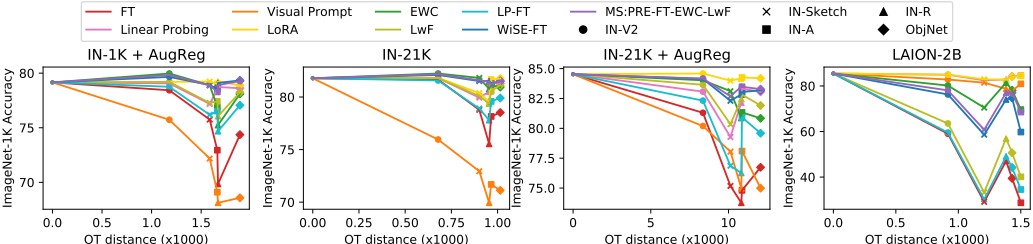

Figure 3: **Relationship between post fine-tuning ImageNet-1K accuracy and the distance between ImageNet-1K and the downstream dataset.** As the distance increases, accuracy generally decreases across fine-tuning methods. We exclude synthetic datasets since it is made from ImageNet-1K validation set that causes interference.

Table 2: Pearson correlation coefficient between the accuracy on ImageNet-1K and the dataset distance between ImageNet-1K and each downstream dataset. There is a negative correlation between accuracy and dataset distance. Notably, FT and Prompter consistently exhibit a strong negative correlation across different pre-trained models.

| Method | FT | LinearProbing | Prompter | LoRA | EWC | LwF | LP-FT | WiSE-FT | MS:PRE-FT-EWC-LwF |
|---|---|---|---|---|---|---|---|---|---|
| IN-1K + AugReg | -0.64 | -0.22 | -0.91 | -0.63 | -0.57 | -0.49 | -0.59 | -0.46 | -0.54 |
| IN-21K | -0.77 | -0.36 | -0.92 | -0.25 | -0.88 | -0.56 | -0.69 | -0.92 | -0.89 |
| IN-21K + AugReg | -0.68 | 0.10 | -0.86 | -0.63 | -0.91 | -0.38 | -0.39 | -0.52 | -0.51 |
| LAION-2B | -0.67 | -0.19 | -0.74 | -0.31 | -0.32 | -0.44 | -0.56 | -0.31 | -0.13 |

2021), Visual Prompt (Bahng et al., 2022), LwF (Li & Hoiem, 2017), and EWC (Kirkpatrick et al., 2017)[2]. We do not use LoRA for ResNet as they are designed for ViT. We also employ robust fine-tuning methods; LP-FT (Kumar et al., 2022), WiSE-FT (Wortsman et al., 2022b), and uniform model soup (Wortsman et al., 2022a), which averages the parameters of pre-trained model, vanilla fine-tuned model (FT), LwF, and EWC. We denote the uniform model soup MS:PRE-FT-EWC-LwF to reveal the source of the parameters.

**Training** Each pre-trained model is fine-tuned on the downstream dataset for 10 epochs with a batch size of 64. We use stochastic gradient descent (SGD) with a learning rate of 0.005 and a momentum of 0.9 with cosine annealing (Loshchilov & Hutter, 2017). Visual Prompt is trained for 10 epochs with a learning rate of 40 without weight decay following Bahng et al. (2022). The experiment was conducted on NVIDIA A100. Please refer to Appendix A.3 and the code repository for detailed implementation.

**Dataset Distance** We measure the distance between datasets by using Optimal Transport Dataset Distance (OTDD) (Alvarez-Melis & Fusi, 2020) and Normalized Compression Distance (NCD) (Cilibrasi & Vitányi, 2005). We measure the distance in the image space and the feature space from ImageNet-1K with AugReg pre-trained ViT-16/B, class tokens before the classifier layer. For NCD, we employ concatenated images and features. We use OTDD in the feature space in the main paper and other distance metrics are discussed in Appendix B.

---

[2]We do not use other continual learning methods as the pre-training dataset is not accessible, and to ensure a fair comparison with other methods.

Table 3: mean Robustness Improvement ($mRI$) of each method with different architectures and pre-training datasets. Model Soup (MS) and WiSE-FT generally achieve the best performance while Linear Probing performs the best with LAION-2B pre-trained models.

| Architecture | ViT-B/16 | | | | | ViT-B/32 | | | | ViT-S/16 | | ViT-S/32 | ViT-L/16 | ResNet-18 | ResNet-50 |
|---|---|---|---|---|---|---|---|---|---|---|---|---|---|---|---|
| Method | IN-1K +AugReg | IN-21K | IN-21K +AugReg | OpenAI | LAION-2B | IN-1K +AugReg | IN-21K +AugReg | OpenAI | LAION-2B | IN-1K +AugReg | IN-21K +AugReg | IN-21K +AugReg | IN-21K +AugReg | IN-1K | IN-1K |
| FT | 1.3 | -0.1 | -5.5 | -38.0 | -38.1 | -0.0 | -0.1 | -28.7 | -31.6 | -3.2 | -2.3 | -2.9 | -2.1 | -5.2 | -5.2 |
| Linear Probing | 0.7 | 0.4 | -0.3 | **-2.0** | **-2.0** | 1.1 | 0.3 | **-1.3** | **-1.4** | 0.3 | -0.2 | -0.1 | -1.3 | -7.3 | -11.2 |
| Visual Prompt | -4.5 | -9.4 | -8.8 | -8.4 | -8.2 | -5.4 | -8.4 | -8.0 | -8.4 | -7.4 | -9.2 | -9.6 | -12.9 | -8.3 | -6.5 |
| LoRA | 0.9 | -0.3 | -2.1 | -3.6 | -3.6 | 0.9 | 0.9 | -1.8 | -1.9 | 0.9 | -1.5 | 0.4 | 1.0 | - | - |
| EWC | 2.8 | 1.4 | 0.6 | -12.7 | -12.5 | 1.3 | 1.6 | -7.0 | -10.0 | 1.6 | 1.6 | 1.0 | 1.1 | -5.7 | -8.9 |
| LwF | 3.1 | 1.6 | -1.0 | -33.1 | -33.9 | 1.8 | 1.7 | -23.9 | -26.7 | 0.6 | 0.5 | 0.3 | -0.2 | -1.9 | -5.8 |
| LP-FT | 2.3 | 0.5 | -2.6 | -36.9 | -37.1 | 1.5 | 1.2 | -27.7 | -30.8 | -1.2 | -0.8 | -1.1 | -3.5 | -4.8 | -5.1 |
| WiSE-FT | 3.6 | 2.5 | 1.7 | -18.1 | -21.6 | 2.5 | **3.0** | -9.7 | -13.5 | 2.9 | 2.8 | **2.3** | 2.3 | **0.7** | **1.2** |
| MS | **3.9** | **2.7** | **2.2** | -16.0 | -17.9 | **2.5** | 2.8 | -8.1 | -10.9 | **3.0** | 2.3 | **2.8** | **2.5** | -0.1 | -0.5 |

Table 4: $RI$ and $mRI$ of ImageNet-1K with AugReg pre-trained ViT-B/16 with different fine-tuning methods and downstream datasets on each OOD dataset in ImageNet-RIB.

| Method | $mRI$ | Realistic Downstream Dataset | | | | | Synthetic Downstream Dataset | | |
|---|---|---|---|---|---|---|---|---|---|
| | | IN-V2 | IN-A | IN-R | IN-Sketch | ObjNet | IN-Cartoon | IN-Drawing | IN-C |
| FT | 1.3 | 2.9 | -4.0 | 2.8 | 4.4 | -2.7 | 0.6 | 0.4 | 5.9 |
| Linear Probing | 0.7 | 0.1 | -0.1 | 0.8 | 1.2 | 0.3 | 0.2 | 0.1 | 3.2 |
| Visual Prompt | -4.5 | -2.3 | -9.1 | -4.9 | -1.6 | -11.2 | -3.9 | -4.3 | 1.7 |
| LoRA | 0.9 | 0.2 | 0.4 | 1.1 | 2.6 | 0.3 | -0.1 | 1.3 | 1.1 |
| EWC | 2.8 | 2.9 | -0.2 | 5.2 | 4.4 | 1.4 | 1.6 | 2.8 | 4.3 |
| LwF | 3.1 | 2.8 | -0.0 | 6.2 | 4.6 | 0.7 | 1.9 | 2.1 | 6.5 |
| LP-FT | 2.3 | **3.0** | -0.9 | 5.2 | 4.5 | -0.1 | 1.2 | 0.6 | 4.7 |
| WiSE-FT | 3.6 | 2.5 | **0.7** | 7.5 | 4.5 | 2.1 | 2.3 | 3.0 | 6.5 |
| MS:PRE-FT-EWC-LwF | **3.9** | 2.7 | 0.7 | **7.8** | **5.0** | **2.2** | **2.4** | **3.3** | **6.7** |

## 4.2 OPTIMAL TRANSPORT DATASET DISTANCE ALIGNS WITH IMAGENET-1K ACCURACY DROP DURING FINE-TUNING

First, we start with the baseline of assessing model performance on the set of OOD tasks without any fine-tuning. Not surprisingly, models pre-trained on larger and more diverse datasets have better performance on both ImageNet-1K and downstream datasets as shown in Table 1. However, the ImageNet-21K with AugReg pre-trained model achieves better performance on ImageNet-C than LAION-2B pre-trained model since AugReg includes several corruptions in ImageNet-C (*e.g.*, brightness and contrast).

We consider how the accuracy on ImageNet-1K changes after fine-tuning on each downstream dataset. We compute the Optimal Transport Dataset Distance (OTDD) (Alvarez-Melis & Fusi, 2020) between downstream datasets and ImageNet-1K in the feature space extracted from pre-trained ViT-B/16 models. Synthetic datasets are excluded from this comparison, as they are generated from the ImageNet-1K validation set. In general, ImageNet-1K accuracy decreases as the distance between the downstream dataset and ImageNet-1K (pre-training dataset) increases as shown in Figure 3. We also calculate the Pearson correlation coefficient between the accuracy and the distance in Table 2 and there is a negative correlation regardless of the method except linear probing. Especially, FT and Prompter show a strong correlation (< -0.5). However, we did not find clear evidence of a correlation between the OTDD and the accuracy on out-of-distribution (OOD) datasets after fine-tuning on downstream datasets. Please refer to Appendix B for comparison with other dataset distances.

## 4.3 MODELS THAT COMBINE CONTINUAL LEARNING WITH ROBUST FINE-TUNING DO BEST

Table 5 presents accuracy on each OOD dataset before and after fine-tuning an ImageNet-1K with AugReg pre-trained ViT-B/16 model with each method on the downstream dataset. We also illustrate performance on each corruption in ImageNet-C in Table 34 in the Appendix. Linear probing (LP) generally changes performance on both ImageNet-1K and OOD datasets less than fine-tuning (FT) as the backbone network is fixed. However, both methods exhibit similar increase and decrease patterns. Visual Prompt reduces performance even on ImageNet-1K after fine-tuning on synthetic datasets of the ImageNet validation set. This is inconsistent with Bahng et al. (2022), which showed its robustness to OOD datasets. Continual learning methods and robust fine-tuning methods generally improve performance on most OOD datasets after fine-tuning on the downstream datasets. A strong correlation exists between ImageNet-R, ImageNet-Sketch, and ImageNet-Drawing, as they share drawing and sketch renditions, and ImageNet-R and ImageNet-Sketch share images. Fine-tuning on ImageNet-C improves performance on other synthetic datasets but the converse does not hold. This is because ImageNet-C contains 15 different corruptions with 5 different severity.

From these results, we see that the combination of a robust fine-tuning method (Wortsman et al., 2022a) with continual learning methods (MS:PRE-FT-EWC-LwF) achieves the highest mean robustness improvement ($mRI$) across different backbones and pre-training datasets as shown in Table 3. Moreover, end-to-end continual learning methods show comparable performance to the multi-stage method (Kumar et al., 2022) or the post-hoc robustness method (Wortsman et al., 2022b). We believe that this shows the potential of continual learning methods in the field of robust fine-tuning. The robustness of linear probing and Visual Prompt remains relatively unchanged since they do not modify the models' weights significantly but their performance on the downstream dataset tends to be worse (see Appendix D.2). Consequently, they have much better performance with LAION-2B pre-trained models compared to other methods, which show a significant robustness decrease.

Individual robustness improvement scores ($RI$) after fine-tuning on each downstream dataset with ImageNet-1K with AugReg pre-trained ViT-B/16 also show that MS:PRE-FT-EWC-LwF consistently performs the highest in most downstream datasets, followed by WiSE-FT as demonstrated in Table 4. This is because they directly use the weights of pre-trained models, thus taking advantage of their robustness. In contrast, Visual Prompt severely deteriorates robustness with all downstream datasets.

### 4.4 Paradoxically, Models Pre-Trained on the Largest Datasets Do Worst After Fine-Tuning

The extent of robustness degradation increases with the size and diversity of the pre-training dataset, as illustrated in Table 3 and Figure 4. As a result, the robustness of fine-tuned models pre-trained on larger datasets (*e.g.*, LAION-2B, OpenAI) exhibit worse robustness compared to those pre-trained on smaller datasets and their corresponding fine-tuned counterparts when using vanilla fine-tuning. One possible explanation is that models pre-trained on the larger, more diverse dataset demonstrate higher robustness to OOD datasets (see Table 1). Consequently, these models have more room for performance degradation from catastrophic forgetting. However, this does not fully explain the pronounced robustness loss observed in LAION-2B pre-trained models and OpenAI CLIPs, particularly when compared to ImageNet-21K with AugReg pre-trained models, which exhibit similar initial robustness. Moreover, Appendix C shows that these models learn downstream datasets slower than the ImageNet-21K with AugReg pre-trained model and the catastrophic robustness degradation happens in the beginning of fine-tuning.

Notably, ImageNet-21K and its variants begin to cause robustness degradation, especially when using vanilla fine-tuning. This could be an early indicator of performance decay in larger pre-trained models. Although ImageNet-21K is the second-largest dataset with 14 million images, it is much smaller than LAION-2B, which contains two billion images. We hypothesize that this discrepancy in dataset size contributes to the difference in robustness degradation. However, further investigation is required to pinpoint when severe robustness degradation begins and to identify its underlying causes.

## 5 Discussion

In this work, we introduced ImageNet-RIB (Robustness Inheritance Benchmark), a comprehensive benchmark designed to assess the robustness of fine-tuned models relative to pre-trained models across diverse out-of-distribution (OOD) datasets. A key distinction of ImageNet-RIB is that it fine-tunes models on multiple downstream datasets and evaluates their performance on various OOD datasets, providing a more holistic understanding of robustness compared to the prior benchmark (Taori et al., 2020), which focused on a single downstream dataset. This expanded framework allows us to better examine how downstream dataset distributions affect OOD performance.

Our results demonstrate that continual learning methods and robust fine-tuning approaches, particularly in combination, are effective in preserving or even improving robustness. Specifically, the combination of Model Soup with continual learning techniques consistently achieved superior performance. This finding underscores the potential of integrating these strategies to mitigate catastrophic forgetting and enhance the robustness to OOD datasets.

We also found that models pre-trained on larger, more diverse datasets, such as LAION-2B, experienced more severe robustness degradation during fine-tuning. While these models exhibited high initial robustness, the performance drop was more prominent compared to models pre-trained on smaller datasets like ImageNet-1K, leading to even worse performance. In these scenarios, simpler

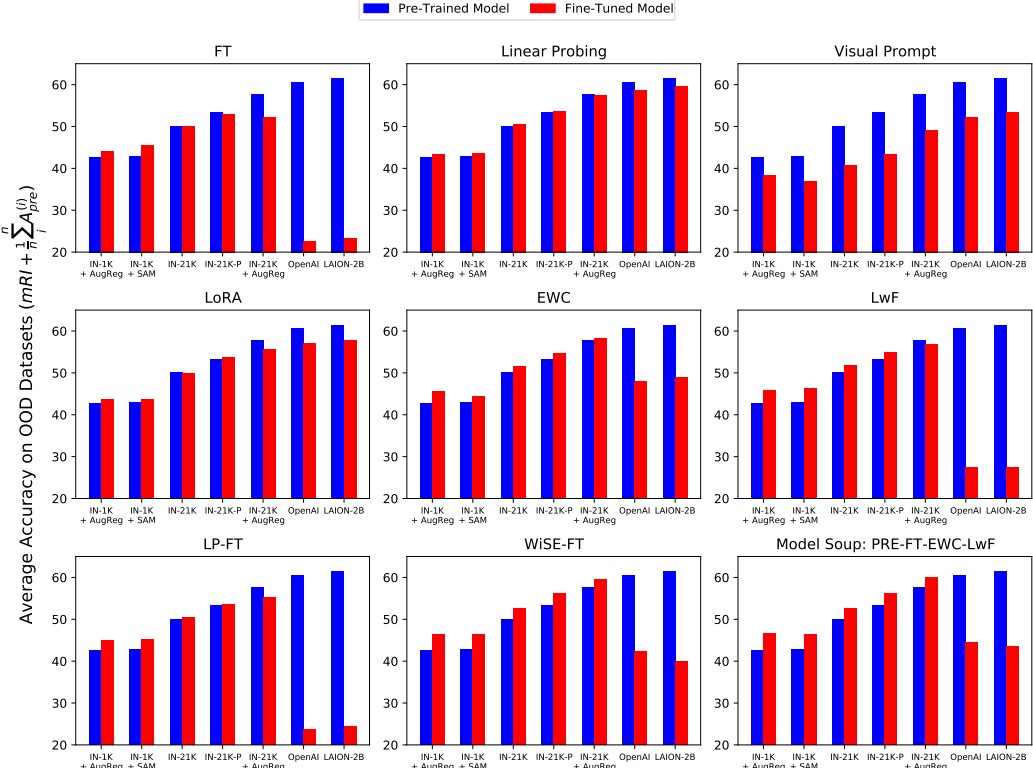

Figure 4: **Fine-tuning LAION-2B pre-trained model and OpenAI CLIP cause severe robustness loss relative to ImageNet-1K pre-trained model.** The average accuracy on OOD datasets before (blue) and after (red) fine-tuning with each method on downstream datasets. The red bar is calculated directly by evaluating pre-trained models on OOD datasets while the blue bar is calculated by adding $mRI$ of each method to the pre-trained models' accuracy. Note that it is identical to the average accuracy on OOD datasets after fine-tuning on each downstream dataset ($mRI + \frac{1}{n}\sum_i^n A_{\text{pre}}^{(i)} = \frac{1}{n}\sum_j \frac{1}{n-1}\sum_{i,i\neq j}^n A_{\text{down}}^{(i)}$). Fine-tuning LAION-2B pre-trained model and OpenAI CLIP on the downstream OOD datasets causes severe robustness loss leading to worse performance than ImageNet-1K with AugReg pre-trained model. Conversely, ImageNet-1K with AugReg pre-trained model improves robustness after fine-tuning. Note that the difference between red and blue bars is $mRI$.

methods such as linear probing, which freeze most of the model's layers, were more effective in maintaining robustness, as more complex methods often led to significant performance degradation. This highlights the nuanced relationship between the size and diversity of the pre-training dataset and the model's ability to generalize after fine-tuning.

Despite these contributions, our work has certain limitations. We primarily focused on fine-tuning, continual learning, and robust fine-tuning methods. Future research could explore the role of advanced data augmentation techniques (Cubuk et al., 2019; Hendrycks et al., 2021a; Wang et al., 2023) in further improving OOD robustness. Moreover, while Optimal Transport Dataset Distance (OTDD) shows promise in predicting performance degradation on the pre-training dataset after fine-tuning, more refined metrics are needed to better capture and address robustness degradation.

Future research should focus on understanding the significant robustness degradation after fine-tuning observed in models pre-trained on larger datasets like LAION-2B. Uncovering why such extensive pre-training leads to worse robustness compared to models pre-trained on smaller datasets could inform more effective robustness fine-tuning strategies. Moreover, expanding the scope of our analysis to include a broader range of model architectures and datasets would further enhance the generalizability of our findings. We believe that ImageNet-RIB offers a valuable framework for studying the impact of fine-tuning on OOD generalization, and we hope this work encourages further research into developing more robust and generalizable machine learning models.

Table 5: The accuracy on each OOD dataset after fine-tuning on ImageNet-1K with AugReg pre-trained ViT-B/16 on the downstream datasets with various methods. Note that ImageNet-Drawing, ImageNet-Cartoon, and ImageNet-C are generated from the ImageNet validation set. Green and red indicate relative performance increases and decreases, respectively, compared to the pre-trained model. Bold indicates the best performance on each evaluation dataset.

| Method | Downstream Dataset | $D_{pre}$ IN | IN-V2 | Realistic OOD IN-A | IN-R | IN-Sketch | ObjNet | Synthetic OOD IN-Cartoon | IN-Drawing | IN-C |
|---|---|---|---|---|---|---|---|---|---|---|
| Pre-Trained | | 79.2 | 66.4 | 15.0 | 38.0 | 28.0 | 25.7 | 66.2 | 39.1 | 56.0 |
| FT | IN-V2 | 78.4 | - | 25.2 | 41.9 | 29.2 | 37.1 | 64.7 | 40.4 | 57.4 |
| | IN-A | 72.9 | 60.6 | - | 36.7 | 24.9 | 35.0 | 55.3 | 32.6 | 53.5 |
| | IN-R | 69.8 | 59.2 | 20.9 | - | 46.7 | 32.0 | 61.3 | 51.4 | 52.0 |
| | IN-Sketch | 75.7 | 63.9 | 17.3 | 59.1 | - | 33.0 | 66.3 | 50.8 | 53.8 |
| | ObjNet | 74.4 | 62.2 | 24.9 | 36.3 | 25.1 | - | 55.6 | 33.6 | 52.3 |
| | IN-Cartoon | 85.2 | 63.5 | 19.9 | 40.5 | 29.5 | 33.5 | - | 41.2 | 51.3 |
| | IN-Drawing | 81.5 | 62.9 | 16.5 | 41.1 | 32.7 | 32.4 | 64.2 | - | 56.0 |
| | IN-C | 99.8 | 61.1 | 13.9 | 37.0 | 25.1 | 27.7 | 92.2 | 70.2 | - |
| Linear Probing | IN-V2 | 79.1 | - | 15.6 | 38.2 | 28.1 | 33.1 | 66.2 | 39.0 | 55.9 |
| | IN-A | 78.6 | 65.9 | - | 38.5 | 27.4 | 34.1 | 65.6 | 38.6 | 55.8 |
| | IN-R | 78.7 | 66.6 | 17.1 | - | 30.2 | 33.4 | 66.1 | 39.8 | 56.2 |
| | IN-Sketch | 77.2 | 64.8 | 16.6 | 46.3 | - | 33.5 | 65.6 | 40.5 | 54.5 |
| | ObjNet | 78.6 | 65.9 | 18.1 | 38.6 | 27.9 | - | 65.1 | 39.3 | 56.1 |
| | IN-Cartoon | 80.5 | 65.4 | 15.1 | 39.2 | 28.1 | 32.2 | - | 40.9 | 55.6 |
| | IN-Drawing | 78.1 | 65.2 | 14.9 | 41.3 | 28.5 | 33.3 | 65.6 | - | 54.3 |
| | IN-C | 97.1 | 61.9 | 15.1 | 36.8 | 25.2 | 28.3 | 83.3 | 57.4 | - |
| Visual Prompt (Bahng et al., 2022) | IN-V2 | 75.7 | - | 12.7 | 39.6 | 27.4 | 34.4 | 60.5 | 36.7 | 47.9 |
| | IN-A | 69.1 | 57.1 | - | 36.3 | 21.9 | 32.7 | 50.6 | 26.1 | 38.0 |
| | IN-R | 68.1 | 55.9 | 9.6 | - | 36.2 | 30.0 | 55.7 | 41.8 | 40.1 |
| | IN-Sketch | 72.2 | 59.5 | 9.4 | 51.6 | - | 32.3 | 60.6 | 44.9 | 44.3 |
| | ObjNet | 68.6 | 56.2 | 13.0 | 33.7 | 22.2 | - | 46.8 | 23.0 | 35.3 |
| | IN-Cartoon | 74.5 | 61.2 | 10.2 | 41.2 | 27.0 | 31.5 | - | 35.2 | 41.8 |
| | IN-Drawing | 72.1 | 59.4 | 8.4 | 42.2 | 28.8 | 30.6 | 59.3 | - | 44.2 |
| | IN-C | 77.9 | 65.2 | 14.8 | 40.1 | 28.3 | 35.7 | 63.5 | 49.8 | - |
| LoRA (Hu et al., 2021) | IN-V2 | 79.2 | - | 15.3 | 38.2 | 28.1 | 33.2 | 66.4 | 39.3 | 56.1 |
| | IN-A | 79.0 | 66.4 | - | 38.9 | 27.8 | 35.5 | 65.2 | 39.3 | 56.5 |
| | IN-R | 79.2 | 66.8 | 16.7 | - | 29.7 | 34.8 | 66.9 | 40.0 | 56.7 |
| | IN-Sketch | 79.2 | 66.8 | 16.5 | 45.9 | - | 34.6 | 67.7 | 44.1 | 56.6 |
| | ObjNet | 78.9 | 66.3 | 18.3 | 39.3 | 27.8 | - | 65.1 | 39.2 | 55.0 |
| | IN-Cartoon | 78.7 | 65.8 | 14.8 | 39.3 | 28.3 | 32.1 | - | 39.8 | 54.6 |
| | IN-Drawing | 77.9 | 66.3 | 15.0 | 43.7 | 32.1 | 33.5 | 66.4 | - | 55.1 |
| | IN-C | 79.9 | 67.4 | 16.3 | 39.2 | 28.1 | 34.1 | 67.5 | 40.8 | - |
| EWC (Kirkpatrick et al., 2017) | IN-V2 | 80.0 | - | 19.7 | 41.8 | 29.4 | 36.8 | 67.1 | 42.8 | 58.2 |
| | IN-A | 76.9 | 64.9 | - | 40.4 | 27.8 | 38.2 | 61.1 | 36.5 | 56.6 |
| | IN-R | 75.2 | 63.9 | 19.0 | - | 43.9 | 33.3 | 66.4 | 57.5 | 56.1 |
| | IN-Sketch | 78.9 | 66.6 | 16.6 | 52.2 | - | 34.2 | 68.3 | 49.6 | 57.2 |
| | ObjNet | 78.1 | 66.2 | 23.1 | 40.9 | 29.0 | - | 62.4 | 39.8 | 56.9 |
| | IN-Cartoon | 79.2 | 66.0 | 16.5 | 42.7 | 29.9 | 33.8 | - | 42.6 | 54.7 |
| | IN-Drawing | 79.3 | 66.7 | 16.3 | 44.5 | 34.0 | 34.7 | 67.9 | - | 58.3 |
| | IN-C | 80.1 | 67.8 | 20.0 | 42.5 | 31.2 | 37.5 | 66.8 | 50.0 | - |
| LwF (Li & Hoiem, 2017) | IN-V2 | 79.2 | - | 22.9 | 41.3 | 29.4 | 36.4 | 65.8 | 41.0 | 57.9 |
| | IN-A | 77.4 | 65.5 | - | 39.4 | 27.5 | 36.7 | 61.8 | 38.3 | 57.2 |
| | IN-R | 76.1 | 64.7 | 21.7 | - | 47.8 | 34.1 | 66.8 | 54.9 | 57.2 |
| | IN-Sketch | 77.3 | 65.2 | 17.3 | 57.8 | - | 33.5 | 67.8 | 49.6 | 55.2 |
| | ObjNet | 78.2 | 66.2 | 24.1 | 38.4 | 27.3 | - | 62.3 | 38.8 | 56.3 |
| | IN-Cartoon | 87.2 | 65.9 | 19.4 | 41.2 | 29.9 | 34.2 | - | 42.7 | 55.6 |
| | IN-Drawing | 84.0 | 65.4 | 17.7 | 41.9 | 33.2 | 33.4 | 67.7 | - | 58.2 |
| | IN-C | 99.2 | 65.8 | 13.5 | 40.7 | 27.8 | 31.4 | 90.6 | 61.7 | - |
| LP-FT (Kumar et al., 2022) | IN-V2 | 78.8 | - | 24.7 | 41.6 | 29.3 | 36.8 | 65.3 | 41.3 | 57.6 |
| | IN-A | 76.5 | 64.6 | - | 38.2 | 27.4 | 37.1 | 60.5 | 36.7 | 56.2 |
| | IN-R | 74.7 | 63.4 | 21.1 | - | 46.9 | 34.7 | 65.4 | 53.1 | 55.3 |
| | IN-Sketch | 76.2 | 64.5 | 18.0 | 58.8 | - | 33.9 | 67.0 | 48.9 | 54.4 |
| | ObjNet | 77.1 | 64.9 | 24.9 | 38.2 | 26.8 | - | 60.7 | 37.7 | 54.9 |
| | IN-Cartoon | 86.3 | 64.2 | 19.5 | 41.0 | 29.9 | 33.5 | - | 43.1 | 52.8 |
| | IN-Drawing | 82.1 | 63.2 | 16.5 | 41.7 | 32.9 | 32.0 | 64.8 | - | 56.0 |
| | IN-C | 98.0 | 61.0 | 13.7 | 37.5 | 25.7 | 27.3 | 87.1 | 66.0 | - |
| WiSE-FT (Wortsman et al., 2022b) | IN-V2 | 79.7 | - | 21.3 | 40.5 | 29.5 | 36.0 | 66.5 | 40.9 | 58.0 |
| | IN-A | 78.6 | 66.4 | - | 39.3 | 28.5 | 37.1 | 64.4 | 38.6 | 57.8 |
| | IN-R | 79.1 | 67.1 | 23.0 | - | 44.7 | 37.4 | 69.5 | 54.7 | 59.6 |
| | IN-Sketch | 78.9 | 66.4 | 17.6 | 52.1 | - | 34.7 | 68.7 | 48.7 | 57.3 |
| | ObjNet | 79.3 | 67.3 | 23.5 | 40.0 | 29.0 | - | 65.2 | 40.5 | 57.6 |
| | IN-Cartoon | 83.8 | 66.5 | 19.3 | 41.0 | 30.4 | 34.9 | - | 43.2 | 56.3 |
| | IN-Drawing | 82.5 | 66.9 | 18.5 | 42.2 | 33.5 | 35.0 | 68.2 | - | 59.5 |
| | IN-C | 93.4 | 66.9 | 18.7 | 41.3 | 29.9 | 34.7 | 82.4 | 57.6 | - |
| Model Soup PRE-FT-EWC-LwF (Wortsman et al., 2022a) | IN-V2 | 79.8 | - | 21.0 | 41.0 | 29.7 | 36.0 | 66.9 | 41.7 | 58.0 |
| | IN-A | 78.3 | 66.4 | - | 39.7 | 28.5 | 37.5 | 63.7 | 38.4 | 57.8 |
| | IN-R | 78.9 | 67.1 | 23.1 | - | 45.9 | 37.2 | 69.6 | 55.8 | 59.6 |
| | IN-Sketch | 78.9 | 66.6 | 17.5 | 54.0 | - | 34.6 | 69.1 | 49.8 | 57.5 |
| | ObjNet | 79.3 | 67.4 | 24.1 | 40.3 | 29.1 | - | 64.9 | 40.6 | 57.7 |
| | IN-Cartoon | 83.7 | 66.4 | 18.9 | 41.8 | 30.6 | 34.7 | - | 43.6 | 56.2 |
| | IN-Drawing | 82.6 | 66.9 | 18.4 | 43.0 | 34.0 | 35.2 | 68.7 | - | 59.7 |
| | IN-C | 92.6 | 67.5 | 18.6 | 42.3 | 30.6 | 35.3 | 81.3 | 57.3 | - |

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

---

**Algorithm 1** Protocol of ImageNet-RIB

---

**Input:** A set of out-of-distribution datasets $\mathbb{D} = \{D_1, D_2, ..., D_n\}$, pre-trained model $f_{\text{pre}}$, fine-tuning method fine-tune$(\cdot)$.
**Output:** Mean robustness improvement, $mRI$.
 1: **procedure** IMAGENET-RIB
 2:     **for** $D_{\text{down}}(= D_i) \in \mathbb{D}$ **do**
 3:         $f_{\text{down}} \leftarrow$ fine-tune$(f_{\text{pre}}; D_{\text{down}})$            $\triangleright$ Fine-tune pre-trained model $f_{\text{pre}}$ on $D_{\text{down}}$.
 4:         **for** $D_j \in \mathbb{D} \setminus D_{\text{down}}$ **do**
 5:             $A_i^{(j)} =$ eval$(f_{\text{down}}; D_j)$                        $\triangleright$ Evaluate $f_{\text{down}}$ on $D_j$.
 6:             $A_{\text{pre}}^{(j)} =$ eval$(f_{\text{pre}}; D_j)$                        $\triangleright$ Evaluate $f_{\text{pre}}$ on $D_j$.
 7:         **end for**
 8:         $RI_i = \frac{1}{n-1} \sum_{j=1, j \neq i}^{n} A_i^{(j)} - A_{\text{pre}}^{(j)}$      $\triangleright$ Calculate Robustness Improvement ($RI$).
 9:     **end for**
10:     $mRI = \sum_i^n RI_i$                            $\triangleright$ Calculate mean $RI$.
11: **end procedure**

---

# APPENDIX

## A  EXPERIMENTAL DETAILS

In this section, we describe the details of the experimental setup.

### A.1  OUT-OF-DISTRIBUTION DATASETS IN IMAGENET-RIB

We leverage all existing ImageNet variants designed to measure the robustness of the trained network during distribution shifts. ImageNet-O (Hendrycks et al., 2021b) is not used since it is an out-of-distribution detection dataset.

**ImageNet-V2 (Recht et al., 2019)**  ImageNet-V2 is designed to have a distribution as similar as possible to the original ImageNet-1K. It has 50,000 images with 1,000 classes same as the original validation set. The dataset is used under the MIT license.

**ImageNet-A (Hendrycks et al., 2021b)**  ImageNet-A is an adversarially filtered test image that ImageNet-1K pre-trained ResNet-50 (He et al., 2016) is difficult to predict correctly. It contains 7,500 images with 200 difficult subclasses from ImageNet-1K. The dataset is used under the MIT license.

**ImageNet-R (Hendrycks et al., 2021a)**  ImageNet-R (Renditions) contains 30,000 images from 200 ImageNet classes with various rendition styles such as painting, sculpture, embroidery, origami, cartoon, toy, and so on. The drawing rendition overlaps with ImageNet-Sketch (Wang et al., 2019). The dataset is used under the MIT license.

**ImageNet-Sketch (Wang et al., 2019)**  ImageNet-Sketch comprises black and white sketch drawings of the ImageNet-1K classes and each class has 50 images. The dataset is used under the MIT license.

**ImageNet-Cartoon and ImageNet-Drawing (Salvador & Oberman, 2022)**  ImageNet-Cartoon and ImageNet-Drawing are to be converted from ImageNet validation set images to cartoon, and drawing styles based on generative adversarial network (Wang & Yu, 2020) and image processing (Lu et al., 2012). These simplified representations test a model's ability to identify objects from minimalistic and abstract visual information. The dataset is used under the Creative Commons Attribution 4.0 International license.

**ObjectNet (Barbu et al., 2019)**  ObjectNet is designed for evaluating object recognition models under more realistic conditions such as various poses, backgrounds, and viewpoints. There are 50,000

Table 6: Python libraries and the names of network weights for each pre-trained model.

| Architecture | $D_{\text{pre}}$ | Library | Weight Name |
|---|---|---|---|
| ViT-B/16 | IN-1K + AugReg | timm | vit_base_patch16_224.augreg_in1k |
| | IN-1K + SAM | timm | vit_base_patch16_224.sam_in1k |
| | IN-21K | timm | vit_base_patch16_224.orig_in21k_ft_in1k |
| | IN-21K + AugReg | timm | vit_base_patch16_224.augreg_in21k_ft_in1k |
| | IN-21K-P | timm | vit_base_patch16_224_miil.in21k_ft_in1k |
| | LAION-2B | timm | vit_base_patch16_clip_224.laion2b_ft_in1k |
| | OpenAI | timm | vit_base_patch16_clip_224.openai_ft_in1k |
| ViT-B/32 | IN-1K + AugReg | timm | vit_base_patch32_224.augreg_in1k |
| | IN-21K + AugReg | timm | vit_base_patch32_224.augreg_in21k_ft_in1k |
| | LAION-2B | timm | vit_base_patch32_clip_224.laion2b_ft_in1k |
| | OpenAI | timm | vit_base_patch32_clip_224.openai_ft_in1k |
| ViT-S/16 | IN-21K + AugReg | timm | vit_small_patch16_224.augreg_in21k_ft_in1k |
| ViT-S/32 | IN-21K + AugReg | timm | vit_small_patch32_224.augreg_in21k_ft_in1k |
| ViT-L/16 | IN-21K + AugReg | timm | vit_large_patch16_224.augreg_in21k_ft_in1k |
| ResNet-18 | IN-1K | torchvision | ResNet18_Weights.DEFAULT |
| ResNet-50 | IN-1K | torchvision | ResNet50_Weights.DEFAULT |

images with 313 object classes and 113 classes are overlapped with ImageNet. We only use ImageNet class objects. The dataset is used under the MIT license.

**ImageNet-C (Hendrycks & Dietterich, 2019)**   ImageNet-C is designed for measuring the robustness of models to common perturbations such as noise, blur, weather, and digital distortions. In the dataset, ImageNet validation set images are perturbed with various severity from 1 to 5. Unlike the original metrics, corruption error compared with AlexNet, we use average accuracy for consistency with other datasets. The dataset is used under the Apache-2.0 license.

## A.2 Pre-Trained Model

Table 6 lists the libraries and corresponding network weight names for each model. We use the entire models in timm and torchvision library, which are finally fine-tuned on ImageNet-1K, with patch sizes of 16 and 32, and input image shape of 224 among ViT small, base, and large. For ResNets, we use the default ImageNet-1K pre-trained weights from the torchvision library.

## A.3 Training and Hyperparameters

Each pre-trained model is fine-tuned on the downstream dataset for 10 epochs where the average accuracy on downstream datasets for each pre-trained ViT-B/16 model achieves more than 90% with vanilla fine-tuning. We applied LoRA on query and value projection layers with rank 8 following the original implementation (Hu et al., 2021). We use 2 as a temperature for calculating KL divergence for LwF following Li & Hoiem (2017). For WiSE-FT, we use the interpolation ratio between pre-trained and fine-tuned models as 0.5 following the recommendation by Wortsman et al. (2022b) instead of finding the best hyperparameters evaluated on the benchmark for the fair comparison.

## B Dataset Distance

We measure the Optimal Transport dataset distance(OTDD) (Alvarez-Melis & Fusi, 2020) between each dataset using both images and the pre-trained model features from ImageNet-1K with AugReg pre-trained ViT-B/16, as shown in Figures 5a and 5b, respectively. Since ImageNet-C comprises multiple corruptions with different severities, we do not measure the distance to ImageNet-C. As shown in Figure 5a, in the image space, ImageNet-Sketch is the farthest from other datasets as it is black and white sketch images. ImageNet-Drawing is the closest to the dataset and the ImageNet-R is the second closest as they share the same styles and images, respectively.

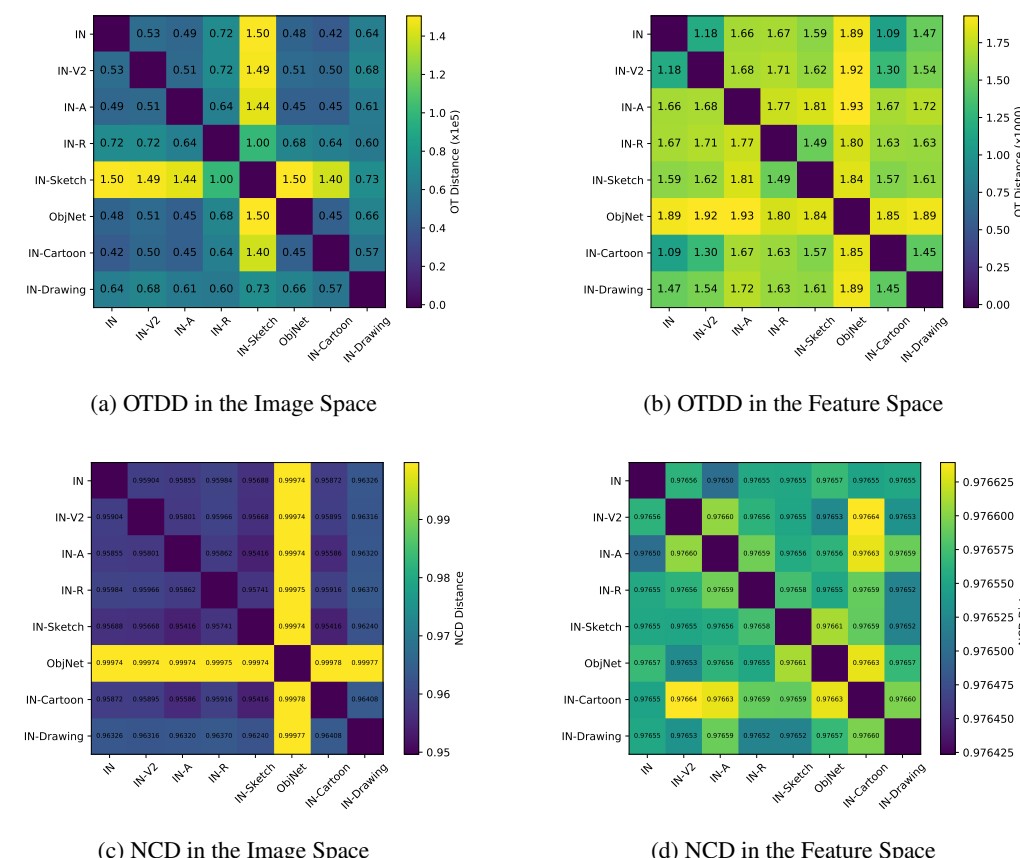

(a) OTDD in the Image Space        (b) OTDD in the Feature Space

(c) NCD in the Image Space        (d) NCD in the Feature Space

Figure 5: **Optimal Transport Dataset Distances (OTDD) in the feature space aligns with each dataset design.** Pairwise OTDD (up) and Normalized Compression Distance (NCD) (down) between datasets using images (left) and features extracted by ImageNet-1K with AugReg pre-trained ViT-B/16 on each dataset (right), respectively.

OTDD in the feature space (Figure 5b) demonstrates a better alignment with the dataset design principles. For example, ImageNet-V2 is designed to replicate the distribution of the ImageNet validation set. It leads ImageNet-V2 the closest to ImageNet-1K among realistic datasets. Moreover, the distances between ImageNet-1K and ImageNet-V2 to other datasets are consistent across both image and feature spaces. This is not true with ImageNet-Cartoon since it is a synthetic dataset based on the ImageNet validation set. As shown in Table 5, ImageNet-Cartoon improves ImageNet-1K accuracy more than ImageNet-Drawing, suggesting that the distribution shift in cartoon-style images is less severe than that of drawing-style images. Similarly, ObjectNet is intentionally collected with different viewpoints and backgrounds and it is the most distant from all other datasets in the feature space.

We also measure Normalized Compression Distance (NCD) using both images and the features from ImageNet-1K with AugReg pre-trained ViT-B/16. However, the distance between each dataset pair is too insignificant to compare with each dataset as shown in Figures 5c and 5d.

## C    Overfitting Does Not Drive Robustness Collapse of LAION-2B Pre-Trained Model and OpenAI CLIP

A potential explanation for the significant robustness decline observed in LAION-2B and OpenAI pre-trained ViT-B/16 during fine-tuning as shown in Section 4.4, is that these models may overfit earlier compared to other models. To investigate this, we analyze the robustness performance change

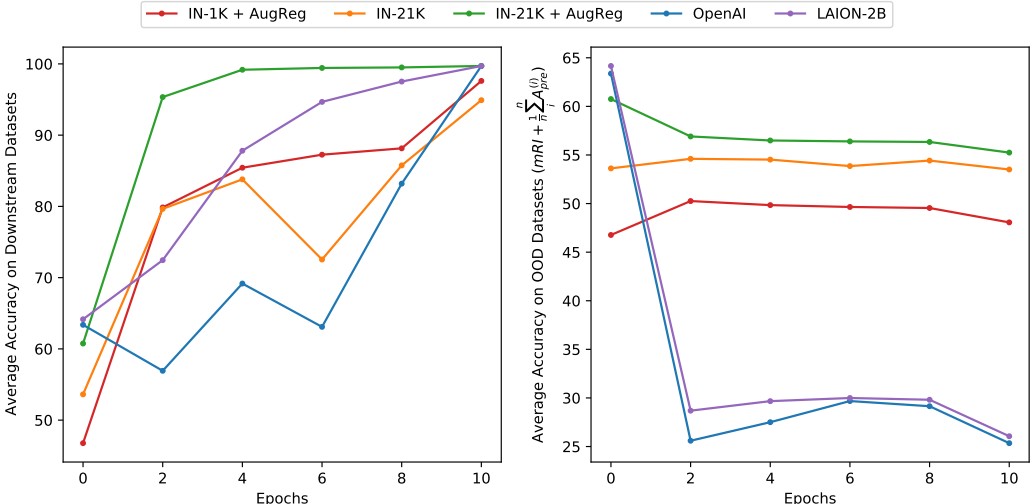

Figure 6: **Fine-tuning LAION-2B and OpenAI pre-trained model cause severe robustness loss while learning slower than ImageNet-21K with AugReg pre-trained model.** The average accuracy on Downstream Datasets (left) and the average accuracy on OOD datasets (right) while fine-tuning on the downstream dataset using vanilla fine-tuning method (FT) with ViT-B/16. Although LAION-2B pre-trained model and OpenAI CLIP learn slower than other methods, they suffer from a huge robustness drop even in the early period of fine-tuning.

throughout the fine-tuning process on the downstream dataset using a standard fine-tuning (FT) approach along with the average accuracy on downstream datasets.

Figure 6 illustrates that the ImageNet-21K model pre-trained with AugReg learns downstream datasets more rapidly than other methods, while the OpenAI CLIP models exhibit the slowest learning pace. However, only the LAION-2B and OpenAI CLIP pre-trained models experience a severe degradation in robustness to out-of-distribution datasets. This suggests that overfitting is not the primary cause of the dramatic performance decline.

## D  ADDITIONAL EXPERIMENTS WITH VARIOUS PRE-TRAINED MODELS

In this section, we also use ViT-B/16 pre-trained on ImageNet-1K with Sharpness Aware Minimization (SAM) (Chen et al., 2022)), ImageNet-21K-P (Ridnik et al., 2021), ViT-B/32 pre-trained on ImageNet-1K with SAM and OpenAI CLIP ViT-B/16 and ViT-B/32 models.

### D.1  ROBUSTNESS OF PRE-TRAINED MODELS

We evaluate pre-trained models mentioned in Appendix A.2 on OOD datasets as shown in Table 7. Larger networks with smaller patch sizes achieve higher accuracy on both ImageNet-1K and OOD datasets. Similarly, models pre-trained on larger, more diverse datasets demonstrate better performance.

### D.2  PERFORMANCE ON DOWNSTREAM DATASET

Table 8  9, and  10 demonstrate the accuracy on downstream datasets (*i.e.*, training accuracy) with ViT base, ViT large and ViT small, and ResNet, respectively. FT, LwF, and LP-FT can overfit to the downstream dataset but WiSE-FT and Model Soup (PRE-FT-LwF-EWC) have worse performance which might be due to using pre-trained model weights. Visual Prompt and LoRA rarely learn a downstream dataset.

Table 7: The average accuracy of various pre-trained models on ImageNet-1K and OOD datasets.

| Arch | $D_{pre}$ | ImageNet | Realistic OOD (Taori et al., 2020) | | | | | Synthetic OOD | | |
|------|-----------|----------|-------|------|------|-----------|--------|------------|------------|------|
| | | | IN-V2 | IN-A | IN-R | IN-Sketch | ObjNet | IN-Cartoon | IN-Drawing | IN-C |
| ViT-B/16 | IN-1K + AugReg | 79.2 | 66.4 | 15.0 | 38.0 | 28.0 | 33.0 | 66.2 | 39.1 | 56.0 |
| | IN-1K + SAM | 80.2 | 68.2 | 9.0 | 40.1 | 27.7 | 34.2 | 66.9 | 42.3 | 54.6 |
| | IN-21K | 81.8 | 71.4 | 32.0 | 47.3 | 35.8 | 42.5 | 69.4 | 44.1 | 58.3 |
| | IN-21K-P | 84.3 | 74.0 | 34.1 | 51.5 | 40.2 | 46.7 | 73.5 | 45.1 | 61.4 |
| | IN-21K + AugReg | 84.5 | 74.0 | 43.2 | 56.8 | 43.2 | 48.4 | 75.1 | 54.9 | 66.5 |
| | OpenAI | 85.3 | 75.7 | 47.3 | 65.9 | 50.9 | 50.7 | 76.3 | 55.7 | 62.6 |
| | LAION-2B | 85.5 | 75.6 | 41.5 | **68.8** | **55.4** | 51.1 | 78.2 | 58.4 | 63.0 |
| ViT-B/32 | IN-1K + SAM | 73.7 | 59.9 | 4.3 | 36.6 | 23.0 | 25.2 | 63.2 | 40.6 | 48.8 |
| | IN-21K + AugReg | 80.7 | 69.0 | 22.4 | 49.3 | 37.1 | 40.7 | 70.6 | 42.5 | 60.5 |
| | OpenAI | 82.0 | 70.9 | 22.6 | 55.8 | 45.0 | 41.5 | 71.1 | 42.5 | 57.9 |
| | LAION-2B | 82.6 | 71.6 | 22.8 | 59.2 | 49.1 | 43.5 | 73.0 | 42.3 | 57.5 |
| ViT-S/16 | IN-21K + AugReg | 81.4 | 70.3 | 27.0 | 46.0 | 32.9 | 32.2 | 67.8 | 37.7 | 58.0 |
| ViT-S/32 | IN-21K + AugReg | 76.0 | 63.9 | 11.5 | 39.7 | 26.2 | 24.8 | 62.9 | 34.3 | 52.0 |
| ViT-L/16 | IN-21K + AugReg | **85.8** | **76.2** | **55.5** | 64.4 | 51.8 | **52.8** | **79.5** | **64.6** | **72.2** |
| ResNet-18 | IN-1K | 69.8 | 57.3 | 1.1 | 33.1 | 20.2 | 18.1 | 48.2 | 20.4 | 31.7 |
| ResNet-50 | IN-1K | 80.3 | 69.5 | 16.7 | 41.6 | 28.4 | 33.0 | 61.1 | 31.1 | 46.6 |

## D.3 ROBUSTNESS IMPROVEMENT RESULTS OF DIFFERENT MODELS

Across the ImageNet pre-trained models, WiSE-FT and Model Soup consistently have better robustness improvement compared to other methods fine-tuning on realistic OOD datasets (Tables 11, 13, and 14). Linear Probing consistently achieves the best robustness improvement using LAION-2B pre-trained models (Table 16) and OpenAI CLIP models (Table 17).

## D.4 ACCURACY OF USING VARIOUS PRE-TRAINED MODELS ON EACH OOD DATASETS

The following tables present the accuracy on each OOD (out-of-distribution) dataset after fine-tuning on various datasets. Specifically:

- Tables 18, 19, 20, 21, and 22 show results for the ViT-B/16 pre-trained on ImageNet-1K with SAM, ImageNet-21K, ImageNet-21K with AugReg, ImageNet-21K-P, OpenAI and LAION-2B, respectively. Table 23 uses OpenAI CLIP ViT-B/16 as a pre-trained model.

- Tables 24, 25, 27, and 26 provide the corresponding accuracy for ViT-B/32 pre-trained on ImageNet-1K with AugReg, ImageNet-21K with AugReg, LAION-2B, respectively. Table 27 uses OpenAI CLIP ViT-B/32 as a pre-trained model.

- Table 31 report the accuracy for ViT-L/16 pre-trained on ImageNet-21K with AugReg.

- Finally, Tables 32 and 33 present results for ResNet-18 and ResNet-50 pre-trained on ImageNet-1K.

## D.5 ACCURACY OF USING VARIOUS PRE-TRAINED MODELS ON EACH CORRUPTION IN IMAGENET-C

Each pre-trained and fine-tuned model is evaluated on ImageNet-C with 15 corruptions at severity levels ranging from 1 to 5. Following the original benchmark (Hendrycks & Dietterich, 2019), we average the performance over the different severity levels. However, for consistency with other datasets, we report the results as accuracy rather than error.

Specifically:

- Table 34 shows the results for ViT-B/16 pre-trained on ImageNet-1K with AugReg, repspectively. Table 35 denotes the results for ViT-B/16 pre-trained on ImageNet-1K with SAM, while Tables 36, 37, 38 and 39 present the performance of ViT-B/16 pre-trained on ImageNet-21K, ImageNet-21K with AugReg, ImageNet-21K-P and LAION-2B, respectively. Table 40 uses OpenAI CLIP ViT-B/16 as a pre-trained model.

Table 8: Accuracy on downstream datasets after fine-tuning with each method using ViT-B/16. FT and LP-FT generally achieve the highest performance, while Visual Prompt and LoRA show the lowest.

| Arch | $D_{pre}$ | Method | Realistic Downstream Dataset | | | | | Synthetic Downstream Dataset | | |
|------|------|--------|------|------|------|------|------|------|------|------|
| | | | IN-V2 | IN-A | IN-R | IN-Sketch | ObjNet | IN-Cartoon | IN-Drawing | IN-C |
| ViT-B/16 | IN-1K + AugReg | FT | 96.3 | **97.5** | 98.4 | 96.0 | 97.7 | 97.5 | 97.6 | **100.0** |
| | | Linear Probing | 71.5 | 42.4 | 60.4 | 58.8 | 57.8 | 76.1 | 61.2 | 89.7 |
| | | Visual Prompt | 66.6 | 28.8 | 58.9 | 45.8 | 46.3 | 72.7 | 64.8 | 64.4 |
| | | LoRA | 66.5 | 18.7 | 41.8 | 39.0 | 36.7 | 69.7 | 62.4 | 58.6 |
| | | EWC | 72.0 | 54.3 | 65.3 | 50.3 | 50.8 | 76.2 | 70.3 | 67.7 |
| | | LwF | 95.8 | 95.5 | 97.3 | 95.1 | 95.1 | 96.7 | 96.5 | 100.0 |
| | | LP-FT | **96.7** | 97.2 | **98.5** | **96.1** | **97.9** | **97.5** | **97.6** | 94.6 |
| | | WiSE-FT | 81.2 | 56.6 | 71.2 | 61.4 | 63.7 | 84.0 | 74.0 | 88.8 |
| | | MS:PRE-FT-EWC-LwF | 82.4 | 67.4 | 75.5 | 66.8 | 66.8 | 85.5 | 78.7 | 88.0 |
| | IN-1K + SAM | FT | 77.9 | 67.2 | 87.2 | **84.3** | 75.1 | 87.1 | 85.7 | **100.0** |
| | | Linear Probing | 68.7 | 14.3 | 50.5 | 38.8 | 41.4 | 71.3 | 53.6 | 80.7 |
| | | Visual Prompt | 64.4 | 17.0 | 50.6 | 37.2 | 40.1 | 69.7 | 56.2 | 57.7 |
| | | LoRA | 68.2 | 10.0 | 44.9 | 32.7 | 36.7 | 69.6 | 49.6 | 67.5 |
| | | EWC | 69.0 | 23.9 | 50.4 | 43.8 | 41.3 | 72.6 | 62.3 | 59.6 |
| | | LwF | 77.6 | 62.5 | 84.2 | 81.7 | 69.7 | 85.9 | 84.0 | 99.9 |
| | | LP-FT | **78.3** | 64.9 | 86.6 | 83.5 | 74.6 | **87.2** | **86.1** | 84.4 |
| | | WiSE-FT | 72.7 | 31.4 | 64.7 | 52.6 | 52.4 | 78.9 | 68.1 | 78.8 |
| | | MS:PRE-FT-EWC-LwF | 72.8 | 36.5 | 66.7 | 55.6 | 53.0 | 79.3 | 70.7 | 80.3 |
| | IN-21K | FT | 92.2 | 94.9 | **96.3** | 92.8 | 94.3 | 94.7 | 94.1 | **100.0** |
| | | Linear Probing | 75.0 | 51.8 | 66.4 | 59.0 | 63.2 | 77.7 | 59.6 | 86.4 |
| | | Visual Prompt | 66.8 | 37.4 | 58.2 | 43.9 | 51.0 | 68.9 | 57.9 | 58.6 |
| | | LoRA | 71.5 | 38.2 | 52.9 | 39.8 | 47.1 | 73.5 | 53.8 | 49.4 |
| | | EWC | 74.5 | 59.7 | 65.6 | 50.1 | 56.1 | 77.3 | 67.6 | 66.5 |
| | | LwF | 91.9 | 92.8 | 94.3 | 90.9 | 91.2 | 93.7 | 92.1 | 99.9 |
| | | LP-FT | **93.4** | **95.1** | 96.2 | **93.1** | **94.7** | **95.1** | **94.3** | 97.3 |
| | | WiSE-FT | 81.8 | 67.7 | 75.1 | 63.5 | 68.7 | 83.2 | 72.8 | 84.7 |
| | | MS:PRE-FT-EWC-LwF | 82.6 | 73.7 | 78.3 | 67.0 | 70.6 | 84.5 | 76.0 | 88.7 |
| | IN-21K-P | FT | 95.4 | 98.7 | 99.3 | 96.7 | 99.2 | 97.3 | 97.6 | **100.0** |
| | | Linear Probing | 78.0 | 57.0 | 70.5 | 67.3 | 68.5 | 81.0 | 64.5 | 88.8 |
| | | Visual Prompt | 70.2 | 43.1 | 63.3 | 49.9 | 56.1 | 74.6 | 63.6 | 63.2 |
| | | LoRA | 74.2 | 37.5 | 53.1 | 47.4 | 48.9 | 75.6 | 67.4 | 63.1 |
| | | EWC | 76.8 | 66.7 | 73.0 | 57.8 | 61.0 | 80.7 | 73.9 | 69.7 |
| | | LwF | 94.2 | 97.2 | 98.5 | 95.7 | 97.0 | 96.1 | 96.2 | 100.0 |
| | | LP-FT | **96.2** | **98.8** | **99.4** | **96.9** | **99.3** | **97.7** | **98.1** | 100.0 |
| | | WiSE-FT | 84.2 | 74.3 | 80.1 | 70.2 | 73.4 | 87.0 | 78.0 | 88.7 |
| | | MS:PRE-FT-EWC-LwF | 84.7 | 80.8 | 82.8 | 73.5 | 75.9 | 87.8 | 80.9 | 89.0 |
| | IN-21K + AugReg | FT | 100.0 | 100.0 | 99.8 | 98.0 | 100.0 | 99.9 | 99.9 | **100.0** |
| | | Linear Probing | 98.3 | 97.3 | 91.7 | 93.2 | 92.0 | 96.5 | 91.1 | 98.6 |
| | | Visual Prompt | 74.2 | 52.0 | 71.7 | 56.6 | 61.1 | 78.7 | 73.2 | 70.2 |
| | | LoRA | 75.1 | 53.1 | 66.5 | 56.5 | 56.4 | 78.6 | 74.4 | 19.2 |
| | | EWC | 91.1 | 97.8 | 91.2 | 73.4 | 93.8 | 86.2 | 84.1 | 76.8 |
| | | LwF | 100.0 | **100.0** | 99.8 | 98.0 | **100.0** | 99.9 | 99.9 | 100.0 |
| | | LP-FT | **100.0** | **100.0** | 99.8 | **98.1** | **100.0** | **99.9** | **99.9** | 100.0 |
| | | WiSE-FT | 95.9 | 97.0 | 94.7 | 88.1 | 91.0 | 95.3 | 92.7 | 96.2 |
| | | MS:PRE-FT-EWC-LwF | 96.8 | 98.6 | 96.5 | 89.9 | 95.9 | 95.3 | 93.9 | 96.8 |
| | OpenAI | FT | **100.0** | **100.0** | 99.8 | **98.0** | 100.0 | **99.9** | 99.9 | **100.0** |
| | | Linear Probing | 82.3 | 78.0 | 86.1 | 74.1 | 79.3 | 86.0 | 79.5 | 92.2 |
| | | Visual Prompt | 77.7 | 54.4 | 78.1 | 56.0 | 60.4 | 80.3 | 71.2 | 66.5 |
| | | LoRA | 79.1 | 65.1 | 79.2 | 60.1 | 62.0 | 83.0 | 76.9 | 41.7 |
| | | EWC | 88.7 | 90.0 | 90.9 | 73.8 | 86.2 | 87.0 | 85.4 | 77.8 |
| | | LwF | **100.0** | 100.0 | 99.8 | 98.0 | 99.9 | 99.9 | 99.9 | **100.0** |
| | | LP-FT | 100.0 | **100.0** | 99.8 | 98.0 | **100.0** | 99.9 | **99.9** | 100.0 |
| | | WiSE-FT | 88.0 | 76.8 | 89.9 | 78.6 | 81.5 | 91.5 | 91.0 | 94.7 |
| | | MS:PRE-FT-EWC-LwF | 88.9 | 81.7 | 91.3 | 79.4 | 83.3 | 91.0 | 91.1 | 93.0 |
| | LAION-2B | FT | 100.0 | 100.0 | **99.8** | 98.0 | 100.0 | 99.9 | **99.9** | **100.0** |
| | | Linear Probing | 82.8 | 77.2 | 88.4 | 79.3 | 80.9 | 87.6 | 80.0 | 93.3 |
| | | Visual Prompt | 77.2 | 49.9 | 79.6 | 62.1 | 63.6 | 81.3 | 72.4 | 68.1 |
| | | LoRA | 78.1 | 58.6 | 79.8 | 62.3 | 61.5 | 83.9 | 76.4 | 39.8 |
| | | EWC | 83.8 | 68.7 | 89.3 | 71.8 | 79.9 | 86.3 | 83.5 | 74.2 |
| | | LwF | 100.0 | 99.9 | 99.8 | 98.0 | 99.9 | 99.9 | 99.9 | 100.0 |
| | | LP-FT | **100.0** | **100.0** | 99.8 | **98.0** | **100.0** | **99.9** | **99.9** | 100.0 |
| | | WiSE-FT | 85.8 | 46.5 | 87.6 | 77.9 | 77.6 | 91.0 | 89.9 | 93.3 |
| | | MS:PRE-FT-EWC-LwF | 87.3 | 64.6 | 89.5 | 79.2 | 80.3 | 90.6 | 90.1 | 94.3 |

- Tables 41, 42, 43, and 44 report the corresponding results for ViT-B/32 pre-trained on ImageNet-1K with AugReg, ImageNet-1K with SAM, ImageNet-21K with AugReg, and LAION-2B.

  Table 45 uses OpenAI CLIP ViT-B/32 as a pre-trained model.

- Tables 46, 47, and 48 show the results for ViT-S/16 pre-trained on ImageNet-1K and ImageNet-21K, and ViT-S/32 pre-trained on ImageNet-21, respectively. All models are pre-trained with AugReg.

- Table 49 provides the results for ViT-L/16 pre-trained on ImageNet-21K with AugReg.

- Finally, Tables 50 and 51 present the accuracy for ResNet-18 and ResNet-50 models, both pre-trained on ImageNet-1K.

Table 9: Accuracy on downstream datasets after fine-tuning with each method using various ViTs. FT and LP-FT generally achieve the highest performance, while Visual Prompt and LoRA show the lowest.

| Arch | $D_{pre}$ | Method | Realistic Downstream Dataset | | | | | Synthetic Downstream Dataset | | |
|---|---|---|---|---|---|---|---|---|---|---|
| | | | IN-V2 | IN-A | IN-R | IN-Sketch | ObjNet | IN-Cartoon | IN-Drawing | IN-C |
| ViT-B/32 | IN-1K + AugReg | FT | 94.6 | 94.0 | 97.3 | 95.3 | 95.9 | 96.3 | 96.3 | 100.0 |
| | | Linear Probing | 68.7 | 34.9 | 63.4 | 62.1 | 54.7 | 75.1 | 62.6 | 90.9 |
| | | Visual Prompt | 59.6 | 15.1 | 54.0 | 41.4 | 38.3 | 68.3 | 60.1 | 59.3 |
| | | LoRA | 61.1 | 10.1 | 42.1 | 31.9 | 31.0 | 67.3 | 53.1 | 66.3 |
| | | EWC | 65.9 | 33.4 | 59.3 | 45.6 | 42.1 | 71.3 | 64.3 | 62.7 |
| | | LwF | 94.0 | 91.2 | 95.7 | 94.2 | 92.6 | 95.5 | 95.0 | 99.9 |
| | | LP-FT | **96.1** | **94.5** | **97.7** | **95.7** | **96.7** | **96.9** | **96.9** | **100.0** |
| | | WiSE-FT | 77.7 | 39.3 | 67.4 | 58.7 | 56.4 | 81.6 | 71.7 | 87.7 |
| | | MS:PRE-FT-EWC-LwF | 79.0 | 49.1 | 71.0 | 63.0 | 60.6 | 82.7 | 75.4 | 86.8 |
| | IN-1K + SAM | FT | 73.5 | **46.8** | **82.5** | **83.8** | 66.4 | 84.3 | 82.5 | **100.0** |
| | | Linear Probing | 60.7 | 8.9 | 48.3 | 35.5 | 33.4 | 67.2 | 51.2 | 83.4 |
| | | Visual Prompt | 57.2 | 8.5 | 44.6 | 31.8 | 29.9 | 63.9 | 50.1 | 52.8 |
| | | LoRA | 59.9 | 5.2 | 41.6 | 28.1 | 28.6 | 65.3 | 46.9 | 62.2 |
| | | EWC | 60.9 | 10.2 | 45.1 | 37.7 | 31.1 | 67.0 | 52.3 | 53.7 |
| | | LwF | 73.2 | 43.1 | 79.2 | 81.3 | 61.2 | 83.0 | 80.4 | 99.9 |
| | | LP-FT | **74.2** | 44.8 | 81.8 | 82.4 | 66.1 | **84.8** | **82.7** | 100.0 |
| | | WiSE-FT | 66.0 | 17.6 | 59.3 | 46.9 | 42.6 | 74.5 | 64.0 | 76.0 |
| | | MS:PRE-FT-EWC-LwF | 66.1 | 20.1 | 60.8 | 51.1 | 43.3 | 74.8 | 65.5 | 76.5 |
| | IN-21K + AugReg | FT | 99.5 | 100.0 | 99.8 | 97.7 | 99.9 | 99.5 | 99.6 | 100.0 |
| | | Linear Probing | 83.5 | 65.6 | 77.5 | 78.2 | 73.5 | 86.0 | 72.9 | 94.4 |
| | | Visual Prompt | 68.2 | 32.0 | 66.3 | 51.4 | 52.4 | 74.0 | 67.3 | 65.2 |
| | | LoRA | 69.1 | 25.0 | 51.5 | 43.0 | 43.4 | 71.9 | 63.2 | 66.1 |
| | | EWC | 76.3 | 69.5 | 72.1 | 56.3 | 70.0 | 78.0 | 72.5 | 68.0 |
| | | LwF | 99.2 | 99.8 | 99.5 | 97.3 | 99.7 | 99.2 | 99.1 | 100.0 |
| | | LP-FT | 99.8 | 100.0 | 99.8 | 97.9 | 100.0 | 99.8 | 99.8 | 100.0 |
| | | WiSE-FT | 87.9 | 72.0 | 80.4 | 72.3 | 73.8 | 88.9 | 79.1 | 92.5 |
| | | MS:PRE-FT-EWC-LwF | 89.0 | 82.8 | 85.0 | 76.8 | 82.1 | 89.6 | 83.2 | 90.6 |
| | OpenAI | FT | 100.0 | 100.0 | 99.8 | 98.0 | 100.0 | 99.9 | **99.9** | **100.0** |
| | | Linear Probing | 74.8 | 47.1 | 75.9 | 64.3 | 63.6 | 80.1 | 71.2 | 89.0 |
| | | Visual Prompt | 71.6 | 29.3 | 65.6 | 50.7 | 47.5 | 75.5 | 64.9 | 62.3 |
| | | LoRA | 72.5 | 34.7 | 67.7 | 53.3 | 48.4 | 77.9 | 69.6 | 71.5 |
| | | EWC | 88.4 | 86.9 | 88.4 | 70.8 | 79.8 | 85.1 | 83.4 | 72.6 |
| | | LwF | 99.9 | 99.8 | 99.8 | 97.9 | 99.8 | 99.8 | 99.9 | 100.0 |
| | | LP-FT | 100.0 | 100.0 | 99.8 | 98.0 | 100.0 | **99.9** | 99.9 | 100.0 |
| | | WiSE-FT | 85.9 | 71.4 | 88.6 | 78.1 | 75.5 | 89.3 | 89.9 | 93.4 |
| | | MS:PRE-FT-EWC-LwF | 87.0 | 76.0 | 89.1 | 77.7 | 76.6 | 88.1 | 89.5 | 91.1 |
| | LAION-2B | FT | **100.0** | **100.0** | 99.8 | **98.0** | **100.0** | 99.9 | **99.9** | **100.0** |
| | | Linear Probing | 75.5 | 47.8 | 78.7 | 67.3 | 66.7 | 81.4 | 71.5 | 88.7 |
| | | Visual Prompt | 72.2 | 30.0 | 69.2 | 54.6 | 51.2 | 76.1 | 65.1 | 62.1 |
| | | LoRA | 72.9 | 35.7 | 70.4 | 56.4 | 51.4 | 79.3 | 69.5 | 73.4 |
| | | EWC | 85.6 | 80.7 | 87.3 | 71.6 | 77.2 | 84.8 | 83.4 | 69.7 |
| | | LwF | 99.9 | 99.8 | 99.8 | 97.9 | 99.8 | 99.8 | 99.8 | 100.0 |
| | | LP-FT | 100.0 | 100.0 | 99.8 | 98.0 | 100.0 | **99.9** | 99.9 | 100.0 |
| | | WiSE-FT | 85.3 | 68.0 | 87.9 | 77.8 | 76.1 | 87.5 | 88.6 | 92.9 |
| | | MS:PRE-FT-EWC-LwF | 85.9 | 73.9 | 88.8 | 77.8 | 77.1 | 86.4 | 88.5 | 91.9 |
| ViT-S/16 | IN-21K + AugReg | FT | 99.9 | **100.0** | 99.8 | 97.9 | 100.0 | 99.7 | 99.7 | 100.0 |
| | | Linear Probing | 84.0 | 67.6 | 73.9 | 75.0 | 73.2 | 84.0 | 69.5 | 88.8 |
| | | Visual Prompt | 69.3 | 40.6 | 63.8 | 49.4 | 56.5 | 74.5 | 65.3 | 62.8 |
| | | LoRA | 70.7 | 29.4 | 49.8 | 45.9 | 45.1 | 71.5 | 65.6 | 16.2 |
| | | EWC | 79.5 | 84.9 | 75.2 | 57.1 | 68.8 | 79.4 | 73.6 | 68.9 |
| | | LwF | 99.7 | 99.9 | 99.7 | 97.6 | 99.9 | 99.4 | 99.3 | 100.0 |
| | | LP-FT | 99.9 | 100.0 | 99.8 | 98.0 | 100.0 | **99.9** | **99.9** | 100.0 |
| | | WiSE-FT | 90.2 | 82.7 | 82.9 | 74.9 | 78.3 | 89.6 | 81.2 | 90.4 |
| | | MS:PRE-FT-EWC-LwF | 91.0 | 91.8 | 87.4 | 79.1 | 85.3 | 90.7 | 85.6 | 92.7 |
| ViT-S/32 | IN-21K + AugReg | FT | 99.9 | **100.0** | 99.8 | 97.8 | 100.0 | 99.6 | 99.7 | 100.0 |
| | | Linear Probing | 78.3 | 50.0 | 68.0 | 68.0 | 63.9 | 79.7 | 64.1 | 83.4 |
| | | Visual Prompt | 60.7 | 21.4 | 54.1 | 40.4 | 43.8 | 66.6 | 57.0 | 54.7 |
| | | LoRA | 64.0 | 12.5 | 42.4 | 39.9 | 35.6 | 65.5 | 57.3 | 36.2 |
| | | EWC | 73.7 | 67.9 | 67.1 | 50.4 | 57.3 | 73.5 | 66.7 | 61.5 |
| | | LwF | 99.6 | 99.9 | 99.6 | 97.5 | 99.9 | 99.3 | 99.4 | 100.0 |
| | | LP-FT | 100.0 | 100.0 | 99.8 | 97.9 | 100.0 | **99.8** | **99.9** | 100.0 |
| | | WiSE-FT | 87.4 | 67.1 | 77.9 | 69.7 | 71.2 | 87.8 | 77.0 | 90.9 |
| | | MS:PRE-FT-EWC-LwF | 88.7 | 81.7 | 83.1 | 75.8 | 78.6 | 88.7 | 82.4 | 88.6 |
| ViT-L/16 | IN-21K + AugReg | FT | 99.9 | **100.0** | **99.8** | 98.0 | **100.0** | 99.9 | 99.9 | **100.0** |
| | | Linear Probing | 98.3 | 98.1 | 94.0 | 93.1 | 92.9 | 96.9 | 91.7 | 99.1 |
| | | Visual Prompt | 71.3 | 48.0 | 69.4 | 50.8 | 59.6 | 76.0 | 66.7 | 67.1 |
| | | LoRA | 76.5 | 59.9 | 66.4 | 54.7 | 56.8 | 80.0 | 68.6 | 72.7 |
| | | EWC | 82.9 | 91.2 | 87.9 | 70.9 | 87.0 | 85.1 | 82.8 | 80.5 |
| | | LwF | 99.9 | 100.0 | 99.8 | 97.8 | 100.0 | 99.8 | 99.8 | 100.0 |
| | | LP-FT | 100.0 | 100.0 | 99.8 | **98.1** | 100.0 | **99.9** | **99.9** | 99.8 |
| | | WiSE-FT | 93.1 | 77.7 | 93.5 | 85.7 | 88.3 | 93.8 | 90.3 | 96.1 |
| | | MS:PRE-FT-EWC-LwF | 93.3 | 90.5 | 92.9 | 87.8 | 93.0 | 93.9 | 90.9 | 97.1 |

Table 10: Accuracy on downstream datasets after fine-tuning with each method. FT and LP-FT generally achieve the highest performance, while EWC shows the lowest.

| Arch | $D_{pre}$ | Method | Realistic Downstream Dataset | | | | | Synthetic Downstream Dataset | | |
|---|---|---|---|---|---|---|---|---|---|---|
| | | | IN-V2 | IN-A | IN-R | IN-Sketch | ObjNet | IN-Cartoon | IN-Drawing | IN-C |
| ResNet-18 | IN-1K | FT | 98.3 | **98.7** | **99.1** | 95.7 | **97.6** | 97.4 | 97.1 | 100.0 |
| | | Linear Probing | 59.9 | 6.5 | 47.0 | 33.3 | 34.2 | 65.1 | 48.3 | 50.8 |
| | | Visual Prompt | 52.4 | 5.8 | 35.9 | 22.9 | 29.1 | 46.8 | 28.2 | 28.1 |
| | | EWC | 63.0 | 17.5 | 50.8 | 37.9 | 38.0 | 66.0 | 54.0 | 42.6 |
| | | LwF | 97.0 | 97.6 | 98.3 | 94.7 | 96.7 | 96.2 | 95.8 | 99.9 |
| | | LP-FT | **98.5** | 98.5 | 98.9 | **95.8** | 97.4 | **97.7** | **97.3** | **100.0** |
| | | WiSE-FT | 80.2 | 30.7 | 69.4 | 53.5 | 58.7 | 75.9 | 58.6 | 75.6 |
| | | MS:PRE-FT-EWC-LwF | 80.8 | 41.8 | 72.9 | 59.9 | 62.4 | 80.3 | 70.3 | 74.9 |
| ResNet-50 | IN-1K | FT | 95.3 | 94.5 | **98.6** | 96.2 | 96.9 | 97.4 | 97.8 | 100.0 |
| | | Linear Probing | 69.8 | 19.8 | 52.3 | 32.9 | 46.2 | 75.0 | 57.0 | 55.5 |
| | | Visual Prompt | 66.0 | 22.4 | 47.3 | 33.3 | 45.6 | 59.3 | 39.3 | 42.1 |
| | | EWC | 72.0 | 43.1 | 58.4 | 44.5 | 49.5 | 76.3 | 63.6 | 52.4 |
| | | LwF | 94.6 | 94.0 | 97.9 | 95.4 | 95.6 | 96.4 | 96.8 | 100.0 |
| | | LP-FT | **95.7** | **94.8** | 98.6 | **96.2** | **97.0** | **97.5** | **97.9** | **100.0** |
| | | WiSE-FT | 82.7 | 56.6 | 73.7 | 56.7 | 68.6 | 82.5 | 65.9 | 83.8 |
| | | MS:PRE-FT-EWC-LwF | 84.1 | 66.7 | 78.9 | 62.4 | 71.6 | 86.4 | 76.0 | 84.8 |

Table 11: $RI$ and $mRI$ of ImageNet-1K with AugReg pre-trained models with different fine-tuning methods and downstream datasets on each dataset in ImageNet-RIB.

| Architecture | Method | $mRI$ | Realistic OOD | | | | | Synthetic OOD | | |
| --- | --- | --- | --- | --- | --- | --- | --- | --- | --- | --- |
| | | | IN-V2 | IN-A | IN-R | IN-Sketch | ObjNet | IN-Cartoon | IN-Drawing | IN-C |
| ViT-B/16 | FT | 1.3 | 2.9 | -4.0 | 2.8 | 4.4 | -2.7 | 0.6 | 0.4 | 5.9 |
| | Linear Probing | 0.7 | 0.1 | -0.1 | 0.8 | 1.2 | 0.3 | 0.2 | 0.1 | 3.2 |
| | Visual Prompt | -4.5 | -2.3 | -9.1 | -4.9 | -1.6 | -11.2 | -3.9 | -4.3 | 1.7 |
| | LoRA | 0.9 | 0.2 | 0.4 | 1.1 | 2.6 | 0.3 | -0.1 | 1.3 | 1.1 |
| | EWC | 2.8 | 2.9 | -0.2 | 5.2 | 4.4 | 1.4 | 1.6 | 2.8 | 4.3 |
| | LwF | 3.1 | 2.8 | -0.0 | 6.2 | 4.6 | 0.7 | 1.9 | 2.1 | 6.5 |
| | LP-FT | 2.3 | **3.0** | -0.9 | 5.2 | 4.5 | -0.1 | 1.2 | 0.6 | 4.7 |
| | WiSE-FT | 3.6 | 2.5 | **0.7** | 7.5 | 4.5 | 2.1 | 2.3 | 3.0 | 6.5 |
| | MS:PRE-FT-EWC-LwF | **3.9** | 2.7 | 0.7 | **7.8** | **5.0** | **2.2** | **2.4** | **3.3** | **6.7** |
| ViT-B/32 | FT | -0.0 | 1.6 | -5.5 | 0.2 | 2.6 | -5.4 | 0.3 | -0.3 | 6.4 |
| | Linear Probing | 1.1 | 0.1 | -0.1 | 0.9 | 1.3 | 0.4 | 1.0 | 1.1 | 3.8 |
| | Visual Prompt | -5.4 | -2.7 | -13.3 | -4.7 | -2.0 | -12.7 | -2.4 | -5.0 | -0.1 |
| | LoRA | 0.9 | 0.3 | **0.3** | 0.5 | 1.0 | **0.7** | 0.7 | 0.5 | 3.1 |
| | EWC | 1.3 | **1.9** | -2.9 | 3.2 | 2.6 | 0.1 | 1.2 | 2.0 | 2.6 |
| | LwF | 1.8 | 1.5 | -2.0 | 3.9 | 3.2 | -1.9 | 1.4 | 1.2 | **6.9** |
| | LP-FT | 1.5 | 1.5 | -1.7 | 3.4 | 2.9 | -1.9 | 1.0 | 0.3 | 6.4 |
| | WiSE-FT | 2.5 | 1.5 | 0.2 | 5.0 | 3.3 | 0.3 | 1.6 | 2.2 | 6.1 |
| | MS:PRE-FT-EWC-LwF | **2.5** | 1.7 | -0.5 | **5.1** | **3.5** | 0.2 | **1.8** | **2.4** | 6.0 |
| ViT-S/16 | FT | -3.2 | -0.0 | -8.2 | -2.9 | 0.3 | -9.7 | -2.4 | -5.3 | 2.9 |
| | Linear Probing | 0.3 | 0.1 | -0.5 | 0.9 | 1.4 | -0.2 | -0.1 | 0.6 | -0.1 |
| | Visual Prompt | -7.4 | -4.6 | -13.3 | -6.1 | -3.5 | -18.1 | -6.3 | -6.0 | -1.4 |
| | LoRA | 0.9 | 0.2 | 0.1 | 1.6 | 3.6 | -0.1 | -0.3 | **1.5** | 0.8 |
| | EWC | 1.6 | **2.6** | -2.2 | 4.2 | **5.5** | -1.9 | 0.6 | 0.9 | 2.7 |
| | LwF | 0.6 | 0.9 | -1.5 | 3.5 | 1.5 | -2.7 | 0.3 | -2.4 | 5.4 |
| | LP-FT | -1.2 | 0.9 | -4.0 | 0.1 | 1.8 | -5.8 | -1.2 | -4.2 | 2.8 |
| | WiSE-FT | 2.9 | 2.2 | **0.7** | 6.5 | 4.7 | 0.1 | 1.9 | 1.4 | 5.8 |
| | MS:PRE-FT-EWC-LwF | **3.0** | 2.2 | 0.3 | **6.7** | 5.3 | **0.1** | 1.9 | 1.3 | **6.0** |
| ResNet-18 | FT | -5.2 | -2.1 | -11.7 | -0.6 | -5.0 | -8.8 | -5.7 | -13.6 | 5.7 |
| | Linear Probing | -7.3 | -1.4 | -2.5 | -1.2 | -26.9 | -3.9 | -4.7 | -15.5 | -2.1 |
| | Visual Prompt | -8.3 | -4.3 | -18.3 | -7.5 | -6.9 | -12.9 | -6.1 | -7.8 | -2.8 |
| | EWC | -5.7 | -0.6 | -9.6 | 2.0 | -11.7 | -4.3 | -4.6 | -15.1 | -1.5 |
| | LwF | -1.9 | -0.9 | -5.5 | 2.6 | -2.7 | -4.7 | -1.4 | -9.0 | **6.7** |
| | LP-FT | -4.8 | -2.2 | -10.0 | 1.0 | -6.2 | -7.1 | -5.7 | -13.9 | 6.1 |
| | WiSE-FT | **0.7** | **-0.1** | **-1.5** | 4.3 | **2.4** | **-1.5** | **-0.7** | **-2.8** | 5.3 |
| | MS:PRE-FT-EWC-LwF | -0.1 | -0.2 | -2.7 | 4.2 | 1.9 | -1.9 | -1.2 | -5.7 | 4.9 |
| ResNet-50 | FT | -5.2 | -0.1 | -2.9 | 2.8 | -10.7 | -4.3 | -6.5 | -22.4 | 2.4 |
| | Linear Probing | -11.2 | -1.5 | -1.2 | -1.2 | -37.0 | -4.2 | -5.6 | -35.2 | -3.9 |
| | Visual Prompt | -6.5 | -5.9 | -7.8 | -6.0 | -5.9 | -9.1 | -6.2 | -6.0 | -5.1 |
| | EWC | -8.9 | -1.1 | -0.5 | 2.2 | -21.7 | -3.2 | -7.2 | -36.2 | -3.3 |
| | LwF | -5.8 | 0.5 | -2.2 | 3.6 | -12.3 | -3.0 | -4.9 | -31.5 | 3.2 |
| | LP-FT | -5.1 | -0.2 | -2.6 | 3.2 | -10.3 | -4.1 | -6.4 | -22.1 | 1.9 |
| | WiSE-FT | **1.2** | **0.7** | **0.9** | 6.1 | **1.2** | **-0.0** | **-0.6** | **-3.0** | 4.3 |
| | MS:PRE-FT-EWC-LwF | -0.5 | 0.5 | 0.6 | **6.1** | 0.2 | -0.6 | -2.1 | -13.1 | **4.6** |

Table 12: $RI$ and $mRI$ of ImageNet-1K with SAM pre-trained models with different fine-tuning methods and downstream datasets on each dataset in ImageNet-RIB.

| Architecture | Method | $mRI$ | Realistic OOD | | | | | Synthetic OOD | | |
| --- | --- | --- | --- | --- | --- | --- | --- | --- | --- | --- |
| | | | IN-V2 | IN-A | IN-R | IN-Sketch | ObjNet | IN-Cartoon | IN-Drawing | IN-C |
| ViT-B/16 | FT | 2.5 | **3.4** | -3.2 | 5.8 | 5.7 | -2.3 | 1.7 | 1.8 | 7.3 |
| | Linear Probing | 0.8 | 0.1 | 0.3 | 0.5 | 1.1 | 0.0 | 0.0 | 0.4 | 4.0 |
| | Visual Prompt | -6.1 | -4.1 | -12.4 | -6.4 | -6.3 | -10.6 | -3.4 | -4.9 | -0.2 |
| | LoRA | 0.9 | 0.1 | 0.4 | 0.8 | 1.1 | 0.3 | 0.1 | 0.5 | 3.6 |
| | EWC | 1.6 | 0.7 | 0.3 | 3.7 | 2.4 | 1.2 | 1.0 | 2.0 | 1.3 |
| | LwF | 3.5 | 3.2 | -1.3 | 6.9 | **5.7** | 0.2 | 2.3 | 2.2 | **8.7** |
| | LP-FT | 2.4 | 3.3 | -2.3 | 6.4 | 5.6 | -1.5 | 1.7 | 1.5 | 4.8 |
| | WiSE-FT | 3.6 | 2.0 | **1.9** | 7.1 | 4.3 | 1.5 | **2.4** | 2.8 | 6.5 |
| | MS:PRE-FT-EWC-LwF | **3.7** | 2.0 | 1.7 | **7.3** | 4.6 | **1.6** | 2.4 | **3.0** | 6.7 |
| ViT-B/32 | FT | 1.4 | **2.4** | -4.6 | 4.2 | 3.8 | -4.0 | 0.9 | 0.8 | 7.6 |
| | Linear Probing | 0.9 | 0.1 | 0.4 | 0.8 | 1.1 | 0.2 | 0.3 | 0.2 | 4.2 |
| | Visual Prompt | -5.9 | -2.5 | -17.1 | -5.3 | -5.0 | -12.6 | -1.9 | -2.6 | -0.1 |
| | LoRA | 0.8 | 0.1 | 0.6 | 1.0 | 1.0 | 0.5 | 0.3 | 0.3 | 2.9 |
| | EWC | 1.0 | 0.6 | -0.3 | 2.5 | 2.1 | 0.6 | 0.6 | 1.1 | 1.0 |
| | LwF | 2.4 | 2.3 | -2.3 | **5.5** | **4.0** | -1.3 | 1.5 | 1.4 | 8.3 |
| | LP-FT | 1.9 | 2.3 | -3.0 | 5.1 | 3.7 | -2.8 | 1.0 | 0.4 | **8.3** |
| | WiSE-FT | 2.6 | 1.5 | **1.1** | 5.2 | 3.1 | **0.7** | **1.7** | 2.0 | 5.6 |
| | MS:PRE-FT-EWC-LwF | **2.6** | 1.5 | 0.8 | 5.4 | 3.4 | 0.7 | 1.7 | **2.1** | 5.5 |

Table 13: $RI$ and $mRI$ of ImageNet-21K pre-trained models with different fine-tuning methods and downstream datasets on each dataset in ImageNet-RIB.

| Architecture | Method | $mRI$ | Realistic OOD | | | | | Synthetic OOD | | |
| | | | IN-V2 | IN-A | IN-R | IN-Sketch | ObjNet | IN-Cartoon | IN-Drawing | IN-C |
| --- | --- | --- | --- | --- | --- | --- | --- | --- | --- | --- |
| | FT | -0.1 | 1.5 | -2.8 | 0.3 | 2.7 | -4.0 | -1.2 | -1.8 | 4.2 |
| | Linear Probing | 0.4 | 0.3 | 0.4 | 0.2 | 0.0 | 0.8 | -0.0 | 1.0 | 0.5 |
| | Visual Prompt | -9.4 | -7.7 | -12.7 | -11.1 | -7.7 | -14.8 | -8.4 | -10.0 | -3.3 |
| | LoRA | -0.3 | 0.2 | 0.5 | -1.6 | -0.5 | **0.9** | -0.4 | 0.6 | -1.9 |
| ViT-B/16 | EWC | 1.4 | 1.5 | 0.2 | 2.4 | 2.9 | 0.1 | 0.5 | 1.2 | 2.6 |
| | LwF | 1.6 | 1.5 | -0.5 | 2.9 | 3.5 | -0.8 | 0.9 | 0.0 | 5.3 |
| | LP-FT | 0.5 | 1.6 | -1.2 | 2.2 | 2.7 | -1.8 | -0.5 | -1.3 | 2.1 |
| | WiSE-FT | 2.5 | 1.7 | **0.8** | **4.9** | 3.8 | 0.6 | 1.3 | 1.7 | 5.5 |
| | MS:PRE-FT-EWC-LwF | **2.7** | **1.7** | 0.7 | 4.7 | **4.2** | 0.6 | **1.4** | **1.8** | **6.0** |

Table 14: $RI$ and $mRI$ of ImageNet-21K with AugReg pre-trained models with different fine-tuning methods and downstream datasets on each dataset in ImageNet-RIB.

| Architecture | Method | $mRI$ | Realistic OOD | | | | | Synthetic OOD | | |
| | | | IN-V2 | IN-A | IN-R | IN-Sketch | ObjNet | IN-Cartoon | IN-Drawing | IN-C |
| --- | --- | --- | --- | --- | --- | --- | --- | --- | --- | --- |
| | FT | -5.5 | -1.4 | -9.1 | -5.4 | -5.3 | -11.1 | -3.6 | -5.7 | -2.5 |
| | Linear Probing | -0.3 | -0.4 | -0.4 | -0.9 | -1.3 | -0.5 | 0.4 | 0.6 | 0.2 |
| | Visual Prompt | -8.0 | -6.1 | -9.2 | -9.3 | -6.1 | -16.6 | -7.1 | -7.0 | -3.0 |
| | LoRA | -2.1 | 0.7 | **0.8** | 2.6 | 2.9 | **0.8** | 0.8 | 1.6 | -27.4 |
| ViT-B/16 | EWC | 0.6 | **2.0** | -2.0 | 2.3 | **3.5** | -3.5 | 0.4 | 0.5 | 1.7 |
| | LwF | -1.0 | -1.0 | -2.3 | 0.5 | -1.5 | -4.2 | 0.3 | -1.3 | 1.7 |
| | LP-FT | -2.6 | 0.3 | -3.5 | -4.7 | -3.0 | -6.2 | -0.6 | -2.5 | -0.5 |
| | WiSE-FT | 1.7 | 1.8 | -0.2 | 4.0 | 2.4 | -0.9 | 2.0 | 1.5 | 3.2 |
| | MS:PRE-FT-EWC-LwF | **2.2** | 1.9 | 0.3 | **4.6** | 2.8 | -0.7 | **2.1** | **1.7** | **5.0** |
| | FT | -0.1 | 0.7 | -3.9 | 0.8 | 2.7 | -4.9 | -0.1 | -0.6 | 4.4 |
| | Linear Probing | 0.3 | -0.1 | -0.8 | 0.1 | 0.7 | -0.0 | 0.7 | 1.2 | 0.7 |
| | Visual Prompt | -8.4 | -4.9 | -13.1 | -7.8 | -5.0 | -20.7 | -6.2 | -7.3 | -2.4 |
| | LoRA | 0.9 | 0.0 | 0.5 | 1.0 | 1.2 | 0.7 | 0.1 | 1.5 | 2.0 |
| ViT-B/32 | EWC | 1.6 | **1.9** | -0.7 | 4.0 | 3.9 | -1.0 | 1.0 | 1.7 | 2.3 |
| | LwF | 1.7 | 1.0 | -0.5 | 3.9 | 2.8 | -1.3 | 1.7 | 1.2 | 5.0 |
| | LP-FT | 1.2 | 1.1 | -1.3 | 3.3 | 2.1 | -0.9 | 1.5 | 0.8 | 3.0 |
| | WiSE-FT | **3.0** | 1.7 | **0.9** | 5.6 | 4.0 | **1.0** | 2.0 | **2.5** | **6.0** |
| | MS:PRE-FT-EWC-LwF | 2.8 | 1.7 | 0.6 | 5.6 | **4.1** | 0.6 | 2.0 | 2.4 | 5.6 |
| | FT | -2.3 | -0.2 | -5.4 | -0.8 | 0.4 | -8.5 | -1.8 | -4.1 | 1.8 |
| | Linear Probing | -0.2 | -0.1 | -0.8 | -0.1 | 0.3 | -0.3 | 0.3 | 0.6 | -1.2 |
| | Visual Prompt | -9.2 | -5.7 | -12.1 | -8.9 | -5.0 | -21.3 | -8.3 | -9.6 | -2.8 |
| | LoRA | -1.5 | 0.1 | 0.4 | 1.5 | 2.8 | **0.5** | 0.3 | 1.6 | -19.5 |
| ViT-S/16 | EWC | 1.6 | **2.0** | -0.8 | 4.2 | **4.8** | -1.7 | 0.8 | 1.0 | 2.7 |
| | LwF | 0.5 | 0.5 | -0.8 | 3.2 | 1.4 | -3.1 | 0.9 | -1.3 | 3.4 |
| | LP-FT | -0.8 | 0.5 | -2.9 | 1.6 | 1.1 | -4.6 | -0.5 | -2.3 | 0.7 |
| | WiSE-FT | 2.8 | 1.8 | **0.8** | 6.1 | 4.4 | -0.1 | 1.9 | **2.0** | 5.1 |
| | MS:PRE-FT-EWC-LwF | **2.8** | 1.7 | 0.6 | **6.3** | 4.6 | -0.2 | **2.0** | 1.8 | **5.9** |
| | FT | -2.9 | -1.2 | -8.1 | -1.3 | 0.1 | -9.3 | -2.5 | -4.9 | 4.2 |
| | Linear Probing | -0.1 | -0.1 | -1.5 | 0.1 | 0.6 | -0.2 | 0.5 | 0.1 | -0.2 |
| | Visual Prompt | -9.6 | -4.7 | -21.6 | -8.5 | -5.6 | -19.1 | -5.7 | -8.6 | -2.7 |
| | LoRA | 0.4 | 0.1 | **0.5** | 1.1 | 2.7 | **0.5** | 0.3 | 1.1 | -3.0 |
| ViT-S/32 | EWC | 1.0 | **1.5** | -3.1 | 3.6 | **4.2** | -1.5 | 0.2 | 0.7 | 2.2 |
| | LwF | 0.3 | -0.1 | -2.1 | 3.2 | 1.6 | -4.0 | 0.5 | -1.5 | 4.8 |
| | LP-FT | -1.1 | -0.5 | -4.5 | 1.5 | 0.8 | -5.1 | -0.9 | -3.0 | 3.0 |
| | WiSE-FT | **2.3** | 1.1 | 0.1 | 5.5 | 3.8 | -0.6 | 1.2 | **1.2** | **6.2** |
| | MS:PRE-FT-EWC-LwF | 2.3 | 1.1 | -0.5 | **5.6** | 4.1 | -0.6 | **1.2** | 1.2 | 6.0 |
| | FT | -2.1 | 0.3 | -8.7 | -3.5 | -0.6 | -3.1 | -0.8 | -0.9 | 0.1 |
| | Linear Probing | -1.3 | -0.5 | -4.1 | -6.1 | -1.2 | -0.5 | 0.7 | 0.7 | 0.7 |
| | Visual Prompt | -12.9 | -10.7 | -13.6 | -13.5 | -15.0 | -17.0 | -10.4 | -14.2 | -9.0 |
| | LoRA | 1.0 | 0.2 | 0.7 | 1.1 | 1.2 | 0.9 | 0.7 | 1.1 | 1.7 |
| ViT-L/16 | EWC | 1.1 | -0.6 | 0.3 | 2.5 | 2.3 | -0.7 | 1.2 | 1.5 | 1.9 |
| | LwF | -0.2 | -0.6 | 0.4 | -1.9 | 0.5 | -0.2 | -0.6 | -1.8 | 2.6 |
| | LP-FT | -3.5 | 0.5 | -14.0 | -16.4 | -0.5 | -0.8 | 1.3 | 0.8 | 0.8 |
| | WiSE-FT | 2.3 | **2.1** | 0.1 | 3.3 | **2.6** | 1.1 | **2.4** | 2.7 | 4.4 |
| | MS:PRE-FT-EWC-LwF | **2.5** | 1.8 | **1.1** | 3.4 | 2.5 | **1.1** | 2.0 | **2.7** | **5.1** |

Table 15: $RI$ and $mRI$ of ImageNet-21K-P pre-trained models with different fine-tuning methods and downstream datasets on each dataset in ImageNet-RIB.

| Architecture | Method | $mRI$ | Realistic OOD | | | | | Synthetic OOD | | |
| | | | IN-V2 | IN-A | IN-R | IN-Sketch | ObjNet | IN-Cartoon | IN-Drawing | IN-C |
|---|---|---|---|---|---|---|---|---|---|---|
| ViT-B/16 | FT | -0.5 | 0.7 | -3.5 | 1.6 | 3.0 | -4.4 | -1.4 | -2.2 | 2.3 |
| | Linear Probing | 0.2 | 0.2 | 0.4 | 0.5 | 1.1 | 0.3 | 0.2 | 0.3 | -1.0 |
| | Visual Prompt | -10.1 | -8.0 | -11.5 | -9.9 | -7.8 | -19.9 | -8.8 | -11.1 | -3.6 |
| | LoRA | 0.4 | 0.1 | 0.3 | 0.5 | 1.2 | 0.5 | -0.2 | 0.9 | -0.1 |
| | EWC | 1.3 | 1.3 | 0.6 | 1.1 | 3.0 | 0.8 | 0.6 | 0.6 | 2.1 |
| | LwF | 1.7 | 1.6 | -0.1 | 4.5 | 3.5 | -0.3 | 1.0 | -0.2 | 3.7 |
| | LP-FT | 0.4 | 0.8 | -1.1 | 3.8 | 3.3 | -1.2 | -0.4 | -1.8 | 0.1 |
| | WiSE-FT | **3.0** | 2.0 | **1.3** | 6.3 | 4.1 | **1.3** | **1.8** | 2.0 | **5.3** |
| | MS:PRE-FT-EWC-LwF | 3.0 | **2.0** | 1.1 | **6.3** | **4.3** | 1.3 | 1.7 | **2.0** | 5.3 |

Table 16: $RI$ and $mRI$ of LAION-2B pre-trained models with different fine-tuning methods and downstream datasets on each dataset in ImageNet-RIB.

| Architecture | Method | $mRI$ | Realistic OOD | | | | | Synthetic OOD | | |
| | | | IN-V2 | IN-A | IN-R | IN-Sketch | ObjNet | IN-Cartoon | IN-Drawing | IN-C |
|---|---|---|---|---|---|---|---|---|---|---|
| ViT-B/16 | FT | -38.1 | -39.4 | -53.7 | -36.9 | -46.9 | -49.2 | -29.0 | -36.6 | -12.9 |
| | Linear Probing | **-2.0** | **-0.6** | **-0.9** | **-1.5** | **-0.8** | **-1.9** | **-2.4** | **-5.5** | -2.0 |
| | Visual Prompt | -8.2 | -6.4 | -8.1 | -8.1 | -6.7 | -16.7 | -8.3 | -8.0 | -2.9 |
| | LoRA | -3.6 | -0.8 | -1.3 | -1.6 | -1.2 | -3.0 | -2.8 | -5.8 | -12.3 |
| | EWC | -12.5 | -16.1 | -27.2 | -2.4 | -17.3 | -19.0 | -6.4 | -9.9 | **-1.7** |
| | LwF | -33.9 | -37.3 | -49.3 | -31.7 | -45.2 | -44.7 | -22.3 | -31.0 | -9.9 |
| | LP-FT | -37.1 | -39.3 | -51.0 | -35.9 | -46.1 | -47.7 | -28.3 | -33.7 | -14.6 |
| | WiSE-FT | -21.6 | -25.3 | -39.1 | -17.6 | -25.4 | -25.4 | -11.3 | -16.3 | -5.5 |
| | MS:PRE-FT-EWC-LwF | -17.9 | -21.1 | -31.3 | -12.9 | -29.7 | -22.1 | -8.6 | -14.6 | -2.7 |
| ViT-B/32 | FT | -31.6 | -31.1 | -47.0 | -28.9 | -37.5 | -41.3 | -24.5 | -32.8 | -9.6 |
| | Linear Probing | **-1.4** | **-0.1** | **-1.5** | 0.2 | 0.5 | **-2.1** | **-2.3** | **-6.0** | 0.5 |
| | Visual Prompt | -8.4 | -6.4 | -12.1 | -6.9 | -6.0 | -21.9 | -6.1 | -6.7 | -1.5 |
| | LoRA | -1.9 | -0.2 | -2.0 | -0.4 | -0.9 | -4.0 | -2.6 | -6.1 | **1.1** |
| | EWC | -10.0 | -10.6 | -25.6 | -1.2 | -11.5 | -15.1 | -3.5 | -11.0 | -1.1 |
| | LwF | -26.7 | -28.5 | -40.5 | -22.8 | -33.7 | -34.4 | -18.5 | -26.8 | -8.6 |
| | LP-FT | -30.8 | -31.3 | -45.7 | -27.9 | -35.9 | -39.7 | -24.4 | -30.8 | -10.3 |
| | WiSE-FT | -13.5 | -15.5 | -22.8 | -8.6 | -17.8 | -17.9 | -8.4 | -14.4 | -2.2 |
| | MS:PRE-FT-EWC-LwF | -10.9 | -12.4 | -19.0 | -5.7 | -16.4 | -14.3 | -5.7 | -12.5 | -1.3 |

Table 17: $RI$ and $mRI$ of OpenAI CLIP models with different fine-tuning methods and downstream datasets on each dataset in ImageNet-RIB.

| Architecture | Method | $mRI$ | Realistic OOD | | | | | Synthetic OOD | | |
| | | | IN-V2 | IN-A | IN-R | IN-Sketch | ObjNet | IN-Cartoon | IN-Drawing | IN-C |
|---|---|---|---|---|---|---|---|---|---|---|
| ViT-B/16 | FT | -38.0 | -38.3 | -51.6 | -35.4 | -48.5 | -50.3 | -28.9 | -35.8 | -15.3 |
| | Linear Probing | **-2.0** | **-0.5** | **-0.8** | **-1.3** | -1.3 | **-1.2** | **-3.4** | **-5.6** | -1.8 |
| | Visual Prompt | -8.4 | -7.4 | -8.1 | -7.6 | -6.3 | -16.3 | -9.4 | -9.9 | -2.7 |
| | LoRA | -3.6 | -0.6 | -1.0 | -1.9 | **-1.0** | -2.8 | -4.0 | -6.4 | -11.3 |
| | EWC | -12.7 | -14.4 | -20.9 | -2.4 | -24.8 | -19.9 | -7.5 | -10.8 | **-0.8** |
| | LwF | -33.1 | -35.5 | -46.4 | -30.6 | -47.1 | -44.3 | -22.7 | -30.2 | -7.9 |
| | LP-FT | -36.9 | -38.3 | -50.0 | -34.4 | -48.5 | -49.0 | -29.8 | -31.7 | -13.3 |
| | WiSE-FT | -18.1 | -19.5 | -26.7 | -11.7 | -31.0 | -23.7 | -11.1 | -15.8 | -5.5 |
| | MS:PRE-FT-EWC-LwF | -16.0 | -17.1 | -24.3 | -9.4 | -30.3 | -20.9 | -9.1 | -14.4 | -2.7 |
| ViT-B/32 | FT | -28.7 | -28.1 | -43.8 | -26.4 | -35.0 | -39.1 | -20.8 | -28.2 | -8.4 |
| | Linear Probing | **-1.3** | 0.2 | **-0.9** | -0.8 | **-0.1** | **-1.8** | **-2.1** | -5.6 | 0.9 |
| | Visual Prompt | -8.0 | -5.4 | -12.5 | -6.2 | -4.6 | -20.8 | -5.9 | -7.0 | -1.4 |
| | LoRA | -1.8 | 0.1 | -1.6 | **-0.8** | -0.6 | -3.7 | -2.3 | **-5.4** | -0.2 |
| | EWC | -7.0 | -5.6 | -17.0 | -1.1 | -11.4 | -13.0 | -3.1 | -6.5 | **1.7** |
| | LwF | -23.9 | -24.8 | -37.0 | -21.1 | -31.3 | -31.7 | -16.5 | -24.3 | -4.4 |
| | LP-FT | -27.7 | -27.4 | -42.2 | -24.3 | -33.7 | -37.7 | -20.2 | -26.9 | -9.0 |
| | WiSE-FT | -9.7 | -10.3 | -16.5 | -5.3 | -14.2 | -12.5 | -5.7 | -11.3 | -1.5 |
| | MS:PRE-FT-EWC-LwF | -8.1 | -8.5 | -14.7 | -3.2 | -13.4 | -10.6 | -4.4 | -9.9 | 0.5 |

Table 18: The accuracy on each OOD dataset after fine-tuning on ImageNet-1K with SAM pre-trained ViT-B/16 on the downstream datasets with various methods. Note that ImageNet-Drawing, ImageNet-Cartoon, and ImageNet-C are generated from the ImageNet validation set. Green and red indicate relative performance increases and decreases, respectively, compared to the pre-trained model. Bold indicates the best performance on each evaluation dataset.

| Method | Downstream Dataset | $D_{pre}$ IN | Realistic OOD | | | | | Synthetic OOD | | |
|---|---|---|---|---|---|---|---|---|---|---|
| | | | IN-V2 | IN-A | IN-R | IN-Sketch | ObjNet | IN-Cartoon | IN-Drawing | IN-C |
| Pre-Trained | | 79.2 | 66.4 | 15.0 | 38.0 | 28.0 | 25.7 | 66.2 | 39.1 | 56.0 |
| Pre-Trained | | 80.2 | 68.2 | 9.0 | 40.1 | 27.7 | 34.2 | 66.9 | 42.3 | 54.6 |
| FT | IN-V2 | 81.1 | - | 17.1 | 42.9 | 29.7 | 38.7 | 69.5 | 44.1 | 56.8 |
| | IN-A | 77.1 | 65.7 | - | 37.9 | 25.7 | 39.8 | 60.4 | 30.9 | 51.3 |
| | IN-R | 75.0 | 64.1 | 19.1 | - | 49.5 | 36.1 | 66.9 | 53.6 | 53.9 |
| | IN-Sketch | 79.2 | 67.3 | 14.2 | 59.5 | - | 35.8 | 71.0 | 52.0 | 55.7 |
| | ObjNet | 77.6 | 66.1 | 21.9 | 37.1 | 25.9 | - | 57.3 | 32.2 | 52.2 |
| | IN-Cartoon | 82.6 | 67.1 | 15.4 | 44.2 | 32.2 | 35.2 | - | 42.8 | 51.3 |
| | IN-Drawing | 79.9 | 65.1 | 12.9 | 45.1 | 34.0 | 33.5 | 67.2 | - | 55.4 |
| | IN-C | 99.7 | 63.0 | 15.5 | 39.3 | 27.1 | 30.4 | 92.5 | 71.4 | - |
| HeadOnly | IN-V2 | 80.3 | - | 9.0 | 40.2 | 27.6 | 34.1 | 67.5 | 42.3 | 54.7 |
| | IN-A | 80.2 | 68.0 | - | 41.0 | 27.8 | 34.9 | 67.4 | 42.4 | 54.5 |
| | IN-R | 80.1 | 68.1 | 9.6 | - | 28.6 | 34.1 | 67.9 | 43.5 | 54.6 |
| | IN-Sketch | 79.5 | 67.3 | 9.3 | 45.2 | - | 34.2 | 67.7 | 44.9 | 54.1 |
| | ObjNet | 80.1 | 67.9 | 9.2 | 40.0 | 27.9 | - | 67.3 | 42.3 | 54.6 |
| | IN-Cartoon | 80.3 | 68.1 | 8.4 | 41.6 | 28.2 | 32.8 | - | 43.5 | 53.8 |
| | IN-Drawing | 79.9 | 67.5 | 8.9 | 42.7 | 29.3 | 32.7 | 67.9 | - | 54.6 |
| | IN-C | 93.4 | 65.6 | 11.4 | 40.6 | 28.1 | 32.1 | 80.6 | 58.1 | - |
| Visual Prompt (Bahng et al., 2022) | IN-V2 | 75.5 | - | 8.1 | 39.9 | 24.1 | 35.5 | 58.5 | 35.0 | 44.7 |
| | IN-A | 67.5 | 55.6 | - | 35.8 | 19.1 | 34.0 | 45.6 | 26.4 | 30.4 |
| | IN-R | 69.9 | 57.4 | 6.3 | - | 32.0 | 29.2 | 56.5 | 39.7 | 36.8 |
| | IN-Sketch | 71.2 | 58.2 | 5.6 | 47.2 | - | 28.3 | 56.4 | 39.5 | 36.3 |
| | ObjNet | 70.7 | 58.7 | 9.0 | 36.2 | 20.1 | - | 48.2 | 27.2 | 35.3 |
| | IN-Cartoon | 75.8 | 62.3 | 6.5 | 42.5 | 27.0 | 31.8 | - | 39.8 | 42.3 |
| | IN-Drawing | 72.8 | 59.8 | 5.4 | 43.0 | 28.1 | 28.6 | 59.0 | - | 42.2 |
| | IN-C | 77.7 | 65.7 | 8.8 | 41.3 | 28.0 | 36.7 | 62.1 | 44.1 | - |
| LoRA (Hu et al., 2021) | IN-V2 | 80.2 | - | 8.9 | 40.1 | 27.7 | 34.2 | 67.4 | 42.3 | 54.7 |
| | IN-A | 80.2 | 68.2 | - | 41.1 | 27.8 | 34.6 | 67.7 | 42.7 | 54.8 |
| | IN-R | 80.2 | 68.2 | 9.7 | - | 29.0 | 34.8 | 68.2 | 43.9 | 54.7 |
| | IN-Sketch | 79.9 | 67.7 | 9.4 | 44.0 | - | 34.7 | 68.1 | 44.8 | 54.6 |
| | ObjNet | 80.1 | 68.0 | 9.2 | 40.9 | 28.2 | - | 67.5 | 42.5 | 54.7 |
| | IN-Cartoon | 80.1 | 68.2 | 8.7 | 41.5 | 28.2 | 33.3 | - | 43.5 | 53.7 |
| | IN-Drawing | 80.0 | 67.8 | 9.1 | 42.6 | 29.0 | 33.1 | 68.1 | - | 54.5 |
| | IN-C | 80.2 | 68.6 | 13.7 | 42.4 | 30.7 | 36.9 | 68.6 | 52.4 | - |
| EWC (Kirkpatrick et al., 2017) | IN-V2 | 80.4 | - | 9.7 | 40.8 | 28.2 | 35.0 | 67.8 | 43.0 | 55.1 |
| | IN-A | 79.5 | 68.2 | - | 41.3 | 27.4 | 39.0 | 64.3 | 41.5 | 54.7 |
| | IN-R | 80.3 | 68.6 | 11.5 | - | 36.2 | 36.5 | 70.2 | 49.6 | 56.3 |
| | IN-Sketch | 80.1 | 68.1 | 9.7 | 47.9 | - | 34.2 | 69.3 | 47.7 | 55.6 |
| | ObjNet | 80.4 | 68.6 | 12.5 | 41.4 | 28.3 | - | 66.7 | 43.0 | 56.5 |
| | IN-Cartoon | 80.4 | 68.2 | 9.3 | 42.8 | 29.3 | 34.4 | - | 44.6 | 54.3 |
| | IN-Drawing | 80.1 | 67.9 | 10.0 | 44.5 | 31.2 | 35.3 | 68.7 | - | 56.9 |
| | IN-C | 80.4 | 68.6 | 11.0 | 41.2 | 28.8 | 36.7 | 66.8 | 44.7 | - |
| LwF (Li & Hoiem, 2017) | IN-V2 | 81.1 | - | 16.1 | 42.8 | 29.7 | 38.3 | 69.5 | 44.3 | 56.8 |
| | IN-A | 78.6 | 67.1 | - | 39.4 | 26.7 | 40.2 | 63.4 | 34.3 | 53.9 |
| | IN-R | 77.2 | 66.1 | 17.9 | - | 49.6 | 37.1 | 69.2 | 55.3 | 56.0 |
| | IN-Sketch | 79.6 | 67.8 | 13.8 | 59.0 | - | 35.7 | 71.3 | 51.6 | 56.3 |
| | ObjNet | 79.4 | 67.7 | 20.2 | 39.7 | 27.5 | - | 62.5 | 37.3 | 55.0 |
| | IN-Cartoon | 82.9 | 68.0 | 14.7 | 44.3 | 32.0 | 35.7 | - | 44.3 | 53.1 |
| | IN-Drawing | 80.6 | 65.9 | 12.3 | 45.2 | 34.4 | 34.2 | 68.1 | - | 56.2 |
| | IN-C | 99.4 | 65.5 | 14.7 | 42.4 | 28.9 | 33.0 | 93.0 | 71.8 | - |
| LP-FT (Kumar et al., 2022) | IN-V2 | 81.1 | - | 16.9 | 42.8 | 29.7 | 38.6 | 69.4 | 44.0 | 56.8 |
| | IN-A | 77.8 | 66.3 | - | 38.5 | 26.1 | 40.2 | 61.7 | 32.7 | 52.3 |
| | IN-R | 76.2 | 65.2 | 18.4 | - | 49.7 | 37.1 | 67.9 | 54.9 | 54.5 |
| | IN-Sketch | 78.9 | 67.1 | 13.9 | 60.0 | - | 35.9 | 70.7 | 51.4 | 55.6 |
| | ObjNet | 78.1 | 66.7 | 21.7 | 37.8 | 26.3 | - | 59.0 | 33.7 | 53.0 |
| | IN-Cartoon | 82.6 | 66.9 | 15.4 | 44.5 | 32.2 | 34.8 | - | 43.3 | 51.1 |
| | IN-Drawing | 79.6 | 64.6 | 12.9 | 46.1 | 34.4 | 32.5 | 66.9 | - | 53.8 |
| | IN-C | 93.9 | 65.1 | 11.6 | 41.2 | 28.9 | 31.4 | 82.5 | 61.4 | - |
| WiSE-FT (Wortsman et al., 2022b) | IN-V2 | 80.9 | - | 12.2 | 41.8 | 29.1 | 36.6 | 68.8 | 43.9 | 56.1 |
| | IN-A | 80.7 | 69.0 | - | 42.0 | 29.0 | 39.7 | 68.2 | 42.6 | 57.0 |
| | IN-R | 80.5 | 69.0 | 15.2 | - | 43.8 | 38.1 | 72.3 | 55.5 | 58.7 |
| | IN-Sketch | 80.5 | 68.5 | 11.9 | 51.4 | - | 36.2 | 70.9 | 49.9 | 56.8 |
| | ObjNet | 80.6 | 69.0 | 15.4 | 41.1 | 28.9 | - | 67.1 | 41.8 | 56.4 |
| | IN-Cartoon | 82.3 | 69.0 | 12.2 | 43.5 | 30.9 | 36.1 | - | 46.0 | 55.2 |
| | IN-Drawing | 81.4 | 68.5 | 11.7 | 44.1 | 32.6 | 35.7 | 69.9 | - | 57.9 |
| | IN-C | 89.1 | 69.2 | 16.6 | 43.5 | 31.5 | 37.1 | 79.5 | 56.3 | - |
| Model Soup PRE-FT-EWC-LwF (Wortsman et al., 2022a) | IN-V2 | 80.8 | - | 12.2 | 41.9 | 29.1 | 36.6 | 68.9 | 44.0 | 56.1 |
| | IN-A | 80.5 | 68.9 | - | 41.9 | 28.6 | 40.4 | 67.5 | 41.6 | 57.0 |
| | IN-R | 80.3 | 69.1 | 15.5 | - | 44.8 | 38.2 | 72.4 | 55.7 | 58.7 |
| | IN-Sketch | 80.5 | 68.5 | 12.0 | 52.6 | - | 36.3 | 71.1 | 50.4 | 57.0 |
| | ObjNet | 80.6 | 69.0 | 15.7 | 41.3 | 28.9 | - | 66.7 | 41.9 | 56.7 |
| | IN-Cartoon | 82.2 | 68.9 | 12.1 | 43.9 | 31.1 | 35.9 | - | 46.1 | 55.0 |
| | IN-Drawing | 81.2 | 68.4 | 11.7 | 44.8 | 33.2 | 35.7 | 70.0 | - | 58.0 |
| | IN-C | 89.3 | 69.3 | 16.2 | 43.9 | 31.7 | 37.3 | 79.5 | 57.2 | - |

Table 19: The accuracy on each OOD dataset after fine-tuning ImageNet-21K pre-trained ViT-B/16 on the downstream datasets with various methods. Note that ImageNet-Drawing, ImageNet-Cartoon, and ImageNet-C are generated from the ImageNet validation set. Green and red indicate relative performance increases and decreases, respectively, compared to the pre-trained model. Bold indicates the best performance on each evaluation dataset.

| Method | Downstream Dataset | $D_{pre}$ IN | Realistic OOD | | | | | Synthetic OOD | | |
| --- | --- | --- | --- | --- | --- | --- | --- | --- | --- | --- |
| | | | IN-V2 | IN-A | IN-R | IN-Sketch | ObjNet | IN-Cartoon | IN-Drawing | IN-C |
| Pre-Trained | | 81.8 | 71.4 | 32.0 | 47.3 | 35.8 | 42.5 | 69.4 | 44.1 | 58.3 |
| FT | IN-V2 | 81.7 | - | **40.4** | 49.6 | 36.2 | 45.4 | 68.5 | 41.6 | 58.5 |
| | IN-A | 78.2 | 68.1 | - | 47.4 | 33.4 | 45.7 | 62.7 | 37.2 | 55.0 |
| | IN-R | 75.5 | 65.2 | 34.2 | - | 50.2 | 39.8 | 64.6 | 48.9 | 52.9 |
| | IN-Sketch | 78.8 | 68.3 | 32.6 | **64.7** | - | 41.7 | 69.7 | 51.3 | 55.5 |
| | ObjNet | 78.5 | 67.8 | 39.3 | 44.3 | 32.5 | - | 60.6 | 33.3 | 52.8 |
| | IN-Cartoon | 85.2 | 68.0 | 33.0 | 49.5 | 36.4 | 41.9 | - | 41.4 | 52.9 |
| | IN-Drawing | 81.5 | 66.6 | 27.0 | 50.3 | 38.4 | 39.4 | 67.0 | - | 55.3 |
| | IN-C | **99.7** | 65.6 | 25.4 | 45.4 | 32.9 | 36.7 | 93.2 | **72.9** | - |
| Linear Probing | IN-V2 | 81.8 | - | 32.8 | 47.6 | 35.9 | 42.9 | 69.7 | 44.2 | 58.5 |
| | IN-A | 81.4 | 71.0 | - | 48.6 | 35.9 | 45.9 | 67.8 | 43.9 | 58.7 |
| | IN-R | 80.5 | 70.1 | 33.5 | - | 38.3 | 42.0 | 69.8 | 44.3 | 56.7 |
| | IN-Sketch | 79.9 | 69.2 | 30.1 | 53.9 | - | 41.1 | 69.9 | 44.5 | 56.4 |
| | ObjNet | 81.3 | 71.1 | 37.6 | 48.3 | 36.1 | - | 67.9 | 44.2 | 58.9 |
| | IN-Cartoon | 82.5 | 70.3 | 31.7 | 49.3 | 36.4 | 41.5 | - | 45.2 | 56.9 |
| | IN-Drawing | 82.0 | 70.2 | 32.8 | 50.0 | 38.1 | 42.5 | 71.0 | - | 59.0 |
| | IN-C | 96.7 | 65.3 | 27.6 | 42.8 | 31.7 | 36.2 | 85.5 | 57.0 | - |
| Visual Prompt (Bahng et al., 2022) | IN-V2 | 76.0 | - | 25.2 | 43.8 | 29.4 | 41.7 | 58.2 | 32.1 | 45.5 |
| | IN-A | 71.7 | 60.6 | - | 41.0 | 24.1 | 39.2 | 50.9 | 25.8 | 38.3 |
| | IN-R | 69.9 | 59.1 | 18.1 | - | 35.0 | 35.0 | 54.9 | 35.8 | 38.3 |
| | IN-Sketch | 72.9 | 61.7 | 18.1 | 53.3 | - | 38.7 | 58.3 | 40.3 | 41.1 |
| | ObjNet | 71.1 | 59.6 | 25.1 | 38.0 | 23.8 | - | 49.0 | 22.6 | 36.9 |
| | IN-Cartoon | 75.6 | 63.4 | 20.9 | 44.9 | 29.9 | 37.9 | - | 33.9 | 41.9 |
| | IN-Drawing | 73.3 | 61.8 | 17.1 | 45.4 | 30.0 | 35.1 | 55.5 | - | 41.8 |
| | IN-C | 78.9 | 67.2 | 25.3 | 45.5 | 31.5 | 42.7 | 63.6 | 43.5 | - |
| LoRA (Hu et al., 2021) | IN-V2 | 81.8 | - | 32.4 | 47.4 | 35.8 | 42.8 | 69.6 | 44.1 | 58.4 |
| | IN-A | 81.7 | 71.2 | - | 48.5 | 35.9 | 45.9 | 67.9 | 43.8 | 58.9 |
| | IN-R | 79.6 | 68.3 | 30.2 | - | 37.4 | 40.2 | 69.7 | 43.3 | 53.6 |
| | IN-Sketch | 80.4 | 69.4 | 30.1 | 51.2 | - | 40.6 | 70.7 | 43.0 | 56.3 |
| | ObjNet | 81.8 | 71.5 | 37.4 | 48.7 | 36.1 | - | 67.7 | 43.9 | 59.2 |
| | IN-Cartoon | 81.1 | 70.1 | 30.9 | 49.4 | 36.4 | 41.3 | - | 44.1 | 56.1 |
| | IN-Drawing | 81.6 | 71.0 | 32.1 | 49.9 | 37.3 | 42.7 | 70.2 | - | 58.0 |
| | IN-C | 81.5 | 70.7 | 29.8 | 45.8 | 33.4 | 42.3 | 69.6 | 37.8 | - |
| EWC (Kirkpatrick et al., 2017) | IN-V2 | 82.2 | - | 35.5 | 49.0 | 36.3 | 44.8 | 70.2 | 44.7 | 59.4 |
| | IN-A | 81.1 | 70.9 | - | 49.1 | 35.4 | **48.4** | 66.7 | 41.0 | 58.8 |
| | IN-R | 80.8 | 69.8 | 34.5 | - | 44.3 | 42.6 | 70.3 | 51.0 | 57.6 |
| | IN-Sketch | 81.8 | 71.2 | 31.9 | 57.1 | - | 42.5 | 71.8 | 51.6 | 59.3 |
| | ObjNet | 80.9 | 69.8 | 39.0 | 47.8 | 35.3 | - | 66.5 | 42.3 | 58.2 |
| | IN-Cartoon | 81.8 | 70.5 | 32.6 | 50.6 | 36.8 | 42.4 | - | 45.2 | 56.7 |
| | IN-Drawing | 81.0 | 70.6 | 31.7 | 52.1 | 38.0 | 44.0 | 69.1 | - | 59.4 |
| | IN-C | 82.2 | **71.6** | 35.2 | 50.3 | 37.7 | 45.4 | 69.1 | 51.1 | - |
| LwF (Li & Hoiem, 2017) | IN-V2 | 81.8 | - | 38.3 | 49.2 | 36.6 | 44.8 | 69.3 | 42.6 | 59.3 |
| | IN-A | 80.5 | 70.3 | - | 48.5 | 34.9 | 45.6 | 66.4 | 41.7 | 58.1 |
| | IN-R | 79.4 | 68.6 | 35.7 | - | 49.6 | 41.6 | 69.1 | 52.0 | 57.3 |
| | IN-Sketch | 80.1 | 70.0 | 33.2 | 64.0 | - | 42.7 | 71.0 | 51.6 | 57.3 |
| | ObjNet | 81.1 | 70.3 | 38.3 | 46.8 | 35.0 | - | 66.0 | 39.6 | 57.0 |
| | IN-Cartoon | 86.7 | 70.5 | 34.7 | 50.4 | 37.3 | 43.3 | - | 43.8 | 57.9 |
| | IN-Drawing | 83.3 | 68.9 | 29.6 | 51.2 | 39.0 | 40.9 | 69.3 | - | 58.1 |
| | IN-C | 99.7 | 67.4 | 25.3 | 48.1 | 35.2 | 38.2 | **93.3** | 72.3 | - |
| LP-FT (Kumar et al., 2022) | IN-V2 | 81.6 | - | 39.8 | 49.3 | 36.3 | 45.2 | 69.0 | 42.2 | 58.7 |
| | IN-A | 79.6 | 69.2 | - | 48.3 | 35.0 | 46.6 | 64.7 | 39.9 | 56.7 |
| | IN-R | 77.8 | 67.4 | 35.0 | - | 50.2 | 41.8 | 68.1 | 51.2 | 55.4 |
| | IN-Sketch | 78.9 | 68.5 | 32.6 | 64.4 | - | 41.7 | 70.4 | 50.5 | 55.8 |
| | ObjNet | 79.9 | 69.2 | 40.2 | 46.2 | 33.9 | - | 63.8 | 37.2 | 55.5 |
| | IN-Cartoon | 86.0 | 68.2 | 33.1 | 49.8 | 36.7 | 42.0 | - | 43.4 | 54.4 |
| | IN-Drawing | 82.2 | 67.1 | 27.8 | 50.9 | 39.1 | 39.3 | 67.7 | - | 56.0 |
| | IN-C | 99.1 | 63.5 | 21.4 | 43.7 | 31.3 | 33.6 | 92.2 | 71.7 | - |
| WiSE-FT (Wortsman et al., 2022b) | IN-V2 | 82.1 | - | 37.6 | 49.0 | 36.6 | 45.2 | 69.7 | 43.9 | 59.3 |
| | IN-A | 81.3 | 71.2 | - | 49.1 | 36.3 | 47.0 | 68.4 | 43.4 | 59.0 |
| | IN-R | 81.5 | 71.3 | 38.2 | - | 47.6 | 45.6 | 71.7 | 53.7 | 60.0 |
| | IN-Sketch | 81.5 | 71.4 | 34.2 | 59.4 | - | 43.9 | 72.1 | 51.7 | 59.2 |
| | ObjNet | 81.5 | 71.1 | 39.0 | 48.0 | 36.3 | - | 68.1 | 42.0 | 58.2 |
| | IN-Cartoon | 84.8 | 71.3 | 34.8 | 49.8 | 37.2 | 44.0 | - | 45.6 | 58.1 |
| | IN-Drawing | 83.1 | 71.0 | 32.9 | 51.0 | 39.6 | 43.3 | 70.6 | - | 59.8 |
| | IN-C | 92.0 | 71.3 | 33.7 | 50.4 | 38.4 | 43.8 | 82.4 | 60.8 | - |
| Model Soup PRE-FT-EWC-LwF (Wortsman et al., 2022a) | IN-V2 | 82.2 | - | 37.4 | 49.1 | 36.6 | 45.1 | 69.8 | 44.0 | 59.5 |
| | IN-A | 81.2 | 71.2 | - | 49.3 | 36.0 | 47.1 | 67.9 | 42.9 | 59.1 |
| | IN-R | 81.4 | 71.2 | 37.6 | - | 48.0 | 44.7 | 71.8 | 53.6 | 59.8 |
| | IN-Sketch | 81.6 | 71.4 | 34.3 | 60.8 | - | 43.8 | 72.4 | 52.5 | 59.4 |
| | ObjNet | 81.5 | 70.9 | 39.2 | 48.1 | 36.2 | - | 67.9 | 42.2 | 58.3 |
| | IN-Cartoon | 84.7 | 71.1 | 34.9 | 50.3 | 37.5 | 44.0 | - | 45.6 | 58.2 |
| | IN-Drawing | 83.1 | 71.1 | 32.8 | 51.7 | 39.8 | 43.5 | 70.6 | - | **60.1** |
| | IN-C | 93.5 | 71.2 | 33.1 | 50.9 | 38.9 | 43.6 | 84.1 | 62.9 | - |

Table 20: The accuracy on each OOD dataset after fine-tuning ImageNet-21K with AugReg pre-trained ViT-B/16 on the downstream datasets with various methods. Note that ImageNet-Drawing, ImageNet-Cartoon, and ImageNet-C are generated from the ImageNet validation set. Green and red indicate relative performance increases and decreases, respectively, compared to the pre-trained model. Bold indicates the best performance on each evaluation dataset.

| Method | Downstream Dataset | $D_{pre}$ IN | Realistic OOD | | | | | Synthetic OOD | | IN-C |
| --- | --- | --- | --- | --- | --- | --- | --- | --- | --- | --- |
| | | | IN-V2 | IN-A | IN-R | IN-Sketch | ObjNet | IN-Cartoon | IN-Drawing | |
| Pre-Trained | | 84.5 | 74.0 | 43.2 | 56.8 | 43.2 | 48.4 | 75.1 | 54.9 | 66.5 |
| FT | IN-V2 | 81.3 | - | 47.5 | 56.8 | 40.0 | 49.7 | 70.6 | 49.3 | 64.4 |
| | IN-A | 74.8 | 64.8 | - | 50.6 | 35.6 | 46.3 | 60.7 | 40.7 | 56.3 |
| | IN-R | 73.8 | 63.7 | 33.9 | - | 52.4 | 42.5 | 65.1 | 54.4 | 55.7 |
| | IN-Sketch | 75.2 | 65.2 | 28.4 | 70.6 | - | 43.7 | 65.3 | 54.2 | 54.8 |
| | ObjNet | 76.7 | 65.6 | 37.1 | 47.6 | 32.5 | - | 62.2 | 36.0 | 54.9 |
| | IN-Cartoon | 95.3 | 67.2 | 35.1 | 54.9 | 39.8 | 44.8 | - | 58.9 | 61.5 |
| | IN-Drawing | 91.1 | 65.2 | 27.6 | 54.2 | 41.1 | 41.1 | 78.4 | - | 59.9 |
| | IN-C | **99.9** | 62.7 | 20.2 | 46.5 | 31.8 | 35.0 | 98.1 | **84.1** | - |
| Linear Probing | IN-V2 | 83.1 | - | 45.1 | 56.3 | 42.8 | 48.3 | 73.7 | 53.3 | 65.7 |
| | IN-A | 83.3 | 73.6 | - | 56.7 | 42.5 | 51.0 | 73.6 | 52.8 | 65.8 |
| | IN-R | 82.2 | 72.1 | 44.6 | - | 45.3 | 49.2 | 71.6 | 52.2 | 64.0 |
| | IN-Sketch | 79.3 | 69.2 | 42.3 | 64.6 | - | 49.0 | 70.0 | 53.0 | 62.1 |
| | ObjNet | 83.1 | 73.2 | 47.8 | 56.3 | 41.4 | - | 73.3 | 52.6 | 65.9 |
| | IN-Cartoon | 91.1 | 71.9 | 43.3 | 56.5 | 42.2 | 47.5 | - | 58.5 | 69.9 |
| | IN-Drawing | 87.6 | 71.9 | 42.0 | 57.5 | 43.5 | 47.0 | 79.7 | - | **70.0** |
| | IN-C | 99.2 | 65.0 | 35.2 | 50.3 | 37.0 | 39.5 | 93.8 | 76.1 | - |
| Visual Prompt (Bahng et al., 2022) | IN-V2 | 81.3 | - | 37.2 | 53.6 | 38.8 | 48.8 | 67.8 | 42.8 | 56.9 |
| | IN-A | 78.5 | 67.3 | - | 52.3 | 35.0 | 51.5 | 63.1 | 34.6 | 51.1 |
| | IN-R | 74.9 | 64.1 | 27.4 | - | 47.4 | 43.3 | 62.2 | 49.2 | 46.4 |
| | IN-Sketch | 78.3 | 67.4 | 28.0 | 64.2 | - | 45.0 | 67.2 | 52.7 | 51.7 |
| | ObjNet | 75.0 | 64.2 | 34.8 | 43.9 | 30.4 | - | 54.1 | 27.8 | 42.5 |
| | IN-Cartoon | 80.8 | 68.6 | 31.8 | 54.3 | 39.3 | 45.2 | - | 45.4 | 53.1 |
| | IN-Drawing | 79.3 | 68.3 | 27.9 | 55.1 | 40.1 | 44.6 | 67.8 | - | 54.5 |
| | IN-C | 82.3 | 71.3 | 38.8 | 53.3 | 38.9 | 48.1 | 70.6 | 53.7 | - |
| LoRA (Hu et al., 2021) | IN-V2 | 84.6 | - | 44.9 | 57.1 | 43.4 | 49.9 | 75.2 | 55.0 | 67.2 |
| | IN-A | 84.3 | **74.5** | - | 57.9 | 42.8 | **52.8** | 74.7 | 54.2 | 67.8 |
| | IN-R | 84.2 | 74.1 | 47.6 | - | 48.5 | 52.4 | 75.4 | 58.8 | 67.0 |
| | IN-Sketch | 84.0 | 73.7 | 44.9 | 66.5 | - | 51.3 | 75.6 | 60.7 | 66.9 |
| | ObjNet | 84.2 | 74.3 | 49.5 | 58.0 | 42.4 | - | 74.5 | 53.6 | 67.2 |
| | IN-Cartoon | 84.5 | 73.8 | 44.9 | 58.3 | 43.7 | 50.0 | - | 55.5 | 66.4 |
| | IN-Drawing | 84.0 | 73.8 | 43.9 | 61.2 | 47.3 | 49.7 | 75.3 | - | 67.1 |
| | IN-C | 64.8 | 52.2 | 7.9 | 36.4 | 24.9 | 24.6 | 39.4 | 18.9 | - |
| EWC (Kirkpatrick et al., 2017) | IN-V2 | 84.0 | - | **52.3** | 59.4 | 43.6 | 52.1 | 73.2 | 54.0 | 67.7 |
| | IN-A | 81.3 | 71.3 | - | 56.8 | 41.8 | 52.4 | 69.8 | 46.8 | 65.8 |
| | IN-R | 80.9 | 70.5 | 47.1 | - | 56.2 | 48.4 | 72.3 | 63.0 | 63.9 |
| | IN-Sketch | 83.1 | 72.9 | 44.8 | 70.3 | - | 50.1 | 75.4 | 63.9 | 66.2 |
| | ObjNet | 80.8 | 70.6 | 49.9 | 53.3 | 40.1 | - | 67.4 | 44.5 | 63.5 |
| | IN-Cartoon | 84.6 | 72.2 | 45.0 | 58.5 | 43.6 | 49.2 | - | 57.4 | 63.9 |
| | IN-Drawing | 84.2 | 73.1 | 42.2 | 59.3 | 45.8 | 49.0 | 75.4 | - | 65.9 |
| | IN-C | 85.4 | 74.2 | 46.5 | 56.4 | 42.8 | 49.6 | 75.9 | 62.5 | - |
| LwF (Li & Hoiem, 2017) | IN-V2 | 83.6 | - | 42.9 | 56.3 | 40.7 | 48.2 | 73.8 | 53.5 | 66.0 |
| | IN-A | 82.8 | 72.1 | - | 55.5 | 39.4 | 48.4 | 72.2 | 51.6 | 64.1 |
| | IN-R | 82.5 | 71.9 | 39.3 | - | 54.3 | 46.5 | 74.1 | 58.7 | 64.1 |
| | IN-Sketch | 80.3 | 70.1 | 33.2 | 69.2 | - | 46.6 | 71.7 | 56.5 | 61.5 |
| | ObjNet | 81.9 | 71.4 | 42.2 | 53.6 | 38.4 | - | 70.7 | 45.9 | 62.2 |
| | IN-Cartoon | 96.6 | 71.8 | 37.9 | 57.1 | 41.7 | 47.0 | - | 62.0 | 71.7 |
| | IN-Drawing | 94.3 | 69.9 | 31.1 | 57.5 | 43.6 | 44.1 | 83.6 | - | 68.5 |
| | IN-C | 99.9 | 69.2 | 27.8 | 53.9 | 38.5 | 41.8 | 97.5 | 79.4 | - |
| LP-FT (Kumar et al., 2022) | IN-V2 | 82.3 | - | 50.8 | 56.9 | 41.9 | 50.3 | 72.4 | 52.5 | 65.6 |
| | IN-A | 80.9 | 70.9 | - | 54.5 | 40.2 | 51.0 | 69.3 | 45.7 | 63.2 |
| | IN-R | 76.3 | 66.2 | 35.3 | - | 49.0 | 45.3 | 66.5 | 52.8 | 57.0 |
| | IN-Sketch | 76.9 | 66.7 | 35.1 | 69.7 | - | 46.8 | 68.0 | 53.6 | 58.5 |
| | ObjNet | 79.6 | 68.9 | 42.2 | 52.5 | 36.1 | - | 67.4 | 43.4 | 59.9 |
| | IN-Cartoon | 96.0 | 69.6 | 40.0 | 55.6 | 40.9 | 47.2 | - | 60.9 | 68.6 |
| | IN-Drawing | 92.8 | 67.5 | 31.0 | 56.4 | 42.5 | 43.7 | 81.7 | - | 66.9 |
| | IN-C | 99.9 | 63.3 | 27.6 | 49.3 | 34.8 | 37.1 | 97.2 | 83.0 | - |
| WiSE-FT (Wortsman et al., 2022b) | IN-V2 | 84.1 | - | 50.0 | 58.5 | 43.5 | 51.4 | 74.6 | 54.9 | 68.0 |
| | IN-A | 83.0 | 73.1 | - | 57.2 | 42.2 | 52.5 | 72.6 | 53.3 | 66.5 |
| | IN-R | 82.8 | 72.8 | 47.7 | - | 56.8 | 50.4 | 75.4 | 63.7 | 66.9 |
| | IN-Sketch | 82.3 | 72.9 | 40.5 | 70.7 | - | 49.8 | 74.5 | 62.5 | 65.1 |
| | ObjNet | 83.2 | 72.9 | 48.0 | 56.1 | 41.9 | - | 73.2 | 50.0 | 65.6 |
| | IN-Cartoon | 92.0 | 72.8 | 45.5 | 58.4 | 44.2 | 50.1 | - | 61.8 | 68.6 |
| | IN-Drawing | 90.2 | 73.0 | 41.1 | 59.0 | 46.4 | 49.1 | 80.3 | - | 69.2 |
| | IN-C | 96.5 | 71.7 | 36.9 | 55.8 | 41.2 | 46.4 | 92.3 | 73.9 | - |
| Model Soup PRE-FT-EWC-LwF (Wortsman et al., 2022a) | IN-V2 | 84.2 | - | 49.9 | 58.6 | 43.7 | 51.1 | 74.7 | 55.3 | 68.0 |
| | IN-A | 83.5 | 73.7 | - | 57.7 | 42.9 | 52.3 | 73.4 | 53.9 | 66.9 |
| | IN-R | 83.5 | 73.6 | 48.4 | - | 57.2 | 50.5 | 76.1 | 64.4 | 67.4 |
| | IN-Sketch | 82.7 | 72.8 | 41.6 | 71.0 | - | 49.8 | 75.0 | 62.6 | 65.6 |
| | ObjNet | 83.3 | 73.1 | 49.4 | 56.0 | 42.1 | - | 72.8 | 49.7 | 65.6 |
| | IN-Cartoon | 91.7 | 73.0 | 44.9 | 58.8 | 44.3 | 49.8 | - | 61.9 | 69.2 |
| | IN-Drawing | 90.2 | 73.0 | 40.9 | 59.7 | 46.8 | 48.6 | 80.7 | - | 69.6 |
| | IN-C | 96.5 | 72.9 | 41.0 | 58.0 | 43.6 | 47.7 | 92.1 | 75.5 | - |

Table 21: The accuracy on each OOD dataset after fine-tuning ImageNet-21K-P pre-trained ViT-B/16 on the downstream datasets with various methods. Note that ImageNet-Drawing, ImageNet-Cartoon, and ImageNet-C are generated from the ImageNet validation set. Green and red indicate relative performance increases and decreases, respectively, compared to the pre-trained model. Bold indicates the best performance on each evaluation dataset.

| Method | Downstream Dataset | $D_{pre}$ IN | Realistic OOD | | | | | Synthetic OOD | | |
|---|---|---|---|---|---|---|---|---|---|---|
| | | | IN-V2 | IN-A | IN-R | IN-Sketch | ObjNet | IN-Cartoon | IN-Drawing | IN-C |
| Pre-Trained | | 84.3 | 74.0 | 34.1 | 51.5 | 40.2 | 46.7 | 73.5 | 45.1 | 61.4 |
| FT | IN-V2 | 83.5 | - | 44.6 | 52.9 | 40.4 | 47.1 | 71.4 | 39.2 | 62.1 |
| | IN-A | 80.8 | 70.7 | - | 50.3 | 38.6 | 47.2 | 65.5 | 35.5 | 60.5 |
| | IN-R | 78.9 | 68.8 | 38.7 | - | 54.6 | 42.8 | 69.9 | 54.2 | 57.5 |
| | IN-Sketch | 81.9 | 71.8 | 35.4 | **67.0** | - | 44.5 | 74.8 | 54.4 | 59.3 |
| | ObjNet | 81.1 | 70.6 | 43.8 | 47.5 | 36.8 | - | 63.1 | 29.6 | 57.7 |
| | IN-Cartoon | 89.0 | 71.0 | 38.0 | 51.9 | 40.2 | 43.4 | - | 42.5 | 56.1 |
| | IN-Drawing | 85.8 | 69.5 | 31.4 | 52.7 | 42.7 | 41.8 | 70.3 | - | 57.9 |
| | IN-C | 99.8 | 67.9 | 25.6 | 47.1 | 34.7 | 37.4 | 95.0 | 73.7 | - |
| HeadOnly | IN-V2 | 84.3 | - | 34.8 | 51.6 | 40.0 | 46.4 | 74.1 | 45.6 | 61.4 |
| | IN-A | 84.0 | 73.9 | - | 51.9 | 40.1 | 48.5 | 73.9 | 45.2 | 61.8 |
| | IN-R | 83.8 | 73.9 | 35.6 | - | 42.0 | 46.5 | 73.3 | 46.1 | 61.0 |
| | IN-Sketch | 83.2 | 73.4 | 34.8 | 57.2 | - | 46.9 | 73.4 | 47.7 | 60.8 |
| | ObjNet | 83.9 | 73.7 | 36.6 | 51.3 | 39.8 | - | 73.7 | 45.1 | 61.9 |
| | IN-Cartoon | 85.6 | 73.8 | 34.5 | 52.3 | 40.3 | 45.7 | - | 46.5 | 61.3 |
| | IN-Drawing | 84.5 | 73.5 | 33.2 | 53.0 | 41.7 | 45.5 | 74.5 | - | 62.0 |
| | IN-C | 97.3 | 68.4 | 30.0 | 44.6 | 33.4 | 39.2 | 86.4 | 56.2 | - |
| Visual Prompt (Bahng et al., 2022) | IN-V2 | 79.7 | - | 27.1 | 46.9 | 32.7 | 45.2 | 62.1 | 32.9 | 49.6 |
| | IN-A | 76.7 | 66.0 | - | 44.0 | 27.5 | 46.4 | 55.6 | 27.6 | 44.6 |
| | IN-R | 74.1 | 62.8 | 20.3 | - | 41.0 | 39.4 | 59.3 | 42.1 | 40.5 |
| | IN-Sketch | 77.0 | 65.5 | 18.3 | 56.6 | - | 40.9 | 62.6 | 43.8 | 44.1 |
| | ObjNet | 72.4 | 61.1 | 22.1 | 34.8 | 23.1 | - | 46.4 | 18.4 | 34.8 |
| | IN-Cartoon | 79.0 | 67.2 | 21.6 | 48.5 | 33.4 | 41.7 | - | 34.8 | 44.2 |
| | IN-Drawing | 75.9 | 63.9 | 17.1 | 48.0 | 33.2 | 38.4 | 58.5 | - | 44.3 |
| | IN-C | 80.9 | 70.2 | 28.5 | 48.5 | 35.4 | 45.1 | 65.6 | 46.9 | - |
| LoRA (Hu et al., 2021) | IN-V2 | 84.2 | - | 34.5 | 51.3 | 40.3 | 46.6 | 73.9 | 45.5 | 61.3 |
| | IN-A | 84.1 | 74.1 | - | 51.4 | 40.1 | 47.9 | 73.7 | 45.4 | 62.1 |
| | IN-R | 84.1 | 73.9 | 35.5 | - | 41.0 | 46.7 | 74.1 | 46.1 | 61.2 |
| | IN-Sketch | 84.1 | 73.8 | 34.1 | 55.0 | - | 46.4 | 74.5 | 49.4 | 61.4 |
| | ObjNet | 84.2 | 74.1 | 36.8 | 51.3 | 40.0 | - | 73.6 | 45.6 | 62.1 |
| | IN-Cartoon | 84.0 | 73.6 | 33.9 | 52.1 | 40.7 | 45.4 | - | 45.6 | 60.3 |
| | IN-Drawing | 84.0 | 73.6 | 33.5 | 54.9 | 43.9 | 45.9 | 73.9 | - | 61.6 |
| | IN-C | 84.5 | 74.3 | 35.3 | 50.6 | 38.6 | 46.7 | 73.9 | 45.1 | - |
| EWC (Kirkpatrick et al., 2017) | IN-V2 | 84.4 | - | 37.9 | 52.6 | 40.9 | 47.8 | 74.1 | 45.8 | 62.4 |
| | IN-A | 83.8 | **74.4** | - | 53.2 | 40.3 | **50.8** | 72.0 | 42.0 | 63.7 |
| | IN-R | 81.4 | 71.4 | 30.2 | - | 52.1 | 43.3 | 72.7 | 57.7 | 55.3 |
| | IN-Sketch | 83.9 | 73.6 | 34.2 | 62.0 | - | 46.2 | 76.0 | 54.4 | 61.4 |
| | ObjNet | 84.0 | 74.1 | 42.0 | 51.9 | 40.5 | - | 71.7 | 42.6 | 62.5 |
| | IN-Cartoon | 84.3 | 73.7 | 35.3 | 54.2 | 41.5 | 46.0 | - | 47.0 | 59.9 |
| | IN-Drawing | 83.4 | 73.0 | 33.3 | 56.1 | 44.0 | 45.0 | 73.3 | - | 60.9 |
| | IN-C | 84.2 | 74.2 | 38.8 | 52.7 | 41.5 | 48.5 | 73.0 | 50.9 | - |
| LwF (Li & Hoiem, 2017) | IN-V2 | 83.9 | - | 44.0 | 53.5 | 41.1 | 47.2 | 72.9 | 42.0 | 62.9 |
| | IN-A | 83.3 | 73.7 | - | 52.4 | 40.2 | 48.4 | 71.2 | 42.5 | 63.5 |
| | IN-R | 82.3 | 72.2 | 40.9 | - | 55.0 | 45.1 | 74.3 | 57.2 | 62.0 |
| | IN-Sketch | 82.9 | 72.7 | 36.7 | 65.9 | - | 45.4 | 75.7 | 53.7 | 61.1 |
| | ObjNet | 83.5 | 73.3 | 44.5 | 50.4 | 39.8 | - | 69.7 | 38.5 | 61.8 |
| | IN-Cartoon | 89.8 | 73.1 | 39.6 | 53.3 | 41.7 | 45.4 | - | 45.6 | 61.3 |
| | IN-Drawing | 87.0 | 71.5 | 32.8 | 53.7 | 43.6 | 44.0 | 73.4 | - | 61.2 |
| | IN-C | 99.7 | 70.5 | 25.6 | 50.3 | 37.6 | 40.6 | 94.7 | 71.9 | - |
| LP-FT (Kumar et al., 2022) | IN-V2 | 83.6 | - | 44.2 | 52.4 | 39.8 | 47.1 | 72.0 | 39.9 | 62.5 |
| | IN-A | 82.5 | 72.8 | - | 51.6 | 39.3 | 48.7 | 69.5 | 40.0 | 62.6 |
| | IN-R | 81.3 | 71.5 | 40.6 | - | 54.3 | 45.3 | 73.1 | 56.2 | 60.5 |
| | IN-Sketch | 82.2 | 71.9 | 37.2 | 66.7 | - | 45.6 | 75.0 | 53.0 | 60.2 |
| | ObjNet | 82.6 | 72.3 | **44.7** | 49.8 | 39.1 | - | 67.9 | 36.9 | 60.6 |
| | IN-Cartoon | 89.8 | 71.4 | 38.4 | 52.5 | 40.5 | 43.9 | - | 44.6 | 58.8 |
| | IN-Drawing | 86.5 | 69.9 | 30.3 | 53.1 | 43.2 | 41.9 | 71.4 | - | 58.7 |
| | IN-C | 99.9 | 64.7 | 19.8 | 43.5 | 30.7 | 33.1 | 96.1 | 78.2 | - |
| WiSE-FT (Wortsman et al., 2022b) | IN-V2 | 84.4 | - | 42.1 | 53.0 | 41.4 | 48.3 | 73.9 | 44.7 | 63.3 |
| | IN-A | 84.0 | 74.3 | - | 53.1 | 41.8 | 50.2 | 73.0 | 44.9 | 64.1 |
| | IN-R | 84.0 | 74.3 | 43.1 | - | 53.6 | 48.2 | 76.8 | 59.0 | 64.3 |
| | IN-Sketch | 84.0 | 74.1 | 37.4 | 62.9 | - | 47.3 | 76.6 | 54.4 | 62.7 |
| | ObjNet | 84.1 | 74.2 | 43.8 | 51.9 | 41.8 | - | 72.3 | 42.3 | 62.9 |
| | IN-Cartoon | 87.4 | 74.0 | 40.1 | 53.5 | 42.1 | 46.9 | - | 47.8 | 61.4 |
| | IN-Drawing | 86.4 | 73.5 | 37.1 | 54.6 | 44.5 | 46.5 | 75.6 | - | 63.5 |
| | IN-C | 94.1 | 73.3 | 36.9 | 53.4 | 41.6 | 46.1 | 87.3 | 63.8 | - |
| Model Soup PRE-FT-EWC-LwF (Wortsman et al., 2022a) | IN-V2 | 84.4 | - | 41.7 | 53.2 | 41.5 | 48.1 | 74.0 | 44.9 | 63.3 |
| | IN-A | 83.9 | 74.3 | - | 53.2 | 41.6 | 50.2 | 72.5 | 44.1 | 64.1 |
| | IN-R | 83.9 | 74.1 | 42.7 | - | 54.2 | 47.9 | 76.8 | 59.4 | 64.1 |
| | IN-Sketch | 84.0 | 74.0 | 37.3 | 64.0 | - | 47.1 | 76.7 | 55.0 | 62.7 |
| | ObjNet | 84.1 | 74.2 | 44.4 | 51.9 | 41.4 | - | 71.9 | 42.1 | 62.9 |
| | IN-Cartoon | 87.2 | 74.0 | 39.5 | 53.8 | 42.1 | 46.6 | - | 47.6 | 61.4 |
| | IN-Drawing | 86.2 | 73.5 | 37.0 | 55.0 | 44.6 | 46.6 | 75.4 | - | 63.4 |
| | IN-C | 93.8 | 73.9 | 36.3 | 53.8 | 42.1 | 46.2 | 86.4 | 63.4 | - |

Table 22: The accuracy on each OOD dataset after fine-tuning LAION-2B pre-trained ViT-B/16 on the downstream datasets with various methods. Note that ImageNet-Drawing, ImageNet-Cartoon, and ImageNet-C are generated from the ImageNet validation set. Green and red indicate relative performance increases and decreases, respectively, compared to the pre-trained model. Bold indicates the best performance on each evaluation dataset.

| Method | Downstream Dataset | $D_{pre}$ IN | Realistic OOD | | | | | Synthetic OOD | | |
|---|---|---|---|---|---|---|---|---|---|---|
| | | | IN-V2 | IN-A | IN-R | IN-Sketch | ObjNet | IN-Cartoon | IN-Drawing | IN-C |
| Pre-Trained | | 85.5 | **75.6** | 41.5 | 68.8 | 55.4 | 51.1 | 78.2 | 58.4 | **63.0** |
| FT | IN-V2 | 59.0 | - | 8.1 | 25.6 | 12.2 | 21.1 | 36.6 | 12.2 | 24.5 |
| | IN-A | 28.7 | 20.4 | - | 11.9 | 4.5 | 10.3 | 14.9 | 4.0 | 8.5 |
| | IN-R | 47.1 | 36.4 | 7.2 | - | 29.5 | 19.4 | 33.7 | 17.0 | 21.4 |
| | IN-Sketch | 29.1 | 20.3 | 2.6 | 37.8 | - | 9.2 | 20.4 | 8.0 | 9.7 |
| | ObjNet | 39.4 | 30.5 | 4.8 | 16.4 | 6.7 | - | 20.2 | 4.7 | 13.5 |
| | IN-Cartoon | 90.2 | 52.8 | 9.9 | 36.4 | 22.0 | 27.6 | - | 29.1 | 32.6 |
| | IN-Drawing | 62.0 | 37.0 | 5.0 | 29.7 | 19.1 | 15.4 | 48.6 | - | 22.5 |
| | IN-C | 99.9 | 54.4 | 8.2 | 40.8 | 27.0 | 24.3 | 98.6 | 85.1 | - |
| Linear Probing | IN-V2 | 84.9 | - | 42.0 | 68.7 | 54.0 | 50.2 | 77.4 | 57.5 | 62.3 |
| | IN-A | 84.5 | 74.9 | - | 68.9 | 53.6 | **53.2** | 76.3 | 54.2 | 62.7 |
| | IN-R | 82.8 | 72.6 | 37.4 | - | 57.6 | 47.9 | 75.7 | 60.5 | 61.2 |
| | IN-Sketch | 82.5 | 72.8 | 37.9 | **74.8** | - | 49.1 | 76.0 | 59.6 | 60.7 |
| | ObjNet | 84.2 | 74.5 | **44.3** | 66.9 | 52.5 | - | 74.6 | 53.7 | 61.2 |
| | IN-Cartoon | 85.7 | 72.6 | 36.7 | 69.8 | 52.9 | 47.5 | - | 57.8 | 59.7 |
| | IN-Drawing | 83.8 | 71.1 | 27.7 | 67.0 | 51.7 | 43.9 | 76.0 | - | 57.7 |
| | IN-C | 97.6 | 67.8 | 25.9 | 61.4 | 47.5 | 40.2 | 93.2 | 78.8 | - |
| Visual Prompt (Bahng et al., 2022) | IN-V2 | 82.8 | - | 32.3 | 64.4 | 51.4 | 49.5 | 72.3 | 47.0 | 54.4 |
| | IN-A | 80.9 | 70.9 | - | 64.1 | 49.0 | 53.0 | 69.5 | 36.8 | 50.7 |
| | IN-R | 78.7 | 68.0 | 30.1 | - | 54.7 | 45.9 | 68.5 | 49.5 | 49.4 |
| | IN-Sketch | 81.4 | 70.6 | 28.6 | 69.5 | - | 48.9 | 71.6 | 49.8 | 50.9 |
| | ObjNet | 77.9 | 66.8 | 30.1 | 55.2 | 41.1 | - | 61.5 | 26.8 | 42.4 |
| | IN-Cartoon | 82.4 | 71.0 | 30.9 | 64.0 | 49.3 | 47.8 | - | 43.7 | 48.7 |
| | IN-Drawing | 81.6 | 70.8 | 26.6 | 63.3 | 49.3 | 47.4 | 69.9 | - | 50.2 |
| | IN-C | 83.8 | 73.3 | 34.1 | 66.6 | 52.2 | 49.2 | 74.5 | 58.9 | - |
| LoRA (Hu et al., 2021) | IN-V2 | 85.0 | - | 40.8 | 68.6 | 53.9 | 50.6 | 77.3 | 57.8 | 62.0 |
| | IN-A | 84.5 | 75.0 | - | 68.6 | 53.6 | 52.8 | 76.2 | 53.1 | 62.2 |
| | IN-R | 82.7 | 72.2 | 36.9 | - | 58.2 | 47.6 | 75.8 | 60.2 | 61.2 |
| | IN-Sketch | 82.9 | 72.5 | 35.2 | 74.6 | - | 48.4 | 76.3 | 60.2 | 60.9 |
| | ObjNet | 84.1 | 74.0 | 42.5 | 67.3 | 52.2 | - | 72.8 | 51.6 | 59.7 |
| | IN-Cartoon | 83.7 | 72.6 | 35.9 | 69.5 | 53.3 | 47.0 | - | 57.3 | 58.4 |
| | IN-Drawing | 81.7 | 70.9 | 28.2 | 67.1 | 52.1 | 44.8 | 74.4 | - | 55.8 |
| | IN-C | 78.9 | 67.8 | 20.3 | 61.4 | 47.3 | 38.7 | 68.8 | 38.3 | - |
| EWC (Kirkpatrick et al., 2017) | IN-V2 | 80.2 | - | 26.4 | 49.4 | 31.9 | 41.2 | 65.2 | 38.0 | 51.4 |
| | IN-A | 69.6 | 60.3 | - | 34.8 | 19.7 | 38.1 | 47.6 | 19.7 | 39.9 |
| | IN-R | 81.5 | 71.2 | 38.3 | - | **58.4** | 47.8 | 73.6 | 58.3 | 58.7 |
| | IN-Sketch | 70.5 | 57.0 | 18.4 | 60.6 | - | 28.0 | 64.0 | 45.8 | 41.7 |
| | ObjNet | 78.1 | 67.0 | 28.5 | 45.5 | 30.7 | - | 59.6 | 28.5 | 48.0 |
| | IN-Cartoon | 84.0 | 71.4 | 34.1 | 64.5 | 47.2 | 46.9 | - | 52.3 | 52.6 |
| | IN-Drawing | 80.9 | 67.7 | 24.8 | 62.8 | 46.8 | 39.5 | 72.1 | - | 50.4 |
| | IN-C | 85.5 | 74.3 | 34.3 | 66.8 | 51.1 | 48.2 | 77.6 | 64.5 | - |
| LwF (Li & Hoiem, 2017) | IN-V2 | 63.5 | - | 8.7 | 28.0 | 14.7 | 22.7 | 41.0 | 12.2 | 28.0 |
| | IN-A | 40.1 | 30.0 | - | 15.1 | 5.7 | 14.5 | 20.6 | 6.1 | 13.6 |
| | IN-R | 56.9 | 46.1 | 7.8 | - | 33.5 | 23.1 | 41.6 | 22.2 | 26.9 |
| | IN-Sketch | 33.3 | 24.7 | 2.6 | 39.0 | - | 11.0 | 22.5 | 8.9 | 11.5 |
| | ObjNet | 50.6 | 39.2 | 6.3 | 19.8 | 9.1 | - | 28.4 | 6.3 | 19.2 |
| | IN-Cartoon | 93.8 | 60.8 | 12.3 | 43.9 | 28.6 | 32.2 | - | 37.7 | 42.0 |
| | IN-Drawing | 70.8 | 42.2 | 5.6 | 37.8 | 26.9 | 17.1 | 58.5 | - | 28.8 |
| | IN-C | 99.9 | 60.4 | 8.9 | 47.8 | 32.8 | 29.1 | 98.2 | 82.6 | - |
| LP-FT (Kumar et al., 2022) | IN-V2 | 59.5 | - | 8.0 | 26.4 | 12.7 | 21.2 | 37.6 | 11.3 | 24.3 |
| | IN-A | 34.5 | 27.2 | - | 13.6 | 5.2 | 13.6 | 17.6 | 4.7 | 11.4 |
| | IN-R | 49.1 | 38.6 | 7.1 | - | 30.2 | 20.3 | 34.5 | 18.5 | 22.5 |
| | IN-Sketch | 30.2 | 22.9 | 2.3 | 38.7 | - | 10.6 | 21.1 | 8.0 | 10.5 |
| | ObjNet | 44.2 | 33.6 | 5.7 | 17.5 | 7.0 | - | 23.4 | 4.8 | 15.4 |
| | IN-Cartoon | 90.9 | 53.5 | 10.7 | 37.6 | 23.0 | 27.7 | - | 31.2 | 32.3 |
| | IN-Drawing | 67.8 | 39.0 | 5.2 | 34.2 | 22.5 | 16.2 | 55.9 | - | 24.4 |
| | IN-C | 99.9 | 51.1 | 6.8 | 38.2 | 23.5 | 20.9 | 98.9 | 87.4 | - |
| WiSE-FT (Wortsman et al., 2022b) | IN-V2 | 76.3 | - | 17.9 | 38.9 | 22.2 | 34.7 | 58.1 | 23.5 | 43.7 |
| | IN-A | 59.8 | 48.0 | - | 24.2 | 10.8 | 22.8 | 34.4 | 10.0 | 26.7 |
| | IN-R | 74.1 | 62.1 | 18.2 | - | 41.1 | 36.2 | 60.9 | 38.1 | 43.3 |
| | IN-Sketch | 58.6 | 47.0 | 7.2 | 48.0 | - | 21.4 | 43.7 | 20.6 | 25.4 |
| | ObjNet | 74.7 | 62.2 | 18.3 | 40.0 | 23.3 | - | 55.4 | 21.4 | 42.3 |
| | IN-Cartoon | 88.5 | 69.5 | 26.3 | 54.7 | 39.0 | 42.6 | - | 49.8 | 53.0 |
| | IN-Drawing | 81.1 | 61.7 | 15.6 | 53.6 | 41.0 | 30.8 | 70.5 | - | 45.9 |
| | IN-C | 94.2 | 67.1 | 19.2 | 58.2 | 44.2 | 39.3 | 90.0 | 74.6 | - |
| Model Soup PRE-FT-EWC-LwF (Wortsman et al., 2022a) | IN-V2 | 78.1 | - | 20.5 | 44.2 | 27.8 | 37.3 | 61.3 | 31.2 | 46.2 |
| | IN-A | 68.5 | 56.6 | - | 31.3 | 16.0 | 30.6 | 45.7 | 15.9 | 35.0 |
| | IN-R | 77.0 | 65.3 | 22.5 | - | 46.9 | 39.7 | 65.4 | 45.6 | 47.8 |
| | IN-Sketch | 60.8 | 49.0 | 8.0 | 50.9 | - | 23.4 | 46.6 | 23.5 | 27.5 |
| | ObjNet | 77.0 | 65.2 | 21.6 | 43.0 | 27.1 | - | 58.5 | 25.4 | 45.0 |
| | IN-Cartoon | 88.7 | 71.1 | 27.9 | 58.4 | 43.2 | 43.9 | - | 54.5 | 54.5 |
| | IN-Drawing | 81.3 | 62.3 | 16.2 | 57.3 | 44.9 | 31.4 | 72.2 | - | 47.0 |
| | IN-C | 94.3 | 69.5 | 22.2 | 62.1 | 47.2 | 40.6 | 90.8 | 77.5 | - |

Table 23: The accuracy on each OOD dataset after fine-tuning OpenAI CLIP ViT-B/16 on the downstream datasets with various methods. Note that ImageNet-Drawing, ImageNet-Cartoon, and ImageNet-C are generated from the ImageNet validation set. Green and red indicate relative performance increases and decreases, respectively, compared to the pre-trained model. Bold indicates the best performance on each evaluation dataset.

| Method | Downstream Dataset | $D_{pre}$ IN | Realistic OOD | | | | | Synthetic OOD | | |
| --- | --- | --- | --- | --- | --- | --- | --- | --- | --- | --- |
| | | | IN-V2 | IN-A | IN-R | IN-Sketch | ObjNet | IN-Cartoon | IN-Drawing | IN-C |
| Pre-Trained | | 85.3 | **75.7** | 47.3 | 65.9 | 50.9 | 50.7 | 76.3 | 55.7 | **62.6** |
| FT | IN-V2 | 60.8 | - | 9.7 | 24.2 | 10.1 | 22.0 | 36.4 | 13.0 | 25.7 |
| | IN-A | 29.8 | 23.3 | - | 10.5 | 4.1 | 11.4 | 14.3 | 4.1 | 8.9 |
| | IN-R | 47.9 | 37.9 | 7.9 | - | 29.4 | 19.3 | 34.4 | 20.1 | 22.2 |
| | IN-Sketch | 24.4 | 18.0 | 1.9 | 34.6 | - | 8.2 | 17.3 | 7.5 | 7.2 |
| | ObjNet | 36.0 | 26.7 | 4.9 | 12.5 | 5.4 | - | 18.3 | 3.7 | 10.6 |
| | IN-Cartoon | 92.3 | 52.7 | 10.5 | 34.0 | 18.1 | 24.5 | - | 33.9 | 32.8 |
| | IN-Drawing | 67.3 | 36.1 | 5.0 | 29.5 | 17.7 | 14.2 | 53.0 | - | 23.0 |
| | IN-C | **99.9** | 51.0 | 7.0 | 35.4 | 19.0 | 20.8 | 98.2 | 84.0 | - |
| HeadOnly | IN-V2 | 84.7 | - | 48.0 | 65.6 | 50.3 | 50.9 | 75.3 | 53.9 | 61.6 |
| | IN-A | 84.1 | 74.5 | - | 66.5 | 49.1 | **53.6** | 74.1 | 51.9 | 62.3 |
| | IN-R | 82.3 | 72.9 | 42.0 | - | 54.7 | 48.5 | 73.8 | 58.5 | 59.9 |
| | IN-Sketch | 82.0 | 72.6 | 41.5 | 72.4 | - | 48.8 | 74.3 | 57.6 | 58.0 |
| | ObjNet | 83.9 | 74.5 | **52.0** | 65.5 | 49.0 | - | 73.7 | 51.2 | 60.1 |
| | IN-Cartoon | 85.0 | 73.1 | 40.7 | 65.6 | 48.9 | 46.5 | - | 53.9 | 56.3 |
| | IN-Drawing | 83.1 | 71.4 | 32.2 | 64.6 | 47.9 | 43.9 | 75.3 | - | 54.7 |
| | IN-C | 97.0 | 68.7 | 32.3 | 59.1 | 44.4 | 40.3 | 90.6 | 74.2 | - |
| Visual Prompt (Bahng et al., 2022) | IN-V2 | 82.4 | - | 38.2 | 60.5 | 46.4 | 48.3 | 68.2 | 42.5 | 53.6 |
| | IN-A | 80.4 | 70.5 | - | 62.1 | 45.1 | 51.9 | 66.2 | 35.1 | 50.5 |
| | IN-R | 79.3 | 68.8 | 38.2 | - | 51.2 | 46.5 | 66.9 | 46.6 | 47.9 |
| | IN-Sketch | 81.2 | 70.5 | 35.6 | 67.3 | - | 47.9 | 69.0 | 48.9 | 50.8 |
| | ObjNet | 77.6 | 67.0 | 37.7 | 53.0 | 38.9 | - | 57.8 | 25.2 | 41.1 |
| | IN-Cartoon | 81.8 | 70.1 | 35.7 | 60.7 | 44.4 | 45.5 | - | 40.2 | 46.2 |
| | IN-Drawing | 80.5 | 69.5 | 27.2 | 60.6 | 45.9 | 43.2 | 66.9 | - | 46.8 |
| | IN-C | 83.1 | 73.1 | 41.5 | 64.4 | 49.3 | 49.5 | 71.2 | 54.6 | - |
| LoRA (Hu et al., 2021) | IN-V2 | 84.7 | - | 47.4 | 65.7 | 50.3 | 51.4 | 75.3 | 53.7 | 61.5 |
| | IN-A | 84.0 | 74.6 | - | 66.4 | 48.9 | 53.5 | 73.8 | 51.7 | 62.0 |
| | IN-R | 81.9 | 71.9 | 40.9 | - | 54.4 | 47.9 | 73.3 | 58.1 | 59.5 |
| | IN-Sketch | 82.7 | 73.1 | 40.2 | **72.7** | - | 48.4 | 74.9 | 59.1 | 58.6 |
| | ObjNet | 83.3 | 74.0 | 49.5 | 65.6 | 48.2 | - | 71.3 | 48.6 | 57.9 |
| | IN-Cartoon | 83.4 | 72.8 | 39.6 | 65.4 | 48.7 | 46.7 | - | 53.0 | 54.6 |
| | IN-Drawing | 81.4 | 71.3 | 31.8 | 64.0 | 48.1 | 43.9 | 73.1 | - | 52.5 |
| | IN-C | 79.5 | 69.6 | 26.9 | 59.6 | 42.7 | 41.5 | 69.3 | 33.6 | - |
| EWC (Kirkpatrick et al., 2017) | IN-V2 | 80.6 | - | 33.0 | 48.4 | 29.3 | 40.3 | 64.2 | 40.7 | 52.9 |
| | IN-A | 73.8 | 62.9 | - | 40.9 | 23.5 | 39.3 | 52.2 | 27.0 | 45.3 |
| | IN-R | 80.5 | 71.2 | 43.5 | - | 55.4 | 46.7 | 71.5 | 56.3 | 58.0 |
| | IN-Sketch | 58.5 | 48.2 | 17.7 | 49.3 | - | 25.4 | 53.7 | 33.6 | 32.8 |
| | ObjNet | 77.8 | 67.1 | 33.1 | 43.6 | 24.8 | - | 57.0 | 23.2 | 46.6 |
| | IN-Cartoon | 84.1 | 71.4 | 37.4 | 61.2 | 42.7 | 45.0 | - | 49.5 | 49.4 |
| | IN-Drawing | 80.5 | 67.7 | 27.7 | 59.2 | 41.7 | 37.6 | 70.6 | - | 49.0 |
| | IN-C | 86.7 | 74.2 | 41.5 | 65.1 | 47.8 | 47.0 | 77.3 | 64.1 | - |
| LwF (Li & Hoiem, 2017) | IN-V2 | 65.3 | - | 10.2 | 28.0 | 12.6 | 24.4 | 42.3 | 14.3 | 28.8 |
| | IN-A | 42.3 | 33.6 | - | 14.8 | 5.1 | 16.1 | 21.4 | 7.0 | 14.7 |
| | IN-R | 57.9 | 46.6 | 8.0 | - | 33.2 | 22.3 | 41.9 | 25.4 | 27.7 |
| | IN-Sketch | 26.8 | 20.1 | 2.0 | 35.8 | - | 9.2 | 19.2 | 9.1 | 8.8 |
| | ObjNet | 52.4 | 40.1 | 6.1 | 18.2 | 7.6 | - | 27.4 | 6.4 | 18.6 |
| | IN-Cartoon | 95.0 | 60.1 | 13.1 | 40.3 | 23.5 | 28.6 | - | 42.5 | 41.4 |
| | IN-Drawing | 75.6 | 43.7 | 5.9 | 36.0 | 23.5 | 17.3 | 61.5 | - | 30.3 |
| | IN-C | 99.5 | 65.7 | 12.4 | 51.0 | 34.9 | 33.5 | 96.1 | 73.7 | - |
| LP-FT (Kumar et al., 2022) | IN-V2 | 60.9 | - | 9.6 | 24.9 | 9.7 | 22.3 | 36.8 | 12.4 | 25.8 |
| | IN-A | 33.6 | 25.8 | - | 11.7 | 4.4 | 13.8 | 17.3 | 4.4 | 10.4 |
| | IN-R | 50.2 | 39.9 | 8.3 | - | 29.9 | 21.4 | 35.5 | 20.1 | 23.3 |
| | IN-Sketch | 23.8 | 17.6 | 2.2 | 34.8 | - | 7.6 | 17.2 | 7.8 | 7.3 |
| | ObjNet | 40.3 | 30.6 | 5.0 | 13.9 | 6.0 | - | 18.9 | 4.4 | 12.7 |
| | IN-Cartoon | 91.6 | 51.8 | 10.4 | 33.6 | 18.2 | 24.1 | - | 31.7 | 30.6 |
| | IN-Drawing | 77.5 | 43.4 | 5.8 | 32.4 | 18.2 | 17.9 | 62.2 | - | 27.7 |
| | IN-C | 99.9 | 53.2 | 7.2 | 38.6 | 22.3 | 21.7 | **98.6** | **87.6** | - |
| WiSE-FT (Wortsman et al., 2022b) | IN-V2 | 79.1 | - | 26.3 | 43.1 | 23.9 | 38.0 | 62.9 | 30.9 | 47.9 |
| | IN-A | 72.2 | 60.0 | - | 32.5 | 15.9 | 34.1 | 49.0 | 20.3 | 39.1 |
| | IN-R | 77.1 | 65.6 | 25.8 | - | 45.0 | 39.5 | 65.8 | 47.2 | 48.5 |
| | IN-Sketch | 55.4 | 44.4 | 8.6 | 47.0 | - | 21.6 | 45.1 | 24.7 | 25.5 |
| | ObjNet | 76.2 | 64.2 | 23.2 | 38.6 | 21.8 | - | 57.3 | 21.8 | 41.7 |
| | IN-Cartoon | 89.1 | 69.9 | 28.4 | 52.6 | 35.1 | 40.7 | - | 52.0 | 52.4 |
| | IN-Drawing | 82.0 | 61.9 | 18.0 | 51.5 | 37.5 | 30.5 | 71.9 | - | 47.4 |
| | IN-C | 94.1 | 67.4 | 21.0 | 54.9 | 37.6 | 36.7 | 90.2 | 76.0 | - |
| Model Soup PRE-FT-EWC-LwF (Wortsman et al., 2022a) | IN-V2 | 79.7 | - | 27.8 | 46.7 | 28.1 | 39.2 | 64.5 | 34.7 | 48.7 |
| | IN-A | 73.1 | 61.6 | - | 36.0 | 19.0 | 35.7 | 52.3 | 23.1 | 39.8 |
| | IN-R | 78.3 | 67.4 | 28.5 | - | 48.2 | 41.2 | 67.6 | 49.9 | 50.4 |
| | IN-Sketch | 55.1 | 44.8 | 9.0 | 48.2 | - | 21.9 | 45.7 | 26.0 | 26.3 |
| | ObjNet | 77.8 | 66.1 | 25.6 | 42.2 | 26.3 | - | 59.5 | 24.5 | 44.2 |
| | IN-Cartoon | 89.0 | 71.3 | 29.7 | 55.5 | 38.7 | 41.2 | - | 55.1 | 53.8 |
| | IN-Drawing | 82.7 | 63.3 | 18.9 | 54.1 | 39.9 | 31.2 | 72.8 | - | 48.6 |
| | IN-C | 93.2 | 70.9 | 26.2 | 59.4 | 42.6 | 40.3 | 89.1 | 74.9 | - |

Table 24: The accuracy on each OOD dataset after fine-tuning ImageNet-1K with AugReg pre-trained ViT-B/32 on the downstream datasets with various methods. Note that ImageNet-Drawing, ImageNet-Cartoon, and ImageNet-C are generated from images in the ImageNet validation set. Green and red indicate relative performance increases and decreases, respectively, compared to the pre-trained model. Bold indicates the best performance on each evaluation dataset.

| Method | Downstream Dataset | $D_{pre}$ IN | Realistic OOD | | | | | Synthetic OOD | | |
|---|---|---|---|---|---|---|---|---|---|---|
| | | | IN-V2 | IN-A | IN-R | IN-Sketch | ObjNet | IN-Cartoon | IN-Drawing | IN-C |
| Pre-Trained | | 74.9 | 61.0 | 8.0 | 37.5 | 27.1 | 28.0 | 65.2 | 40.5 | 53.5 |
| FT | IN-V2 | 73.6 | - | 12.8 | 41.0 | 28.4 | 31.4 | 63.1 | 40.9 | 53.4 |
| | IN-A | 66.5 | 54.1 | - | 35.4 | 24.0 | 28.0 | 53.5 | 31.9 | 47.5 |
| | IN-R | 64.7 | 52.8 | 11.0 | - | 43.4 | 26.5 | 57.2 | 47.0 | 47.0 |
| | IN-Sketch | 70.8 | 57.9 | 8.9 | 55.8 | - | 28.2 | 63.1 | 47.2 | 51.1 |
| | ObjNet | 68.5 | 55.9 | 12.7 | 33.7 | 23.3 | - | 52.4 | 29.1 | 47.8 |
| | IN-Cartoon | 82.7 | 58.0 | 11.2 | 39.8 | 28.3 | 28.6 | - | 41.8 | 50.1 |
| | IN-Drawing | 77.1 | 56.9 | 9.0 | 40.8 | 29.1 | 26.5 | 63.2 | - | 52.8 |
| | IN-C | 99.6 | 54.8 | 8.1 | 35.3 | 23.5 | 24.1 | 93.5 | 72.7 | - |
| Linear Probing | IN-V2 | 74.8 | - | 8.3 | 37.8 | 27.1 | 28.1 | 65.1 | 40.5 | 53.5 |
| | IN-A | 73.8 | 60.3 | - | 38.0 | 26.7 | 29.6 | 64.1 | 40.2 | 53.2 |
| | IN-R | 74.0 | 60.4 | 8.9 | - | 29.7 | 29.1 | 65.6 | 43.4 | 52.8 |
| | IN-Sketch | 72.3 | 58.9 | 8.5 | 47.6 | - | 28.8 | 64.8 | 43.4 | 51.0 |
| | ObjNet | 74.3 | 60.4 | 9.9 | 38.7 | 27.4 | - | 64.8 | 41.3 | 53.3 |
| | IN-Cartoon | 77.7 | 60.5 | 8.2 | 39.6 | 27.2 | 28.6 | - | 44.5 | 54.2 |
| | IN-Drawing | 75.7 | 59.8 | 8.5 | 41.7 | 28.3 | 28.3 | 67.1 | - | 54.4 |
| | IN-C | 97.9 | 54.3 | 7.9 | 35.2 | 23.2 | 21.5 | 88.5 | 63.6 | - |
| Visual Prompt (Bahng et al., 2022) | IN-V2 | 69.8 | - | 7.0 | 38.4 | 25.1 | 28.8 | 58.3 | 38.1 | 45.5 |
| | IN-A | 58.2 | 46.7 | - | 32.9 | 18.2 | 24.5 | 44.4 | 23.4 | 29.9 |
| | IN-R | 62.1 | 50.5 | 6.5 | - | 33.7 | 25.2 | 52.9 | 42.1 | 39.4 |
| | IN-Sketch | 66.1 | 53.4 | 5.8 | 48.8 | - | 26.5 | 57.1 | 45.0 | 43.1 |
| | ObjNet | 60.5 | 49.0 | 7.1 | 29.8 | 17.6 | - | 44.0 | 23.7 | 33.1 |
| | IN-Cartoon | 70.6 | 56.7 | 6.5 | 40.1 | 26.3 | 27.0 | - | 39.6 | 42.4 |
| | IN-Drawing | 66.2 | 53.2 | 4.8 | 39.5 | 26.0 | 23.9 | 55.6 | - | 42.3 |
| | IN-C | 72.1 | 58.8 | 8.0 | 37.6 | 25.8 | 29.1 | 61.1 | 46.3 | - |
| LoRA (Hu et al., 2021) | IN-V2 | 75.0 | - | 8.1 | 37.9 | 27.3 | 28.5 | 65.4 | 40.9 | 53.9 |
| | IN-A | 74.8 | 61.0 | - | 38.7 | 27.1 | 30.0 | 64.5 | 40.6 | 53.4 |
| | IN-R | 73.8 | 60.1 | 8.7 | - | 28.6 | 29.1 | 65.9 | 43.5 | 50.6 |
| | IN-Sketch | 73.9 | 60.1 | 8.1 | 42.2 | - | 28.9 | 66.3 | 43.7 | 51.6 |
| | ObjNet | 74.8 | 61.2 | 9.9 | 39.2 | 27.5 | - | 65.1 | 41.6 | 53.4 |
| | IN-Cartoon | 74.7 | 60.8 | 7.9 | 39.6 | 27.7 | 29.0 | - | 43.1 | 52.6 |
| | IN-Drawing | 73.6 | 60.6 | 8.5 | 41.7 | 28.0 | 28.5 | 64.4 | - | 52.1 |
| | IN-C | 75.6 | 61.9 | 9.9 | 41.1 | 28.7 | 31.2 | 66.0 | 50.6 | - |
| EWC (Kirkpatrick et al., 2017) | IN-V2 | 75.4 | - | 9.8 | 40.2 | 28.1 | 30.8 | 65.7 | 43.0 | 55.3 |
| | IN-A | 69.7 | 56.9 | - | 38.8 | 25.9 | 28.8 | 58.1 | 34.4 | 49.9 |
| | IN-R | 71.2 | 58.4 | 10.9 | - | 39.7 | 29.4 | 63.9 | 51.2 | 52.5 |
| | IN-Sketch | 73.7 | 60.2 | 8.7 | 48.8 | - | 28.5 | 65.2 | 46.6 | 53.8 |
| | ObjNet | 74.1 | 61.0 | 11.2 | 38.1 | 26.9 | - | 62.4 | 39.7 | 54.0 |
| | IN-Cartoon | 75.0 | 60.9 | 8.6 | 41.2 | 28.5 | 28.8 | - | 43.2 | 52.8 |
| | IN-Drawing | 74.3 | 60.6 | 9.0 | 43.9 | 31.0 | 29.5 | 65.5 | - | 55.1 |
| | IN-C | 75.5 | 62.0 | 9.7 | 40.5 | 28.4 | 31.3 | 65.4 | 48.3 | - |
| LwF (Li & Hoiem, 2017) | IN-V2 | 74.3 | - | 11.9 | 40.6 | 28.2 | 30.7 | 64.1 | 41.0 | 54.0 |
| | IN-A | 71.1 | 58.5 | - | 37.2 | 25.5 | 29.8 | 59.3 | 36.8 | 52.0 |
| | IN-R | 71.2 | 58.5 | 11.2 | - | 44.7 | 28.8 | 63.4 | 51.1 | 53.0 |
| | IN-Sketch | 72.3 | 59.1 | 8.7 | 55.3 | - | 28.6 | 64.7 | 47.2 | 52.5 |
| | ObjNet | 73.1 | 59.5 | 12.2 | 36.5 | 25.2 | - | 59.8 | 34.6 | 52.1 |
| | IN-Cartoon | 84.6 | 59.9 | 10.7 | 39.9 | 28.3 | 29.2 | - | 43.0 | 54.1 |
| | IN-Drawing | 79.3 | 58.9 | 9.3 | 41.3 | 29.6 | 27.5 | 66.6 | - | 55.2 |
| | IN-C | 99.0 | 59.2 | 7.6 | 38.7 | 26.7 | 26.4 | 92.3 | 64.7 | - |
| LP-FT (Kumar et al., 2022) | IN-V2 | 73.8 | - | 12.7 | 40.7 | 28.1 | 31.1 | 63.7 | 40.6 | 53.7 |
| | IN-A | 71.7 | 58.9 | - | 36.8 | 25.9 | 30.1 | 60.5 | 36.7 | 52.2 |
| | IN-R | 70.4 | 57.9 | 11.6 | - | 43.6 | 29.8 | 62.4 | 50.1 | 51.4 |
| | IN-Sketch | 71.1 | 58.2 | 9.0 | 56.3 | - | 28.5 | 63.8 | 47.3 | 51.1 |
| | ObjNet | 72.3 | 58.9 | 12.7 | 36.8 | 25.7 | - | 59.4 | 34.9 | 51.3 |
| | IN-Cartoon | 84.7 | 58.7 | 11.0 | 40.3 | 28.2 | 28.5 | - | 43.7 | 52.6 |
| | IN-Drawing | 78.1 | 57.1 | 8.6 | 41.9 | 30.1 | 26.1 | 64.8 | - | 53.6 |
| | IN-C | 99.9 | 51.8 | 7.2 | 35.1 | 22.7 | 20.0 | 96.7 | 78.4 | - |
| WiSE-FT (Wortsman et al., 2022b) | IN-V2 | 75.1 | - | 10.9 | 39.7 | 28.1 | 30.5 | 65.2 | 41.6 | 54.7 |
| | IN-A | 74.0 | 60.9 | - | 38.6 | 27.3 | 31.2 | 63.3 | 38.7 | 54.3 |
| | IN-R | 74.2 | 61.2 | 10.9 | - | 41.0 | 30.9 | 66.7 | 52.0 | 55.4 |
| | IN-Sketch | 74.3 | 61.0 | 9.0 | 49.7 | - | 29.6 | 66.5 | 46.9 | 54.4 |
| | ObjNet | 74.7 | 61.4 | 11.6 | 38.3 | 27.2 | - | 63.3 | 38.8 | 54.3 |
| | IN-Cartoon | 80.3 | 61.1 | 10.4 | 40.0 | 28.5 | 29.8 | - | 42.9 | 54.2 |
| | IN-Drawing | 78.4 | 61.0 | 9.7 | 41.3 | 30.3 | 29.5 | 67.2 | - | 56.9 |
| | IN-C | 92.0 | 60.8 | 10.2 | 40.0 | 28.1 | 29.4 | 83.0 | 58.8 | - |
| Model Soup PRE-FT-EWC-LwF (Wortsman et al., 2022a) | IN-V2 | 75.2 | - | 10.8 | 40.0 | 28.2 | 30.6 | 65.3 | 42.0 | 54.9 |
| | IN-A | 73.1 | 60.0 | - | 38.4 | 26.8 | 31.1 | 62.0 | 37.2 | 53.6 |
| | IN-R | 74.1 | 61.1 | 11.2 | - | 41.8 | 30.7 | 66.6 | 52.5 | 55.3 |
| | IN-Sketch | 74.4 | 60.9 | 8.9 | 51.0 | - | 29.5 | 66.4 | 47.2 | 54.4 |
| | ObjNet | 74.6 | 61.4 | 11.9 | 38.0 | 27.0 | - | 62.9 | 38.5 | 54.3 |
| | IN-Cartoon | 80.2 | 61.2 | 10.1 | 40.6 | 28.8 | 29.8 | - | 43.4 | 54.3 |
| | IN-Drawing | 78.2 | 61.0 | 9.6 | 42.2 | 30.9 | 29.5 | 67.3 | - | 56.9 |
| | IN-C | 91.0 | 61.6 | 9.9 | 40.4 | 28.6 | 29.7 | 81.6 | 58.0 | - |

Table 25: The accuracy on each OOD dataset after fine-tuning ImageNet-21K pre-trained ViT-B/32 with AugReg on the downstream datasets with various methods. Note that ImageNet-Drawing, ImageNet-Cartoon, and ImageNet-C are generated from the ImageNet validation set. Green and red indicate relative performance increases and decreases, respectively, compared to the pre-trained model. Bold indicates the best performance on each evaluation dataset.

| Method | Downstream Dataset | $D_{pre}$ IN | Realistic OOD | | | | | Synthetic OOD | | |
|---|---|---|---|---|---|---|---|---|---|---|
| | | | IN-V2 | IN-A | IN-R | IN-Sketch | ObjNet | IN-Cartoon | IN-Drawing | IN-C |
| Pre-Trained | | 80.7 | 69.0 | 22.4 | 49.3 | 37.1 | 40.7 | 70.6 | 42.5 | 60.5 |
| FT | IN-V2 | 78.8 | - | 31.5 | 50.5 | 36.1 | 42.9 | 67.0 | 39.5 | 60.2 |
| | IN-A | 75.0 | 64.1 | - | 47.6 | 34.4 | 42.5 | 60.8 | 36.1 | 56.9 |
| | IN-R | 72.2 | 62.0 | 25.0 | - | 51.3 | 38.3 | 63.7 | 52.8 | 55.3 |
| | IN-Sketch | 76.0 | 65.2 | 23.6 | 65.3 | - | 40.1 | 68.1 | 54.4 | 57.0 |
| | ObjNet | 74.9 | 63.5 | 29.1 | 44.3 | 32.1 | - | 57.4 | 35.5 | 55.3 |
| | IN-Cartoon | 88.0 | 65.3 | 25.1 | 50.4 | 36.5 | 39.5 | - | 47.4 | 56.7 |
| | IN-Drawing | 84.3 | 64.1 | 21.7 | 51.5 | 39.3 | 38.2 | 71.1 | - | 59.5 |
| | IN-C | 99.8 | 62.2 | 18.7 | 45.5 | 32.0 | 34.1 | 94.3 | 75.5 | - |
| Linear Probing | IN-V2 | 80.3 | - | 23.7 | 49.5 | 36.7 | 40.3 | 70.1 | 41.9 | 60.1 |
| | IN-A | 79.3 | 67.7 | - | 49.1 | 36.4 | 40.9 | 69.2 | 41.5 | 59.4 |
| | IN-R | 79.7 | 68.6 | 23.9 | - | 39.4 | 39.9 | 69.4 | 42.7 | 59.7 |
| | IN-Sketch | 77.9 | 66.6 | 23.8 | 57.0 | - | 40.6 | 68.3 | 45.1 | 58.4 |
| | ObjNet | 79.7 | 68.3 | 24.4 | 49.5 | 36.9 | - | 69.5 | 42.3 | 60.3 |
| | IN-Cartoon | 85.6 | 67.8 | 22.0 | 50.1 | 36.7 | 40.2 | - | 46.8 | 62.6 |
| | IN-Drawing | 82.6 | 68.1 | 23.3 | 51.1 | 38.4 | 40.4 | 73.5 | - | 63.0 |
| | IN-C | 98.3 | 59.2 | 20.1 | 43.4 | 30.7 | 31.7 | 89.5 | 62.0 | - |
| Visual Prompt (Bahng et al., 2022) | IN-V2 | 76.1 | - | 18.9 | 46.6 | 33.2 | 40.5 | 62.6 | 35.7 | 51.1 |
| | IN-A | 67.6 | 55.5 | - | 40.6 | 25.7 | 38.7 | 50.2 | 27.3 | 39.7 |
| | IN-R | 67.2 | 56.8 | 13.9 | - | 42.5 | 33.4 | 56.2 | 42.5 | 42.6 |
| | IN-Sketch | 72.2 | 60.3 | 14.0 | 57.2 | - | 37.0 | 61.3 | 44.2 | 46.2 |
| | ObjNet | 62.6 | 51.1 | 13.1 | 32.0 | 20.7 | - | 40.8 | 18.0 | 30.9 |
| | IN-Cartoon | 75.4 | 62.8 | 15.4 | 46.7 | 32.4 | 37.9 | - | 37.5 | 45.5 |
| | IN-Drawing | 73.1 | 61.0 | 12.6 | 48.3 | 34.6 | 34.1 | 61.3 | - | 46.7 |
| | IN-C | 77.3 | 65.2 | 20.7 | 45.9 | 31.8 | 41.1 | 64.3 | 45.6 | - |
| LoRA (Hu et al., 2021) | IN-V2 | 80.7 | - | 22.5 | 49.2 | 37.1 | 40.7 | 70.6 | 42.5 | 60.5 |
| | IN-A | 80.7 | 69.4 | - | 49.9 | 37.2 | 42.4 | 70.3 | 43.0 | 61.2 |
| | IN-R | 80.7 | 69.3 | 24.9 | - | 37.6 | 42.3 | 70.6 | 43.9 | 60.9 |
| | IN-Sketch | 80.6 | 69.1 | 22.9 | 52.7 | - | 41.2 | 70.9 | 46.0 | 60.8 |
| | ObjNet | 80.7 | 69.3 | 25.4 | 50.1 | 37.1 | - | 70.0 | 43.2 | 61.3 |
| | IN-Cartoon | 80.7 | 69.1 | 22.3 | 49.8 | 37.0 | 40.8 | - | 42.9 | 60.2 |
| | IN-Drawing | 80.3 | 69.3 | 23.8 | 53.2 | 40.4 | 41.5 | 70.7 | - | 61.0 |
| | IN-C | 80.0 | 68.8 | 24.7 | 51.5 | 37.9 | 41.5 | 68.3 | 53.1 | - |
| EWC (Kirkpatrick et al., 2017) | IN-V2 | 81.0 | - | 28.0 | 51.2 | 37.6 | 44.0 | 70.2 | 43.4 | 62.1 |
| | IN-A | 78.8 | 67.7 | - | 50.7 | 36.4 | 45.2 | 66.2 | 37.8 | 60.9 |
| | IN-R | 78.7 | 67.6 | 26.5 | - | 49.2 | 40.8 | 69.9 | 56.8 | 60.0 |
| | IN-Sketch | 80.2 | 68.8 | 24.1 | 60.8 | - | 41.7 | 71.9 | 54.0 | 61.1 |
| | ObjNet | 77.9 | 67.3 | 30.7 | 48.0 | 36.2 | - | 62.2 | 39.9 | 60.1 |
| | IN-Cartoon | 80.4 | 68.1 | 23.5 | 52.1 | 37.9 | 41.1 | - | 46.6 | 59.0 |
| | IN-Drawing | 80.3 | 68.8 | 23.1 | 53.5 | 41.6 | 41.5 | 71.9 | - | 61.2 |
| | IN-C | 81.0 | 69.5 | 26.6 | 51.0 | 38.3 | 43.4 | 69.6 | 49.1 | - |
| LwF (Li & Hoiem, 2017) | IN-V2 | 80.1 | - | 28.0 | 50.6 | 36.9 | 42.3 | 69.4 | 41.4 | 61.1 |
| | IN-A | 79.3 | 68.2 | - | 49.7 | 36.2 | 42.5 | 67.7 | 41.4 | 60.6 |
| | IN-R | 79.2 | 67.8 | 25.9 | - | 50.3 | 39.8 | 70.9 | 54.6 | 60.9 |
| | IN-Sketch | 78.1 | 67.1 | 23.4 | 63.2 | - | 40.6 | 70.1 | 51.1 | 59.1 |
| | ObjNet | 78.5 | 67.2 | 30.8 | 47.5 | 34.7 | - | 64.8 | 38.4 | 59.1 |
| | IN-Cartoon | 89.8 | 68.3 | 25.2 | 51.1 | 37.3 | 40.9 | - | 47.3 | 63.0 |
| | IN-Drawing | 86.3 | 67.0 | 22.7 | 51.9 | 39.9 | 39.4 | 73.8 | - | 63.1 |
| | IN-C | 99.4 | 66.5 | 20.0 | 49.4 | 35.9 | 38.2 | 93.1 | 63.7 | - |
| LP-FT (Kumar et al., 2022) | IN-V2 | 79.4 | - | 30.3 | 50.9 | 36.7 | 42.6 | 68.3 | 41.2 | 60.7 |
| | IN-A | 77.8 | 67.0 | - | 48.7 | 36.0 | 43.3 | 65.8 | 40.0 | 59.5 |
| | IN-R | 77.3 | 66.5 | 26.4 | - | 50.5 | 40.4 | 68.0 | 54.7 | 59.0 |
| | IN-Sketch | 76.3 | 65.3 | 23.2 | 64.2 | - | 40.4 | 68.3 | 50.5 | 57.5 |
| | ObjNet | 78.0 | 67.2 | 30.2 | 48.3 | 34.8 | - | 65.4 | 39.9 | 58.9 |
| | IN-Cartoon | 90.5 | 66.5 | 26.0 | 51.2 | 37.1 | 40.2 | - | 49.6 | 61.3 |
| | IN-Drawing | 86.3 | 65.4 | 22.9 | 51.9 | 39.8 | 38.8 | 73.8 | - | 62.6 |
| | IN-C | 99.9 | 57.5 | 15.0 | 43.1 | 29.8 | 29.7 | 96.7 | 80.4 | - |
| WiSE-FT (Wortsman et al., 2022b) | IN-V2 | 80.7 | - | 28.7 | 51.0 | 37.8 | 43.3 | 70.1 | 42.6 | 61.7 |
| | IN-A | 80.1 | 69.0 | - | 50.9 | 37.7 | 45.1 | 69.0 | 42.5 | 61.9 |
| | IN-R | 80.0 | 69.1 | 28.7 | - | 49.8 | 43.4 | 71.9 | 56.8 | 62.4 |
| | IN-Sketch | 79.9 | 69.0 | 24.5 | 61.4 | - | 42.1 | 71.9 | 53.1 | 61.0 |
| | ObjNet | 80.1 | 69.0 | 29.8 | 49.8 | 37.3 | - | 68.2 | 43.1 | 61.2 |
| | IN-Cartoon | 86.0 | 69.0 | 25.9 | 51.5 | 38.2 | 41.7 | - | 48.2 | 60.9 |
| | IN-Drawing | 84.4 | 68.8 | 25.1 | 52.6 | 41.3 | 41.7 | 73.6 | - | 63.6 |
| | IN-C | 95.3 | 68.6 | 24.9 | 51.0 | 37.8 | 41.3 | 87.5 | 62.2 | - |
| Model Soup PRE-FT-EWC-LwF (Wortsman et al., 2022a) | IN-V2 | 80.7 | - | 28.3 | 51.1 | 37.7 | 43.3 | 70.1 | 42.8 | 61.7 |
| | IN-A | 79.9 | 68.9 | - | 50.7 | 37.3 | 44.8 | 68.5 | 41.8 | 61.6 |
| | IN-R | 79.9 | 69.1 | 28.6 | - | 50.4 | 42.6 | 71.9 | 57.3 | 62.2 |
| | IN-Sketch | 79.8 | 68.9 | 24.1 | 62.5 | - | 41.9 | 71.9 | 53.6 | 61.0 |
| | ObjNet | 79.7 | 68.6 | 30.7 | 49.2 | 36.9 | - | 67.2 | 42.1 | 60.9 |
| | IN-Cartoon | 85.7 | 68.8 | 25.4 | 51.9 | 38.2 | 41.5 | - | 48.1 | 61.1 |
| | IN-Drawing | 84.3 | 68.5 | 24.7 | 52.9 | 41.5 | 41.4 | 73.7 | - | 63.3 |
| | IN-C | 94.1 | 69.0 | 24.9 | 51.5 | 38.3 | 41.5 | 85.7 | 60.0 | - |

Table 26: The accuracy on each OOD datasets after fine-tuning LAION-2B pre-trained ViT-B/32 on the downstream datasets with various methods. Note that ImageNet-Drawing, ImageNet-Cartoon, and ImageNet-C are generated from the ImageNet validation set. Green and red indicate relative performance increases and decreases, respectively, compared to the pre-trained model. Bold indicates the best performance on each evaluation dataset.

| Method | Downstream Dataset | $D_{pre}$ IN | Realistic OOD | | | | | Synthetic OOD | | |
| | | | IN-V2 | IN-A | IN-R | IN-Sketch | ObjNet | IN-Cartoon | IN-Drawing | IN-C |
|---|---|---|---|---|---|---|---|---|---|---|
| Pre-Trained | | 82.6 | **71.6** | 22.8 | 59.2 | 49.1 | 43.5 | 73.0 | 42.3 | 57.5 |
| FT | IN-V2 | 50.6 | - | 5.1 | 24.7 | 11.4 | 16.9 | 34.5 | 15.1 | 21.7 |
| | IN-A | 21.2 | 18.6 | - | 10.2 | 4.1 | 10.1 | 12.8 | 4.1 | 7.0 |
| | IN-R | 42.0 | 32.5 | 4.5 | - | 31.9 | 16.7 | 31.6 | 19.4 | 20.9 |
| | IN-Sketch | 24.7 | 20.2 | 1.6 | 37.5 | - | 8.8 | 19.3 | 10.3 | 9.7 |
| | ObjNet | 31.9 | 23.7 | 3.6 | 15.0 | 6.9 | - | 18.6 | 6.2 | 12.2 |
| | IN-Cartoon | 85.0 | 42.6 | 5.8 | 30.6 | 17.1 | 18.6 | - | 32.0 | 27.3 |
| | IN-Drawing | 52.2 | 29.3 | 3.0 | 25.6 | 15.3 | 10.6 | 41.8 | - | 21.6 |
| | IN-C | 99.8 | 44.1 | 4.7 | 31.7 | 18.4 | 16.8 | 97.3 | 81.2 | - |
| Linear Probing | IN-V2 | 82.4 | - | 24.2 | 58.7 | 46.8 | 44.3 | 72.1 | 42.5 | 58.1 |
| | IN-A | 81.2 | 70.6 | - | 59.0 | 45.7 | **46.5** | 70.6 | 35.2 | **58.2** |
| | IN-R | 79.3 | 68.0 | 21.2 | - | 52.6 | 40.8 | 71.1 | 51.1 | 56.1 |
| | IN-Sketch | 79.5 | 68.0 | 21.2 | **66.8** | - | 42.0 | 70.8 | 49.1 | 55.6 |
| | ObjNet | 81.0 | 69.7 | **26.7** | 56.9 | 43.7 | - | 67.8 | 38.8 | 56.6 |
| | IN-Cartoon | 81.3 | 68.2 | 19.5 | 59.7 | 45.7 | 39.7 | - | 45.6 | 51.2 |
| | IN-Drawing | 78.0 | 64.5 | 13.4 | 58.0 | 44.4 | 35.5 | 68.7 | - | 49.8 |
| | IN-C | 96.0 | 64.2 | 15.7 | 53.3 | 41.0 | 33.9 | 88.4 | 68.5 | - |
| Visual Prompt (Bahng et al., 2022) | IN-V2 | 78.7 | - | 16.4 | 54.2 | 44.3 | 41.5 | 65.1 | 33.2 | 47.9 |
| | IN-A | 73.9 | 62.1 | - | 52.1 | 36.7 | 42.2 | 57.7 | 23.7 | 36.9 |
| | IN-R | 74.3 | 62.5 | 15.0 | - | 48.7 | 38.4 | 62.1 | 41.0 | 43.9 |
| | IN-Sketch | 76.2 | 63.9 | 13.5 | 60.7 | - | 39.0 | 64.5 | 41.3 | 45.1 |
| | ObjNet | 68.3 | 55.9 | 14.6 | 34.1 | 28.6 | - | 47.4 | 13.9 | 27.7 |
| | IN-Cartoon | 78.4 | 65.5 | 14.6 | 55.9 | 43.9 | 38.9 | - | 39.2 | 44.9 |
| | IN-Drawing | 77.6 | 65.3 | 13.3 | 56.5 | 44.8 | 38.1 | 64.7 | - | 47.2 |
| | IN-C | 80.0 | 68.2 | 18.6 | 57.9 | 46.0 | 41.7 | 68.8 | 50.0 | - |
| LoRA (Hu et al., 2021) | IN-V2 | 82.3 | - | 23.7 | 58.6 | 46.5 | 44.4 | 72.2 | 42.8 | 58.1 |
| | IN-A | 81.0 | 70.2 | - | 58.8 | 45.2 | 46.1 | 70.3 | 33.8 | 57.9 |
| | IN-R | 78.8 | 67.2 | 20.3 | - | 52.2 | 40.2 | 71.1 | 50.9 | 54.8 |
| | IN-Sketch | 78.6 | 66.8 | 17.6 | 64.9 | - | 39.4 | 70.1 | 51.1 | 53.4 |
| | ObjNet | 80.3 | 68.9 | 24.0 | 55.6 | 42.2 | - | 65.4 | 36.4 | 54.9 |
| | IN-Cartoon | 79.9 | 67.8 | 19.2 | 59.4 | 45.6 | 39.7 | - | 45.4 | 50.3 |
| | IN-Drawing | 76.7 | 64.3 | 13.6 | 57.9 | 44.4 | 35.9 | 67.7 | - | 49.8 |
| | IN-C | 83.1 | 70.4 | 23.8 | 58.9 | 46.7 | 41.9 | 73.1 | 53.9 | - |
| EWC (Kirkpatrick et al., 2017) | IN-V2 | 76.5 | - | 16.1 | 46.3 | 31.0 | 35.6 | 61.4 | 34.0 | 48.4 |
| | IN-A | 60.0 | 48.3 | - | 31.6 | 18.8 | 28.5 | 40.6 | 18.0 | 30.7 |
| | IN-R | 76.1 | 65.1 | 21.5 | - | 53.9 | 39.1 | 67.5 | 51.0 | 53.0 |
| | IN-Sketch | 66.6 | 54.9 | 12.8 | 55.4 | - | 26.8 | 60.0 | 40.1 | 39.7 |
| | ObjNet | 72.9 | 60.1 | 18.5 | 41.8 | 29.0 | - | 50.5 | 26.3 | 43.0 |
| | IN-Cartoon | 81.0 | 67.1 | 19.6 | 57.7 | 43.7 | 39.1 | - | 46.1 | 48.1 |
| | IN-Drawing | 72.6 | 59.0 | 12.2 | 53.4 | 41.0 | 28.8 | 62.0 | - | 43.6 |
| | IN-C | 81.2 | 68.3 | 18.6 | 56.9 | 43.2 | 40.3 | 70.5 | 55.9 | - |
| LwF (Li & Hoiem, 2017) | IN-V2 | 56.9 | - | 5.4 | 26.5 | 13.2 | 19.6 | 40.2 | 16.8 | 26.1 |
| | IN-A | 37.9 | 29.4 | - | 16.6 | 7.2 | 13.2 | 23.7 | 8.3 | 14.5 |
| | IN-R | 54.5 | 42.8 | 5.2 | - | 36.0 | 20.0 | 41.9 | 25.5 | 28.9 |
| | IN-Sketch | 33.3 | 26.0 | 2.4 | 40.9 | - | 11.3 | 26.4 | 13.4 | 13.7 |
| | ObjNet | 49.0 | 37.4 | 3.8 | 20.3 | 10.1 | - | 30.0 | 12.3 | 20.7 |
| | IN-Cartoon | 89.8 | 51.9 | 6.9 | 36.2 | 22.2 | 23.5 | - | 39.0 | 36.6 |
| | IN-Drawing | 63.9 | 37.5 | 3.7 | 32.1 | 20.4 | 14.9 | 52.0 | - | 28.2 |
| | IN-C | 99.6 | 49.5 | 4.6 | 35.3 | 21.0 | 19.8 | 96.1 | 75.1 | - |
| LP-FT (Kumar et al., 2022) | IN-V2 | 50.4 | - | 5.7 | 24.1 | 11.3 | 16.2 | 34.6 | 14.2 | 22.0 |
| | IN-A | 25.0 | 19.7 | - | 11.5 | 5.2 | 10.4 | 15.0 | 5.1 | 9.0 |
| | IN-R | 44.2 | 34.3 | 4.5 | - | 32.8 | 17.6 | 33.5 | 19.9 | 22.2 |
| | IN-Sketch | 28.5 | 22.4 | 2.1 | 39.3 | - | 9.9 | 21.9 | 11.3 | 11.5 |
| | ObjNet | 35.7 | 27.3 | 4.0 | 16.1 | 7.7 | - | 20.6 | 7.6 | 14.4 |
| | IN-Cartoon | 85.9 | 43.5 | 5.4 | 30.8 | 17.1 | 18.3 | - | 31.1 | 28.6 |
| | IN-Drawing | 57.3 | 31.6 | 3.1 | 27.2 | 17.0 | 12.2 | 46.9 | - | 22.7 |
| | IN-C | 99.9 | 42.0 | 4.2 | 29.8 | 17.1 | 14.1 | 98.2 | 83.4 | - |
| WiSE-FT (Wortsman et al., 2022b) | IN-V2 | 72.8 | - | 12.8 | 40.0 | 25.0 | 31.1 | 54.9 | 31.4 | 43.5 |
| | IN-A | 65.3 | 53.5 | - | 32.9 | 19.7 | 28.0 | 44.5 | 20.8 | 37.0 |
| | IN-R | 72.3 | 60.4 | 12.9 | - | 44.1 | 33.1 | 59.5 | 42.8 | 46.5 |
| | IN-Sketch | 60.9 | 49.6 | 7.4 | 52.0 | - | 23.4 | 49.3 | 30.3 | 33.3 |
| | ObjNet | 70.7 | 57.0 | 13.5 | 37.2 | 22.7 | - | 49.7 | 28.0 | 41.6 |
| | IN-Cartoon | 84.8 | 63.5 | 14.4 | 47.4 | 32.0 | 33.4 | - | 47.6 | 48.8 |
| | IN-Drawing | 75.5 | 55.3 | 9.5 | 45.7 | 32.6 | 25.1 | 63.2 | - | 44.7 |
| | IN-C | 91.7 | 61.2 | 10.5 | 51.0 | 36.6 | 30.6 | 86.4 | 69.8 | - |
| Model Soup PRE-FT-EWC-LwF (Wortsman et al., 2022a) | IN-V2 | 74.7 | - | 14.1 | 43.6 | 28.6 | 33.3 | 57.5 | 37.0 | 46.5 |
| | IN-A | 68.8 | 56.3 | - | 36.5 | 22.9 | 31.2 | 48.6 | 26.8 | 40.5 |
| | IN-R | 74.5 | 63.0 | 14.9 | - | 47.9 | 35.1 | 62.2 | 47.3 | 49.8 |
| | IN-Sketch | 61.6 | 50.4 | 7.8 | 53.6 | - | 23.7 | 51.1 | 34.0 | 34.7 |
| | ObjNet | 73.8 | 61.2 | 14.6 | 41.1 | 26.8 | - | 53.5 | 33.0 | 45.1 |
| | IN-Cartoon | 85.3 | 65.5 | 15.3 | 50.9 | 36.3 | 35.1 | - | 51.9 | 50.9 |
| | IN-Drawing | 76.0 | 56.6 | 9.7 | 49.1 | 37.0 | 26.4 | 63.9 | - | 46.2 |
| | IN-C | 91.1 | 62.8 | 11.1 | 53.0 | 38.3 | 31.6 | 85.4 | 69.9 | - |

Table 27: The accuracy on each OOD dataset after fine-tuning OpenAI CLIP ViT-B/32 on the downstream datasets with various methods. Note that ImageNet-Drawing, ImageNet-Cartoon, and ImageNet-C are generated from the ImageNet validation set. Green and red indicate relative performance increases and decreases, respectively, compared to the pre-trained model. Bold indicates the best performance on each evaluation dataset.

| Method | Downstream Dataset | $D_{pre}$ IN | Realistic OOD | | | | | Synthetic OOD | | |
|---|---|---|---|---|---|---|---|---|---|---|
| | | | IN-V2 | IN-A | IN-R | IN-Sketch | ObjNet | IN-Cartoon | IN-Drawing | IN-C |
| Pre-Trained | | 82.0 | **70.9** | 22.6 | 55.8 | 45.0 | 41.5 | 71.1 | 42.5 | 57.9 |
| FT | IN-V2 | 54.8 | - | 6.5 | 24.5 | 10.4 | 18.4 | 38.3 | 16.2 | 25.9 |
| | IN-A | 26.3 | 21.0 | - | 11.0 | 3.9 | 12.3 | 14.8 | 5.1 | 10.1 |
| | IN-R | 44.0 | 33.9 | 5.7 | - | 32.1 | 16.4 | 34.0 | 21.9 | 23.1 |
| | IN-Sketch | 28.0 | 21.1 | 2.2 | 39.9 | - | 9.1 | 21.8 | 12.6 | 11.1 |
| | ObjNet | 33.8 | 25.0 | 3.9 | 15.0 | 8.6 | - | 19.6 | 6.8 | 13.4 |
| | IN-Cartoon | 88.7 | 47.2 | 6.7 | 31.9 | 17.6 | 19.3 | - | 36.8 | 31.4 |
| | IN-Drawing | 60.3 | 32.9 | 3.5 | 28.0 | 16.7 | 12.4 | 49.4 | - | 24.6 |
| | IN-C | 99.8 | 44.5 | 4.5 | 30.2 | 17.2 | 16.2 | 97.1 | 80.7 | - |
| HeadOnly | IN-V2 | 81.6 | - | 23.5 | 56.1 | 43.4 | 42.4 | 71.1 | 42.4 | **59.0** |
| | IN-A | 80.6 | 69.9 | - | 55.9 | 42.2 | **44.5** | 69.8 | 37.8 | 58.5 |
| | IN-R | 77.7 | 66.3 | 18.2 | - | 48.1 | 37.5 | 69.8 | 50.8 | 55.5 |
| | IN-Sketch | 78.0 | 66.3 | 19.1 | **63.5** | - | 39.2 | 69.7 | 48.6 | 55.4 |
| | ObjNet | 80.3 | 69.4 | **26.6** | 54.6 | 41.2 | - | 68.2 | 36.9 | 56.7 |
| | IN-Cartoon | 80.8 | 67.5 | 20.0 | 57.1 | 42.4 | 37.9 | - | 43.8 | 53.1 |
| | IN-Drawing | 77.4 | 64.0 | 14.9 | 55.6 | 41.7 | 34.1 | 66.8 | - | 48.6 |
| | IN-C | 95.8 | 62.9 | 16.4 | 50.8 | 38.7 | 31.7 | 87.7 | 67.6 | - |
| Visual Prompt (Bahng et al., 2022) | IN-V2 | 78.5 | - | 16.3 | 52.1 | 41.2 | 40.0 | 64.3 | 35.1 | 50.1 |
| | IN-A | 72.9 | 61.3 | - | 47.3 | 31.3 | 40.9 | 55.7 | 21.4 | 39.6 |
| | IN-R | 73.9 | 62.7 | 15.5 | - | 44.8 | 37.3 | 61.8 | 41.7 | 44.2 |
| | IN-Sketch | 76.4 | 64.3 | 14.5 | 58.0 | - | 38.7 | 64.5 | 43.0 | 47.2 |
| | ObjNet | 65.9 | 53.5 | 14.8 | 41.9 | 24.9 | - | 41.4 | 14.5 | 29.5 |
| | IN-Cartoon | 77.7 | 65.3 | 16.2 | 53.3 | 39.7 | 37.7 | - | 37.5 | 45.4 |
| | IN-Drawing | 77.0 | 64.6 | 12.8 | 53.1 | 39.7 | 36.2 | 63.2 | - | 46.1 |
| | IN-C | 79.2 | 67.8 | 20.6 | 55.5 | 41.9 | 41.0 | 66.8 | 46.3 | - |
| LoRA (Hu et al., 2021) | IN-V2 | 81.5 | - | 23.2 | 56.1 | 43.5 | 42.3 | 71.1 | 42.4 | 58.9 |
| | IN-A | 80.3 | 69.6 | - | 55.5 | 42.1 | 43.8 | 69.0 | 35.9 | 58.1 |
| | IN-R | 77.5 | 66.0 | 18.4 | - | 48.0 | 37.5 | 69.5 | 51.8 | 55.1 |
| | IN-Sketch | 77.8 | 65.7 | 18.5 | 61.9 | - | 38.4 | 68.9 | 51.4 | 53.5 |
| | ObjNet | 79.3 | 67.8 | 23.3 | 54.1 | 39.9 | - | 66.8 | 34.6 | 53.4 |
| | IN-Cartoon | 79.5 | 67.4 | 19.6 | 56.8 | 42.4 | 38.1 | - | 43.7 | 52.0 |
| | IN-Drawing | 76.3 | 64.6 | 15.5 | 55.6 | 42.1 | 34.7 | 66.1 | - | 48.4 |
| | IN-C | 81.1 | 69.5 | 21.8 | 57.7 | 44.5 | 38.6 | 73.4 | 43.0 | - |
| EWC (Kirkpatrick et al., 2017) | IN-V2 | 78.2 | - | 20.3 | 49.0 | 33.0 | 37.2 | 65.5 | 38.8 | 53.9 |
| | IN-A | 67.1 | 55.4 | - | 38.3 | 24.5 | 31.9 | 49.3 | 25.7 | 40.9 |
| | IN-R | 75.2 | 63.8 | 21.7 | - | 51.2 | 36.7 | 66.9 | 51.2 | 52.5 |
| | IN-Sketch | 64.9 | 53.1 | 13.4 | 52.7 | - | 24.9 | 59.3 | 40.3 | 38.8 |
| | ObjNet | 73.1 | 61.2 | 20.9 | 43.0 | 29.5 | - | 52.9 | 23.3 | 44.7 |
| | IN-Cartoon | 80.8 | 67.2 | 20.3 | 55.4 | 41.0 | 37.9 | - | 44.7 | 48.1 |
| | IN-Drawing | 76.8 | 63.3 | 14.9 | 53.8 | 40.6 | 31.6 | 66.6 | - | 48.7 |
| | IN-C | 82.9 | 70.0 | 23.6 | 56.0 | 42.4 | 40.5 | 72.3 | 56.6 | - |
| LwF (Li & Hoiem, 2017) | IN-V2 | 60.5 | - | 6.8 | 27.8 | 13.9 | 20.0 | 44.4 | 20.2 | 30.2 |
| | IN-A | 42.3 | 33.0 | - | 16.9 | 7.1 | 14.8 | 26.4 | 10.1 | 17.6 |
| | IN-R | 55.1 | 43.5 | 5.4 | - | 34.5 | 19.0 | 43.3 | 28.2 | 30.1 |
| | IN-Sketch | 36.1 | 27.6 | 2.4 | 43.3 | - | 10.8 | 27.8 | 16.0 | 15.3 |
| | ObjNet | 52.1 | 40.1 | 4.9 | 20.0 | 10.7 | - | 32.3 | 12.9 | 23.1 |
| | IN-Cartoon | 91.7 | 53.0 | 6.8 | 35.3 | 21.1 | 22.0 | - | 43.0 | 39.6 |
| | IN-Drawing | 67.6 | 38.8 | 3.7 | 32.6 | 19.5 | 14.8 | 56.0 | - | 29.6 |
| | IN-C | 98.7 | 59.1 | 5.9 | 42.7 | 29.3 | 25.4 | 93.3 | 62.8 | - |
| LP-FT (Kumar et al., 2022) | IN-V2 | 55.4 | - | 6.3 | 26.2 | 11.7 | 17.6 | 39.0 | 17.8 | 26.0 |
| | IN-A | 30.7 | 23.6 | - | 13.0 | 4.9 | 12.4 | 17.9 | 6.2 | 11.6 |
| | IN-R | 47.6 | 37.3 | 5.6 | - | 33.1 | 17.9 | 37.1 | 25.1 | 25.7 |
| | IN-Sketch | 30.1 | 23.4 | 2.6 | 41.4 | - | 10.0 | 23.1 | 13.9 | 12.3 |
| | ObjNet | 38.6 | 28.8 | 4.5 | 15.1 | 7.4 | - | 21.5 | 8.3 | 16.3 |
| | IN-Cartoon | 89.2 | 47.1 | 6.5 | 32.6 | 18.8 | 19.2 | - | 39.2 | 31.3 |
| | IN-Drawing | 63.1 | 34.8 | 4.2 | 29.0 | 17.3 | 12.8 | 52.8 | - | 25.7 |
| | IN-C | 99.9 | 42.1 | 4.7 | 29.1 | 16.0 | 14.5 | 98.0 | 82.5 | - |
| WiSE-FT (Wortsman et al., 2022b) | IN-V2 | 75.6 | - | 15.2 | 42.9 | 26.9 | 32.7 | 60.7 | 37.0 | 49.1 |
| | IN-A | 69.8 | 57.5 | - | 37.0 | 21.5 | 31.3 | 51.7 | 27.2 | 43.3 |
| | IN-R | 73.4 | 61.5 | 14.6 | - | 45.8 | 32.7 | 62.7 | 47.4 | 49.5 |
| | IN-Sketch | 64.2 | 52.2 | 8.2 | 54.6 | - | 23.4 | 52.8 | 35.8 | 36.3 |
| | ObjNet | 73.5 | 61.3 | 15.0 | 40.9 | 28.5 | - | 55.7 | 30.1 | 46.7 |
| | IN-Cartoon | 85.8 | 64.7 | 15.6 | 48.4 | 33.0 | 33.4 | - | 50.2 | 50.8 |
| | IN-Drawing | 77.3 | 56.8 | 10.0 | 47.0 | 34.6 | 23.8 | 66.8 | - | 46.9 |
| | IN-C | 91.9 | 61.4 | 11.1 | 47.6 | 32.7 | 29.6 | 86.6 | 69.7 | - |
| Model Soup PRE-FT-EWC-LwF (Wortsman et al., 2022a) | IN-V2 | 76.5 | - | 15.6 | 45.7 | 29.6 | 33.5 | 62.1 | 40.1 | 50.2 |
| | IN-A | 71.2 | 59.3 | - | 38.8 | 24.3 | 31.9 | 53.4 | 29.8 | 44.3 |
| | IN-R | 74.9 | 63.6 | 16.2 | - | 48.4 | 34.1 | 64.5 | 50.6 | 51.8 |
| | IN-Sketch | 64.8 | 52.8 | 8.1 | 55.7 | - | 23.4 | 54.0 | 37.2 | 37.2 |
| | ObjNet | 75.2 | 63.1 | 16.5 | 42.7 | 30.3 | - | 57.7 | 32.8 | 48.7 |
| | IN-Cartoon | 85.8 | 65.6 | 15.9 | 50.1 | 35.1 | 33.9 | - | 53.0 | 51.7 |
| | IN-Drawing | 77.9 | 58.3 | 10.3 | 49.4 | 36.9 | 25.0 | 67.7 | - | 47.9 |
| | IN-C | 90.8 | 65.2 | 13.2 | 51.7 | 37.1 | 32.5 | 85.1 | 68.1 | - |

Table 28: The accuracy on each OOD dataset after fine-tuning ImageNet-1K with AugReg pre-trained ViT-S/16 on the downstream datasets with various methods. Note that ImageNet-Drawing, ImageNet-Cartoon, and ImageNet-C are generated from the ImageNet validation set. Green and red indicate relative performance increases and decreases, respectively, compared to the pre-trained model. Bold indicates the best performance on each evaluation dataset.

| Method | Downstream Dataset | $D_{pre}$ IN | IN-V2 | IN-A | Realistic OOD IN-R | IN-Sketch | ObjNet | Synthetic OOD IN-Cartoon | IN-Drawing | IN-C |
|---|---|---|---|---|---|---|---|---|---|---|
| Pre-Trained | | 78.8 | 66.7 | 13.4 | 37.1 | 25.9 | 33.6 | 63.3 | 37.2 | 53.2 |
| FT | IN-V2 | 75.5 | - | 19.8 | 38.1 | 25.0 | 35.3 | 58.2 | 35.9 | 51.2 |
| | IN-A | 66.5 | 55.9 | - | 33.5 | 22.0 | 31.0 | 46.5 | 26.6 | 44.1 |
| | IN-R | 62.8 | 52.6 | 14.5 | - | 41.2 | 28.8 | 53.0 | 40.4 | 42.7 |
| | IN-Sketch | 70.6 | 59.2 | 11.5 | 56.6 | - | 31.4 | 58.9 | 43.7 | 45.5 |
| | ObjNet | 67.3 | 55.8 | 15.7 | 30.1 | 18.5 | - | 43.9 | 23.0 | 41.9 |
| | IN-Cartoon | 86.8 | 59.3 | 15.1 | 37.0 | 23.9 | 31.7 | - | 38.7 | 44.4 |
| | IN-Drawing | 77.6 | 55.9 | 10.2 | 35.8 | 24.8 | 26.7 | 56.5 | - | 46.0 |
| | IN-C | 99.9 | 55.0 | 8.3 | 31.1 | 18.3 | 24.0 | 91.8 | 68.8 | - |
| Linear Probing | IN-V2 | 78.6 | - | 14.3 | 37.4 | 25.8 | 33.9 | 63.0 | 36.8 | 53.1 |
| | IN-A | 77.7 | 65.5 | - | 37.1 | 25.4 | 35.1 | 61.4 | 36.3 | 53.0 |
| | IN-R | 78.2 | 66.2 | 15.3 | - | 29.0 | 34.3 | 63.3 | 38.5 | 53.0 |
| | IN-Sketch | 76.1 | 64.0 | 15.0 | 47.2 | - | 34.6 | 62.6 | 39.5 | 50.9 |
| | ObjNet | 78.1 | 66.1 | 16.2 | 37.0 | 25.2 | - | 61.1 | 36.6 | 53.4 |
| | IN-Cartoon | 80.4 | 64.9 | 13.0 | 38.5 | 25.8 | 32.9 | - | 38.8 | 52.6 |
| | IN-Drawing | 78.6 | 65.2 | 13.8 | 39.7 | 27.2 | 33.3 | 63.6 | - | 54.3 |
| | IN-C | 93.5 | 58.7 | 10.9 | 32.6 | 21.5 | 27.5 | 75.4 | 50.0 | - |
| Visual Prompt (Bahng et al., 2022) | IN-V2 | 74.7 | - | 10.8 | 36.3 | 22.3 | 34.1 | 54.9 | 29.2 | 43.9 |
| | IN-A | 65.4 | 53.7 | - | 29.7 | 15.9 | 33.0 | 41.7 | 18.0 | 32.1 |
| | IN-R | 65.4 | 53.9 | 8.9 | - | 33.9 | 28.7 | 52.8 | 37.5 | 35.0 |
| | IN-Sketch | 70.5 | 58.0 | 8.3 | 48.3 | - | 31.4 | 55.7 | 39.0 | 39.2 |
| | ObjNet | 62.6 | 51.0 | 8.6 | 24.2 | 13.3 | - | 34.7 | 11.1 | 26.8 |
| | IN-Cartoon | 73.3 | 59.6 | 8.4 | 37.2 | 22.0 | 31.1 | - | 28.4 | 36.0 |
| | IN-Drawing | 71.0 | 58.5 | 5.9 | 38.3 | 24.3 | 28.0 | 55.8 | - | 40.7 |
| | IN-C | 76.1 | 63.5 | 11.9 | 36.3 | 23.2 | 35.0 | 57.0 | 40.1 | - |
| LoRA (Hu et al., 2021) | IN-V2 | 78.9 | - | 13.8 | 37.3 | 26.0 | 34.0 | 63.3 | 37.4 | 53.5 |
| | IN-A | 78.6 | 66.6 | - | 37.7 | 25.6 | 36.2 | 61.4 | 36.9 | 53.5 |
| | IN-R | 78.6 | 66.5 | 16.2 | - | 28.9 | 36.0 | 64.2 | 40.1 | 52.8 |
| | IN-Sketch | 78.7 | 66.5 | 14.8 | 47.7 | - | 34.9 | 65.9 | 46.7 | 53.5 |
| | ObjNet | 78.6 | 66.5 | 16.8 | 37.5 | 24.8 | - | 61.0 | 36.5 | 53.0 |
| | IN-Cartoon | 77.8 | 65.1 | 12.5 | 39.3 | 26.3 | 32.8 | - | 37.4 | 51.3 |
| | IN-Drawing | 78.1 | 66.2 | 13.4 | 42.0 | 29.4 | 34.1 | 63.7 | - | 54.6 |
| | IN-C | 79.2 | 66.8 | 13.9 | 38.1 | 25.9 | 34.6 | 64.5 | 38.6 | - |
| EWC (Kirkpatrick et al., 2017) | IN-V2 | 79.1 | - | 19.8 | 40.1 | 26.8 | 37.6 | 62.7 | 39.5 | 55.4 |
| | IN-A | 74.3 | 62.8 | - | 38.3 | 25.2 | 37.2 | 54.5 | 32.2 | 51.4 |
| | IN-R | 74.0 | 62.6 | 16.4 | - | 42.3 | 33.6 | 64.4 | 51.6 | 51.8 |
| | IN-Sketch | 77.7 | 65.7 | 14.9 | 54.0 | - | 35.7 | 66.9 | 52.1 | 54.0 |
| | ObjNet | 74.8 | 63.8 | 22.0 | 37.1 | 24.3 | - | 52.3 | 32.4 | 51.7 |
| | IN-Cartoon | 77.9 | 64.7 | 14.0 | 41.6 | 27.7 | 34.0 | - | 39.8 | 49.1 |
| | IN-Drawing | 77.4 | 65.3 | 12.5 | 41.7 | 29.8 | 33.3 | 63.8 | - | 53.2 |
| | IN-C | 79.4 | 67.0 | 17.2 | 39.4 | 26.6 | 37.1 | 62.2 | 46.6 | - |
| LwF (Li & Hoiem, 2017) | IN-V2 | 77.8 | - | 17.5 | 38.6 | 26.0 | 35.2 | 61.5 | 37.5 | 53.5 |
| | IN-A | 76.0 | 64.5 | - | 37.7 | 25.0 | 33.7 | 58.5 | 35.0 | 51.7 |
| | IN-R | 75.1 | 63.2 | 16.0 | - | 42.9 | 32.3 | 63.7 | 48.1 | 51.8 |
| | IN-Sketch | 74.5 | 62.3 | 11.6 | 55.3 | - | 31.9 | 62.1 | 43.5 | 48.6 |
| | ObjNet | 76.1 | 64.2 | 17.1 | 35.2 | 22.6 | - | 56.8 | 31.9 | 50.4 |
| | IN-Cartoon | 89.9 | 64.3 | 15.8 | 38.4 | 24.8 | 34.0 | - | 39.4 | 52.6 |
| | IN-Drawing | 83.1 | 60.6 | 11.4 | 36.9 | 26.1 | 29.5 | 60.8 | - | 50.8 |
| | IN-C | 99.5 | 63.0 | 10.4 | 37.0 | 23.7 | 29.9 | 90.1 | 60.9 | - |
| LP-FT (Kumar et al., 2022) | IN-V2 | 76.4 | - | 20.4 | 38.6 | 25.6 | 36.1 | 59.3 | 37.5 | 52.3 |
| | IN-A | 72.5 | 61.4 | - | 35.2 | 23.9 | 34.2 | 53.4 | 32.3 | 48.6 |
| | IN-R | 69.4 | 57.7 | 15.2 | - | 41.0 | 31.7 | 57.9 | 43.6 | 46.6 |
| | IN-Sketch | 72.7 | 60.9 | 13.5 | 57.3 | - | 32.6 | 61.3 | 43.3 | 47.9 |
| | ObjNet | 72.5 | 60.4 | 17.1 | 33.6 | 20.7 | - | 51.3 | 26.4 | 46.6 |
| | IN-Cartoon | 88.3 | 60.9 | 15.1 | 37.7 | 24.3 | 32.6 | - | 40.9 | 47.2 |
| | IN-Drawing | 79.7 | 57.5 | 10.4 | 36.7 | 25.6 | 27.1 | 58.5 | - | 48.2 |
| | IN-C | 99.9 | 52.8 | 7.1 | 30.5 | 17.7 | 21.5 | 93.5 | 73.3 | - |
| WiSE-FT (Wortsman et al., 2022b) | IN-V2 | 78.8 | - | 18.9 | 39.5 | 27.2 | 36.2 | 63.0 | 39.4 | 55.0 |
| | IN-A | 77.6 | 65.8 | - | 39.2 | 26.9 | 37.4 | 61.2 | 36.9 | 54.2 |
| | IN-R | 77.3 | 65.7 | 18.8 | - | 43.1 | 36.6 | 67.2 | 52.0 | 55.3 |
| | IN-Sketch | 77.6 | 66.1 | 14.3 | 53.7 | - | 35.3 | 65.9 | 48.8 | 53.6 |
| | ObjNet | 77.8 | 65.7 | 19.2 | 37.7 | 25.5 | - | 60.3 | 35.3 | 53.5 |
| | IN-Cartoon | 85.2 | 65.7 | 16.6 | 39.7 | 27.2 | 35.0 | - | 43.1 | 52.9 |
| | IN-Drawing | 82.5 | 65.0 | 14.0 | 40.2 | 29.9 | 33.5 | 64.7 | - | 55.4 |
| | IN-C | 94.2 | 64.9 | 13.9 | 38.7 | 26.0 | 33.1 | 82.7 | 58.0 | - |
| Model Soup PRE-FT-EWC-LwF (Wortsman et al., 2022a) | IN-V2 | 78.9 | - | 19.1 | 39.5 | 27.2 | 36.4 | 62.8 | 39.4 | 55.1 |
| | IN-A | 77.1 | 65.4 | - | 39.2 | 26.7 | 37.1 | 60.1 | 36.4 | 53.9 |
| | IN-R | 77.2 | 65.5 | 19.0 | - | 44.4 | 36.0 | 67.5 | 52.7 | 55.2 |
| | IN-Sketch | 77.6 | 66.1 | 14.6 | 55.6 | - | 35.4 | 66.4 | 49.7 | 53.6 |
| | ObjNet | 77.7 | 65.9 | 20.1 | 37.9 | 25.5 | - | 59.5 | 35.1 | 53.5 |
| | IN-Cartoon | 84.9 | 65.9 | 16.2 | 40.4 | 27.4 | 34.9 | - | 42.7 | 52.7 |
| | IN-Drawing | 82.3 | 64.8 | 13.7 | 40.6 | 30.1 | 33.3 | 64.9 | - | 55.1 |
| | IN-C | 93.0 | 66.2 | 14.4 | 39.8 | 27.1 | 34.3 | 80.8 | 56.8 | - |

Table 29: The accuracy on each OOD dataset after fine-tuning ImageNet-21K pre-trained ViT-S/16 with AugReg on the downstream datasets with various methods. Note that ImageNet-Drawing, ImageNet-Cartoon, and ImageNet-C are generated from the ImageNet validation set. Green and red indicate relative performance increases and decreases, respectively, compared to the pre-trained model. Bold indicates the best performance on each evaluation dataset.

| Method | Downstream Dataset | $D_{pre}$ IN | Realistic OOD | | | | | Synthetic OOD | | |
|---|---|---|---|---|---|---|---|---|---|---|
| | | | IN-V2 | IN-A | IN-R | IN-Sketch | ObjNet | IN-Cartoon | IN-Drawing | IN-C |
| Pre-Trained | | 81.4 | 70.3 | 27.0 | 46.0 | 32.9 | 41.3 | 67.8 | 37.7 | 58.0 |
| FT | IN-V2 | 78.9 | - | 34.3 | 46.1 | 30.5 | 43.2 | 63.9 | 34.4 | 56.8 |
| | IN-A | 73.2 | 62.8 | - | 43.7 | 30.4 | 41.5 | 55.4 | 29.7 | 52.7 |
| | IN-R | 70.5 | 60.5 | 26.4 | - | 48.0 | 37.0 | 60.5 | 46.2 | 50.6 |
| | IN-Sketch | 74.4 | 63.1 | 25.2 | 63.3 | - | 39.1 | 62.5 | 47.2 | 50.7 |
| | ObjNet | 73.3 | 61.6 | 28.9 | 38.3 | 26.3 | - | 52.7 | 24.0 | 48.5 |
| | IN-Cartoon | 87.6 | 64.4 | 26.5 | 45.1 | 31.1 | 40.0 | - | 41.2 | 52.2 |
| | IN-Drawing | 83.1 | 62.2 | 20.3 | 45.1 | 33.1 | 36.3 | 64.9 | - | 52.8 |
| | IN-C | 99.8 | 59.8 | 15.9 | 39.4 | 25.2 | 32.0 | 91.7 | 71.3 | - |
| Linear Probing | IN-V2 | 80.9 | - | 28.2 | 46.1 | 32.5 | 41.0 | 67.1 | 37.1 | 57.7 |
| | IN-A | 80.0 | 69.1 | - | 45.6 | 32.2 | 41.6 | 66.2 | 36.4 | 57.2 |
| | IN-R | 80.1 | 69.5 | 27.9 | - | 35.5 | 40.5 | 66.3 | 38.2 | 56.8 |
| | IN-Sketch | 77.5 | 67.0 | 27.7 | 54.5 | - | 40.8 | 64.6 | 40.4 | 55.1 |
| | ObjNet | 80.2 | 69.5 | 28.9 | 45.6 | 32.0 | - | 66.5 | 37.1 | 57.9 |
| | IN-Cartoon | 85.1 | 68.8 | 26.4 | 46.5 | 32.5 | 40.7 | - | 41.0 | 59.3 |
| | IN-Drawing | 82.5 | 68.8 | 26.9 | 47.4 | 34.1 | 40.0 | 70.4 | - | 60.0 |
| | IN-C | 95.6 | 60.4 | 21.4 | 39.8 | 25.9 | 32.8 | 81.7 | 52.3 | - |
| Visual Prompt (Bahng et al., 2022) | IN-V2 | 77.0 | - | 22.2 | 42.7 | 28.1 | 41.8 | 58.4 | 29.9 | 47.9 |
| | IN-A | 69.3 | 58.7 | - | 38.5 | 22.2 | 40.2 | 48.1 | 21.8 | 39.6 |
| | IN-R | 66.9 | 56.4 | 15.7 | - | 38.4 | 32.3 | 53.0 | 38.8 | 38.1 |
| | IN-Sketch | 73.0 | 61.3 | 15.3 | 55.7 | - | 38.3 | 58.3 | 42.1 | 42.1 |
| | ObjNet | 64.1 | 52.7 | 15.1 | 27.2 | 16.5 | - | 35.4 | 14.4 | 29.1 |
| | IN-Cartoon | 75.2 | 62.2 | 17.8 | 41.6 | 26.8 | 37.7 | - | 29.3 | 39.7 |
| | IN-Drawing | 72.9 | 60.9 | 13.2 | 42.2 | 27.5 | 35.7 | 55.8 | - | 41.0 |
| | IN-C | 78.1 | 66.6 | 23.1 | 42.7 | 28.1 | 41.1 | 60.9 | 41.0 | - |
| LoRA (Hu et al., 2021) | IN-V2 | 81.5 | - | 27.2 | 45.9 | 32.8 | 41.6 | 67.8 | 37.7 | 58.2 |
| | IN-A | 81.5 | 70.7 | - | 46.2 | 32.8 | 43.0 | 67.8 | 37.7 | 58.9 |
| | IN-R | 81.4 | 70.6 | 29.1 | - | 34.9 | 43.3 | 68.5 | 40.7 | 58.2 |
| | IN-Sketch | 81.3 | 70.3 | 28.1 | 54.0 | - | 42.1 | 68.6 | 46.0 | 58.3 |
| | ObjNet | 81.5 | 70.8 | 29.2 | 46.3 | 32.3 | - | 67.6 | 38.0 | 58.8 |
| | IN-Cartoon | 81.4 | 70.2 | 27.3 | 47.0 | 33.1 | 41.6 | - | 38.6 | 57.7 |
| | IN-Drawing | 81.4 | 70.5 | 27.7 | 49.7 | 36.7 | 42.4 | 68.5 | - | 59.0 |
| | IN-C | 68.9 | 56.6 | 8.0 | 26.4 | 15.9 | 27.3 | 42.4 | 9.4 | - |
| EWC (Kirkpatrick et al., 2017) | IN-V2 | 81.6 | - | 33.2 | 48.2 | 33.3 | 44.6 | 67.1 | 38.4 | 59.7 |
| | IN-A | 78.9 | 68.3 | - | 47.2 | 33.0 | 45.4 | 62.7 | 33.9 | 58.0 |
| | IN-R | 78.1 | 67.3 | 30.3 | - | 47.5 | 41.1 | 68.1 | 53.4 | 56.4 |
| | IN-Sketch | 80.6 | 69.7 | 29.1 | 60.0 | - | 42.9 | 69.8 | 52.4 | 58.0 |
| | ObjNet | 78.8 | 68.0 | 34.5 | 44.5 | 30.9 | - | 59.2 | 34.3 | 56.7 |
| | IN-Cartoon | 81.0 | 68.9 | 28.4 | 48.6 | 32.9 | 42.0 | - | 42.6 | 55.2 |
| | IN-Drawing | 80.7 | 69.0 | 26.7 | 50.0 | 37.1 | 41.3 | 68.3 | - | 58.1 |
| | IN-C | 81.8 | 71.0 | 31.2 | 47.5 | 33.2 | 43.2 | 67.4 | 48.6 | - |
| LwF (Li & Hoiem, 2017) | IN-V2 | 80.7 | - | 31.4 | 46.3 | 31.8 | 42.6 | 67.1 | 36.7 | 58.3 |
| | IN-A | 80.0 | 69.3 | - | 46.2 | 31.6 | 42.1 | 64.5 | 37.0 | 57.8 |
| | IN-R | 79.5 | 68.3 | 28.4 | - | 47.5 | 39.4 | 68.4 | 48.3 | 57.2 |
| | IN-Sketch | 77.8 | 66.2 | 25.4 | 60.9 | - | 40.9 | 66.1 | 43.8 | 54.5 |
| | ObjNet | 79.1 | 67.3 | 30.7 | 42.2 | 29.6 | - | 61.7 | 31.1 | 55.1 |
| | IN-Cartoon | 90.4 | 68.8 | 26.9 | 46.9 | 33.2 | 41.7 | - | 42.2 | 60.0 |
| | IN-Drawing | 86.7 | 66.5 | 21.9 | 46.0 | 34.7 | 38.6 | 68.5 | - | 57.8 |
| | IN-C | 99.8 | 64.7 | 17.1 | 43.4 | 29.0 | 36.0 | 91.6 | 65.1 | - |
| LP-FT (Kumar et al., 2022) | IN-V2 | 79.4 | - | 34.2 | 46.6 | 31.3 | 43.1 | 65.2 | 36.3 | 57.6 |
| | IN-A | 77.0 | 67.0 | - | 44.3 | 31.5 | 42.7 | 60.3 | 32.2 | 56.0 |
| | IN-R | 75.4 | 65.0 | 27.6 | - | 47.4 | 39.5 | 64.3 | 49.1 | 53.5 |
| | IN-Sketch | 75.0 | 63.9 | 27.5 | 62.8 | - | 40.6 | 64.1 | 44.9 | 52.3 |
| | ObjNet | 77.4 | 66.1 | 31.5 | 41.4 | 27.9 | - | 59.2 | 28.0 | 53.4 |
| | IN-Cartoon | 89.3 | 65.3 | 27.2 | 45.8 | 31.8 | 40.9 | - | 42.8 | 56.0 |
| | IN-Drawing | 85.3 | 63.4 | 21.5 | 46.7 | 34.9 | 37.5 | 67.9 | - | 55.5 |
| | IN-C | 99.9 | 56.4 | 12.0 | 38.3 | 23.3 | 27.6 | 94.5 | 75.6 | - |
| WiSE-FT (Wortsman et al., 2022b) | IN-V2 | 81.3 | - | 33.4 | 47.6 | 33.4 | 44.0 | 67.6 | 38.1 | 59.3 |
| | IN-A | 80.6 | 69.8 | - | 47.7 | 33.9 | 45.4 | 66.1 | 38.0 | 58.9 |
| | IN-R | 79.9 | 69.8 | 32.6 | - | 48.5 | 43.8 | 70.4 | 53.2 | 59.6 |
| | IN-Sketch | 80.0 | 69.2 | 29.3 | 60.7 | - | 42.6 | 69.6 | 49.8 | 58.0 |
| | ObjNet | 80.5 | 69.3 | 33.1 | 45.6 | 32.5 | - | 65.7 | 35.0 | 57.6 |
| | IN-Cartoon | 86.7 | 69.5 | 30.2 | 47.7 | 34.0 | 43.1 | - | 43.9 | 58.4 |
| | IN-Drawing | 85.1 | 69.0 | 27.7 | 49.4 | 37.8 | 42.5 | 70.8 | - | 60.4 |
| | IN-C | 93.8 | 68.7 | 26.5 | 47.1 | 32.5 | 41.5 | 83.6 | 58.7 | - |
| Model Soup PRE-FT-EWC-LwF (Wortsman et al., 2022a) | IN-V2 | 81.4 | - | 33.3 | 47.6 | 33.3 | 43.8 | 67.6 | 38.0 | 59.3 |
| | IN-A | 80.4 | 69.8 | - | 47.7 | 33.6 | 45.1 | 65.6 | 37.6 | 58.9 |
| | IN-R | 80.3 | 69.9 | 32.6 | - | 49.1 | 43.2 | 70.7 | 53.8 | 59.4 |
| | IN-Sketch | 80.0 | 69.2 | 29.2 | 61.5 | - | 42.6 | 69.7 | 49.9 | 57.9 |
| | ObjNet | 80.4 | 69.2 | 33.8 | 45.3 | 32.3 | - | 64.6 | 35.3 | 57.6 |
| | IN-Cartoon | 86.4 | 69.6 | 29.7 | 48.2 | 34.1 | 42.9 | - | 43.9 | 58.6 |
| | IN-Drawing | 85.0 | 69.2 | 27.0 | 49.2 | 37.8 | 41.9 | 70.6 | - | 60.2 |
| | IN-C | 94.2 | 69.3 | 27.2 | 48.0 | 34.0 | 41.9 | 84.1 | 59.7 | - |

Table 30: The accuracy on each OOD dataset after fine-tuning ImageNet-21K pre-trained ViT-S/32 with AugReg on the downstream datasets with various methods. Note that ImageNet-Drawing, ImageNet-Cartoon, and ImageNet-C are generated from the ImageNet validation set. Green and red indicate relative performance increases and decreases, respectively, compared to the pre-trained model. Bold indicates the best performance on each evaluation dataset.

| Method | Downstream Dataset | $D_{pre}$ IN | Realistic OOD | | | | | Synthetic OOD | | |
| --- | --- | --- | --- | --- | --- | --- | --- | --- | --- | --- |
| | | | IN-V2 | IN-A | IN-R | IN-Sketch | ObjNet | IN-Cartoon | IN-Drawing | IN-C |
| Pre-Trained | | 76.0 | 63.9 | 11.5 | 39.7 | 26.2 | 33.1 | 62.9 | 34.3 | 52.0 |
| FT | IN-V2 | 72.2 | - | 17.3 | 38.2 | 23.4 | 34.0 | 56.9 | 31.2 | 50.1 |
| | IN-A | 64.3 | 53.7 | - | 33.3 | 20.9 | 31.9 | 46.9 | 24.7 | 44.0 |
| | IN-R | 62.4 | 51.6 | 14.1 | - | 42.7 | 29.1 | 53.2 | 40.5 | 43.4 |
| | IN-Sketch | 67.2 | 55.1 | 12.0 | 56.9 | - | 30.4 | 56.7 | 41.7 | 45.5 |
| | ObjNet | 64.9 | 53.4 | 14.2 | 31.7 | 18.3 | - | 44.7 | 21.5 | 41.7 |
| | IN-Cartoon | 84.6 | 56.2 | 12.8 | 38.7 | 24.6 | 30.3 | - | 35.6 | 45.2 |
| | IN-Drawing | 75.4 | 53.0 | 9.6 | 38.0 | 26.1 | 25.8 | 55.9 | - | 46.9 |
| | IN-C | 99.7 | 52.1 | 8.1 | 32.9 | 19.8 | 24.2 | 92.9 | 71.0 | - |
| Linear Probing | IN-V2 | 75.3 | - | 12.7 | 39.4 | 26.0 | 32.7 | 62.3 | 34.1 | 51.5 |
| | IN-A | 73.5 | 61.6 | - | 38.5 | 25.2 | 32.3 | 60.7 | 32.6 | 50.4 |
| | IN-R | 74.6 | 62.5 | 13.0 | - | 28.8 | 32.4 | 62.1 | 34.5 | 51.5 |
| | IN-Sketch | 71.5 | 60.1 | 13.3 | 49.2 | - | 33.1 | 59.6 | 36.6 | 49.6 |
| | ObjNet | 74.5 | 62.9 | 12.7 | 40.1 | 26.3 | - | 61.6 | 33.1 | 52.1 |
| | IN-Cartoon | 80.6 | 61.6 | 11.4 | 40.8 | 26.1 | 32.4 | - | 38.2 | 53.5 |
| | IN-Drawing | 77.3 | 61.1 | 11.3 | 40.8 | 27.4 | 30.3 | 65.5 | - | 53.6 |
| | IN-C | 93.6 | 52.2 | 10.7 | 33.5 | 20.6 | 25.3 | 79.2 | 49.0 | - |
| Visual Prompt (Bahng et al., 2022) | IN-V2 | 69.9 | - | 9.6 | 36.9 | 22.5 | 33.1 | 53.6 | 28.5 | 42.6 |
| | IN-A | 48.5 | 38.5 | - | 24.9 | 8.8 | 22.0 | 31.2 | 12.4 | 23.2 |
| | IN-R | 58.5 | 48.0 | 6.7 | - | 31.1 | 25.5 | 46.2 | 34.4 | 32.3 |
| | IN-Sketch | 64.5 | 52.5 | 6.3 | 48.1 | - | 28.2 | 51.0 | 37.0 | 35.1 |
| | ObjNet | 53.5 | 42.9 | 7.2 | 23.9 | 12.2 | - | 31.8 | 14.1 | 24.5 |
| | IN-Cartoon | 69.0 | 56.1 | 7.4 | 38.0 | 22.3 | 30.0 | - | 30.6 | 36.0 |
| | IN-Drawing | 63.6 | 51.3 | 5.5 | 37.8 | 22.9 | 25.3 | 49.5 | - | 36.6 |
| | IN-C | 71.2 | 59.4 | 9.8 | 36.9 | 22.5 | 33.5 | 55.2 | 35.6 | - |
| LoRA (Hu et al., 2021) | IN-V2 | 76.0 | - | 11.6 | 39.7 | 26.2 | 33.2 | 62.9 | 34.3 | 52.1 |
| | IN-A | 76.1 | 64.0 | - | 40.3 | 26.5 | 34.1 | 63.0 | 34.1 | 53.1 |
| | IN-R | 76.1 | **64.1** | 12.4 | - | 27.8 | 34.7 | 63.8 | 35.8 | 52.8 |
| | IN-Sketch | 75.7 | 63.5 | 12.7 | 48.1 | - | 34.1 | 63.8 | 41.5 | 52.9 |
| | ObjNet | 76.1 | 64.0 | 12.5 | 40.6 | 26.6 | - | 63.1 | 33.9 | 53.3 |
| | IN-Cartoon | 75.9 | 63.7 | 11.7 | 40.7 | 26.6 | 33.7 | - | 35.0 | 51.4 |
| | IN-Drawing | 75.5 | 63.5 | 10.9 | 43.7 | 30.2 | 32.8 | 63.4 | - | 52.3 |
| | IN-C | 74.1 | 62.0 | 9.3 | 36.9 | 23.5 | 30.2 | 58.4 | 30.5 | - |
| EWC (Kirkpatrick et al., 2017) | IN-V2 | 76.0 | - | 15.8 | 41.0 | 26.1 | 36.0 | 62.1 | 35.4 | 53.6 |
| | IN-A | 70.9 | 59.1 | - | 38.3 | 24.0 | 35.8 | 54.9 | 28.1 | 50.2 |
| | IN-R | 71.8 | 60.4 | 14.5 | - | 41.4 | 32.6 | 61.9 | 47.4 | 50.7 |
| | IN-Sketch | 74.9 | 62.9 | 13.2 | 53.4 | - | 34.6 | 64.7 | 45.5 | 52.6 |
| | ObjNet | 72.5 | 60.9 | 17.9 | 38.7 | 24.8 | - | 55.2 | 31.6 | 50.7 |
| | IN-Cartoon | 75.0 | 61.8 | 11.7 | 42.4 | 26.9 | 33.1 | - | 37.3 | 48.8 |
| | IN-Drawing | 74.2 | 61.9 | 11.6 | 43.7 | 29.8 | 33.0 | 61.8 | - | 52.3 |
| | IN-C | 76.2 | 64.0 | 14.3 | 41.4 | 27.0 | 35.8 | 62.3 | 42.6 | - |
| LwF (Li & Hoiem, 2017) | IN-V2 | 75.0 | - | 14.9 | 39.4 | 25.3 | 33.6 | 60.8 | 32.9 | 51.8 |
| | IN-A | 73.1 | 61.1 | - | 38.2 | 24.4 | 33.6 | 57.7 | 31.5 | 50.8 |
| | IN-R | 72.9 | 60.7 | 14.1 | - | 42.6 | 31.6 | 62.4 | 44.2 | 50.9 |
| | IN-Sketch | 71.2 | 59.1 | 11.9 | 55.3 | - | 32.6 | 60.0 | 40.5 | 48.9 |
| | ObjNet | 71.1 | 59.7 | 15.5 | 36.2 | 21.6 | - | 54.3 | 27.2 | 48.2 |
| | IN-Cartoon | 88.0 | 61.5 | 12.6 | 40.3 | 26.4 | 32.1 | - | 37.8 | 53.7 |
| | IN-Drawing | 81.4 | 58.2 | 10.6 | 39.1 | 27.4 | 28.8 | 62.2 | - | 52.1 |
| | IN-C | 98.7 | 60.0 | 8.9 | 38.2 | 24.1 | 29.1 | 89.2 | 55.7 | - |
| LP-FT (Kumar et al., 2022) | IN-V2 | 73.2 | - | 17.4 | 39.0 | 24.5 | 33.9 | 58.4 | 32.1 | 50.7 |
| | IN-A | 69.2 | 58.2 | - | 35.0 | 22.9 | 33.4 | 53.9 | 28.6 | 48.2 |
| | IN-R | 68.5 | 56.8 | 14.2 | - | 42.5 | 32.3 | 58.0 | 43.2 | 47.5 |
| | IN-Sketch | 67.8 | 56.5 | 13.0 | 57.3 | - | 32.0 | 57.6 | 39.9 | 46.9 |
| | ObjNet | 70.2 | 58.2 | 14.8 | 35.8 | 21.4 | - | 53.1 | 24.6 | 47.0 |
| | IN-Cartoon | 87.3 | 57.7 | 13.2 | 39.2 | 25.2 | 31.2 | - | 37.8 | 50.1 |
| | IN-Drawing | 79.1 | 55.2 | 10.7 | 38.5 | 26.9 | 26.3 | 60.0 | - | 50.3 |
| | IN-C | **99.8** | 48.0 | 6.9 | 30.8 | 18.0 | 20.7 | 94.6 | 73.5 | - |
| WiSE-FT (Wortsman et al., 2022b) | IN-V2 | 75.7 | - | 15.7 | 40.3 | 26.3 | 35.3 | 62.0 | 35.0 | 53.1 |
| | IN-A | 74.5 | 62.9 | - | 40.0 | 26.1 | **36.6** | 60.2 | 34.2 | 52.8 |
| | IN-R | 74.0 | 62.8 | 16.2 | - | 42.2 | 35.3 | 64.7 | 47.5 | 53.3 |
| | IN-Sketch | 74.2 | 62.2 | 13.1 | 54.4 | - | 34.1 | 63.9 | 44.2 | 52.1 |
| | ObjNet | 74.5 | 62.6 | 15.9 | 39.1 | 25.4 | - | 59.5 | 32.2 | 51.8 |
| | IN-Cartoon | 83.0 | 62.6 | 13.6 | 41.0 | 27.1 | 33.7 | - | 39.0 | 51.8 |
| | IN-Drawing | 79.8 | 61.8 | 12.2 | 42.4 | 30.0 | 32.4 | 64.5 | - | **54.3** |
| | IN-C | 93.0 | 61.8 | 12.8 | 40.4 | 26.7 | 32.4 | 83.5 | 56.9 | - |
| Model Soup PRE-FT-EWC-LwF (Wortsman et al., 2022a) | IN-V2 | 75.7 | - | 15.6 | 40.4 | 26.3 | 35.3 | 61.9 | 35.0 | 53.1 |
| | IN-A | 74.2 | 62.4 | - | 39.6 | 25.7 | 36.3 | 59.2 | 33.0 | 52.5 |
| | IN-R | 74.3 | 62.8 | 16.0 | - | **43.4** | 34.8 | 64.9 | 48.0 | 53.2 |
| | IN-Sketch | 74.2 | 62.3 | 13.0 | 55.5 | - | 34.4 | 64.0 | 44.6 | 52.2 |
| | ObjNet | 74.1 | 62.6 | 17.2 | 39.1 | 25.3 | - | 58.4 | 32.0 | 51.7 |
| | IN-Cartoon | 82.5 | 62.6 | 13.1 | 41.6 | 27.3 | 33.4 | - | 39.0 | 52.0 |
| | IN-Drawing | 79.6 | 61.7 | 12.3 | 42.5 | 30.1 | 32.1 | 64.4 | - | 54.2 |
| | IN-C | 91.5 | 63.1 | 13.2 | 41.2 | 27.3 | 33.3 | 81.0 | 54.5 | - |

Table 31: The accuracy on each OOD dataset after fine-tuning ImageNet-21K pre-trained ViT-L/16 with AugReg on the downstream datasets with various methods. Note that ImageNet-Drawing, ImageNet-Cartoon, and ImageNet-C are generated from the ImageNet validation set. Green and red indicate relative performance increases and decreases, respectively, compared to the pre-trained model. Bold indicates the best performance on each evaluation dataset.

| Method | Downstream Dataset | $D_{pre}$ IN | Realistic OOD IN-V2 | IN-A | IN-R | IN-Sketch | ObjNet | Synthetic OOD IN-Cartoon | IN-Drawing | IN-C |
|---|---|---|---|---|---|---|---|---|---|---|
| Pre-Trained | | 85.8 | 76.2 | 55.5 | 64.4 | 51.8 | 52.8 | 79.5 | 64.6 | 72.2 |
| FT | IN-V2 | 83.8 | - | 59.7 | 66.2 | 51.0 | 54.1 | 77.0 | 63.9 | 71.0 |
| | IN-A | 81.9 | 24.4 | - | 66.1 | 51.0 | 52.3 | 75.1 | 61.4 | 70.4 |
| | IN-R | 79.4 | 69.8 | 48.8 | - | 58.8 | 48.9 | 72.6 | 63.8 | 65.4 |
| | IN-Sketch | 81.6 | 72.8 | 52.0 | **76.3** | - | 52.5 | 75.6 | 64.7 | 67.6 |
| | ObjNet | 82.3 | 73.2 | 58.2 | 62.0 | 50.4 | - | 74.5 | 56.2 | 68.4 |
| | IN-Cartoon | 96.1 | 72.7 | 52.1 | 63.2 | 50.8 | 50.5 | - | 70.4 | 72.6 |
| | IN-Drawing | 93.8 | 72.2 | 46.4 | 65.4 | 52.5 | 49.9 | 85.1 | - | 74.7 |
| | IN-C | 99.9 | 70.1 | 37.9 | 58.7 | 45.8 | 44.0 | 98.8 | **89.9** | - |
| Linear Probing | IN-V2 | 85.0 | - | 57.1 | 64.1 | 51.2 | 51.9 | 78.4 | 63.5 | 71.4 |
| | IN-A | 84.9 | 49.8 | - | 64.8 | 51.3 | 53.4 | 78.5 | 63.5 | 71.4 |
| | IN-R | 83.9 | 43.5 | 56.3 | - | 50.7 | 51.1 | 76.5 | 61.7 | 69.8 |
| | IN-Sketch | 82.1 | 72.9 | 55.3 | 70.0 | - | 53.2 | 75.6 | 61.3 | 68.4 |
| | ObjNet | 84.8 | 75.0 | 58.2 | 63.3 | 50.9 | - | 78.3 | 63.6 | 71.5 |
| | IN-Cartoon | 92.6 | 74.8 | 56.0 | 63.8 | 50.6 | 52.7 | - | 68.5 | 76.5 |
| | IN-Drawing | 89.1 | 75.1 | 55.0 | 64.3 | 51.2 | 51.7 | 84.1 | - | 76.0 |
| | IN-C | 99.6 | 69.6 | 49.3 | 59.2 | 45.6 | 45.6 | 96.7 | 84.1 | - |
| Visual Prompt (Bahng et al., 2022) | IN-V2 | 80.3 | - | 39.4 | 56.9 | 42.1 | 49.8 | 70.2 | 50.1 | 57.4 |
| | IN-A | 77.2 | 66.8 | - | 52.6 | 37.2 | 51.1 | 64.8 | 43.4 | 50.5 |
| | IN-R | 75.3 | 64.8 | 34.8 | - | 47.4 | 45.2 | 64.9 | 50.7 | 50.3 |
| | IN-Sketch | 73.3 | 62.8 | 25.7 | 63.5 | - | 43.8 | 64.5 | 51.2 | 48.6 |
| | ObjNet | 77.4 | 66.9 | 40.8 | 52.1 | 36.1 | - | 62.8 | 38.8 | 47.7 |
| | IN-Cartoon | 80.2 | 69.3 | 37.9 | 58.9 | 43.1 | 47.5 | - | 52.2 | 55.8 |
| | IN-Drawing | 76.0 | 65.2 | 28.3 | 57.0 | 40.5 | 44.0 | 65.8 | - | 52.2 |
| | IN-C | 81.1 | 70.5 | 42.8 | 54.3 | 41.5 | 51.3 | 69.9 | 51.8 | - |
| LoRA (Hu et al., 2021) | IN-V2 | 85.9 | - | 56.2 | 64.5 | 51.9 | 53.3 | 79.5 | 64.8 | 72.3 |
| | IN-A | 85.9 | **76.6** | - | 65.1 | 52.0 | 55.4 | 79.5 | 65.0 | 72.9 |
| | IN-R | 85.9 | **76.6** | 58.8 | - | 52.5 | 55.2 | 79.5 | 65.3 | 72.6 |
| | IN-Sketch | 85.9 | 76.4 | 58.0 | 67.0 | - | 54.7 | 79.6 | 65.6 | 72.5 |
| | ObjNet | 85.8 | 76.3 | 59.9 | 65.3 | 52.0 | - | 79.3 | 64.9 | 72.9 |
| | IN-Cartoon | 85.9 | 76.3 | 57.7 | 65.0 | 51.6 | 54.4 | - | 64.8 | 72.5 |
| | IN-Drawing | 85.8 | 76.4 | 58.1 | 65.8 | 52.4 | 54.8 | 79.7 | - | 73.0 |
| | IN-C | 86.7 | 76.5 | 58.0 | 65.3 | 52.3 | 55.0 | 80.6 | 69.2 | - |
| EWC (Kirkpatrick et al., 2017) | IN-V2 | 84.3 | - | 52.7 | 66.7 | 51.3 | 51.9 | 77.6 | 65.1 | 71.0 |
| | IN-A | 84.2 | 75.8 | - | 67.9 | 51.8 | **57.0** | 77.0 | 62.4 | 72.0 |
| | IN-R | 84.6 | 75.3 | 62.7 | - | 60.0 | 55.0 | 77.9 | 67.7 | 72.0 |
| | IN-Sketch | 85.2 | 76.0 | 57.0 | 74.3 | - | 54.8 | 79.3 | 67.8 | 72.4 |
| | ObjNet | 83.6 | 74.4 | 61.4 | 64.3 | 51.6 | - | 75.3 | 61.4 | 71.2 |
| | IN-Cartoon | 86.2 | 76.1 | 58.7 | 66.0 | 52.1 | 54.2 | - | 66.9 | 71.8 |
| | IN-Drawing | 86.2 | 76.5 | 57.4 | 67.1 | 53.2 | 55.1 | 80.2 | - | 73.5 |
| | IN-C | 87.4 | 76.4 | 57.5 | 65.8 | 52.5 | 53.1 | 81.7 | 71.2 | - |
| LwF (Li & Hoiem, 2017) | IN-V2 | 84.9 | - | 55.8 | 65.1 | 50.6 | 52.6 | 78.2 | 63.1 | 71.3 |
| | IN-A | 85.2 | 76.0 | - | 66.8 | 51.5 | 54.9 | 78.8 | 64.5 | 72.0 |
| | IN-R | 85.1 | 56.9 | 57.0 | - | 59.8 | 48.0 | 78.9 | 67.5 | 71.4 |
| | IN-Sketch | 83.6 | 74.2 | 53.4 | 74.5 | - | 53.1 | 77.9 | 65.4 | 70.2 |
| | ObjNet | 85.0 | 75.4 | 60.5 | 64.6 | 51.2 | - | 77.8 | 61.7 | 71.5 |
| | IN-Cartoon | 97.2 | 74.0 | 47.3 | 65.0 | 50.4 | 49.9 | - | 71.2 | 75.5 |
| | IN-Drawing | 94.5 | 70.0 | 43.2 | 66.6 | 53.1 | 42.7 | 87.6 | - | 76.6 |
| | IN-C | **99.9** | 72.4 | 41.5 | 63.1 | 50.0 | 47.4 | 98.9 | 89.4 | - |
| LP-FT (Kumar et al., 2022) | IN-V2 | 84.6 | - | 59.9 | 65.8 | 52.1 | 53.6 | 78.0 | 63.6 | 71.4 |
| | IN-A | 67.4 | 58.1 | - | 57.0 | 40.5 | 50.8 | 59.2 | 44.5 | 53.7 |
| | IN-R | 63.1 | 55.1 | 43.9 | - | 40.6 | 46.5 | 55.3 | 47.2 | 49.4 |
| | IN-Sketch | 81.5 | 72.5 | 54.4 | 74.6 | - | 53.7 | 75.6 | 62.7 | 67.8 |
| | ObjNet | 83.9 | 74.1 | 61.1 | 64.2 | 50.7 | - | 76.9 | 60.8 | 70.6 |
| | IN-Cartoon | 96.2 | 74.2 | 55.7 | 64.4 | 51.5 | 52.6 | - | 71.4 | 77.0 |
| | IN-Drawing | 93.4 | 73.7 | 51.2 | 65.2 | 52.5 | 51.2 | 87.1 | - | **77.5** |
| | IN-C | 99.9 | 69.6 | 44.5 | 59.0 | 45.0 | 44.6 | 98.3 | 89.7 | - |
| WiSE-FT (Wortsman et al., 2022b) | IN-V2 | 85.7 | - | 61.9 | 66.3 | 52.2 | 55.9 | 79.3 | 66.0 | 73.5 |
| | IN-A | 84.7 | 75.3 | - | 66.2 | 51.6 | 55.4 | 78.6 | 63.7 | 71.5 |
| | IN-R | 85.1 | 76.1 | 61.5 | - | 60.9 | 55.7 | 79.2 | 69.4 | 73.0 |
| | IN-Sketch | 85.0 | 76.2 | 57.9 | 74.4 | - | 55.2 | 79.5 | 67.8 | 72.1 |
| | ObjNet | 85.3 | 76.2 | 62.5 | 65.4 | 52.8 | - | 78.7 | 63.9 | 72.6 |
| | IN-Cartoon | 92.3 | 76.1 | 59.3 | 65.3 | 52.7 | 54.9 | - | 70.5 | 75.6 |
| | IN-Drawing | 91.2 | 76.0 | 57.1 | 67.0 | 54.4 | 55.1 | 84.8 | - | 76.8 |
| | IN-C | 97.0 | 75.5 | 54.7 | 65.7 | 51.8 | 53.2 | 93.4 | 81.3 | - |
| Model Soup PRE-FT-EWC-LwF (Wortsman et al., 2022a) | IN-V2 | 85.7 | - | 60.8 | 66.5 | 52.2 | 55.2 | 79.4 | 66.1 | 73.4 |
| | IN-A | 85.4 | 76.3 | - | 67.6 | 52.6 | 56.0 | 79.1 | 65.3 | 72.7 |
| | IN-R | 85.3 | 76.4 | 61.7 | - | **61.1** | 55.1 | 79.4 | 69.4 | 73.3 |
| | IN-Sketch | 85.0 | 76.0 | 57.2 | 75.0 | - | 55.0 | 79.5 | 68.0 | 72.3 |
| | ObjNet | 85.4 | 76.3 | **63.0** | 65.5 | 52.7 | - | 78.4 | 63.6 | 72.7 |
| | IN-Cartoon | 92.3 | 76.0 | 57.2 | 65.8 | 52.3 | 53.8 | - | 70.7 | 75.8 |
| | IN-Drawing | 91.2 | 76.4 | 56.5 | 67.3 | 54.4 | 54.5 | 84.9 | - | 77.4 |
| | IN-C | 97.2 | 76.0 | 55.7 | 65.9 | 53.0 | 53.3 | 94.0 | 82.5 | - |

Table 32: The accuracy on each OOD dataset after fine-tuning ImageNet-1K pre-trained ResNet-18 on the downstream datasets with various methods. Note that ImageNet-Drawing, ImageNet-Cartoon, and ImageNet-C are generated from the ImageNet validation set. Green and red indicate relative performance increases and decreases, respectively, compared to the pre-trained model. Bold indicates the best performance on each evaluation dataset.

| Method | Downstream Dataset | $D_{pre}$ IN | IN-V2 | Realistic OOD IN-A | IN-R | IN-Sketch | ObjNet | Synthetic OOD IN-Cartoon | IN-Drawing | IN-C |
|---|---|---|---|---|---|---|---|---|---|---|
| Pre-Trained | | 69.8 | **57.3** | 1.1 | 33.1 | 20.2 | 26.0 | 48.2 | 20.4 | 31.7 |
| FT | IN-V2 | 67.1 | - | 2.2 | 31.0 | 17.7 | 25.1 | 42.7 | 17.2 | 29.9 |
| | IN-A | 50.4 | 40.6 | - | 22.8 | 12.1 | 19.7 | 30.0 | 10.1 | 19.9 |
| | IN-R | 53.1 | 42.6 | 3.4 | - | 37.1 | 21.3 | 43.4 | 27.6 | 25.6 |
| | IN-Sketch | 48.1 | 37.4 | 2.1 | 47.4 | - | 16.5 | 36.9 | 22.9 | 19.2 |
| | ObjNet | 55.2 | 44.3 | 3.7 | 25.3 | 13.7 | - | 32.8 | 9.9 | 20.8 |
| | IN-Cartoon | 75.7 | 45.2 | 2.2 | 28.8 | 17.3 | 20.3 | - | 16.3 | 20.0 |
| | IN-Drawing | 46.4 | 32.3 | 2.1 | 24.0 | 15.2 | 10.8 | 23.1 | - | 15.1 |
| | IN-C | 99.2 | 38.3 | 2.5 | 26.6 | 13.4 | 14.3 | 85.8 | 65.3 | - |
| Linear Probing | IN-V2 | 69.0 | - | 1.1 | 31.6 | 18.8 | 24.6 | 45.4 | 19.7 | 29.8 |
| | IN-A | 66.1 | 53.5 | - | 30.8 | 18.7 | 23.6 | 43.5 | 20.5 | 28.7 |
| | IN-R | 64.3 | 51.7 | 1.2 | - | 23.9 | 21.6 | 48.6 | 23.7 | 25.9 |
| | IN-Sketch | 5.9 | 4.4 | 1.0 | 12.5 | - | 3.4 | 4.7 | 1.9 | 1.7 |
| | ObjNet | 64.7 | 51.8 | 1.7 | 30.0 | 16.6 | - | 44.5 | 15.4 | 25.0 |
| | IN-Cartoon | 63.7 | 48.1 | 1.3 | 30.7 | 18.5 | 20.4 | - | 16.0 | 21.8 |
| | IN-Drawing | 38.2 | 29.0 | 1.5 | 22.8 | 13.5 | 9.7 | 21.2 | - | 11.4 |
| | IN-C | 77.2 | 48.6 | 2.5 | 28.1 | 16.1 | 20.5 | 50.3 | 25.2 | - |
| Visual Prompt (Bahng et al., 2022) | IN-V2 | 60.6 | - | 2.2 | 30.7 | 16.7 | 24.5 | 37.8 | 17.1 | 21.4 |
| | IN-A | 39.5 | 30.1 | - | 19.6 | 8.8 | 18.2 | 19.2 | 5.6 | 7.3 |
| | IN-R | 53.4 | 42.6 | 2.3 | - | 18.3 | 22.6 | 33.8 | 16.3 | 16.6 |
| | IN-Sketch | 54.6 | 42.9 | 2.3 | 32.4 | - | 22.6 | 34.5 | 17.2 | 17.1 |
| | ObjNet | 47.3 | 36.0 | 2.9 | 23.6 | 11.6 | - | 25.6 | 9.1 | 12.9 |
| | IN-Cartoon | 58.2 | 45.4 | 2.2 | 29.8 | 15.7 | 22.8 | - | 13.6 | 17.8 |
| | IN-Drawing | 54.0 | 43.0 | 1.9 | 29.7 | 17.1 | 20.6 | 33.1 | - | 17.3 |
| | IN-C | 61.8 | 50.5 | 2.2 | 31.1 | 18.2 | 25.4 | 38.7 | 20.5 | - |
| EWC (Kirkpatrick et al., 2017) | IN-V2 | 69.7 | - | 1.1 | 32.2 | 19.2 | 25.7 | 45.8 | 20.7 | 31.5 |
| | IN-A | 56.0 | 45.5 | - | 23.5 | 8.5 | 22.1 | 35.4 | 12.6 | 22.0 |
| | IN-R | 64.5 | 52.2 | 1.7 | - | 31.3 | 24.2 | 53.0 | 28.0 | 29.0 |
| | IN-Sketch | 39.5 | 29.5 | 2.0 | 28.2 | - | 13.3 | 31.5 | 17.9 | 13.3 |
| | ObjNet | 63.9 | 51.6 | 2.7 | 29.5 | 15.5 | - | 42.4 | 15.0 | 24.9 |
| | IN-Cartoon | 62.0 | 47.6 | 1.3 | 31.9 | 19.5 | 20.2 | - | 16.3 | 20.7 |
| | IN-Drawing | 36.7 | 29.2 | 1.5 | 24.5 | 15.4 | 9.9 | 19.2 | - | 12.0 |
| | IN-C | 66.2 | 54.4 | 2.3 | 31.2 | 18.5 | **26.8** | 40.4 | 22.0 | - |
| LwF (Li & Hoiem, 2017) | IN-V2 | 68.7 | - | 1.9 | 32.3 | 19.1 | 25.8 | 45.4 | 18.8 | 31.0 |
| | IN-A | 61.4 | 50.3 | - | 28.0 | 16.0 | 23.0 | 38.9 | 15.7 | 26.7 |
| | IN-R | 62.3 | 50.6 | 2.6 | - | 36.2 | 23.8 | 50.0 | 29.6 | 30.3 |
| | IN-Sketch | 54.7 | 43.1 | 1.7 | **47.8** | - | 19.1 | 42.0 | 23.6 | 21.8 |
| | ObjNet | 61.9 | 50.0 | 3.3 | 29.8 | 16.9 | - | 39.3 | 14.1 | 25.6 |
| | IN-Cartoon | 81.6 | 52.5 | 1.8 | 32.2 | 19.5 | 23.7 | - | 21.5 | 29.0 |
| | IN-Drawing | 60.3 | 41.0 | 2.0 | 27.0 | 16.8 | 15.5 | 31.3 | - | 20.8 |
| | IN-C | 97.2 | 47.2 | 1.6 | 31.0 | 18.1 | 18.4 | 83.2 | 53.4 | - |
| LP-FT (Kumar et al., 2022) | IN-V2 | 67.0 | - | 2.3 | 31.0 | 17.7 | 25.1 | 42.8 | 17.0 | 29.8 |
| | IN-A | 54.4 | 44.2 | - | 23.4 | 12.6 | 20.8 | 30.9 | 12.2 | 22.6 |
| | IN-R | 56.2 | 45.5 | 3.3 | - | 37.9 | 21.9 | 46.2 | 29.5 | 27.5 |
| | IN-Sketch | 45.8 | 36.1 | 2.2 | 45.2 | - | 15.5 | 35.9 | 21.6 | 17.5 |
| | ObjNet | 58.1 | 46.8 | 3.6 | 27.0 | 14.8 | - | 36.4 | 11.5 | 22.2 |
| | IN-Cartoon | 76.3 | 45.2 | 2.2 | 28.8 | 17.2 | 20.2 | - | 16.6 | 19.7 |
| | IN-Drawing | 46.4 | 31.5 | 2.1 | 24.0 | 14.8 | 10.8 | 22.6 | - | 14.5 |
| | IN-C | **99.4** | 37.4 | 2.5 | 26.6 | 13.0 | 13.5 | **88.3** | 67.6 | - |
| WiSE-FT (Wortsman et al., 2022b) | IN-V2 | 69.6 | - | 1.6 | 32.9 | 19.9 | 26.3 | 47.0 | 19.8 | 32.3 |
| | IN-A | 66.5 | 54.7 | - | 31.5 | 18.9 | 26.5 | 45.5 | 18.4 | 30.7 |
| | IN-R | 66.2 | 54.2 | 1.9 | - | 33.9 | 25.7 | 54.0 | 31.6 | **33.6** |
| | IN-Sketch | 64.9 | 52.3 | 1.5 | 46.6 | - | 24.1 | 50.3 | 30.2 | 29.8 |
| | ObjNet | 67.2 | 54.9 | 2.3 | 32.6 | 19.3 | - | 44.9 | 17.6 | 30.4 |
| | IN-Cartoon | 76.5 | 54.4 | 1.5 | 33.5 | 20.5 | 25.0 | - | 21.2 | 28.8 |
| | IN-Drawing | 68.6 | 51.4 | 1.6 | 32.8 | 20.8 | 21.1 | 41.9 | - | 28.8 |
| | IN-C | 86.0 | 52.9 | 1.9 | 35.4 | 20.9 | 24.2 | 68.0 | 40.0 | - |
| Model Soup PRE-FT-EWC-LwF (Wortsman et al., 2022a) | IN-V2 | 69.6 | - | 1.6 | 32.8 | 19.7 | 26.2 | 46.7 | 20.1 | 32.2 |
| | IN-A | 65.1 | 53.5 | - | 30.7 | 17.6 | 25.8 | 44.3 | 16.7 | 29.2 |
| | IN-R | 65.9 | 54.0 | 2.1 | - | 34.7 | 25.4 | 53.8 | 31.6 | 32.9 |
| | IN-Sketch | 63.2 | 50.9 | 1.5 | 47.4 | - | 23.0 | 49.7 | 30.6 | 28.2 |
| | ObjNet | 66.3 | 54.2 | 2.6 | 32.3 | 18.6 | - | 44.4 | 17.0 | 29.3 |
| | IN-Cartoon | 74.8 | 53.1 | 1.5 | 33.6 | 20.6 | 24.3 | - | 20.9 | 27.5 |
| | IN-Drawing | 62.6 | 46.8 | 1.7 | 31.1 | 19.8 | 18.0 | 35.7 | - | 24.2 |
| | IN-C | 84.3 | 53.9 | 1.7 | 35.5 | 21.2 | 25.0 | 64.9 | 38.5 | - |

Table 33: The accuracy on each OOD dataset after fine-tuning ImageNet-1K pre-trained ResNet-50 on the downstream datasets with various methods. Note that ImageNet-Drawing, ImageNet-Cartoon, and ImageNet-C are generated from the ImageNet validation set. Green and red indicate relative performance increases and decreases, respectively, compared to the pre-trained model. Bold indicates the best performance on each evaluation dataset.

| Method | Downstream Dataset | $D_{pre}$ IN | Realistic OOD IN-V2 | IN-A | IN-R | IN-Sketch | ObjNet | Synthetic OOD IN-Cartoon | IN-Drawing | IN-C |
|---|---|---|---|---|---|---|---|---|---|---|
| Pre-Trained | | 80.3 | **69.5** | 16.7 | 41.6 | 28.4 | 42.7 | 61.1 | 31.1 | 46.6 |
| FT | IN-V2 | 79.6 | - | 18.2 | 42.4 | 28.7 | 41.8 | 58.1 | 31.0 | 47.0 |
| | IN-A | 75.7 | 65.2 | - | 40.6 | 28.0 | 42.6 | 52.4 | 25.8 | 46.3 |
| | IN-R | 72.5 | 61.7 | 16.5 | - | 49.1 | 37.2 | 61.7 | 46.0 | 43.7 |
| | IN-Sketch | 57.3 | 45.7 | 5.5 | 55.0 | - | 25.6 | 48.9 | 30.3 | 23.3 |
| | ObjNet | 75.3 | 63.5 | 22.0 | 38.5 | 23.3 | - | 53.0 | 22.7 | 41.8 |
| | IN-Cartoon | 81.4 | 58.8 | 12.9 | 39.6 | 28.0 | 34.6 | - | 25.1 | 31.9 |
| | IN-Drawing | 42.9 | 32.6 | 4.9 | 30.3 | 24.2 | 11.9 | 30.0 | - | 16.3 |
| | IN-C | 99.7 | 56.5 | 6.6 | 38.2 | 23.1 | 28.3 | 89.3 | 66.0 | - |
| Linear Probing | IN-V2 | 79.6 | - | 12.4 | 41.0 | 27.5 | 40.1 | 56.5 | 34.4 | 45.6 |
| | IN-A | 78.2 | 66.8 | - | 40.8 | 28.0 | 41.1 | 55.0 | 35.2 | 45.4 |
| | IN-R | 76.2 | 64.5 | 16.7 | - | 31.5 | 39.2 | 60.0 | 35.2 | 40.6 |
| | IN-Sketch | 10.5 | 8.3 | 1.7 | 15.9 | - | 7.1 | 9.5 | 4.9 | 2.8 |
| | ObjNet | 76.1 | 63.9 | 17.0 | 39.4 | 25.3 | - | 55.2 | 25.7 | 39.4 |
| | IN-Cartoon | 73.7 | 60.3 | 11.5 | 39.9 | 26.0 | 32.4 | - | 31.6 | 35.6 |
| | IN-Drawing | 10.4 | 7.9 | 1.8 | 15.5 | 17.7 | 2.9 | 5.3 | - | 8.8 |
| | IN-C | 82.5 | 62.5 | 16.3 | 35.3 | 23.6 | 38.1 | 56.1 | 31.6 | - |
| Visual Prompt (Bahng et al., 2022) | IN-V2 | 75.4 | - | 11.9 | 37.5 | 24.2 | 39.9 | 51.8 | 24.9 | 36.4 |
| | IN-A | 73.1 | 61.3 | - | 37.0 | 23.4 | 41.3 | 48.8 | 21.9 | 32.8 |
| | IN-R | 73.2 | 61.1 | 13.1 | - | 28.9 | 38.5 | 51.5 | 28.0 | 32.8 |
| | IN-Sketch | 73.8 | 61.5 | 12.3 | 43.4 | - | 39.0 | 50.9 | 28.1 | 33.0 |
| | ObjNet | 72.7 | 60.5 | 13.5 | 34.6 | 22.9 | - | 47.2 | 19.9 | 33.1 |
| | IN-Cartoon | 74.5 | 62.3 | 11.3 | 38.1 | 24.3 | 38.1 | - | 24.7 | 34.5 |
| | IN-Drawing | 74.2 | 62.3 | 11.3 | 39.6 | 26.4 | 37.9 | 52.3 | - | 35.1 |
| | IN-C | 75.0 | 63.4 | 11.9 | 37.0 | 23.7 | 39.8 | 51.8 | 28.1 | - |
| EWC (Kirkpatrick et al., 2017) | IN-V2 | 80.2 | - | 13.6 | 41.5 | 28.3 | 41.3 | 58.0 | 31.6 | 46.0 |
| | IN-A | 78.3 | 67.5 | - | 42.9 | 28.8 | 43.1 | 56.7 | 31.9 | 46.8 |
| | IN-R | 77.0 | 65.4 | 17.4 | - | 40.3 | 40.2 | 64.8 | 40.3 | 42.9 |
| | IN-Sketch | 40.9 | 32.1 | 4.0 | 32.7 | - | 17.8 | 36.0 | 21.4 | 13.2 |
| | ObjNet | 77.1 | 65.5 | 18.7 | 40.7 | 25.7 | - | 56.8 | 24.0 | 41.3 |
| | IN-Cartoon | 72.1 | 58.8 | 11.4 | 39.1 | 25.5 | 33.2 | - | 25.4 | 33.1 |
| | IN-Drawing | 8.1 | 6.8 | 1.5 | 14.4 | 17.2 | 2.8 | 3.1 | - | 7.5 |
| | IN-C | 76.1 | 64.9 | 18.0 | 38.0 | 25.3 | 40.6 | 51.1 | 30.2 | - |
| LwF (Li & Hoiem, 2017) | IN-V2 | 79.7 | - | 19.4 | 43.1 | 29.0 | 42.3 | 58.7 | 31.8 | 47.5 |
| | IN-A | 76.4 | 65.8 | - | 41.3 | 28.6 | 42.8 | 53.5 | 26.6 | 46.8 |
| | IN-R | 73.7 | 63.0 | 17.0 | - | 49.3 | 38.0 | 62.8 | 46.6 | 44.9 |
| | IN-Sketch | 54.3 | 43.1 | 5.5 | 51.8 | - | 24.5 | 47.5 | 29.7 | 21.4 |
| | ObjNet | 76.7 | 65.3 | 21.9 | 39.7 | 24.6 | - | 55.3 | 24.1 | 43.1 |
| | IN-Cartoon | 82.4 | 60.4 | 14.5 | 41.0 | 29.1 | 35.8 | - | 26.9 | 34.6 |
| | IN-Drawing | 17.5 | 13.8 | 2.7 | 21.2 | 22.4 | 5.7 | 8.2 | - | 12.3 |
| | IN-C | 99.7 | 58.4 | 5.9 | 38.9 | 23.5 | 29.7 | 90.9 | 66.5 | - |
| LP-FT (Kumar et al., 2022) | IN-V2 | 79.6 | - | 17.7 | 42.3 | 28.5 | 41.4 | 58.3 | 31.6 | 47.1 |
| | IN-A | 76.3 | 65.8 | - | 40.2 | 28.1 | 43.0 | 52.6 | 26.8 | 46.4 |
| | IN-R | 73.4 | 62.3 | 16.3 | - | 48.9 | 37.9 | 62.5 | 46.6 | 44.2 |
| | IN-Sketch | 57.6 | 46.0 | 5.1 | 55.5 | - | 25.9 | 49.1 | 31.6 | 24.2 |
| | ObjNet | 75.5 | 63.6 | 21.5 | 39.1 | 23.7 | - | 54.0 | 22.9 | 41.8 |
| | IN-Cartoon | 81.7 | 59.0 | 12.8 | 39.6 | 28.1 | 34.5 | - | 25.8 | 32.2 |
| | IN-Drawing | 46.4 | 35.2 | 5.1 | 29.9 | 22.9 | 13.4 | 28.4 | - | 16.7 |
| | IN-C | 99.6 | 56.9 | 6.3 | 37.8 | 22.9 | 28.4 | 88.7 | 63.7 | - |
| WiSE-FT (Wortsman et al., 2022b) | IN-V2 | 80.7 | - | 17.3 | 42.5 | 29.2 | 42.6 | 60.9 | 32.3 | 48.1 |
| | IN-A | 80.1 | 69.5 | - | 43.0 | 30.1 | **44.5** | 60.1 | 31.6 | 48.9 |
| | IN-R | 79.2 | 68.5 | 18.6 | - | 45.2 | 42.5 | 67.7 | 47.1 | **49.3** |
| | IN-Sketch | 76.6 | 65.3 | 9.9 | 56.7 | - | 38.3 | 63.4 | 42.6 | 41.5 |
| | ObjNet | 79.7 | 68.4 | 20.5 | 42.0 | 27.4 | - | 60.1 | 29.5 | 46.8 |
| | IN-Cartoon | 83.8 | 67.5 | 16.4 | 43.3 | 29.7 | 40.9 | - | 31.8 | 42.8 |
| | IN-Drawing | 78.8 | 64.6 | 12.4 | 42.6 | 30.5 | 35.0 | 58.6 | - | 41.8 |
| | IN-C | 91.4 | 67.4 | 11.3 | 44.2 | 29.4 | 39.0 | 79.1 | 50.8 | - |
| Model Soup PRE-FT-EWC-LwF (Wortsman et al., 2022a) | IN-V2 | 80.6 | - | 16.8 | 42.7 | 29.4 | 42.3 | 60.1 | 32.2 | 48.0 |
| | IN-A | 79.5 | 68.6 | - | 43.1 | 30.0 | 44.2 | 58.8 | 31.2 | 48.9 |
| | IN-R | 78.4 | 67.7 | 18.9 | - | 47.2 | 41.8 | 67.3 | 47.6 | 48.4 |
| | IN-Sketch | 74.0 | 62.7 | 8.9 | **58.4** | - | 36.0 | 60.9 | 45.0 | 38.7 |
| | ObjNet | 79.1 | 67.7 | 21.1 | 41.8 | 27.1 | - | 59.3 | 27.9 | 45.7 |
| | IN-Cartoon | 82.1 | 65.2 | 15.0 | 42.9 | 29.5 | 39.6 | - | 30.0 | 39.5 |
| | IN-Drawing | 62.2 | 49.5 | 8.9 | 36.8 | 28.6 | 20.5 | 41.7 | - | 28.9 |
| | IN-C | 91.2 | 67.3 | 12.1 | 44.3 | 29.6 | 39.5 | 78.4 | 52.1 | - |

Table 34: Accuracy of ImageNet-1K with AugReg pre-trained ViT-B/16 with different fine-tuning methods and downstream datasets on each ImageNet-C corruption. For each corruption, accuracy is averaged across 5 levels of severity.

| Dataset | Method | Avg. | Noise | | | Blur | | | | Weather | | | | Digital | | | |
|---|---|---|---|---|---|---|---|---|---|---|---|---|---|---|---|---|---|
| | | | Gauss. | Shot | Impulse | Defocus | Glass | Motion | Zoom | Snow | Frost | Fog | Bright | Contrast | Elastic | Pixel | JPEG |
| Pre-Trained | | 56.0 | 57 | 54 | 54 | 49 | 42 | 53 | 46 | 48 | 55 | 61 | 74 | 56 | 59 | 67 | 66 |
| FT | IN-V2 | 57.4 | 56 | 54 | 53 | 51 | 40 | 55 | 46 | 53 | 59 | 65 | 74 | 59 | 58 | 68 | 67 |
| | IN-A | 53.5 | 53 | 51 | 50 | 50 | 38 | 52 | 39 | 50 | 56 | 57 | 70 | 56 | 51 | 65 | 64 |
| | IN-R | 52.0 | 52 | 50 | 49 | 46 | 44 | 49 | 37 | 49 | 55 | 57 | 66 | 53 | 51 | 62 | 61 |
| | IN-Sketch | 53.8 | 55 | 53 | 52 | 46 | 39 | 49 | 43 | 51 | 56 | 58 | 70 | 55 | 55 | 63 | 62 |
| | ObjNet | 52.3 | 52 | 48 | 48 | 46 | 37 | 51 | 38 | 50 | 55 | 58 | 70 | 51 | 52 | 64 | 63 |
| | IN-Cartoon | 51.3 | 53 | 50 | 50 | 44 | 35 | 48 | 35 | 48 | 50 | 54 | 74 | 53 | 53 | 65 | 58 |
| | IN-Drawing | 56.0 | 58 | 56 | 55 | 46 | 43 | 52 | 40 | 55 | 62 | 61 | 74 | 53 | 57 | 66 | 62 |
| Linear Probing | IN-V2 | 55.9 | 56 | 54 | 54 | 49 | 42 | 53 | 46 | 48 | 55 | 61 | 73 | 56 | 59 | 66 | 65 |
| | IN-A | 55.8 | 56 | 53 | 53 | 49 | 42 | 54 | 46 | 48 | 55 | 61 | 73 | 57 | 59 | 66 | 65 |
| | IN-R | 56.2 | 56 | 54 | 54 | 49 | 44 | 54 | 47 | 49 | 55 | 61 | 73 | 56 | 60 | 66 | 66 |
| | IN-Sketch | 54.5 | 54 | 52 | 52 | 48 | 41 | 51 | 45 | 48 | 54 | 59 | 72 | 55 | 58 | 65 | 64 |
| | ObjNet | 56.1 | 56 | 54 | 54 | 49 | 43 | 54 | 48 | 48 | 56 | 62 | 73 | 53 | 60 | 66 | 65 |
| | IN-Cartoon | 55.6 | 56 | 54 | 53 | 48 | 42 | 52 | 46 | 48 | 55 | 59 | 75 | 54 | 59 | 67 | 67 |
| | IN-Drawing | 54.3 | 57 | 55 | 55 | 43 | 43 | 50 | 44 | 51 | 61 | 49 | 74 | 39 | 59 | 66 | 67 |
| Visual Prompt (Bahng et al., 2022) | IN-V2 | 47.9 | 44 | 42 | 41 | 41 | 35 | 46 | 42 | 42 | 46 | 51 | 69 | 48 | 55 | 59 | 57 |
| | IN-A | 38.0 | 33 | 31 | 29 | 31 | 24 | 36 | 31 | 35 | 38 | 43 | 60 | 37 | 46 | 48 | 49 |
| | IN-R | 40.1 | 39 | 38 | 36 | 33 | 28 | 36 | 30 | 36 | 41 | 41 | 61 | 38 | 45 | 50 | 50 |
| | IN-Sketch | 44.3 | 43 | 41 | 40 | 37 | 29 | 40 | 36 | 39 | 45 | 46 | 65 | 47 | 49 | 54 | 55 |
| | ObjNet | 35.3 | 28 | 26 | 24 | 28 | 22 | 33 | 29 | 32 | 35 | 41 | 61 | 37 | 44 | 45 | 44 |
| | IN-Cartoon | 41.8 | 39 | 37 | 36 | 34 | 27 | 38 | 33 | 36 | 38 | 42 | 66 | 43 | 50 | 55 | 53 |
| | IN-Drawing | 44.2 | 45 | 43 | 43 | 33 | 32 | 38 | 32 | 41 | 51 | 42 | 65 | 39 | 50 | 56 | 52 |
| LoRA (Hu et al., 2021) | IN-V2 | 56.1 | 57 | 54 | 54 | 49 | 43 | 53 | 46 | 48 | 55 | 61 | 74 | 57 | 59 | 67 | 66 |
| | IN-A | 56.5 | 57 | 54 | 54 | 49 | 44 | 55 | 48 | 49 | 57 | 61 | 74 | 52 | 60 | 67 | 66 |
| | IN-R | 56.7 | 57 | 54 | 54 | 50 | 44 | 54 | 48 | 49 | 56 | 62 | 74 | 56 | 60 | 67 | 66 |
| | IN-Sketch | 56.6 | 56 | 54 | 54 | 51 | 43 | 53 | 47 | 50 | 56 | 62 | 74 | 57 | 59 | 67 | 66 |
| | ObjNet | 55.0 | 57 | 54 | 54 | 48 | 43 | 54 | 47 | 48 | 55 | 55 | 74 | 44 | 60 | 67 | 66 |
| | IN-Cartoon | 54.6 | 56 | 53 | 53 | 48 | 43 | 50 | 45 | 48 | 54 | 56 | 73 | 50 | 58 | 66 | 65 |
| | IN-Drawing | 55.1 | 58 | 56 | 56 | 44 | 45 | 51 | 43 | 51 | 63 | 54 | 74 | 43 | 59 | 66 | 66 |
| EWC (Kirkpatrick et al., 2017) | IN-V2 | 58.2 | 58 | 55 | 55 | 52 | 44 | 56 | 49 | 52 | 58 | 64 | 75 | 59 | 61 | 68 | 67 |
| | IN-A | 56.6 | 55 | 53 | 52 | 52 | 42 | 56 | 46 | 52 | 58 | 62 | 73 | 59 | 57 | 67 | 66 |
| | IN-R | 56.1 | 55 | 54 | 53 | 50 | 44 | 53 | 43 | 53 | 59 | 62 | 72 | 58 | 56 | 64 | 65 |
| | IN-Sketch | 57.2 | 57 | 56 | 55 | 50 | 44 | 54 | 47 | 52 | 57 | 61 | 74 | 57 | 59 | 67 | 67 |
| | ObjNet | 56.9 | 56 | 53 | 53 | 51 | 43 | 56 | 47 | 52 | 58 | 62 | 74 | 58 | 59 | 67 | 66 |
| | IN-Cartoon | 54.7 | 55 | 52 | 52 | 48 | 40 | 52 | 43 | 48 | 54 | 60 | 73 | 56 | 58 | 66 | 64 |
| | IN-Drawing | 58.3 | 59 | 57 | 57 | 50 | 44 | 55 | 45 | 54 | 63 | 65 | 74 | 59 | 60 | 68 | 66 |
| LwF (Li & Hoiem, 2017) | IN-V2 | 57.9 | 57 | 55 | 54 | 51 | 42 | 55 | 47 | 53 | 59 | 65 | 75 | 60 | 59 | 69 | 68 |
| | IN-A | 57.2 | 56 | 54 | 54 | 52 | 42 | 55 | 45 | 53 | 60 | 62 | 73 | 59 | 57 | 68 | 66 |
| | IN-R | 57.2 | 57 | 56 | 55 | 50 | 48 | 54 | 43 | 54 | 59 | 62 | 72 | 57 | 57 | 67 | 66 |
| | IN-Sketch | 55.2 | 56 | 54 | 53 | 48 | 40 | 51 | 45 | 52 | 57 | 60 | 72 | 56 | 57 | 65 | 64 |
| | ObjNet | 56.3 | 56 | 53 | 53 | 51 | 41 | 55 | 44 | 52 | 57 | 63 | 73 | 57 | 57 | 67 | 66 |
| | IN-Cartoon | 55.6 | 56 | 53 | 53 | 49 | 40 | 52 | 41 | 51 | 55 | 59 | 77 | 57 | 58 | 68 | 65 |
| | IN-Drawing | 58.2 | 59 | 56 | 56 | 50 | 45 | 55 | 43 | 55 | 63 | 64 | 77 | 56 | 59 | 69 | 65 |
| LP-FT (Kumar et al., 2022) | IN-V2 | 57.6 | 57 | 54 | 54 | 51 | 41 | 55 | 46 | 53 | 59 | 65 | 74 | 60 | 59 | 68 | 67 |
| | IN-A | 56.2 | 55 | 52 | 52 | 51 | 41 | 55 | 43 | 53 | 59 | 62 | 73 | 59 | 56 | 67 | 65 |
| | IN-R | 55.3 | 55 | 54 | 52 | 48 | 47 | 52 | 41 | 52 | 58 | 60 | 70 | 56 | 56 | 65 | 64 |
| | IN-Sketch | 54.4 | 54 | 53 | 52 | 48 | 40 | 50 | 44 | 51 | 55 | 59 | 70 | 56 | 56 | 64 | 63 |
| | ObjNet | 54.9 | 54 | 51 | 51 | 48 | 40 | 54 | 43 | 51 | 57 | 61 | 72 | 54 | 56 | 66 | 64 |
| | IN-Cartoon | 52.8 | 53 | 50 | 50 | 46 | 37 | 49 | 38 | 49 | 52 | 55 | 75 | 54 | 55 | 66 | 61 |
| | IN-Drawing | 56.0 | 59 | 56 | 56 | 44 | 44 | 52 | 40 | 56 | 63 | 57 | 76 | 49 | 58 | 67 | 64 |
| WiSE-FT (Wortsman et al., 2022b) | IN-V2 | 58.0 | 58 | 55 | 55 | 51 | 42 | 55 | 47 | 52 | 58 | 65 | 75 | 60 | 60 | 69 | 68 |
| | IN-A | 57.8 | 57 | 55 | 55 | 52 | 43 | 56 | 46 | 53 | 59 | 64 | 74 | 60 | 59 | 68 | 66 |
| | IN-R | 59.6 | 59 | 58 | 57 | 53 | 49 | 57 | 48 | 55 | 61 | 65 | 75 | 60 | 61 | 69 | 68 |
| | IN-Sketch | 57.3 | 58 | 56 | 56 | 50 | 42 | 53 | 47 | 53 | 59 | 63 | 74 | 59 | 59 | 67 | 66 |
| | ObjNet | 57.6 | 57 | 54 | 54 | 51 | 43 | 56 | 46 | 53 | 58 | 64 | 74 | 59 | 59 | 68 | 67 |
| | IN-Cartoon | 56.3 | 57 | 54 | 55 | 50 | 41 | 53 | 43 | 51 | 55 | 61 | 76 | 58 | 59 | 68 | 65 |
| | IN-Drawing | 59.5 | 61 | 59 | 59 | 51 | 45 | 56 | 46 | 55 | 63 | 65 | 77 | 59 | 61 | 69 | 67 |
| Model Soup PRE-FT-EWC-LwF (Wortsman et al., 2022a) | IN-V2 | 58.0 | 58 | 55 | 55 | 51 | 43 | 55 | 47 | 52 | 59 | 64 | 75 | 60 | 60 | 69 | 68 |
| | IN-A | 57.8 | 57 | 55 | 54 | 52 | 43 | 56 | 46 | 53 | 59 | 64 | 74 | 60 | 58 | 68 | 67 |
| | IN-R | 59.6 | 59 | 58 | 57 | 53 | 49 | 57 | 47 | 55 | 61 | 65 | 74 | 60 | 61 | 69 | 68 |
| | IN-Sketch | 57.5 | 58 | 56 | 56 | 50 | 42 | 53 | 47 | 53 | 59 | 63 | 74 | 59 | 59 | 67 | 66 |
| | ObjNet | 57.7 | 57 | 54 | 54 | 52 | 43 | 56 | 47 | 53 | 58 | 64 | 74 | 59 | 59 | 68 | 67 |
| | IN-Cartoon | 56.2 | 57 | 54 | 54 | 50 | 41 | 53 | 43 | 51 | 55 | 61 | 76 | 58 | 59 | 68 | 65 |
| | IN-Drawing | 59.7 | 61 | 59 | 59 | 51 | 45 | 56 | 46 | 55 | 63 | 66 | 77 | 59 | 61 | 69 | 67 |

Table 35: Accuracy of ImageNet-1K with SAM pre-trained ViT-B/16 with different fine-tuning methods and downstream datasets on each ImageNet-C corruption. For each corruption, accuracy is averaged across 5 levels of severity.

| Dataset | Method | Avg. | Noise | | | Blur | | | | Weather | | | | Digital | | | |
|---|---|---|---|---|---|---|---|---|---|---|---|---|---|---|---|---|---|
| | | | Gauss. | Shot | Impulse | Defocus | Glass | Motion | Zoom | Snow | Frost | Fog | Bright | Contrast | Elastic | Pixel | JPEG |
| Pre-Trained | | 54.6 | 53 | 50 | 51 | 50 | 48 | 55 | 47 | 47 | 51 | 51 | 73 | 46 | 64 | 68 | 67 |
| FT | IN-V2 | 56.8 | 51 | 49 | 48 | 54 | 49 | 59 | 50 | 51 | 53 | 59 | 75 | 52 | 65 | 69 | 68 |
| | IN-A | 51.3 | 42 | 39 | 36 | 56 | 45 | 56 | 43 | 46 | 46 | 56 | 71 | 52 | 58 | 61 | 61 |
| | IN-R | 53.9 | 49 | 48 | 47 | 53 | 51 | 54 | 42 | 50 | 54 | 55 | 70 | 52 | 58 | 62 | 64 |
| | IN-Sketch | 55.7 | 53 | 52 | 51 | 52 | 46 | 55 | 49 | 50 | 52 | 59 | 72 | 52 | 60 | 66 | 66 |
| | ObjNet | 52.2 | 43 | 41 | 38 | 56 | 46 | 55 | 47 | 47 | 48 | 54 | 71 | 49 | 60 | 64 | 64 |
| | IN-Cartoon | 51.3 | 44 | 42 | 39 | 50 | 40 | 53 | 42 | 46 | 45 | 54 | 74 | 52 | 59 | 66 | 65 |
| | IN-Drawing | 55.4 | 54 | 53 | 52 | 47 | 48 | 54 | 43 | 51 | 59 | 55 | 74 | 47 | 61 | 67 | 66 |
| HeadOnly | IN-V2 | 54.7 | 52 | 50 | 50 | 50 | 49 | 55 | 47 | 47 | 51 | 52 | 73 | 47 | 64 | 68 | 67 |
| | IN-A | 54.5 | 52 | 50 | 49 | 49 | 48 | 55 | 47 | 47 | 51 | 51 | 73 | 47 | 64 | 68 | 67 |
| | IN-R | 54.6 | 52 | 50 | 49 | 49 | 49 | 54 | 47 | 47 | 51 | 51 | 73 | 46 | 64 | 68 | 67 |
| | IN-Sketch | 54.1 | 52 | 50 | 50 | 49 | 48 | 53 | 47 | 48 | 51 | 50 | 72 | 46 | 63 | 67 | 66 |
| | ObjNet | 54.6 | 52 | 50 | 50 | 50 | 48 | 54 | 47 | 47 | 51 | 52 | 73 | 47 | 64 | 68 | 67 |
| | IN-Cartoon | 53.8 | 52 | 50 | 49 | 49 | 48 | 54 | 46 | 46 | 50 | 46 | 73 | 43 | 64 | 69 | 67 |
| | IN-Drawing | 54.6 | 53 | 51 | 51 | 48 | 49 | 53 | 46 | 48 | 53 | 50 | 73 | 46 | 64 | 68 | 67 |
| Visual Prompt (Bahng et al., 2022) | IN-V2 | 44.7 | 42 | 41 | 40 | 39 | 37 | 42 | 39 | 36 | 40 | 38 | 66 | 34 | 57 | 58 | 61 |
| | IN-A | 30.4 | 26 | 25 | 22 | 22 | 24 | 27 | 25 | 26 | 29 | 26 | 53 | 22 | 43 | 41 | 44 |
| | IN-R | 36.8 | 38 | 37 | 36 | 27 | 28 | 31 | 28 | 31 | 35 | 26 | 60 | 22 | 48 | 51 | 56 |
| | IN-Sketch | 36.3 | 36 | 35 | 34 | 27 | 27 | 30 | 27 | 33 | 36 | 24 | 61 | 21 | 48 | 49 | 57 |
| | ObjNet | 35.3 | 31 | 30 | 27 | 29 | 29 | 32 | 31 | 28 | 32 | 30 | 59 | 25 | 49 | 46 | 50 |
| | IN-Cartoon | 42.3 | 43 | 41 | 40 | 34 | 36 | 39 | 38 | 35 | 39 | 25 | 67 | 23 | 54 | 62 | 62 |
| | IN-Drawing | 42.2 | 46 | 45 | 44 | 31 | 37 | 36 | 34 | 39 | 46 | 18 | 64 | 16 | 55 | 61 | 62 |
| LoRA (Hu et al., 2021) | IN-V2 | 54.7 | 52 | 50 | 50 | 50 | 49 | 55 | 47 | 47 | 51 | 52 | 73 | 47 | 64 | 68 | 67 |
| | IN-A | 54.8 | 52 | 50 | 50 | 50 | 49 | 55 | 47 | 47 | 51 | 52 | 73 | 47 | 64 | 68 | 67 |
| | IN-R | 54.7 | 52 | 50 | 50 | 50 | 49 | 55 | 47 | 47 | 51 | 51 | 73 | 46 | 64 | 68 | 67 |
| | IN-Sketch | 54.6 | 52 | 50 | 50 | 50 | 48 | 54 | 47 | 48 | 51 | 51 | 73 | 47 | 64 | 68 | 67 |
| | ObjNet | 54.7 | 52 | 50 | 50 | 50 | 48 | 55 | 47 | 47 | 51 | 52 | 73 | 47 | 64 | 68 | 67 |
| | IN-Cartoon | 53.7 | 52 | 50 | 50 | 49 | 49 | 54 | 47 | 46 | 51 | 44 | 73 | 42 | 64 | 69 | 67 |
| | IN-Drawing | 54.5 | 53 | 51 | 50 | 49 | 49 | 53 | 47 | 48 | 53 | 48 | 73 | 45 | 64 | 68 | 67 |
| EWC (Kirkpatrick et al., 2017) | IN-V2 | 55.1 | 52 | 50 | 50 | 50 | 49 | 55 | 48 | 48 | 51 | 53 | 74 | 47 | 64 | 68 | 67 |
| | IN-A | 54.7 | 48 | 46 | 44 | 55 | 50 | 57 | 48 | 49 | 52 | 54 | 73 | 48 | 64 | 66 | 67 |
| | IN-R | 56.3 | 53 | 51 | 50 | 53 | 50 | 57 | 48 | 49 | 53 | 55 | 74 | 50 | 65 | 68 | 69 |
| | IN-Sketch | 55.6 | 53 | 52 | 51 | 51 | 50 | 55 | 47 | 48 | 52 | 52 | 73 | 48 | 64 | 69 | 68 |
| | ObjNet | 56.5 | 51 | 49 | 48 | 55 | 51 | 58 | 51 | 49 | 52 | 57 | 74 | 50 | 65 | 68 | 69 |
| | IN-Cartoon | 54.3 | 51 | 49 | 48 | 50 | 47 | 54 | 46 | 47 | 50 | 51 | 74 | 48 | 63 | 69 | 67 |
| | IN-Drawing | 56.9 | 55 | 54 | 54 | 51 | 50 | 56 | 47 | 50 | 56 | 55 | 74 | 50 | 65 | 69 | 68 |
| LwF (Li & Hoiem, 2017) | IN-V2 | 56.8 | 52 | 50 | 49 | 54 | 49 | 58 | 50 | 51 | 53 | 58 | 75 | 52 | 65 | 69 | 68 |
| | IN-A | 53.9 | 45 | 42 | 40 | 57 | 48 | 58 | 47 | 49 | 49 | 58 | 73 | 54 | 61 | 64 | 65 |
| | IN-R | 56.0 | 51 | 51 | 49 | 54 | 52 | 56 | 45 | 52 | 55 | 58 | 72 | 53 | 61 | 65 | 66 |
| | IN-Sketch | 56.3 | 53 | 52 | 51 | 52 | 47 | 56 | 49 | 51 | 53 | 59 | 73 | 53 | 61 | 67 | 67 |
| | ObjNet | 55.0 | 47 | 45 | 43 | 57 | 48 | 58 | 50 | 49 | 50 | 57 | 73 | 52 | 63 | 67 | 66 |
| | IN-Cartoon | 53.1 | 46 | 44 | 42 | 51 | 43 | 54 | 44 | 47 | 47 | 55 | 73 | 53 | 61 | 67 | 66 |
| | IN-Drawing | 56.2 | 55 | 53 | 53 | 48 | 48 | 55 | 44 | 52 | 60 | 56 | 74 | 48 | 62 | 68 | 67 |
| LP-FT (Kumar et al., 2022) | IN-V2 | 56.8 | 51 | 49 | 48 | 54 | 49 | 59 | 50 | 51 | 53 | 58 | 75 | 52 | 65 | 69 | 68 |
| | IN-A | 52.3 | 43 | 40 | 37 | 57 | 46 | 57 | 45 | 47 | 47 | 57 | 71 | 53 | 59 | 62 | 63 |
| | IN-R | 54.5 | 50 | 50 | 48 | 53 | 52 | 55 | 43 | 51 | 55 | 55 | 71 | 48 | 60 | 63 | 65 |
| | IN-Sketch | 55.6 | 53 | 52 | 51 | 52 | 47 | 55 | 48 | 50 | 52 | 58 | 72 | 52 | 60 | 66 | 66 |
| | ObjNet | 53.0 | 44 | 42 | 39 | 56 | 47 | 56 | 47 | 48 | 49 | 54 | 72 | 49 | 61 | 64 | 65 |
| | IN-Cartoon | 51.1 | 44 | 42 | 40 | 49 | 40 | 52 | 41 | 46 | 46 | 52 | 74 | 51 | 59 | 66 | 65 |
| | IN-Drawing | 53.8 | 54 | 52 | 51 | 44 | 48 | 52 | 43 | 51 | 59 | 48 | 73 | 38 | 61 | 67 | 65 |
| WiSE-FT (Wortsman et al., 2022b) | IN-V2 | 56.1 | 52 | 50 | 50 | 52 | 49 | 57 | 49 | 50 | 52 | 56 | 75 | 50 | 65 | 69 | 68 |
| | IN-A | 57.0 | 51 | 49 | 47 | 54 | 50 | 59 | 50 | 52 | 53 | 59 | 75 | 53 | 65 | 69 | 68 |
| | IN-R | 58.7 | 56 | 55 | 54 | 54 | 52 | 58 | 49 | 53 | 56 | 60 | 75 | 54 | 65 | 70 | 69 |
| | IN-Sketch | 56.8 | 55 | 53 | 53 | 52 | 48 | 56 | 49 | 52 | 54 | 57 | 74 | 51 | 64 | 68 | 68 |
| | ObjNet | 56.4 | 51 | 49 | 48 | 54 | 49 | 58 | 50 | 51 | 52 | 58 | 74 | 51 | 65 | 69 | 68 |
| | IN-Cartoon | 55.2 | 50 | 48 | 47 | 51 | 46 | 56 | 47 | 49 | 50 | 56 | 75 | 52 | 64 | 69 | 68 |
| | IN-Drawing | 57.9 | 57 | 55 | 55 | 51 | 50 | 57 | 47 | 53 | 58 | 57 | 75 | 51 | 65 | 70 | 68 |
| Model Soup PRE-FT-EWC-LwF (Wortsman et al., 2022a) | IN-V2 | 56.1 | 52 | 50 | 50 | 52 | 49 | 57 | 49 | 50 | 52 | 56 | 75 | 50 | 65 | 69 | 68 |
| | IN-A | 57.0 | 50 | 47 | 46 | 56 | 51 | 60 | 50 | 52 | 53 | 60 | 75 | 54 | 65 | 68 | 69 |
| | IN-R | 58.7 | 56 | 55 | 54 | 55 | 52 | 58 | 49 | 53 | 56 | 60 | 75 | 54 | 65 | 69 | 69 |
| | IN-Sketch | 57.0 | 53 | 53 | 53 | 52 | 48 | 56 | 49 | 52 | 54 | 57 | 74 | 51 | 64 | 69 | 68 |
| | ObjNet | 56.7 | 51 | 48 | 48 | 55 | 50 | 59 | 51 | 51 | 52 | 58 | 75 | 52 | 65 | 68 | 68 |
| | IN-Cartoon | 55.0 | 50 | 48 | 47 | 51 | 46 | 55 | 46 | 49 | 50 | 55 | 75 | 52 | 63 | 69 | 68 |
| | IN-Drawing | 58.0 | 57 | 55 | 55 | 51 | 50 | 57 | 47 | 53 | 58 | 57 | 75 | 51 | 65 | 70 | 68 |

Table 36: Accuracy of ImageNet-21K pre-trained ViT-B/16 with different fine-tuning methods and downstream datasets on each ImageNet-C corruption For each corruption, accuracy is averaged across 5 levels of severity.

| Dataset | Method | Avg. | Noise | | | Blur | | | | Weather | | | | Digital | | | |
|---|---|---|---|---|---|---|---|---|---|---|---|---|---|---|---|---|---|
| | | | Gauss. | Shot | Impulse | Defocus | Glass | Motion | Zoom | Snow | Frost | Fog | Bright | Contrast | Elastic | Pixel | JPEG |
| Pre-Trained | | 58.3 | 55 | 53 | 53 | 56 | 50 | 60 | 52 | 53 | 54 | 57 | 75 | 52 | 64 | 72 | 69 |
| FT | IN-V2 | 58.5 | 51 | 49 | 48 | 60 | 49 | 62 | 53 | 54 | 53 | 62 | 76 | 56 | 63 | 72 | 70 |
| | IN-A | 55.0 | 47 | 45 | 43 | 56 | 45 | 60 | 50 | 51 | 50 | 57 | 73 | 55 | 58 | 69 | 66 |
| | IN-R | 52.9 | 48 | 47 | 46 | 51 | 49 | 53 | 44 | 48 | 52 | 53 | 70 | 50 | 56 | 65 | 63 |
| | IN-Sketch | 55.5 | 50 | 49 | 48 | 53 | 47 | 56 | 50 | 51 | 54 | 57 | 72 | 53 | 59 | 69 | 66 |
| | ObjNet | 52.8 | 45 | 42 | 40 | 55 | 41 | 57 | 48 | 47 | 47 | 55 | 72 | 52 | 57 | 68 | 66 |
| | IN-Cartoon | 52.9 | 44 | 41 | 39 | 53 | 41 | 57 | 46 | 48 | 46 | 54 | 76 | 53 | 58 | 71 | 67 |
| | IN-Drawing | 55.3 | 52 | 51 | 51 | 48 | 47 | 55 | 44 | 52 | 59 | 55 | 74 | 49 | 60 | 69 | 64 |
| Linear Probing | IN-V2 | 58.5 | 55 | 53 | 53 | 56 | 50 | 61 | 52 | 53 | 54 | 58 | 75 | 53 | 64 | 72 | 69 |
| | IN-A | 58.7 | 55 | 53 | 52 | 56 | 50 | 61 | 54 | 54 | 54 | 59 | 75 | 53 | 65 | 71 | 70 |
| | IN-R | 56.7 | 55 | 53 | 53 | 52 | 49 | 56 | 50 | 52 | 54 | 52 | 74 | 48 | 63 | 70 | 69 |
| | IN-Sketch | 56.4 | 55 | 53 | 53 | 53 | 49 | 57 | 49 | 51 | 53 | 52 | 73 | 48 | 62 | 71 | 69 |
| | ObjNet | 58.9 | 55 | 53 | 53 | 57 | 50 | 61 | 54 | 53 | 54 | 59 | 75 | 53 | 65 | 71 | 70 |
| | IN-Cartoon | 56.9 | 54 | 53 | 52 | 52 | 47 | 58 | 50 | 52 | 52 | 54 | 76 | 50 | 63 | 72 | 70 |
| | IN-Drawing | 59.0 | 56 | 55 | 55 | 54 | 50 | 60 | 52 | 54 | 57 | 58 | 76 | 51 | 65 | 72 | 70 |
| Visual Prompt (Bahng et al., 2022) | IN-V2 | 45.5 | 42 | 41 | 39 | 43 | 34 | 45 | 43 | 38 | 40 | 42 | 66 | 37 | 55 | 59 | 59 |
| | IN-A | 38.3 | 35 | 33 | 31 | 32 | 25 | 37 | 34 | 32 | 33 | 38 | 59 | 31 | 49 | 52 | 52 |
| | IN-R | 38.3 | 36 | 35 | 33 | 34 | 28 | 35 | 33 | 32 | 36 | 32 | 60 | 28 | 47 | 52 | 52 |
| | IN-Sketch | 41.1 | 39 | 38 | 36 | 37 | 29 | 38 | 36 | 35 | 38 | 36 | 63 | 30 | 50 | 55 | 57 |
| | ObjNet | 36.9 | 33 | 31 | 29 | 32 | 25 | 35 | 34 | 29 | 31 | 35 | 59 | 29 | 48 | 50 | 53 |
| | IN-Cartoon | 41.9 | 37 | 36 | 34 | 39 | 29 | 41 | 38 | 36 | 35 | 37 | 65 | 32 | 52 | 58 | 57 |
| | IN-Drawing | 41.8 | 40 | 40 | 38 | 36 | 32 | 38 | 36 | 38 | 43 | 32 | 63 | 25 | 52 | 57 | 57 |
| LoRA (Hu et al., 2021) | IN-V2 | 58.4 | 55 | 53 | 53 | 56 | 50 | 60 | 52 | 53 | 54 | 57 | 75 | 52 | 64 | 72 | 69 |
| | IN-A | 58.9 | 55 | 53 | 52 | 56 | 50 | 61 | 54 | 54 | 54 | 59 | 76 | 53 | 65 | 72 | 70 |
| | IN-R | 53.6 | 54 | 52 | 52 | 46 | 46 | 50 | 46 | 51 | 53 | 44 | 73 | 40 | 61 | 68 | 68 |
| | IN-Sketch | 56.3 | 54 | 53 | 52 | 52 | 50 | 56 | 49 | 51 | 53 | 50 | 73 | 48 | 62 | 71 | 69 |
| | ObjNet | 59.2 | 55 | 53 | 53 | 57 | 50 | 62 | 55 | 53 | 54 | 59 | 76 | 53 | 65 | 72 | 71 |
| | IN-Cartoon | 56.1 | 54 | 52 | 52 | 51 | 47 | 57 | 49 | 51 | 52 | 52 | 75 | 49 | 64 | 71 | 69 |
| | IN-Drawing | 58.0 | 56 | 54 | 54 | 53 | 48 | 59 | 51 | 53 | 56 | 57 | 75 | 48 | 64 | 72 | 69 |
| EWC (Kirkpatrick et al., 2017) | IN-V2 | 59.4 | 55 | 53 | 52 | 58 | 51 | 62 | 54 | 54 | 54 | 59 | 76 | 54 | 65 | 73 | 70 |
| | IN-A | 58.8 | 51 | 49 | 49 | 60 | 49 | 63 | 54 | 55 | 54 | 62 | 76 | 56 | 63 | 72 | 69 |
| | IN-R | 57.6 | 54 | 53 | 52 | 55 | 50 | 58 | 50 | 52 | 55 | 59 | 74 | 54 | 62 | 68 | 67 |
| | IN-Sketch | 59.3 | 56 | 55 | 54 | 57 | 51 | 61 | 53 | 54 | 55 | 59 | 76 | 55 | 64 | 72 | 69 |
| | ObjNet | 58.2 | 53 | 51 | 51 | 58 | 49 | 62 | 53 | 53 | 52 | 60 | 74 | 53 | 63 | 71 | 69 |
| | IN-Cartoon | 56.7 | 51 | 49 | 49 | 55 | 46 | 59 | 49 | 52 | 51 | 57 | 76 | 54 | 62 | 72 | 68 |
| | IN-Drawing | 59.4 | 55 | 54 | 54 | 57 | 51 | 62 | 52 | 53 | 59 | 61 | 75 | 55 | 65 | 71 | 68 |
| LwF (Li & Hoiem, 2017) | IN-V2 | 59.3 | 53 | 52 | 51 | 59 | 51 | 62 | 54 | 54 | 54 | 61 | 76 | 55 | 64 | 73 | 70 |
| | IN-A | 58.1 | 51 | 49 | 48 | 57 | 48 | 62 | 53 | 54 | 54 | 61 | 75 | 56 | 62 | 71 | 68 |
| | IN-R | 57.3 | 53 | 52 | 51 | 56 | 53 | 58 | 49 | 52 | 55 | 57 | 73 | 53 | 61 | 69 | 67 |
| | IN-Sketch | 57.3 | 52 | 51 | 50 | 55 | 48 | 58 | 51 | 53 | 55 | 58 | 74 | 55 | 61 | 70 | 68 |
| | ObjNet | 57.0 | 50 | 48 | 47 | 57 | 47 | 60 | 52 | 51 | 52 | 59 | 74 | 55 | 62 | 71 | 68 |
| | IN-Cartoon | 57.9 | 51 | 48 | 47 | 57 | 47 | 61 | 51 | 53 | 52 | 59 | 78 | 54 | 64 | 74 | 70 |
| | IN-Drawing | 58.1 | 55 | 53 | 53 | 52 | 50 | 59 | 48 | 54 | 61 | 58 | 76 | 52 | 63 | 72 | 67 |
| LP-FT (Kumar et al., 2022) | IN-V2 | 58.7 | 52 | 50 | 49 | 60 | 49 | 62 | 53 | 54 | 54 | 61 | 76 | 55 | 63 | 73 | 70 |
| | IN-A | 56.7 | 49 | 47 | 46 | 57 | 46 | 61 | 52 | 53 | 52 | 59 | 74 | 56 | 61 | 70 | 67 |
| | IN-R | 55.4 | 51 | 50 | 49 | 53 | 51 | 55 | 46 | 51 | 54 | 55 | 72 | 52 | 59 | 67 | 66 |
| | IN-Sketch | 55.8 | 52 | 51 | 50 | 53 | 46 | 56 | 48 | 51 | 54 | 55 | 72 | 52 | 59 | 69 | 67 |
| | ObjNet | 55.5 | 48 | 45 | 43 | 57 | 44 | 60 | 51 | 51 | 51 | 57 | 74 | 54 | 61 | 70 | 68 |
| | IN-Cartoon | 54.4 | 46 | 44 | 42 | 54 | 42 | 58 | 47 | 50 | 48 | 55 | 77 | 54 | 60 | 73 | 68 |
| | IN-Drawing | 56.0 | 53 | 52 | 52 | 48 | 48 | 55 | 45 | 53 | 60 | 56 | 75 | 48 | 61 | 70 | 66 |
| WiSE-FT (Wortsman et al., 2022b) | IN-V2 | 59.3 | 54 | 52 | 51 | 58 | 50 | 62 | 53 | 55 | 55 | 60 | 76 | 55 | 64 | 73 | 70 |
| | IN-A | 59.0 | 53 | 51 | 50 | 58 | 50 | 62 | 54 | 55 | 55 | 61 | 76 | 56 | 64 | 72 | 69 |
| | IN-R | 60.0 | 55 | 54 | 53 | 58 | 54 | 61 | 53 | 55 | 58 | 61 | 76 | 56 | 64 | 72 | 69 |
| | IN-Sketch | 59.2 | 55 | 54 | 53 | 56 | 50 | 60 | 53 | 55 | 57 | 60 | 75 | 56 | 63 | 72 | 69 |
| | ObjNet | 58.2 | 52 | 50 | 49 | 58 | 49 | 61 | 53 | 53 | 53 | 60 | 75 | 55 | 63 | 72 | 69 |
| | IN-Cartoon | 58.1 | 52 | 49 | 49 | 57 | 47 | 61 | 51 | 54 | 52 | 59 | 77 | 56 | 64 | 74 | 70 |
| | IN-Drawing | 59.8 | 57 | 55 | 55 | 55 | 51 | 61 | 51 | 56 | 60 | 60 | 77 | 54 | 65 | 73 | 69 |
| Model Soup PRE-FT-EWC-LwF (Wortsman et al., 2022a) | IN-V2 | 59.5 | 54 | 52 | 52 | 59 | 51 | 62 | 54 | 55 | 55 | 60 | 76 | 55 | 64 | 73 | 70 |
| | IN-A | 59.1 | 52 | 50 | 49 | 58 | 49 | 63 | 54 | 55 | 55 | 62 | 76 | 57 | 64 | 72 | 69 |
| | IN-R | 59.8 | 55 | 54 | 53 | 58 | 54 | 61 | 52 | 55 | 58 | 60 | 76 | 56 | 64 | 71 | 69 |
| | IN-Sketch | 59.4 | 55 | 54 | 53 | 57 | 50 | 61 | 53 | 55 | 57 | 60 | 76 | 56 | 64 | 72 | 69 |
| | ObjNet | 58.3 | 53 | 50 | 50 | 58 | 49 | 61 | 53 | 53 | 53 | 60 | 75 | 55 | 63 | 72 | 69 |
| | IN-Cartoon | 58.2 | 52 | 49 | 49 | 57 | 47 | 61 | 51 | 53 | 52 | 59 | 77 | 56 | 64 | 74 | 70 |
| | IN-Drawing | 60.1 | 57 | 55 | 55 | 55 | 51 | 61 | 51 | 56 | 61 | 61 | 77 | 55 | 65 | 73 | 69 |

Table 37: Accuracy of ImageNet-21K with AugReg pre-trained ViT-B/16 with different fine-tuning methods and downstream datasets on each ImageNet-C corruption For each corruption, accuracy is averaged across 5 levels of severity.

| Dataset | Method | Avg. | Noise | | | Blur | | | | Weather | | | | Digital | | | |
|---|---|---|---|---|---|---|---|---|---|---|---|---|---|---|---|---|---|
| | | | Gauss. | Shot | Impulse | Defocus | Glass | Motion | Zoom | Snow | Frost | Fog | Bright | Contrast | Elastic | Pixel | JPEG |
| Pre-Trained | | 66.5 | 67 | 66 | 66 | 63 | 54 | 67 | 59 | 66 | 62 | 69 | 80 | 65 | 66 | 75 | 74 |
| FT | IN-V2 | 64.4 | 64 | 62 | 63 | 61 | 56 | 64 | 55 | 64 | 61 | 66 | 77 | 63 | 63 | 73 | 73 |
| | IN-A | 56.3 | 57 | 56 | 56 | 52 | 48 | 56 | 46 | 53 | 51 | 56 | 70 | 53 | 55 | 67 | 68 |
| | IN-R | 55.7 | 54 | 53 | 53 | 53 | 51 | 54 | 43 | 52 | 56 | 57 | 69 | 54 | 56 | 65 | 65 |
| | IN-Sketch | 54.8 | 54 | 53 | 52 | 52 | 44 | 53 | 43 | 52 | 55 | 57 | 70 | 55 | 53 | 64 | 64 |
| | ObjNet | 54.9 | 55 | 53 | 52 | 53 | 43 | 56 | 41 | 52 | 50 | 54 | 72 | 52 | 56 | 67 | 67 |
| | IN-Cartoon | 61.5 | 61 | 59 | 58 | 56 | 46 | 63 | 48 | 60 | 57 | 60 | 84 | 60 | 63 | 75 | 72 |
| | IN-Drawing | 59.9 | 57 | 54 | 54 | 51 | 48 | 60 | 45 | 58 | 65 | 64 | 81 | 56 | 62 | 74 | 70 |
| Linear Probing | IN-V2 | 65.7 | 66 | 65 | 64 | 62 | 54 | 66 | 58 | 65 | 62 | 68 | 79 | 65 | 65 | 74 | 73 |
| | IN-A | 65.8 | 65 | 64 | 64 | 62 | 54 | 66 | 58 | 65 | 62 | 68 | 79 | 65 | 65 | 74 | 73 |
| | IN-R | 64.0 | 63 | 62 | 62 | 62 | 53 | 64 | 56 | 63 | 61 | 66 | 77 | 63 | 64 | 72 | 71 |
| | IN-Sketch | 62.1 | 62 | 61 | 60 | 60 | 50 | 62 | 54 | 61 | 59 | 65 | 75 | 62 | 61 | 70 | 69 |
| | ObjNet | 65.9 | 65 | 64 | 64 | 64 | 54 | 67 | 59 | 65 | 62 | 69 | 79 | 64 | 65 | 74 | 73 |
| | IN-Cartoon | 69.9 | 70 | 69 | 68 | 67 | 55 | 71 | 62 | 69 | 64 | 72 | 85 | 69 | 69 | 79 | 78 |
| | IN-Drawing | 70.0 | 70 | 69 | 69 | 67 | 59 | 71 | 61 | 70 | 69 | 73 | 83 | 69 | 69 | 77 | 76 |
| Visual Prompt (Bahng et al., 2022) | IN-V2 | 56.9 | 53 | 51 | 50 | 55 | 43 | 57 | 51 | 54 | 52 | 59 | 75 | 57 | 60 | 68 | 68 |
| | IN-A | 51.1 | 44 | 41 | 40 | 47 | 34 | 52 | 46 | 50 | 47 | 57 | 71 | 53 | 54 | 64 | 65 |
| | IN-R | 46.4 | 43 | 42 | 40 | 40 | 35 | 42 | 36 | 46 | 48 | 46 | 68 | 40 | 51 | 58 | 60 |
| | IN-Sketch | 51.7 | 48 | 46 | 44 | 48 | 37 | 49 | 44 | 50 | 52 | 54 | 72 | 50 | 55 | 62 | 64 |
| | ObjNet | 42.5 | 33 | 32 | 29 | 40 | 25 | 42 | 36 | 42 | 40 | 47 | 67 | 43 | 47 | 56 | 58 |
| | IN-Cartoon | 53.1 | 51 | 49 | 49 | 50 | 37 | 52 | 45 | 51 | 49 | 51 | 74 | 50 | 57 | 66 | 65 |
| | IN-Drawing | 54.5 | 55 | 53 | 53 | 48 | 43 | 51 | 45 | 51 | 58 | 53 | 73 | 46 | 57 | 66 | 64 |
| LoRA (Hu et al., 2021) | IN-V2 | 67.2 | 67 | 66 | 66 | 64 | 55 | 68 | 60 | 66 | 63 | 70 | 80 | 67 | 67 | 75 | 74 |
| | IN-A | 67.8 | 67 | 66 | 66 | 66 | 56 | 69 | 61 | 67 | 64 | 71 | 80 | 67 | 67 | 76 | 75 |
| | IN-R | 67.0 | 67 | 66 | 66 | 65 | 56 | 67 | 58 | 66 | 65 | 69 | 79 | 66 | 66 | 75 | 74 |
| | IN-Sketch | 66.9 | 67 | 66 | 65 | 65 | 55 | 67 | 59 | 67 | 64 | 70 | 79 | 67 | 66 | 74 | 74 |
| | ObjNet | 67.2 | 67 | 66 | 65 | 66 | 54 | 68 | 60 | 67 | 63 | 70 | 80 | 65 | 66 | 75 | 74 |
| | IN-Cartoon | 66.4 | 66 | 65 | 65 | 64 | 54 | 67 | 59 | 66 | 62 | 69 | 80 | 65 | 65 | 74 | 74 |
| | IN-Drawing | 67.1 | 66 | 65 | 65 | 64 | 57 | 67 | 57 | 68 | 67 | 71 | 80 | 67 | 66 | 74 | 73 |
| EWC (Kirkpatrick et al., 2017) | IN-V2 | 67.7 | 67 | 66 | 65 | 65 | 56 | 68 | 59 | 68 | 65 | 71 | 80 | 68 | 66 | 76 | 75 |
| | IN-A | 65.8 | 65 | 64 | 64 | 63 | 55 | 67 | 57 | 66 | 62 | 69 | 78 | 67 | 63 | 74 | 73 |
| | IN-R | 63.9 | 62 | 61 | 60 | 62 | 58 | 63 | 52 | 62 | 64 | 66 | 77 | 65 | 63 | 71 | 72 |
| | IN-Sketch | 66.2 | 65 | 65 | 64 | 64 | 55 | 66 | 58 | 65 | 65 | 70 | 78 | 67 | 64 | 74 | 73 |
| | ObjNet | 63.5 | 63 | 62 | 61 | 62 | 51 | 65 | 52 | 64 | 60 | 67 | 77 | 63 | 61 | 73 | 72 |
| | IN-Cartoon | 63.9 | 63 | 62 | 62 | 62 | 49 | 66 | 55 | 64 | 58 | 66 | 79 | 65 | 62 | 73 | 71 |
| | IN-Drawing | 65.9 | 64 | 63 | 62 | 61 | 54 | 66 | 56 | 66 | 68 | 70 | 79 | 66 | 64 | 75 | 73 |
| LwF (Li & Hoiem, 2017) | IN-V2 | 66.0 | 66 | 65 | 65 | 63 | 55 | 66 | 57 | 65 | 62 | 67 | 79 | 64 | 66 | 75 | 75 |
| | IN-A | 64.1 | 64 | 63 | 64 | 61 | 55 | 65 | 55 | 62 | 59 | 65 | 78 | 62 | 64 | 73 | 73 |
| | IN-R | 64.1 | 63 | 62 | 62 | 61 | 55 | 64 | 55 | 62 | 63 | 66 | 78 | 62 | 64 | 72 | 72 |
| | IN-Sketch | 61.5 | 61 | 60 | 60 | 58 | 50 | 61 | 52 | 59 | 60 | 64 | 76 | 60 | 61 | 70 | 70 |
| | ObjNet | 62.2 | 63 | 61 | 61 | 60 | 50 | 63 | 50 | 61 | 57 | 63 | 77 | 59 | 63 | 73 | 72 |
| | IN-Cartoon | 71.7 | 71 | 69 | 69 | 67 | 59 | 71 | 60 | 70 | 68 | 73 | 90 | 71 | 73 | 82 | 82 |
| | IN-Drawing | 68.5 | 66 | 64 | 64 | 64 | 57 | 70 | 56 | 66 | 71 | 72 | 87 | 62 | 69 | 82 | 79 |
| LP-FT (Kumar et al., 2022) | IN-V2 | 65.6 | 65 | 64 | 64 | 62 | 54 | 65 | 57 | 66 | 62 | 68 | 78 | 66 | 65 | 74 | 73 |
| | IN-A | 63.2 | 62 | 61 | 61 | 60 | 53 | 63 | 55 | 62 | 59 | 65 | 77 | 63 | 62 | 72 | 71 |
| | IN-R | 57.0 | 54 | 54 | 53 | 54 | 51 | 56 | 47 | 55 | 57 | 59 | 71 | 55 | 58 | 67 | 65 |
| | IN-Sketch | 58.5 | 57 | 56 | 56 | 56 | 48 | 58 | 50 | 57 | 57 | 61 | 72 | 58 | 57 | 66 | 66 |
| | ObjNet | 59.9 | 59 | 58 | 58 | 57 | 47 | 61 | 48 | 58 | 56 | 61 | 75 | 58 | 60 | 70 | 70 |
| | IN-Cartoon | 68.6 | 69 | 67 | 67 | 63 | 52 | 70 | 58 | 69 | 64 | 70 | 88 | 68 | 68 | 80 | 77 |
| | IN-Drawing | 66.9 | 65 | 62 | 63 | 64 | 55 | 69 | 54 | 65 | 68 | 71 | 84 | 64 | 67 | 80 | 75 |
| WiSE-FT (Wortsman et al., 2022b) | IN-V2 | 68.0 | 68 | 67 | 67 | 65 | 57 | 68 | 60 | 68 | 65 | 70 | 80 | 67 | 67 | 76 | 76 |
| | IN-A | 66.5 | 67 | 66 | 66 | 62 | 55 | 67 | 57 | 66 | 62 | 68 | 79 | 65 | 66 | 75 | 74 |
| | IN-R | 66.9 | 66 | 65 | 65 | 64 | 59 | 66 | 57 | 66 | 67 | 69 | 79 | 67 | 67 | 74 | 74 |
| | IN-Sketch | 65.1 | 65 | 64 | 64 | 61 | 53 | 64 | 55 | 65 | 64 | 68 | 78 | 65 | 64 | 73 | 73 |
| | ObjNet | 65.6 | 66 | 64 | 64 | 63 | 53 | 67 | 55 | 65 | 62 | 68 | 79 | 65 | 65 | 74 | 74 |
| | IN-Cartoon | 68.6 | 69 | 67 | 67 | 64 | 54 | 69 | 58 | 69 | 64 | 70 | 85 | 68 | 69 | 78 | 78 |
| | IN-Drawing | 69.2 | 70 | 68 | 68 | 63 | 56 | 70 | 57 | 69 | 70 | 72 | 84 | 66 | 68 | 79 | 77 |
| Model Soup PRE-FT-EWC-LwF (Wortsman et al., 2022a) | IN-V2 | 68.0 | 68 | 67 | 67 | 65 | 57 | 68 | 60 | 68 | 65 | 71 | 80 | 67 | 67 | 76 | 75 |
| | IN-A | 66.9 | 68 | 67 | 67 | 63 | 56 | 67 | 58 | 67 | 62 | 69 | 80 | 66 | 66 | 75 | 74 |
| | IN-R | 67.4 | 66 | 66 | 65 | 64 | 59 | 67 | 58 | 66 | 67 | 70 | 79 | 67 | 67 | 75 | 74 |
| | IN-Sketch | 65.6 | 66 | 65 | 65 | 62 | 53 | 65 | 56 | 65 | 64 | 68 | 78 | 66 | 64 | 73 | 73 |
| | ObjNet | 65.6 | 66 | 64 | 65 | 63 | 52 | 67 | 55 | 65 | 61 | 68 | 79 | 65 | 65 | 74 | 74 |
| | IN-Cartoon | 69.2 | 70 | 68 | 68 | 65 | 55 | 70 | 59 | 69 | 65 | 71 | 85 | 69 | 69 | 79 | 78 |
| | IN-Drawing | 69.6 | 70 | 68 | 69 | 64 | 57 | 70 | 58 | 69 | 71 | 72 | 84 | 66 | 69 | 80 | 78 |

Table 38: Accuracy of ViT-B/16 pre-trained on ImageNet-21K-P, using different fine-tuning methods and downstream datasets on each ImageNet-C corruption For each corruption, accuracy is averaged across 5 levels of severity.

| Dataset | Method | Avg. | Noise | | | Blur | | | | Weather | | | | Digital | | | |
|---|---|---|---|---|---|---|---|---|---|---|---|---|---|---|---|---|---|
| | | | Gauss. | Shot | Impulse | Defocus | Glass | Motion | Zoom | Snow | Frost | Fog | Bright | Contrast | Elastic | Pixel | JPEG |
| Pre-Trained | | 61.4 | 60 | 59 | 58 | 57 | 46 | 60 | 53 | 55 | 55 | 65 | 79 | 69 | 64 | 70 | 71 |
| FT | IN-V2 | 62.1 | 58 | 57 | 55 | 62 | 48 | 63 | 55 | 57 | 55 | 67 | 79 | 69 | 62 | 73 | 73 |
| | IN-A | 60.5 | 56 | 55 | 52 | 63 | 48 | 63 | 53 | 54 | 51 | 64 | 77 | 69 | 60 | 71 | 72 |
| | IN-R | 57.5 | 55 | 55 | 53 | 56 | 50 | 57 | 47 | 52 | 55 | 59 | 73 | 57 | 58 | 66 | 68 |
| | IN-Sketch | 59.3 | 57 | 57 | 56 | 54 | 44 | 58 | 52 | 54 | 56 | 61 | 76 | 65 | 60 | 70 | 70 |
| | ObjNet | 57.7 | 54 | 54 | 50 | 58 | 43 | 60 | 50 | 49 | 48 | 60 | 76 | 65 | 58 | 69 | 71 |
| | IN-Cartoon | 56.1 | 53 | 52 | 49 | 55 | 38 | 58 | 47 | 51 | 47 | 59 | 79 | 63 | 58 | 65 | 66 |
| | IN-Drawing | 57.9 | 60 | 58 | 58 | 46 | 46 | 57 | 46 | 54 | 61 | 57 | 78 | 57 | 60 | 66 | 65 |
| HeadOnly | IN-V2 | 61.4 | 59 | 59 | 58 | 57 | 46 | 60 | 53 | 55 | 55 | 65 | 79 | 69 | 64 | 70 | 71 |
| | IN-A | 61.8 | 59 | 59 | 58 | 58 | 46 | 61 | 54 | 56 | 55 | 66 | 79 | 70 | 64 | 70 | 71 |
| | IN-R | 61.0 | 59 | 58 | 57 | 56 | 46 | 60 | 52 | 55 | 55 | 64 | 79 | 68 | 64 | 70 | 71 |
| | IN-Sketch | 60.8 | 58 | 57 | 56 | 57 | 45 | 59 | 52 | 55 | 55 | 64 | 78 | 69 | 64 | 70 | 71 |
| | ObjNet | 61.9 | 59 | 59 | 58 | 58 | 47 | 61 | 54 | 55 | 55 | 66 | 79 | 70 | 64 | 70 | 71 |
| | IN-Cartoon | 61.3 | 59 | 59 | 58 | 56 | 45 | 60 | 52 | 55 | 55 | 65 | 80 | 69 | 64 | 71 | 72 |
| | IN-Drawing | 62.0 | 60 | 59 | 58 | 56 | 47 | 60 | 52 | 57 | 59 | 64 | 80 | 66 | 66 | 72 | 73 |
| Visual Prompt (Bahng et al., 2022) | IN-V2 | 49.6 | 45 | 44 | 43 | 45 | 34 | 46 | 45 | 43 | 44 | 54 | 72 | 52 | 56 | 59 | 61 |
| | IN-A | 44.6 | 38 | 36 | 35 | 40 | 29 | 43 | 40 | 37 | 39 | 52 | 68 | 49 | 51 | 54 | 58 |
| | IN-R | 40.5 | 35 | 35 | 33 | 35 | 26 | 35 | 33 | 35 | 41 | 45 | 65 | 42 | 48 | 45 | 53 |
| | IN-Sketch | 44.1 | 40 | 40 | 38 | 37 | 27 | 39 | 37 | 39 | 44 | 48 | 67 | 46 | 51 | 50 | 57 |
| | ObjNet | 34.8 | 27 | 26 | 25 | 28 | 22 | 32 | 30 | 27 | 30 | 41 | 62 | 37 | 43 | 43 | 49 |
| | IN-Cartoon | 44.2 | 37 | 36 | 34 | 40 | 28 | 42 | 39 | 37 | 38 | 49 | 70 | 48 | 52 | 57 | 57 |
| | IN-Drawing | 44.3 | 44 | 44 | 42 | 37 | 32 | 38 | 36 | 39 | 48 | 42 | 66 | 38 | 52 | 53 | 55 |
| LoRA (Hu et al., 2021) | IN-V2 | 61.3 | 59 | 59 | 57 | 57 | 46 | 60 | 53 | 55 | 55 | 64 | 79 | 69 | 64 | 70 | 72 |
| | IN-A | 62.1 | 59 | 59 | 57 | 58 | 48 | 61 | 54 | 55 | 55 | 66 | 79 | 71 | 65 | 71 | 71 |
| | IN-R | 61.2 | 59 | 58 | 57 | 56 | 47 | 60 | 53 | 55 | 55 | 65 | 79 | 68 | 64 | 70 | 72 |
| | IN-Sketch | 61.4 | 59 | 59 | 57 | 58 | 46 | 60 | 53 | 55 | 56 | 64 | 79 | 68 | 64 | 71 | 72 |
| | ObjNet | 62.1 | 59 | 59 | 58 | 59 | 48 | 61 | 54 | 55 | 55 | 66 | 79 | 71 | 65 | 71 | 72 |
| | IN-Cartoon | 60.3 | 58 | 58 | 57 | 55 | 45 | 58 | 52 | 54 | 54 | 62 | 79 | 66 | 63 | 70 | 72 |
| | IN-Drawing | 61.6 | 59 | 59 | 58 | 56 | 47 | 59 | 51 | 57 | 59 | 64 | 79 | 67 | 65 | 71 | 72 |
| EWC (Kirkpatrick et al., 2017) | IN-V2 | 62.4 | 60 | 59 | 58 | 58 | 47 | 62 | 54 | 57 | 56 | 66 | 80 | 70 | 65 | 72 | 72 |
| | IN-A | 63.7 | 59 | 59 | 57 | 63 | 49 | 65 | 57 | 58 | 56 | 69 | 80 | 72 | 65 | 73 | 73 |
| | IN-R | 55.3 | 58 | 58 | 56 | 41 | 45 | 52 | 45 | 51 | 56 | 50 | 75 | 50 | 60 | 63 | 69 |
| | IN-Sketch | 61.4 | 60 | 60 | 58 | 55 | 46 | 60 | 54 | 55 | 57 | 64 | 78 | 67 | 64 | 70 | 72 |
| | ObjNet | 62.5 | 59 | 59 | 58 | 60 | 47 | 62 | 55 | 56 | 55 | 67 | 80 | 70 | 64 | 72 | 72 |
| | IN-Cartoon | 59.9 | 57 | 56 | 55 | 56 | 42 | 59 | 51 | 54 | 53 | 64 | 79 | 70 | 63 | 70 | 71 |
| | IN-Drawing | 60.9 | 60 | 60 | 58 | 53 | 48 | 60 | 51 | 57 | 62 | 62 | 78 | 61 | 64 | 68 | 70 |
| LwF (Li & Hoiem, 2017) | IN-V2 | 62.9 | 59 | 58 | 57 | 62 | 48 | 63 | 55 | 58 | 56 | 67 | 80 | 71 | 64 | 73 | 73 |
| | IN-A | 63.5 | 59 | 59 | 57 | 63 | 49 | 65 | 57 | 58 | 57 | 68 | 79 | 71 | 64 | 73 | 73 |
| | IN-R | 62.0 | 59 | 59 | 57 | 59 | 53 | 61 | 53 | 57 | 59 | 65 | 77 | 66 | 63 | 71 | 71 |
| | IN-Sketch | 61.1 | 59 | 59 | 57 | 57 | 45 | 60 | 54 | 56 | 57 | 64 | 77 | 68 | 62 | 71 | 71 |
| | ObjNet | 61.8 | 59 | 59 | 56 | 61 | 46 | 63 | 54 | 55 | 54 | 66 | 79 | 69 | 62 | 72 | 72 |
| | IN-Cartoon | 61.3 | 58 | 57 | 55 | 60 | 44 | 62 | 53 | 56 | 54 | 65 | 82 | 68 | 64 | 70 | 71 |
| | IN-Drawing | 61.2 | 61 | 60 | 58 | 51 | 48 | 61 | 50 | 58 | 63 | 62 | 80 | 64 | 63 | 70 | 69 |
| LP-FT (Kumar et al., 2022) | IN-V2 | 62.5 | 58 | 58 | 56 | 62 | 47 | 63 | 55 | 57 | 55 | 67 | 79 | 71 | 63 | 73 | 73 |
| | IN-A | 62.6 | 58 | 57 | 55 | 63 | 50 | 65 | 56 | 57 | 55 | 67 | 79 | 70 | 63 | 73 | 72 |
| | IN-R | 60.5 | 57 | 57 | 55 | 57 | 52 | 60 | 51 | 55 | 58 | 63 | 76 | 64 | 62 | 70 | 70 |
| | IN-Sketch | 60.2 | 58 | 57 | 56 | 56 | 45 | 59 | 52 | 56 | 56 | 63 | 77 | 66 | 61 | 70 | 71 |
| | ObjNet | 60.6 | 57 | 57 | 54 | 60 | 46 | 62 | 52 | 54 | 53 | 64 | 78 | 67 | 61 | 71 | 72 |
| | IN-Cartoon | 58.8 | 56 | 55 | 53 | 58 | 40 | 60 | 50 | 54 | 51 | 62 | 80 | 65 | 61 | 68 | 69 |
| | IN-Drawing | 58.7 | 60 | 59 | 58 | 48 | 46 | 58 | 46 | 56 | 62 | 58 | 79 | 60 | 61 | 67 | 66 |
| WiSE-FT (Wortsman et al., 2022b) | IN-V2 | 63.3 | 60 | 59 | 58 | 61 | 48 | 63 | 55 | 58 | 57 | 68 | 80 | 72 | 65 | 73 | 73 |
| | IN-A | 64.1 | 60 | 60 | 58 | 63 | 49 | 65 | 57 | 59 | 58 | 69 | 80 | 73 | 65 | 73 | 73 |
| | IN-R | 64.3 | 61 | 61 | 60 | 61 | 52 | 63 | 55 | 59 | 61 | 68 | 80 | 71 | 66 | 73 | 73 |
| | IN-Sketch | 62.7 | 61 | 60 | 59 | 58 | 46 | 61 | 55 | 58 | 58 | 66 | 79 | 71 | 64 | 72 | 73 |
| | ObjNet | 62.9 | 60 | 59 | 58 | 61 | 47 | 63 | 55 | 57 | 56 | 67 | 80 | 71 | 64 | 73 | 73 |
| | IN-Cartoon | 61.4 | 59 | 58 | 56 | 59 | 43 | 62 | 52 | 56 | 54 | 65 | 81 | 70 | 64 | 70 | 71 |
| | IN-Drawing | 63.5 | 63 | 62 | 61 | 56 | 48 | 63 | 53 | 59 | 63 | 66 | 81 | 69 | 65 | 71 | 71 |
| Model Soup PRE-FT-EWC-LwF (Wortsman et al., 2022a) | IN-V2 | 63.3 | 60 | 59 | 58 | 61 | 47 | 63 | 55 | 58 | 57 | 68 | 80 | 72 | 65 | 73 | 73 |
| | IN-A | 64.1 | 60 | 60 | 58 | 63 | 49 | 65 | 57 | 59 | 57 | 69 | 80 | 73 | 65 | 73 | 73 |
| | IN-R | 64.1 | 61 | 61 | 59 | 61 | 51 | 63 | 55 | 59 | 61 | 68 | 79 | 71 | 66 | 73 | 73 |
| | IN-Sketch | 62.7 | 61 | 60 | 59 | 58 | 46 | 61 | 55 | 58 | 58 | 66 | 79 | 71 | 64 | 72 | 73 |
| | ObjNet | 62.9 | 60 | 59 | 58 | 61 | 47 | 63 | 55 | 57 | 56 | 67 | 80 | 71 | 64 | 72 | 73 |
| | IN-Cartoon | 61.4 | 58 | 58 | 56 | 59 | 43 | 61 | 53 | 56 | 54 | 65 | 81 | 70 | 64 | 70 | 71 |
| | IN-Drawing | 63.4 | 63 | 62 | 61 | 56 | 48 | 63 | 53 | 59 | 63 | 66 | 81 | 69 | 65 | 71 | 71 |

Table 39: Accuracy of LAION-2B pre-trained ViT-B/16 with different fine-tuning methods and downstream datasets on each ImageNet-C corruption. For each corruption, accuracy is averaged across 5 levels of severity.

| Dataset | Method | Avg. | Noise | | | Blur | | | | Weather | | | | Digital | | | |
|---|---|---|---|---|---|---|---|---|---|---|---|---|---|---|---|---|---|
| | | | Gauss. | Shot | Impulse | Defocus | Glass | Motion | Zoom | Snow | Frost | Fog | Bright | Contrast | Elastic | Pixel | JPEG |
| Pre-Trained | | 63.0 | 59 | 59 | 59 | 58 | 44 | 49 | 64 | 52 | 64 | 61 | 70 | 81 | 70 | 61 | 67 | 71 |
| FT | IN-V2 | 24.5 | 18 | 17 | 16 | 15 | 18 | 19 | 18 | 19 | 21 | 35 | 45 | 33 | 30 | 31 | 32 |
| | IN-A | 8.5 | 8 | 7 | 7 | 5 | 5 | 6 | 6 | 6 | 6 | 12 | 17 | 11 | 11 | 11 | 10 |
| | IN-R | 21.4 | 16 | 15 | 14 | 15 | 18 | 18 | 16 | 15 | 20 | 29 | 38 | 31 | 24 | 26 | 25 |
| | IN-Sketch | 9.7 | 9 | 8 | 8 | 3 | 4 | 5 | 6 | 9 | 11 | 15 | 21 | 16 | 9 | 8 | 14 |

Table 40: Accuracy of OpenAI CLIP ViT-B/16 with different fine-tuning methods and downstream datasets on each ImageNet-C corruption. For each corruption, accuracy is averaged across 5 levels of severity.

| Dataset | Method | Avg. | Noise | | | Blur | | | | Weather | | | | Digital | | | |
|---|---|---|---|---|---|---|---|---|---|---|---|---|---|---|---|---|---|
| | | | Gauss. | Shot | Impulse | Defocus | Glass | Motion | Zoom | Snow | Frost | Fog | Bright | Contrast | Elastic | Pixel | JPEG |
| Pre-Trained | | 62.6 | 58 | 58 | 58 | 57 | 46 | 63 | 54 | 63 | 59 | 69 | 80 | 70 | 62 | 71 | 70 |
| FT | IN-V2 | 25.7 | 18 | 17 | 16 | 15 | 21 | 21 | 19 | 20 | 23 | 36 | 45 | 33 | 34 | 33 | 36 |
| | IN-A | 8.9 | 7 | 7 | 5 | 5 | 6 | 6 | 6 | 7 | 7 | 13 | 17 | 12 | 11 | 12 | 12 |
| | IN-R | 22.2 | 17 | 16 | 15 | 16 | 18 | 19 | 16 | 16 | 22 | 28 | 38 | 29 | 26 | 28 | 28 |
| | IN-Sketch | 7.2 | 7 | 7 | 6 | 2 | 3 | 4 | 4 | 7 | 9 | 11 | 14 | 10 | 7 | 7 | 12 |
| | ObjNet | 10.6 | 7 | 7 | 6 | 7 | 7 | 9 | 9 | 5 | 7 | 17 | 20 | 17 | 13 | 14 | 14 |
| | IN-Cartoon | 32.8 | 20 | 19 | 15 | 20 | 24 | 27 | 22 | 29 | 32 | 41 | 64 | 42 | 43 | 48 | 43 |
| | IN-Drawing | 23.0 | 18 | 14 | 15 | 8 | 13 | 14 | 13 | 24 | 39 | 25 | 56 | 23 | 27 | 30 | 26 |
| HeadOnly | IN-V2 | 61.6 | 57 | 55 | 56 | 57 | 47 | 64 | 54 | 61 | 59 | 70 | 79 | 69 | 60 | 69 | 68 |
| | IN-A | 62.3 | 57 | 56 | 56 | 59 | 47 | 65 | 54 | 62 | 59 | 71 | 80 | 69 | 60 | 70 | 68 |
| | IN-R | 59.9 | 56 | 55 | 54 | 55 | 49 | 61 | 50 | 58 | 59 | 67 | 77 | 65 | 58 | 68 | 69 |
| | IN-Sketch | 58.0 | 54 | 53 | 53 | 53 | 45 | 58 | 48 | 56 | 56 | 66 | 76 | 63 | 56 | 67 | 66 |
| | ObjNet | 60.1 | 56 | 55 | 55 | 57 | 43 | 63 | 51 | 61 | 57 | 69 | 78 | 64 | 59 | 65 | 66 |
| | IN-Cartoon | 56.3 | 53 | 51 | 50 | 50 | 39 | 56 | 44 | 56 | 53 | 65 | 79 | 63 | 57 | 67 | 63 |
| | IN-Drawing | 54.7 | 50 | 50 | 50 | 47 | 43 | 53 | 41 | 53 | 59 | 62 | 75 | 58 | 56 | 64 | 58 |
| Visual Prompt (Bahng et al., 2022) | IN-V2 | 53.6 | 51 | 51 | 50 | 47 | 35 | 53 | 47 | 52 | 49 | 59 | 74 | 58 | 54 | 61 | 62 |
| | IN-A | 50.5 | 44 | 43 | 42 | 44 | 32 | 51 | 45 | 50 | 45 | 59 | 73 | 58 | 53 | 59 | 59 |
| | IN-R | 47.9 | 45 | 44 | 43 | 41 | 30 | 46 | 39 | 47 | 48 | 53 | 71 | 48 | 50 | 55 | 59 |
| | IN-Sketch | 50.8 | 49 | 47 | 46 | 44 | 34 | 49 | 44 | 48 | 49 | 56 | 73 | 53 | 53 | 56 | 60 |
| | ObjNet | 41.1 | 36 | 34 | 32 | 36 | 25 | 41 | 36 | 41 | 37 | 47 | 66 | 45 | 45 | 46 | 50 |
| | IN-Cartoon | 46.2 | 43 | 42 | 40 | 39 | 28 | 45 | 40 | 43 | 40 | 50 | 71 | 48 | 50 | 57 | 57 |
| | IN-Drawing | 46.8 | 48 | 48 | 47 | 38 | 35 | 41 | 38 | 44 | 49 | 44 | 69 | 42 | 50 | 54 | 55 |
| LoRA (Hu et al., 2021) | IN-V2 | 61.5 | 57 | 55 | 56 | 57 | 46 | 64 | 54 | 61 | 58 | 70 | 79 | 69 | 60 | 69 | 68 |
| | IN-A | 62.0 | 57 | 56 | 56 | 59 | 46 | 64 | 54 | 62 | 59 | 71 | 80 | 68 | 60 | 70 | 68 |
| | IN-R | 59.5 | 56 | 55 | 54 | 55 | 48 | 60 | 49 | 56 | 58 | 67 | 77 | 64 | 57 | 68 | 69 |
| | IN-Sketch | 58.6 | 54 | 53 | 53 | 53 | 45 | 58 | 48 | 57 | 58 | 68 | 77 | 63 | 57 | 68 | 67 |
| | ObjNet | 57.9 | 55 | 55 | 54 | 55 | 41 | 61 | 49 | 58 | 54 | 67 | 77 | 61 | 56 | 62 | 64 |
| | IN-Cartoon | 54.6 | 51 | 49 | 49 | 48 | 37 | 54 | 42 | 54 | 52 | 63 | 77 | 61 | 55 | 64 | 61 |
| | IN-Drawing | 52.5 | 48 | 47 | 48 | 46 | 40 | 51 | 39 | 50 | 56 | 60 | 73 | 57 | 53 | 62 | 56 |
| EWC (Kirkpatrick et al., 2017) | IN-V2 | 52.9 | 43 | 41 | 42 | 49 | 41 | 55 | 41 | 50 | 50 | 67 | 74 | 64 | 53 | 62 | 62 |
| | IN-A | 45.3 | 33 | 31 | 32 | 44 | 36 | 48 | 35 | 39 | 41 | 61 | 65 | 57 | 46 | 56 | 56 |
| | IN-R | 58.0 | 53 | 52 | 52 | 52 | 49 | 57 | 46 | 56 | 57 | 66 | 75 | 64 | 57 | 66 | 67 |
| | IN-Sketch | 32.8 | 31 | 30 | 30 | 20 | 17 | 25 | 19 | 44 | 44 | 58 | 37 | 38 | 28 | 33 | 38 |
| | ObjNet | 46.6 | 37 | 35 | 35 | 45 | 36 | 50 | 38 | 40 | 40 | 59 | 69 | 58 | 48 | 55 | 56 |
| | IN-Cartoon | 49.4 | 43 | 41 | 41 | 43 | 33 | 48 | 36 | 51 | 45 | 59 | 73 | 60 | 50 | 60 | 58 |
| | IN-Drawing | 49.0 | 46 | 42 | 45 | 38 | 36 | 44 | 35 | 51 | 57 | 58 | 73 | 55 | 48 | 56 | 52 |
| LwF (Li & Hoiem, 2017) | IN-V2 | 28.8 | 19 | 17 | 14 | 19 | 24 | 25 | 22 | 23 | 27 | 41 | 50 | 37 | 38 | 38 | 40 |
| | IN-A | 14.7 | 12 | 11 | 10 | 9 | 11 | 12 | 11 | 9 | 11 | 21 | 26 | 20 | 20 | 19 | 20 |
| | IN-R | 27.7 | 21 | 20 | 17 | 20 | 23 | 24 | 21 | 22 | 28 | 35 | 47 | 34 | 33 | 35 | 35 |
| | IN-Sketch | 8.8 | 8 | 8 | 7 | 3 | 3 | 4 | 4 | 8 | 12 | 14 | 16 | 12 | 8 | 8 | 14 |
| | ObjNet | 18.6 | 12 | 11 | 10 | 14 | 14 | 19 | 17 | 9 | 12 | 27 | 34 | 25 | 24 | 24 | 26 |
| | IN-Cartoon | 41.1 | 30 | 29 | 24 | 25 | 31 | 35 | 29 | 38 | 42 | 50 | 75 | 48 | 54 | 58 | 54 |
| | IN-Drawing | 30.3 | 23 | 19 | 19 | 15 | 20 | 22 | 20 | 32 | 45 | 34 | 66 | 28 | 35 | 41 | 36 |
| LP-FT (Kumar et al., 2022) | IN-V2 | 25.8 | 19 | 17 | 16 | 15 | 20 | 21 | 18 | 21 | 23 | 37 | 46 | 33 | 33 | 33 | 36 |
| | IN-A | 10.4 | 8 | 7 | 7 | 7 | 8 | 8 | 8 | 7 | 7 | 15 | 19 | 14 | 13 | 14 | 13 |
| | IN-R | 23.3 | 18 | 17 | 15 | 17 | 19 | 20 | 16 | 18 | 24 | 30 | 40 | 30 | 27 | 29 | 29 |
| | IN-Sketch | 7.3 | 7 | 7 | 6 | 2 | 3 | 4 | 4 | 7 | 10 | 11 | 14 | 10 | 7 | 7 | 12 |
| | ObjNet | 12.7 | 9 | 8 | 7 | 9 | 9 | 11 | 11 | 7 | 9 | 19 | 24 | 19 | 16 | 17 | 16 |
| | IN-Cartoon | 30.6 | 20 | 19 | 15 | 17 | 21 | 24 | 20 | 25 | 28 | 39 | 63 | 40 | 41 | 45 | 40 |
| | IN-Drawing | 27.7 | 21 | 18 | 17 | 13 | 18 | 20 | 18 | 25 | 41 | 33 | 59 | 29 | 35 | 38 | 30 |
| WiSE-FT (Wortsman et al., 2022b) | IN-V2 | 47.9 | 39 | 37 | 38 | 40 | 39 | 47 | 36 | 42 | 42 | 61 | 69 | 59 | 53 | 56 | 59 |
| | IN-A | 39.1 | 30 | 28 | 29 | 34 | 30 | 38 | 28 | 31 | 32 | 54 | 59 | 51 | 44 | 48 | 50 |
| | IN-R | 48.5 | 41 | 40 | 39 | 41 | 40 | 46 | 34 | 45 | 48 | 60 | 70 | 59 | 52 | 55 | 56 |
| | IN-Sketch | 25.5 | 24 | 23 | 23 | 13 | 12 | 17 | 13 | 28 | 33 | 44 | 39 | 33 | 24 | 22 | 33 |
| | ObjNet | 41.7 | 33 | 31 | 30 | 38 | 30 | 42 | 35 | 30 | 33 | 56 | 65 | 55 | 46 | 50 | 52 |
| | IN-Cartoon | 52.4 | 43 | 42 | 41 | 43 | 40 | 47 | 36 | 54 | 50 | 63 | 78 | 62 | 58 | 65 | 64 |
| | IN-Drawing | 47.4 | 42 | 37 | 41 | 30 | 34 | 39 | 31 | 54 | 59 | 58 | 76 | 49 | 50 | 56 | 55 |
| Model Soup PRE-FT-EWC-LwF (Wortsman et al., 2022a) | IN-V2 | 48.7 | 41 | 39 | 40 | 38 | 38 | 46 | 35 | 46 | 43 | 61 | 70 | 59 | 54 | 58 | 60 |
| | IN-A | 39.8 | 33 | 30 | 31 | 33 | 30 | 38 | 27 | 31 | 31 | 55 | 60 | 52 | 45 | 48 | 52 |
| | IN-R | 50.4 | 45 | 43 | 43 | 42 | 42 | 48 | 37 | 48 | 49 | 60 | 71 | 59 | 54 | 57 | 58 |
| | IN-Sketch | 26.3 | 25 | 24 | 24 | 14 | 13 | 17 | 14 | 30 | 35 | 45 | 38 | 33 | 25 | 24 | 34 |
| | ObjNet | 44.2 | 37 | 35 | 34 | 39 | 32 | 44 | 36 | 35 | 35 | 57 | 67 | 56 | 49 | 53 | 55 |
| | IN-Cartoon | 53.8 | 47 | 45 | 45 | 42 | 40 | 48 | 37 | 57 | 52 | 64 | 78 | 62 | 59 | 66 | 65 |
| | IN-Drawing | 48.6 | 44 | 39 | 43 | 31 | 36 | 39 | 33 | 56 | 60 | 59 | 77 | 49 | 51 | 57 | 55 |

Table 41: Accuracy of ImageNet-1K with AugReg pre-trained ViT-B/32 with different fine-tuning methods and downstream datasets on each ImageNet-C corruption. For each corruption, accuracy is averaged across 5 levels of severity.

| Dataset | Method | Avg. | Noise | | | Blur | | | | Weather | | | | Digital | | | |
|---|---|---|---|---|---|---|---|---|---|---|---|---|---|---|---|---|---|
| | | | Gauss. | Shot | Impulse | Defocus | Glass | Motion | Zoom | Snow | Frost | Fog | Bright | Contrast | Elastic | Pixel | JPEG |
| Pre-Trained | | 53.5 | 55 | 54 | 54 | 46 | 43 | 49 | 41 | 43 | 53 | 54 | 69 | 52 | 57 | 67 | 64 |
| FT | IN-V2 | 53.4 | 53 | 51 | 51 | 48 | 43 | 50 | 39 | 47 | 55 | 56 | 70 | 54 | 57 | 66 | 64 |
| | IN-A | 47.5 | 48 | 46 | 46 | 45 | 38 | 46 | 32 | 40 | 48 | 48 | 62 | 49 | 48 | 58 | 58 |
| | IN-R | 47.0 | 47 | 46 | 45 | 42 | 43 | 44 | 31 | 42 | 50 | 46 | 61 | 44 | 49 | 57 | 57 |
| | IN-Sketch | 51.1 | 53 | 52 | 51 | 45 | 41 | 45 | 39 | 45 | 52 | 51 | 65 | 50 | 53 | 63 | 61 |
| | ObjNet | 47.8 | 47 | 46 | 46 | 44 | 39 | 45 | 33 | 40 | 49 | 49 | 64 | 44 | 50 | 61 | 59 |
| | IN-Cartoon | 50.1 | 52 | 50 | 50 | 42 | 36 | 46 | 33 | 42 | 50 | 50 | 72 | 48 | 53 | 68 | 61 |
| | IN-Drawing | 52.8 | 57 | 55 | 55 | 42 | 44 | 48 | 37 | 45 | 58 | 52 | 70 | 47 | 56 | 65 | 63 |
| Linear Probing | IN-V2 | 53.5 | 55 | 53 | 54 | 46 | 43 | 49 | 41 | 43 | 53 | 54 | 69 | 53 | 57 | 66 | 64 |
| | IN-A | 53.2 | 54 | 53 | 53 | 47 | 45 | 50 | 41 | 43 | 53 | 52 | 69 | 48 | 58 | 66 | 64 |
| | IN-R | 52.8 | 55 | 53 | 54 | 47 | 45 | 49 | 42 | 43 | 53 | 50 | 69 | 44 | 58 | 66 | 65 |
| | IN-Sketch | 51.0 | 54 | 52 | 53 | 44 | 41 | 46 | 40 | 41 | 51 | 48 | 67 | 47 | 55 | 64 | 62 |
| | ObjNet | 53.3 | 55 | 53 | 54 | 48 | 46 | 51 | 42 | 43 | 53 | 51 | 70 | 45 | 58 | 67 | 65 |
| | IN-Cartoon | 54.2 | 57 | 55 | 55 | 46 | 44 | 49 | 42 | 43 | 54 | 53 | 72 | 49 | 59 | 68 | 67 |
| | IN-Drawing | 54.4 | 57 | 55 | 56 | 47 | 47 | 50 | 42 | 46 | 57 | 52 | 71 | 45 | 59 | 67 | 67 |
| Visual Prompt (Bahng et al., 2022) | IN-V2 | 45.5 | 46 | 44 | 44 | 39 | 36 | 41 | 37 | 37 | 44 | 43 | 63 | 41 | 53 | 58 | 56 |
| | IN-A | 29.9 | 28 | 26 | 25 | 22 | 19 | 26 | 21 | 23 | 29 | 32 | 48 | 31 | 37 | 40 | 43 |
| | IN-R | 39.4 | 40 | 39 | 38 | 33 | 30 | 35 | 29 | 33 | 40 | 36 | 55 | 38 | 44 | 50 | 51 |
| | IN-Sketch | 43.1 | 46 | 45 | 44 | 35 | 31 | 37 | 32 | 37 | 43 | 39 | 59 | 40 | 47 | 54 | 57 |
| | ObjNet | 33.1 | 29 | 27 | 26 | 26 | 23 | 28 | 24 | 25 | 33 | 37 | 51 | 37 | 41 | 44 | 44 |
| | IN-Cartoon | 42.4 | 47 | 46 | 46 | 34 | 30 | 36 | 31 | 32 | 39 | 34 | 63 | 36 | 49 | 57 | 56 |
| | IN-Drawing | 42.3 | 46 | 44 | 44 | 32 | 32 | 35 | 30 | 35 | 47 | 36 | 60 | 36 | 47 | 54 | 53 |
| LoRA (Hu et al., 2021) | IN-V2 | 53.9 | 55 | 54 | 54 | 47 | 44 | 49 | 42 | 43 | 53 | 55 | 70 | 54 | 58 | 67 | 64 |
| | IN-A | 53.4 | 55 | 54 | 54 | 48 | 47 | 51 | 42 | 44 | 54 | 48 | 70 | 44 | 59 | 67 | 66 |
| | IN-R | 50.6 | 54 | 53 | 53 | 44 | 44 | 47 | 40 | 41 | 51 | 40 | 69 | 37 | 57 | 65 | 65 |
| | IN-Sketch | 51.6 | 55 | 53 | 54 | 45 | 43 | 47 | 41 | 41 | 51 | 46 | 69 | 44 | 56 | 66 | 64 |
| | ObjNet | 53.4 | 55 | 54 | 54 | 48 | 47 | 51 | 42 | 44 | 54 | 49 | 70 | 43 | 59 | 67 | 65 |
| | IN-Cartoon | 52.6 | 55 | 54 | 54 | 46 | 44 | 48 | 42 | 42 | 52 | 51 | 69 | 46 | 57 | 66 | 64 |
| | IN-Drawing | 52.1 | 56 | 55 | 55 | 43 | 45 | 47 | 40 | 44 | 56 | 43 | 70 | 38 | 58 | 66 | 65 |
| EWC (Kirkpatrick et al., 2017) | IN-V2 | 55.3 | 55 | 54 | 54 | 49 | 45 | 52 | 43 | 46 | 55 | 57 | 71 | 55 | 60 | 68 | 66 |
| | IN-A | 49.9 | 50 | 48 | 49 | 45 | 41 | 48 | 35 | 42 | 50 | 49 | 65 | 50 | 52 | 61 | 61 |
| | IN-R | 52.5 | 54 | 52 | 52 | 47 | 46 | 50 | 39 | 46 | 54 | 51 | 68 | 47 | 55 | 64 | 64 |
| | IN-Sketch | 53.8 | 55 | 54 | 54 | 47 | 44 | 49 | 40 | 46 | 54 | 54 | 69 | 52 | 57 | 66 | 65 |
| | ObjNet | 54.0 | 55 | 53 | 53 | 50 | 45 | 50 | 40 | 44 | 54 | 55 | 70 | 53 | 57 | 66 | 65 |
| | IN-Cartoon | 52.8 | 55 | 53 | 53 | 46 | 42 | 48 | 40 | 43 | 52 | 54 | 70 | 51 | 56 | 66 | 64 |
| | IN-Drawing | 55.1 | 56 | 54 | 54 | 48 | 46 | 51 | 40 | 48 | 59 | 55 | 70 | 54 | 58 | 67 | 65 |
| LwF (Li & Hoiem, 2017) | IN-V2 | 54.0 | 54 | 52 | 52 | 48 | 43 | 50 | 40 | 46 | 55 | 56 | 70 | 55 | 57 | 66 | 65 |
| | IN-A | 52.0 | 52 | 50 | 50 | 48 | 42 | 49 | 38 | 45 | 53 | 53 | 67 | 53 | 53 | 63 | 63 |
| | IN-R | 53.0 | 53 | 52 | 51 | 48 | 48 | 50 | 38 | 47 | 55 | 53 | 67 | 51 | 56 | 64 | 63 |
| | IN-Sketch | 52.5 | 54 | 53 | 53 | 46 | 42 | 47 | 40 | 46 | 54 | 53 | 67 | 51 | 55 | 64 | 62 |
| | ObjNet | 52.1 | 52 | 50 | 50 | 48 | 42 | 49 | 39 | 43 | 52 | 53 | 68 | 51 | 55 | 65 | 63 |
| | IN-Cartoon | 54.1 | 55 | 53 | 53 | 46 | 42 | 50 | 39 | 45 | 54 | 54 | 74 | 53 | 58 | 70 | 65 |
| | IN-Drawing | 55.2 | 58 | 56 | 56 | 45 | 46 | 51 | 40 | 47 | 59 | 54 | 73 | 50 | 58 | 68 | 65 |
| LP-FT (Kumar et al., 2022) | IN-V2 | 53.7 | 53 | 51 | 52 | 48 | 43 | 50 | 40 | 46 | 55 | 56 | 70 | 55 | 57 | 66 | 65 |
| | IN-A | 52.2 | 52 | 51 | 51 | 48 | 42 | 50 | 38 | 45 | 53 | 52 | 68 | 52 | 54 | 64 | 62 |
| | IN-R | 51.4 | 52 | 51 | 50 | 46 | 47 | 49 | 36 | 45 | 54 | 50 | 66 | 47 | 55 | 63 | 61 |
| | IN-Sketch | 51.1 | 53 | 52 | 52 | 44 | 41 | 45 | 40 | 45 | 53 | 49 | 66 | 50 | 53 | 63 | 61 |
| | ObjNet | 51.3 | 51 | 49 | 49 | 47 | 42 | 49 | 38 | 43 | 53 | 51 | 68 | 48 | 55 | 64 | 63 |
| | IN-Cartoon | 52.6 | 54 | 52 | 52 | 44 | 39 | 48 | 36 | 44 | 52 | 53 | 74 | 51 | 56 | 70 | 64 |
| | IN-Drawing | 53.6 | 58 | 56 | 57 | 42 | 45 | 48 | 37 | 48 | 60 | 50 | 72 | 44 | 57 | 66 | 65 |
| WiSE-FT (Wortsman et al., 2022b) | IN-V2 | 54.7 | 55 | 53 | 53 | 48 | 44 | 50 | 41 | 46 | 55 | 57 | 70 | 56 | 58 | 67 | 65 |
| | IN-A | 54.3 | 54 | 52 | 53 | 49 | 44 | 51 | 40 | 46 | 55 | 56 | 70 | 56 | 57 | 66 | 65 |
| | IN-R | 55.4 | 55 | 54 | 54 | 50 | 48 | 52 | 41 | 48 | 57 | 57 | 70 | 55 | 58 | 67 | 65 |
| | IN-Sketch | 54.4 | 56 | 55 | 55 | 48 | 44 | 49 | 42 | 47 | 55 | 55 | 69 | 55 | 57 | 66 | 64 |
| | ObjNet | 54.3 | 54 | 53 | 53 | 48 | 44 | 51 | 41 | 45 | 54 | 56 | 70 | 54 | 58 | 67 | 65 |
| | IN-Cartoon | 54.2 | 55 | 53 | 54 | 47 | 42 | 50 | 40 | 45 | 53 | 55 | 72 | 55 | 58 | 69 | 65 |
| | IN-Drawing | 56.9 | 58 | 57 | 57 | 48 | 47 | 53 | 42 | 49 | 60 | 58 | 73 | 55 | 60 | 69 | 67 |
| Model Soup PRE-FT-EWC-LwF (Wortsman et al., 2022a) | IN-V2 | 54.9 | 55 | 53 | 54 | 49 | 44 | 51 | 41 | 46 | 55 | 57 | 71 | 55 | 59 | 67 | 65 |
| | IN-A | 53.6 | 53 | 52 | 52 | 49 | 43 | 51 | 39 | 45 | 54 | 55 | 69 | 55 | 56 | 65 | 64 |
| | IN-R | 55.3 | 55 | 54 | 54 | 50 | 48 | 52 | 41 | 48 | 57 | 56 | 70 | 55 | 58 | 67 | 65 |
| | IN-Sketch | 54.4 | 56 | 55 | 55 | 48 | 44 | 49 | 42 | 47 | 55 | 55 | 69 | 54 | 57 | 66 | 64 |
| | ObjNet | 54.3 | 54 | 53 | 53 | 49 | 44 | 51 | 41 | 45 | 54 | 56 | 70 | 54 | 58 | 67 | 65 |
| | IN-Cartoon | 54.3 | 55 | 53 | 54 | 47 | 42 | 50 | 40 | 45 | 54 | 55 | 73 | 54 | 58 | 69 | 65 |
| | IN-Drawing | 56.9 | 59 | 57 | 57 | 49 | 47 | 53 | 42 | 49 | 60 | 57 | 73 | 55 | 60 | 69 | 67 |

Table 42: Accuracy of ImageNet-1K with SAM pre-trained ViT-B/32 with different fine-tuning methods and downstream datasets on each ImageNet-C corruption. For each corruption, accuracy is averaged across 5 levels of severity.

| Dataset | Method | Avg. | Noise | | | Blur | | | | Weather | | | | Digital | | | |
|---|---|---|---|---|---|---|---|---|---|---|---|---|---|---|---|---|---|
| | | | Gauss. | Shot | Impulse | Defocus | Glass | Motion | Zoom | Snow | Frost | Fog | Bright | Contrast | Elastic | Pixel | JPEG |
| Pre-Trained | | 48.8 | 49 | 48 | 47 | 45 | 47 | 46 | 41 | 35 | 45 | 36 | 65 | 35 | 59 | 68 | 65 |
| FT | IN-V2 | 51.1 | 48 | 46 | 45 | 50 | 48 | 50 | 42 | 41 | 47 | 45 | 68 | 41 | 61 | 67 | 67 |
| | IN-A | 43.1 | 36 | 35 | 33 | 50 | 42 | 45 | 32 | 36 | 38 | 34 | 59 | 35 | 53 | 59 | 59 |
| | IN-R | 49.2 | 49 | 49 | 48 | 49 | 47 | 46 | 34 | 42 | 48 | 44 | 62 | 44 | 55 | 61 | 62 |
| | IN-Sketch | 50.1 | 49 | 48 | 47 | 48 | 46 | 47 | 42 | 41 | 46 | 46 | 64 | 42 | 57 | 65 | 64 |
| | ObjNet | 44.7 | 38 | 38 | 35 | 49 | 43 | 46 | 36 | 36 | 40 | 35 | 61 | 34 | 56 | 62 | 61 |
| | IN-Cartoon | 47.3 | 44 | 43 | 40 | 46 | 41 | 46 | 38 | 37 | 41 | 40 | 67 | 42 | 55 | 66 | 65 |
| | IN-Drawing | 49.3 | 55 | 53 | 53 | 41 | 45 | 43 | 35 | 40 | 53 | 37 | 65 | 37 | 55 | 65 | 63 |
| HeadOnly | IN-V2 | 48.8 | 49 | 47 | 46 | 45 | 47 | 46 | 41 | 35 | 45 | 36 | 65 | 36 | 60 | 68 | 65 |
| | IN-A | 49.0 | 48 | 47 | 46 | 46 | 48 | 47 | 40 | 36 | 46 | 37 | 65 | 35 | 60 | 68 | 66 |
| | IN-R | 49.2 | 49 | 48 | 47 | 47 | 48 | 47 | 41 | 36 | 46 | 36 | 66 | 35 | 60 | 67 | 66 |
| | IN-Sketch | 48.5 | 49 | 47 | 47 | 47 | 47 | 47 | 41 | 37 | 46 | 33 | 65 | 33 | 59 | 66 | 65 |
| | ObjNet | 49.2 | 49 | 47 | 46 | 48 | 48 | 47 | 41 | 36 | 46 | 36 | 65 | 34 | 60 | 67 | 66 |
| | IN-Cartoon | 48.2 | 49 | 47 | 47 | 44 | 46 | 46 | 40 | 35 | 45 | 33 | 66 | 33 | 59 | 68 | 65 |
| | IN-Drawing | 49.0 | 52 | 50 | 50 | 43 | 47 | 44 | 39 | 37 | 48 | 36 | 65 | 36 | 59 | 67 | 64 |
| Visual Prompt (Bahng et al., 2022) | IN-V2 | 41.5 | 43 | 42 | 41 | 35 | 36 | 36 | 35 | 31 | 38 | 25 | 59 | 25 | 54 | 60 | 61 |
| | IN-A | 19.2 | 21 | 20 | 18 | 8 | 13 | 11 | 12 | 13 | 18 | 6 | 37 | 6 | 32 | 33 | 38 |
| | IN-R | 34.3 | 39 | 38 | 37 | 25 | 27 | 26 | 24 | 25 | 31 | 19 | 54 | 19 | 45 | 50 | 54 |
| | IN-Sketch | 34.2 | 36 | 36 | 35 | 27 | 28 | 27 | 26 | 26 | 32 | 19 | 54 | 18 | 45 | 52 | 54 |
| | ObjNet | 27.4 | 26 | 26 | 24 | 22 | 23 | 22 | 22 | 18 | 24 | 15 | 45 | 14 | 40 | 44 | 47 |
| | IN-Cartoon | 41.1 | 43 | 42 | 41 | 36 | 37 | 37 | 36 | 29 | 37 | 21 | 61 | 21 | 53 | 62 | 62 |
| | IN-Drawing | 41.1 | 46 | 45 | 45 | 33 | 35 | 33 | 31 | 31 | 40 | 24 | 59 | 24 | 52 | 59 | 60 |
| LoRA (Hu et al., 2021) | IN-V2 | 48.8 | 49 | 47 | 47 | 45 | 47 | 46 | 41 | 35 | 46 | 36 | 65 | 36 | 60 | 68 | 65 |
| | IN-A | 49.4 | 49 | 47 | 46 | 47 | 48 | 47 | 41 | 36 | 46 | 38 | 66 | 35 | 60 | 68 | 66 |
| | IN-R | 49.4 | 49 | 48 | 47 | 47 | 48 | 47 | 41 | 36 | 46 | 36 | 66 | 35 | 60 | 68 | 66 |
| | IN-Sketch | 48.8 | 49 | 48 | 47 | 47 | 48 | 47 | 41 | 36 | 46 | 33 | 65 | 33 | 59 | 67 | 66 |
| | ObjNet | 49.4 | 49 | 47 | 46 | 48 | 49 | 48 | 41 | 36 | 46 | 36 | 66 | 34 | 61 | 68 | 67 |
| | IN-Cartoon | 48.1 | 49 | 47 | 46 | 45 | 46 | 46 | 41 | 35 | 45 | 32 | 65 | 32 | 59 | 68 | 65 |
| | IN-Drawing | 49.1 | 51 | 50 | 49 | 44 | 47 | 45 | 40 | 37 | 47 | 35 | 65 | 35 | 59 | 67 | 64 |
| EWC (Kirkpatrick et al., 2017) | IN-V2 | 49.5 | 49 | 47 | 46 | 46 | 48 | 47 | 41 | 36 | 46 | 38 | 66 | 37 | 60 | 68 | 66 |
| | IN-A | 47.7 | 46 | 44 | 43 | 48 | 48 | 46 | 39 | 35 | 43 | 34 | 63 | 33 | 59 | 66 | 66 |
| | IN-R | 50.1 | 50 | 49 | 48 | 48 | 48 | 47 | 40 | 37 | 46 | 39 | 66 | 38 | 60 | 67 | 67 |
| | IN-Sketch | 49.8 | 50 | 48 | 48 | 48 | 48 | 47 | 41 | 38 | 46 | 39 | 66 | 37 | 60 | 67 | 66 |
| | ObjNet | 49.9 | 48 | 47 | 45 | 49 | 49 | 48 | 42 | 37 | 46 | 40 | 66 | 37 | 61 | 68 | 67 |
| | IN-Cartoon | 48.4 | 48 | 47 | 45 | 45 | 46 | 46 | 40 | 35 | 44 | 36 | 66 | 37 | 59 | 68 | 65 |
| | IN-Drawing | 49.9 | 53 | 51 | 51 | 44 | 47 | 45 | 39 | 38 | 49 | 37 | 65 | 38 | 59 | 67 | 65 |
| LwF (Li & Hoiem, 2017) | IN-V2 | 51.1 | 48 | 47 | 45 | 49 | 48 | 50 | 43 | 41 | 47 | 44 | 67 | 42 | 61 | 68 | 67 |
| | IN-A | 46.3 | 40 | 39 | 37 | 51 | 45 | 48 | 36 | 38 | 41 | 38 | 62 | 38 | 57 | 62 | 62 |
| | IN-R | 51.6 | 52 | 51 | 50 | 51 | 49 | 49 | 38 | 43 | 50 | 46 | 66 | 44 | 58 | 64 | 64 |
| | IN-Sketch | 50.6 | 49 | 48 | 47 | 48 | 46 | 47 | 43 | 41 | 47 | 46 | 65 | 43 | 58 | 66 | 64 |
| | ObjNet | 48.1 | 43 | 42 | 40 | 50 | 47 | 49 | 40 | 38 | 43 | 39 | 64 | 37 | 59 | 65 | 64 |
| | IN-Cartoon | 48.8 | 45 | 44 | 42 | 47 | 43 | 47 | 40 | 38 | 43 | 42 | 68 | 43 | 58 | 66 | 66 |
| | IN-Drawing | 50.2 | 56 | 54 | 54 | 42 | 46 | 44 | 37 | 40 | 53 | 38 | 66 | 38 | 57 | 66 | 64 |
| LP-FT (Kumar et al., 2022) | IN-V2 | 51.1 | 48 | 46 | 45 | 50 | 48 | 50 | 42 | 41 | 47 | 44 | 67 | 42 | 61 | 67 | 67 |
| | IN-A | 44.9 | 39 | 37 | 35 | 51 | 44 | 47 | 34 | 38 | 41 | 34 | 62 | 35 | 55 | 61 | 61 |
| | IN-R | 50.4 | 51 | 51 | 50 | 50 | 48 | 48 | 36 | 43 | 49 | 43 | 64 | 41 | 57 | 63 | 63 |
| | IN-Sketch | 49.4 | 49 | 48 | 46 | 48 | 46 | 47 | 41 | 41 | 46 | 43 | 64 | 39 | 56 | 65 | 63 |
| | ObjNet | 45.9 | 41 | 40 | 38 | 50 | 44 | 47 | 36 | 38 | 42 | 35 | 62 | 35 | 57 | 63 | 62 |
| | IN-Cartoon | 47.7 | 45 | 44 | 41 | 46 | 42 | 46 | 38 | 37 | 42 | 39 | 68 | 41 | 56 | 67 | 65 |
| | IN-Drawing | 48.4 | 55 | 53 | 53 | 40 | 44 | 42 | 34 | 40 | 53 | 32 | 65 | 33 | 55 | 65 | 63 |
| WiSE-FT (Wortsman et al., 2022b) | IN-V2 | 50.4 | 49 | 47 | 46 | 47 | 48 | 48 | 42 | 39 | 47 | 41 | 67 | 39 | 61 | 68 | 66 |
| | IN-A | 50.4 | 47 | 45 | 44 | 49 | 49 | 50 | 42 | 40 | 47 | 42 | 66 | 40 | 61 | 67 | 67 |
| | IN-R | 52.8 | 54 | 53 | 52 | 49 | 50 | 49 | 42 | 43 | 51 | 45 | 68 | 43 | 61 | 68 | 67 |
| | IN-Sketch | 50.7 | 50 | 49 | 48 | 47 | 47 | 48 | 43 | 40 | 48 | 42 | 66 | 41 | 59 | 68 | 65 |
| | ObjNet | 50.1 | 47 | 45 | 44 | 49 | 49 | 49 | 42 | 39 | 46 | 42 | 66 | 39 | 61 | 68 | 66 |
| | IN-Cartoon | 50.0 | 48 | 47 | 45 | 47 | 46 | 48 | 41 | 38 | 46 | 41 | 68 | 41 | 60 | 69 | 66 |
| | IN-Drawing | 51.5 | 55 | 53 | 54 | 44 | 47 | 46 | 40 | 41 | 53 | 40 | 67 | 39 | 59 | 68 | 66 |
| Model Soup PRE-FT-EWC-LwF (Wortsman et al., 2022a) | IN-V2 | 50.5 | 49 | 47 | 46 | 47 | 48 | 49 | 42 | 39 | 47 | 41 | 67 | 39 | 61 | 68 | 66 |
| | IN-A | 50.0 | 46 | 45 | 43 | 51 | 49 | 50 | 41 | 40 | 46 | 42 | 65 | 39 | 61 | 67 | 67 |
| | IN-R | 52.8 | 54 | 53 | 52 | 49 | 50 | 49 | 41 | 42 | 50 | 45 | 68 | 43 | 61 | 67 | 67 |
| | IN-Sketch | 50.9 | 50 | 49 | 48 | 47 | 48 | 48 | 43 | 41 | 48 | 43 | 66 | 41 | 60 | 67 | 65 |
| | ObjNet | 50.2 | 47 | 45 | 44 | 49 | 49 | 50 | 42 | 39 | 46 | 42 | 66 | 39 | 61 | 68 | 66 |
| | IN-Cartoon | 49.8 | 48 | 46 | 45 | 47 | 46 | 47 | 41 | 38 | 45 | 41 | 68 | 41 | 60 | 69 | 66 |
| | IN-Drawing | 51.5 | 56 | 54 | 54 | 44 | 47 | 46 | 39 | 41 | 53 | 39 | 67 | 39 | 59 | 68 | 66 |

Table 43: Accuracy of ImageNet-21K with AugReg pre-trained ViT-B/32 with different fine-tuning methods and downstream datasets on each ImageNet-C corruption For each corruption, accuracy is averaged across 5 levels of severity.

| Dataset | Method | Avg. | Noise | | | Blur | | | | Weather | | | | Digital | | | |
|---|---|---|---|---|---|---|---|---|---|---|---|---|---|---|---|---|---|
| | | | Gauss. | Shot | Impulse | Defocus | Glass | Motion | Zoom | Snow | Frost | Fog | Bright | Contrast | Elastic | Pixel | JPEG |
| Pre-Trained | | 60.5 | 61 | 60 | 60 | 56 | 49 | 61 | 50 | 54 | 56 | 60 | 75 | 59 | 62 | 72 | 70 |
| FT | IN-V2 | 60.2 | 59 | 57 | 58 | 58 | 48 | 62 | 50 | 55 | 57 | 62 | 75 | 61 | 60 | 72 | 70 |
| | IN-A | 56.9 | 54 | 52 | 53 | 56 | 46 | 59 | 46 | 51 | 53 | 58 | 71 | 59 | 56 | 69 | 68 |
| | IN-R | 55.3 | 54 | 53 | 53 | 54 | 50 | 55 | 43 | 50 | 55 | 56 | 68 | 55 | 55 | 66 | 65 |
| | IN-Sketch | 57.0 | 58 | 57 | 56 | 54 | 44 | 56 | 48 | 52 | 55 | 58 | 71 | 56 | 56 | 68 | 66 |
| | ObjNet | 55.3 | 54 | 52 | 52 | 54 | 46 | 58 | 44 | 49 | 52 | 54 | 71 | 54 | 56 | 69 | 67 |
| | IN-Cartoon | 56.7 | 57 | 55 | 56 | 52 | 41 | 58 | 45 | 52 | 51 | 55 | 76 | 55 | 57 | 72 | 67 |
| | IN-Drawing | 59.5 | 60 | 58 | 59 | 53 | 48 | 57 | 43 | 56 | 61 | 60 | 76 | 59 | 60 | 73 | 69 |
| Linear Probing | IN-V2 | 60.1 | 61 | 59 | 60 | 56 | 49 | 61 | 50 | 54 | 56 | 60 | 75 | 58 | 62 | 72 | 70 |
| | IN-A | 59.4 | 60 | 59 | 59 | 55 | 48 | 60 | 49 | 53 | 55 | 60 | 74 | 58 | 61 | 71 | 69 |
| | IN-R | 59.7 | 60 | 59 | 59 | 55 | 49 | 60 | 50 | 53 | 55 | 60 | 74 | 57 | 62 | 72 | 69 |
| | IN-Sketch | 58.4 | 59 | 58 | 58 | 54 | 47 | 58 | 49 | 53 | 54 | 58 | 73 | 57 | 60 | 70 | 68 |
| | ObjNet | 60.3 | 61 | 59 | 60 | 57 | 49 | 62 | 51 | 54 | 55 | 61 | 74 | 58 | 62 | 72 | 69 |
| | IN-Cartoon | 62.6 | 63 | 62 | 62 | 58 | 50 | 64 | 52 | 55 | 57 | 62 | 79 | 61 | 64 | 76 | 74 |
| | IN-Drawing | 63.0 | 63 | 62 | 62 | 57 | 51 | 63 | 52 | 58 | 61 | 64 | 78 | 60 | 65 | 75 | 73 |
| Visual Prompt (Bahng et al., 2022) | IN-V2 | 51.1 | 51 | 49 | 49 | 48 | 39 | 50 | 44 | 43 | 46 | 48 | 69 | 47 | 57 | 64 | 63 |
| | IN-A | 39.7 | 37 | 35 | 35 | 35 | 27 | 40 | 33 | 33 | 35 | 40 | 58 | 37 | 45 | 53 | 53 |
| | IN-R | 42.6 | 41 | 39 | 39 | 40 | 34 | 41 | 33 | 35 | 42 | 40 | 61 | 39 | 47 | 54 | 54 |
| | IN-Sketch | 46.2 | 46 | 45 | 45 | 42 | 33 | 43 | 37 | 39 | 44 | 45 | 65 | 42 | 50 | 59 | 59 |
| | ObjNet | 30.9 | 27 | 25 | 24 | 26 | 19 | 30 | 25 | 23 | 27 | 29 | 51 | 28 | 38 | 45 | 46 |
| | IN-Cartoon | 45.5 | 44 | 43 | 42 | 42 | 31 | 45 | 38 | 39 | 39 | 40 | 67 | 40 | 52 | 62 | 59 |
| | IN-Drawing | 46.7 | 48 | 46 | 45 | 39 | 37 | 41 | 35 | 41 | 51 | 40 | 66 | 39 | 52 | 62 | 59 |
| LoRA (Hu et al., 2021) | IN-V2 | 60.5 | 61 | 60 | 60 | 56 | 49 | 61 | 50 | 54 | 56 | 60 | 75 | 59 | 62 | 72 | 70 |
| | IN-A | 61.2 | 61 | 60 | 60 | 58 | 50 | 63 | 52 | 55 | 56 | 62 | 76 | 59 | 63 | 73 | 70 |
| | IN-R | 60.9 | 61 | 60 | 60 | 57 | 50 | 62 | 51 | 54 | 56 | 61 | 75 | 57 | 63 | 73 | 70 |
| | IN-Sketch | 60.8 | 62 | 60 | 60 | 57 | 49 | 61 | 51 | 54 | 57 | 61 | 76 | 59 | 62 | 73 | 70 |
| | ObjNet | 61.3 | 61 | 60 | 60 | 58 | 50 | 63 | 52 | 55 | 56 | 62 | 76 | 58 | 63 | 73 | 71 |
| | IN-Cartoon | 60.2 | 61 | 60 | 60 | 57 | 49 | 61 | 50 | 53 | 55 | 60 | 75 | 57 | 62 | 72 | 70 |
| | IN-Drawing | 61.0 | 61 | 60 | 60 | 56 | 50 | 61 | 50 | 56 | 59 | 62 | 76 | 58 | 63 | 73 | 70 |
| EWC (Kirkpatrick et al., 2017) | IN-V2 | 62.1 | 62 | 60 | 61 | 60 | 50 | 64 | 53 | 56 | 58 | 63 | 76 | 60 | 63 | 74 | 71 |
| | IN-A | 60.9 | 59 | 57 | 58 | 60 | 50 | 63 | 51 | 55 | 57 | 63 | 75 | 62 | 61 | 73 | 70 |
| | IN-R | 60.0 | 60 | 58 | 58 | 58 | 52 | 60 | 49 | 54 | 58 | 60 | 74 | 59 | 61 | 70 | 69 |
| | IN-Sketch | 61.1 | 62 | 61 | 61 | 57 | 49 | 61 | 51 | 55 | 58 | 62 | 75 | 60 | 62 | 72 | 70 |
| | ObjNet | 60.1 | 59 | 58 | 58 | 60 | 49 | 62 | 49 | 54 | 57 | 60 | 74 | 59 | 60 | 72 | 71 |
| | IN-Cartoon | 59.0 | 59 | 58 | 58 | 55 | 46 | 60 | 48 | 53 | 54 | 60 | 75 | 58 | 60 | 72 | 69 |
| | IN-Drawing | 61.2 | 62 | 61 | 61 | 55 | 49 | 61 | 47 | 58 | 62 | 62 | 76 | 60 | 62 | 73 | 70 |
| LwF (Li & Hoiem, 2017) | IN-V2 | 61.1 | 61 | 59 | 60 | 58 | 49 | 62 | 51 | 55 | 57 | 62 | 75 | 60 | 62 | 73 | 71 |
| | IN-A | 60.6 | 60 | 58 | 58 | 58 | 49 | 62 | 51 | 55 | 56 | 62 | 75 | 60 | 61 | 73 | 70 |
| | IN-R | 60.9 | 61 | 60 | 60 | 57 | 53 | 61 | 50 | 55 | 59 | 62 | 74 | 59 | 62 | 72 | 69 |
| | IN-Sketch | 59.1 | 60 | 59 | 59 | 56 | 46 | 59 | 49 | 54 | 56 | 60 | 73 | 57 | 59 | 70 | 68 |
| | ObjNet | 59.1 | 59 | 57 | 57 | 58 | 48 | 61 | 49 | 53 | 54 | 59 | 74 | 57 | 60 | 72 | 70 |
| | IN-Cartoon | 63.0 | 63 | 61 | 61 | 59 | 50 | 64 | 53 | 56 | 58 | 63 | 80 | 62 | 65 | 77 | 74 |
| | IN-Drawing | 63.1 | 64 | 61 | 62 | 59 | 51 | 63 | 50 | 58 | 62 | 63 | 79 | 61 | 64 | 77 | 73 |
| LP-FT (Kumar et al., 2022) | IN-V2 | 60.7 | 60 | 59 | 59 | 58 | 49 | 62 | 51 | 55 | 57 | 62 | 75 | 60 | 61 | 72 | 70 |
| | IN-A | 59.5 | 58 | 57 | 57 | 57 | 48 | 61 | 50 | 54 | 55 | 62 | 73 | 61 | 59 | 71 | 68 |
| | IN-R | 59.0 | 58 | 57 | 57 | 56 | 52 | 59 | 48 | 53 | 58 | 60 | 72 | 58 | 60 | 70 | 67 |
| | IN-Sketch | 57.5 | 58 | 57 | 57 | 54 | 45 | 57 | 48 | 53 | 54 | 59 | 71 | 56 | 58 | 68 | 66 |
| | ObjNet | 58.9 | 59 | 57 | 57 | 57 | 47 | 61 | 49 | 53 | 55 | 59 | 73 | 57 | 59 | 71 | 69 |
| | IN-Cartoon | 61.3 | 61 | 59 | 60 | 57 | 46 | 63 | 50 | 56 | 56 | 60 | 80 | 61 | 62 | 76 | 72 |
| | IN-Drawing | 62.6 | 63 | 61 | 62 | 57 | 51 | 62 | 48 | 59 | 63 | 63 | 78 | 61 | 63 | 76 | 72 |
| WiSE-FT (Wortsman et al., 2022b) | IN-V2 | 61.7 | 62 | 60 | 61 | 58 | 50 | 63 | 52 | 56 | 58 | 63 | 76 | 61 | 63 | 73 | 71 |
| | IN-A | 61.9 | 61 | 59 | 60 | 59 | 50 | 64 | 52 | 56 | 58 | 64 | 76 | 62 | 63 | 74 | 71 |
| | IN-R | 62.4 | 62 | 61 | 61 | 59 | 54 | 63 | 52 | 56 | 60 | 64 | 76 | 61 | 63 | 73 | 71 |
| | IN-Sketch | 61.0 | 62 | 61 | 61 | 57 | 48 | 61 | 51 | 56 | 58 | 62 | 75 | 60 | 61 | 72 | 70 |
| | ObjNet | 61.2 | 61 | 60 | 60 | 58 | 50 | 63 | 51 | 55 | 57 | 62 | 75 | 60 | 62 | 73 | 71 |
| | IN-Cartoon | 60.9 | 62 | 60 | 61 | 56 | 47 | 62 | 50 | 56 | 56 | 61 | 78 | 60 | 62 | 74 | 71 |
| | IN-Drawing | 63.6 | 65 | 63 | 64 | 59 | 51 | 64 | 51 | 59 | 62 | 64 | 78 | 62 | 64 | 75 | 73 |
| Model Soup PRE-FT-EWC-LwF (Wortsman et al., 2022a) | IN-V2 | 61.7 | 62 | 60 | 61 | 59 | 50 | 63 | 52 | 56 | 58 | 63 | 76 | 61 | 63 | 73 | 71 |
| | IN-A | 61.6 | 60 | 59 | 59 | 59 | 50 | 64 | 52 | 56 | 57 | 64 | 75 | 62 | 62 | 74 | 71 |
| | IN-R | 62.2 | 62 | 61 | 61 | 59 | 54 | 63 | 51 | 56 | 60 | 63 | 76 | 61 | 63 | 73 | 71 |
| | IN-Sketch | 61.0 | 62 | 61 | 61 | 57 | 48 | 61 | 51 | 56 | 58 | 62 | 75 | 60 | 61 | 72 | 70 |
| | ObjNet | 60.9 | 61 | 59 | 59 | 58 | 49 | 63 | 51 | 55 | 56 | 61 | 75 | 59 | 62 | 73 | 71 |
| | IN-Cartoon | 61.1 | 62 | 60 | 60 | 57 | 47 | 62 | 50 | 56 | 56 | 61 | 78 | 60 | 62 | 74 | 72 |
| | IN-Drawing | 63.3 | 65 | 63 | 64 | 58 | 51 | 64 | 51 | 59 | 62 | 63 | 78 | 62 | 64 | 75 | 73 |

Table 44: Accuracy of LAION-2B pre-trained ViT-B/32 with different fine-tuning methods and downstream datasets on each ImageNet-C corruption. For each corruption, accuracy is averaged across 5 levels of severity.

| Dataset | Method | Avg. | Noise | | | Blur | | | | Weather | | | | Digital | | | |
|---|---|---|---|---|---|---|---|---|---|---|---|---|---|---|---|---|---|
| | | | Gauss. | Shot | Impulse | Defocus | Glass | Motion | Zoom | Snow | Frost | Fog | Bright | Contrast | Elastic | Pixel | JPEG |
| Pre-Trained | | 57.5 | 56 | 55 | 55 | 53 | 42 | 57 | 46 | 52 | 51 | 61 | 75 | 64 | 59 | 68 | 69 |
| FT | IN-V2 | 21.7 | 20 | 19 | 18 | 11 | 15 | 15 | 15 | 14 | 20 | 23 | 36 | 25 | 28 | 31 | 36 |
| | IN-A | 7.0 | 8 | 7 | 7 | 4 | 5 | 4 | 5 | 4 | 5 | 7 | 12 | 8 | 8 | 10 | 11 |
| | IN-R | 20.9 | 20 | 19 | 18 | 14 | 17 | 17 | 15 | 12 | 20 | 22 | 33 | 25 | 23 | 28 | 31 |
| | IN-Sketch | 9.7 | 10 | 10 | 9 | 4 | 5 | 5 | 6 | 7 | 11 | 11 | 17 | 13 | 9 | 11 | 17 |
| | ObjNet | 12.2 | 12 | 11 | 11 | 7 | 9 | 10 | 10 | 5 | 9 | 13 | 20 | 15 | 14 | 18 | 20 |
| | IN-Cartoon | 27.3 | 25 | 24 | 21 | 10 | 16 | 17 | 15 | 21 | 25 | 27 | 51 | 31 | 35 | 46 | 47 |
| | IN-Drawing | 21.6 | 20 | 17 | 19 | 9 | 13 | 14 | 13 | 15 | 32 | 20 | 46 | 22 | 24 | 29 | 30 |
| Linear Probing | IN-V2 | 58.1 | 57 | 56 | 56 | 53 | 42 | 58 | 46 | 53 | 52 | 63 | 76 | 65 | 59 | 68 | 68 |
| | IN-A | 58.2 | 56 | 54 | 55 | 52 | 44 | 58 | 47 | 54 | 53 | 64 | 76 | 66 | 59 | 68 | 68 |
| | IN-R | 56.1 | 54 | 53 | 53 | 50 | 44 | 55 | 43 | 51 | 54 | 61 | 73 | 64 | 57 | 65 | 66 |
| | IN-Sketch | 55.6 | 55 | 54 | 53 | 50 | 42 | 54 | 43 | 51 | 52 | 60 | 73 | 61 | 55 | 65 | 67 |
| | ObjNet | 56.6 | 55 | 53 | 53 | 51 | 42 | 57 | 44 | 51 | 51 | 62 | 75 | 63 | 58 | 67 | 67 |
| | IN-Cartoon | 51.2 | 50 | 47 | 48 | 43 | 36 | 49 | 38 | 46 | 45 | 57 | 72 | 59 | 53 | 62 | 61 |
| | IN-Drawing | 49.8 | 53 | 52 | 52 | 37 | 38 | 45 | 34 | 45 | 52 | 54 | 67 | 53 | 51 | 57 | 57 |
| Visual Prompt (Bahng et al., 2022) | IN-V2 | 47.9 | 49 | 48 | 47 | 39 | 32 | 45 | 38 | 41 | 42 | 49 | 68 | 50 | 51 | 58 | 61 |
| | IN-A | 36.9 | 34 | 33 | 31 | 28 | 20 | 33 | 26 | 31 | 31 | 42 | 62 | 42 | 42 | 46 | 51 |
| | IN-R | 43.9 | 42 | 42 | 40 | 39 | 29 | 41 | 34 | 38 | 41 | 46 | 65 | 44 | 48 | 52 | 58 |
| | IN-Sketch | 45.1 | 44 | 43 | 42 | 39 | 28 | 42 | 36 | 39 | 41 | 48 | 66 | 47 | 48 | 54 | 59 |
| | ObjNet | 27.7 | 27 | 26 | 25 | 21 | 14 | 23 | 19 | 21 | 22 | 31 | 50 | 30 | 32 | 37 | 40 |
| | IN-Cartoon | 44.9 | 44 | 43 | 42 | 39 | 28 | 41 | 34 | 39 | 38 | 44 | 68 | 46 | 49 | 57 | 60 |
| | IN-Drawing | 47.2 | 50 | 50 | 48 | 36 | 33 | 43 | 35 | 42 | 47 | 44 | 67 | 43 | 52 | 57 | 61 |
| LoRA (Hu et al., 2021) | IN-V2 | 58.1 | 57 | 56 | 56 | 53 | 42 | 58 | 46 | 53 | 52 | 63 | 76 | 65 | 59 | 68 | 68 |
| | IN-A | 57.9 | 56 | 54 | 55 | 51 | 44 | 58 | 46 | 54 | 53 | 64 | 76 | 65 | 58 | 68 | 67 |
| | IN-R | 54.8 | 53 | 52 | 52 | 48 | 43 | 53 | 41 | 49 | 53 | 60 | 73 | 62 | 55 | 64 | 65 |
| | IN-Sketch | 53.4 | 53 | 53 | 52 | 45 | 40 | 51 | 41 | 49 | 50 | 58 | 70 | 58 | 52 | 63 | 64 |
| | ObjNet | 54.9 | 53 | 52 | 52 | 47 | 41 | 54 | 42 | 48 | 49 | 61 | 74 | 61 | 56 | 67 | 66 |
| | IN-Cartoon | 50.3 | 49 | 47 | 47 | 43 | 35 | 49 | 37 | 46 | 45 | 56 | 72 | 58 | 52 | 61 | 60 |
| | IN-Drawing | 49.8 | 52 | 51 | 51 | 38 | 38 | 46 | 34 | 45 | 52 | 54 | 67 | 53 | 51 | 58 | 56 |
| EWC (Kirkpatrick et al., 2017) | IN-V2 | 48.4 | 47 | 45 | 46 | 40 | 34 | 44 | 35 | 36 | 45 | 58 | 69 | 56 | 50 | 58 | 62 |
| | IN-A | 30.7 | 28 | 26 | 27 | 24 | 19 | 27 | 21 | 19 | 23 | 41 | 46 | 44 | 31 | 38 | 44 |
| | IN-R | 53.0 | 51 | 50 | 50 | 46 | 44 | 50 | 38 | 48 | 52 | 58 | 70 | 58 | 53 | 61 | 65 |
| | IN-Sketch | 39.7 | 40 | 39 | 39 | 27 | 23 | 32 | 26 | 43 | 46 | 56 | 51 | 41 | 37 | 44 | 51 |
| | ObjNet | 43.0 | 38 | 36 | 37 | 38 | 33 | 42 | 32 | 31 | 39 | 50 | 65 | 51 | 45 | 53 | 56 |
| | IN-Cartoon | 48.1 | 46 | 44 | 44 | 41 | 32 | 45 | 33 | 45 | 43 | 53 | 71 | 54 | 50 | 61 | 59 |
| | IN-Drawing | 43.6 | 47 | 42 | 46 | 26 | 30 | 35 | 27 | 44 | 52 | 48 | 68 | 46 | 45 | 48 | 50 |
| LwF (Li & Hoiem, 2017) | IN-V2 | 26.1 | 23 | 21 | 20 | 14 | 19 | 20 | 18 | 17 | 23 | 30 | 43 | 30 | 33 | 37 | 43 |
| | IN-A | 14.5 | 15 | 14 | 14 | 8 | 10 | 10 | 9 | 8 | 10 | 15 | 25 | 16 | 17 | 21 | 24 |
| | IN-R | 28.9 | 26 | 24 | 23 | 22 | 24 | 25 | 22 | 18 | 28 | 30 | 45 | 32 | 33 | 39 | 42 |
| | IN-Sketch | 13.7 | 15 | 14 | 13 | 5 | 7 | 8 | 8 | 10 | 16 | 15 | 24 | 16 | 14 | 16 | 24 |
| | ObjNet | 20.7 | 19 | 17 | 17 | 13 | 16 | 18 | 16 | 9 | 16 | 22 | 33 | 24 | 25 | 31 | 33 |
| | IN-Cartoon | 36.6 | 32 | 31 | 26 | 17 | 25 | 26 | 21 | 30 | 36 | 35 | 64 | 39 | 48 | 57 | 62 |
| | IN-Drawing | 28.2 | 25 | 22 | 22 | 14 | 20 | 20 | 17 | 22 | 38 | 29 | 55 | 27 | 32 | 39 | 40 |
| LP-FT (Kumar et al., 2022) | IN-V2 | 22.0 | 21 | 19 | 19 | 12 | 16 | 16 | 15 | 13 | 18 | 23 | 36 | 24 | 28 | 32 | 36 |
| | IN-A | 9.0 | 10 | 10 | 10 | 5 | 6 | 6 | 6 | 5 | 6 | 9 | 15 | 10 | 11 | 13 | 14 |
| | IN-R | 22.2 | 21 | 20 | 19 | 15 | 18 | 18 | 16 | 13 | 21 | 22 | 35 | 26 | 25 | 30 | 33 |
| | IN-Sketch | 11.5 | 12 | 12 | 11 | 5 | 6 | 6 | 7 | 8 | 13 | 12 | 21 | 15 | 12 | 13 | 20 |
| | ObjNet | 14.4 | 14 | 13 | 12 | 9 | 12 | 12 | 11 | 7 | 11 | 15 | 23 | 17 | 17 | 21 | 23 |
| | IN-Cartoon | 28.6 | 26 | 25 | 21 | 12 | 18 | 19 | 16 | 22 | 26 | 26 | 53 | 32 | 37 | 47 | 50 |
| | IN-Drawing | 22.7 | 21 | 18 | 19 | 9 | 13 | 14 | 13 | 17 | 34 | 22 | 47 | 25 | 26 | 30 | 30 |
| WiSE-FT (Wortsman et al., 2022b) | IN-V2 | 43.5 | 40 | 37 | 38 | 33 | 33 | 37 | 31 | 33 | 41 | 52 | 63 | 52 | 49 | 55 | 59 |
| | IN-A | 37.0 | 35 | 33 | 33 | 29 | 29 | 32 | 26 | 24 | 31 | 48 | 53 | 47 | 39 | 45 | 51 |
| | IN-R | 46.5 | 44 | 42 | 42 | 39 | 40 | 41 | 33 | 36 | 46 | 53 | 65 | 53 | 50 | 55 | 60 |
| | IN-Sketch | 33.3 | 35 | 33 | 33 | 20 | 20 | 23 | 19 | 28 | 38 | 45 | 49 | 40 | 34 | 35 | 47 |
| | ObjNet | 41.6 | 39 | 37 | 37 | 33 | 32 | 38 | 31 | 27 | 37 | 49 | 60 | 49 | 45 | 53 | 57 |
| | IN-Cartoon | 48.8 | 48 | 46 | 45 | 32 | 34 | 40 | 30 | 45 | 47 | 54 | 73 | 52 | 56 | 64 | 65 |
| | IN-Drawing | 44.7 | 45 | 40 | 43 | 28 | 33 | 35 | 27 | 42 | 54 | 49 | 71 | 47 | 48 | 54 | 56 |
| Model Soup PRE-FT-EWC-LwF (Wortsman et al., 2022a) | IN-V2 | 46.5 | 45 | 43 | 43 | 35 | 36 | 39 | 32 | 38 | 44 | 54 | 65 | 53 | 52 | 57 | 62 |
| | IN-A | 40.5 | 39 | 36 | 37 | 33 | 32 | 35 | 28 | 28 | 34 | 50 | 58 | 49 | 43 | 50 | 55 |
| | IN-R | 49.8 | 49 | 48 | 48 | 42 | 43 | 45 | 35 | 41 | 49 | 54 | 67 | 54 | 53 | 58 | 62 |
| | IN-Sketch | 34.7 | 38 | 36 | 36 | 21 | 21 | 24 | 19 | 32 | 41 | 45 | 50 | 39 | 35 | 37 | 48 |
| | ObjNet | 45.1 | 44 | 42 | 42 | 36 | 35 | 42 | 33 | 32 | 40 | 51 | 64 | 50 | 49 | 56 | 60 |
| | IN-Cartoon | 50.9 | 52 | 50 | 49 | 34 | 35 | 41 | 30 | 50 | 50 | 56 | 75 | 53 | 58 | 64 | 67 |
| | IN-Drawing | 46.2 | 48 | 43 | 47 | 28 | 34 | 36 | 27 | 46 | 56 | 51 | 72 | 46 | 50 | 53 | 57 |

Table 45: Accuracy of OpenAI CLIP ViT-B/32 with different fine-tuning methods and downstream datasets on each ImageNet-C corruption. For each corruption, accuracy is averaged across 5 levels of severity.

| Dataset | Method | Avg. | Noise | | | Blur | | | | Weather | | | | Digital | | | |
|---|---|---|---|---|---|---|---|---|---|---|---|---|---|---|---|---|---|
| | | | Gauss. | Shot | Impulse | Defocus | Glass | Motion | Zoom | Snow | Frost | Fog | Bright | Contrast | Elastic | Pixel | JPEG |
| Pre-Trained | | 57.9 | 58 | 56 | 56 | 53 | 41 | 57 | 46 | 52 | 53 | 61 | 76 | 64 | 60 | 68 | 68 |
| FT | IN-V2 | 25.9 | 23 | 22 | 21 | 16 | 20 | 20 | 19 | 17 | 23 | 29 | 39 | 30 | 33 | 37 | 39 |
| | IN-A | 10.1 | 10 | 9 | 10 | 7 | 8 | 8 | 7 | 6 | 7 | 10 | 15 | 12 | 13 | 15 | 15 |
| | IN-R | 23.1 | 21 | 20 | 20 | 17 | 20 | 19 | 17 | 15 | 23 | 23 | 37 | 26 | 26 | 30 | 32 |
| | IN-Sketch | 11.1 | 12 | 11 | 11 | 4 | 5 | 6 | 6 | 8 | 12 | 12 | 21 | 14 | 11 | 14 | 20 |
| | ObjNet | 13.4 | 12 | 11 | 11 | 9 | 11 | 11 | 11 | 6 | 10 | 15 | 20 | 16 | 16 | 20 | 20 |
| | IN-Cartoon | 31.4 | 27 | 26 | 23 | 13 | 20 | 22 | 18 | 25 | 30 | 29 | 57 | 34 | 41 | 52 | 52 |
| | IN-Drawing | 24.6 | 23 | 20 | 21 | 11 | 15 | 16 | 15 | 20 | 37 | 24 | 51 | 24 | 28 | 33 | 32 |
| HeadOnly | IN-V2 | 59.0 | 58 | 57 | 57 | 54 | 43 | 60 | 48 | 53 | 54 | 64 | 76 | 65 | 60 | 68 | 68 |
| | IN-A | 58.5 | 57 | 55 | 55 | 52 | 43 | 61 | 48 | 53 | 53 | 64 | 76 | 66 | 60 | 68 | 67 |
| | IN-R | 55.5 | 56 | 55 | 55 | 48 | 44 | 55 | 43 | 49 | 54 | 59 | 72 | 59 | 56 | 64 | 65 |
| | IN-Sketch | 55.4 | 55 | 53 | 53 | 51 | 42 | 55 | 43 | 48 | 52 | 60 | 72 | 61 | 55 | 66 | 66 |
| | ObjNet | 56.7 | 56 | 54 | 54 | 52 | 41 | 59 | 45 | 50 | 50 | 62 | 75 | 63 | 59 | 65 | 66 |
| | IN-Cartoon | 53.1 | 52 | 49 | 50 | 46 | 38 | 53 | 39 | 46 | 46 | 58 | 74 | 60 | 55 | 66 | 64 |
| | IN-Drawing | 48.6 | 49 | 47 | 47 | 38 | 39 | 45 | 34 | 43 | 52 | 52 | 67 | 52 | 52 | 58 | 53 |
| Visual Prompt (Bahng et al., 2022) | IN-V2 | 50.1 | 52 | 50 | 50 | 42 | 34 | 49 | 40 | 43 | 44 | 50 | 70 | 51 | 54 | 60 | 61 |
| | IN-A | 39.6 | 35 | 34 | 32 | 33 | 25 | 38 | 32 | 33 | 33 | 44 | 63 | 44 | 47 | 50 | 51 |
| | IN-R | 44.2 | 44 | 43 | 42 | 38 | 30 | 41 | 34 | 37 | 42 | 45 | 65 | 43 | 48 | 53 | 56 |
| | IN-Sketch | 47.2 | 47 | 46 | 45 | 42 | 31 | 45 | 39 | 39 | 42 | 49 | 67 | 49 | 51 | 57 | 59 |
| | ObjNet | 29.5 | 25 | 24 | 23 | 26 | 18 | 27 | 22 | 23 | 23 | 31 | 53 | 31 | 37 | 39 | 41 |
| | IN-Cartoon | 45.4 | 46 | 44 | 44 | 39 | 28 | 43 | 34 | 38 | 39 | 43 | 68 | 45 | 51 | 59 | 60 |
| | IN-Drawing | 46.1 | 50 | 50 | 48 | 37 | 33 | 41 | 34 | 40 | 47 | 40 | 66 | 40 | 51 | 56 | 58 |
| LoRA (Hu et al., 2021) | IN-V2 | 58.9 | 59 | 57 | 57 | 54 | 43 | 60 | 48 | 52 | 54 | 63 | 76 | 65 | 60 | 68 | 68 |
| | IN-A | 58.1 | 57 | 55 | 55 | 52 | 43 | 61 | 47 | 52 | 53 | 64 | 75 | 65 | 58 | 67 | 66 |
| | IN-R | 55.1 | 56 | 54 | 55 | 48 | 44 | 55 | 42 | 48 | 53 | 58 | 72 | 57 | 55 | 64 | 65 |
| | IN-Sketch | 53.5 | 53 | 52 | 51 | 46 | 41 | 52 | 40 | 49 | 51 | 59 | 70 | 57 | 53 | 64 | 64 |
| | ObjNet | 53.4 | 53 | 51 | 51 | 46 | 38 | 55 | 41 | 46 | 46 | 60 | 72 | 59 | 55 | 63 | 65 |
| | IN-Cartoon | 52.0 | 51 | 48 | 49 | 45 | 37 | 51 | 38 | 45 | 46 | 57 | 73 | 59 | 54 | 64 | 63 |
| | IN-Drawing | 48.4 | 48 | 47 | 47 | 40 | 39 | 46 | 34 | 43 | 51 | 51 | 66 | 52 | 51 | 58 | 54 |
| EWC (Kirkpatrick et al., 2017) | IN-V2 | 53.9 | 53 | 52 | 52 | 47 | 41 | 52 | 37 | 45 | 50 | 61 | 72 | 60 | 55 | 65 | 65 |
| | IN-A | 40.9 | 39 | 37 | 38 | 36 | 31 | 40 | 28 | 30 | 34 | 49 | 57 | 50 | 41 | 51 | 52 |
| | IN-R | 52.5 | 51 | 50 | 50 | 45 | 45 | 50 | 38 | 48 | 52 | 57 | 69 | 57 | 53 | 60 | 63 |
| | IN-Sketch | 38.8 | 41 | 40 | 40 | 25 | 22 | 31 | 23 | 44 | 45 | 55 | 47 | 41 | 36 | 44 | 49 |
| | ObjNet | 44.7 | 42 | 40 | 40 | 40 | 34 | 46 | 32 | 35 | 39 | 50 | 64 | 51 | 45 | 55 | 57 |
| | IN-Cartoon | 48.1 | 47 | 45 | 45 | 39 | 30 | 46 | 33 | 44 | 42 | 53 | 71 | 55 | 51 | 62 | 58 |
| | IN-Drawing | 48.7 | 51 | 48 | 49 | 34 | 36 | 43 | 33 | 47 | 55 | 52 | 70 | 52 | 51 | 56 | 53 |
| LwF (Li & Hoiem, 2017) | IN-V2 | 30.2 | 28 | 26 | 26 | 19 | 25 | 23 | 21 | 19 | 26 | 32 | 46 | 33 | 38 | 44 | 46 |
| | IN-A | 17.6 | 16 | 15 | 15 | 11 | 14 | 13 | 13 | 10 | 14 | 19 | 28 | 21 | 22 | 26 | 27 |
| | IN-R | 30.1 | 26 | 25 | 24 | 22 | 26 | 26 | 22 | 21 | 30 | 30 | 46 | 33 | 35 | 41 | 42 |
| | IN-Sketch | 15.3 | 17 | 16 | 16 | 6 | 8 | 9 | 8 | 11 | 17 | 17 | 27 | 18 | 16 | 19 | 27 |
| | ObjNet | 23.1 | 22 | 20 | 20 | 16 | 20 | 21 | 18 | 11 | 17 | 24 | 36 | 26 | 28 | 34 | 34 |
| | IN-Cartoon | 39.6 | 33 | 32 | 27 | 19 | 29 | 30 | 24 | 34 | 41 | 38 | 67 | 41 | 52 | 63 | 63 |
| | IN-Drawing | 29.6 | 26 | 23 | 24 | 15 | 21 | 21 | 20 | 24 | 42 | 28 | 58 | 28 | 34 | 41 | 39 |
| LP-FT (Kumar et al., 2022) | IN-V2 | 26.0 | 23 | 22 | 22 | 14 | 20 | 21 | 18 | 17 | 23 | 29 | 41 | 30 | 33 | 38 | 40 |
| | IN-A | 11.6 | 11 | 11 | 11 | 8 | 10 | 9 | 8 | 6 | 8 | 13 | 18 | 14 | 15 | 17 | 17 |
| | IN-R | 25.7 | 24 | 23 | 22 | 19 | 22 | 22 | 18 | 17 | 26 | 27 | 40 | 28 | 29 | 34 | 36 |
| | IN-Sketch | 12.3 | 14 | 13 | 13 | 5 | 6 | 7 | 6 | 8 | 14 | 13 | 23 | 15 | 12 | 15 | 21 |
| | ObjNet | 16.3 | 15 | 13 | 13 | 11 | 15 | 15 | 13 | 8 | 14 | 17 | 25 | 19 | 19 | 25 | 23 |
| | IN-Cartoon | 31.3 | 27 | 25 | 22 | 14 | 21 | 22 | 18 | 24 | 30 | 28 | 58 | 33 | 42 | 53 | 51 |
| | IN-Drawing | 25.7 | 24 | 21 | 22 | 11 | 16 | 17 | 15 | 20 | 37 | 26 | 53 | 27 | 30 | 36 | 33 |
| WiSE-FT (Wortsman et al., 2022b) | IN-V2 | 49.1 | 48 | 46 | 46 | 39 | 41 | 44 | 35 | 39 | 45 | 55 | 66 | 54 | 54 | 62 | 63 |
| | IN-A | 43.3 | 43 | 41 | 42 | 35 | 36 | 38 | 30 | 30 | 36 | 50 | 60 | 50 | 47 | 55 | 57 |
| | IN-R | 49.5 | 48 | 47 | 47 | 40 | 43 | 44 | 34 | 42 | 50 | 52 | 67 | 54 | 54 | 59 | 61 |
| | IN-Sketch | 36.3 | 39 | 38 | 38 | 21 | 22 | 25 | 20 | 32 | 40 | 44 | 54 | 40 | 38 | 42 | 51 |
| | ObjNet | 46.7 | 45 | 43 | 44 | 38 | 39 | 44 | 34 | 35 | 42 | 51 | 64 | 51 | 51 | 59 | 60 |
| | IN-Cartoon | 50.8 | 48 | 46 | 45 | 35 | 38 | 43 | 33 | 47 | 49 | 55 | 74 | 55 | 60 | 68 | 67 |
| | IN-Drawing | 46.9 | 48 | 43 | 47 | 28 | 34 | 35 | 29 | 47 | 57 | 51 | 73 | 48 | 50 | 56 | 56 |
| Model Soup PRE-FT-EWC-LwF (Wortsman et al., 2022a) | IN-V2 | 50.2 | 51 | 49 | 50 | 39 | 42 | 44 | 35 | 42 | 46 | 55 | 67 | 54 | 56 | 62 | 63 |
| | IN-A | 44.3 | 45 | 42 | 44 | 36 | 36 | 39 | 30 | 32 | 36 | 50 | 61 | 51 | 49 | 56 | 58 |
| | IN-R | 51.8 | 51 | 50 | 50 | 43 | 45 | 46 | 36 | 46 | 52 | 54 | 69 | 55 | 56 | 61 | 63 |
| | IN-Sketch | 37.2 | 41 | 40 | 40 | 22 | 22 | 26 | 20 | 34 | 42 | 46 | 54 | 39 | 39 | 42 | 52 |
| | ObjNet | 48.7 | 48 | 46 | 47 | 40 | 41 | 46 | 35 | 38 | 43 | 52 | 66 | 52 | 53 | 61 | 62 |
| | IN-Cartoon | 51.7 | 48 | 46 | 45 | 35 | 39 | 44 | 33 | 50 | 51 | 56 | 75 | 56 | 61 | 69 | 67 |
| | IN-Drawing | 47.9 | 50 | 45 | 49 | 29 | 35 | 36 | 30 | 49 | 59 | 52 | 74 | 49 | 51 | 57 | 56 |

Table 46: Accuracy of ImageNet-1K with AugReg pre-trained ViT-S/16 with with different fine-tuning methods and downstream datasets on each ImageNet-C corruption. For each corruption, accuracy is averaged across 5 levels of severity.

| Dataset | Method | Avg. | Noise | | | Blur | | | | Weather | | | | Digital | | | |
|---|---|---|---|---|---|---|---|---|---|---|---|---|---|---|---|---|---|
| | | | Gauss. | Shot | Impulse | Defocus | Glass | Motion | Zoom | Snow | Frost | Fog | Bright | Contrast | Elastic | Pixel | JPEG |
| Pre-Trained | | 53.2 | 54 | 52 | 52 | 46 | 37 | 50 | 42 | 48 | 52 | 58 | 73 | 53 | 55 | 63 | 63 |
| FT | IN-V2 | 51.2 | 49 | 47 | 46 | 46 | 36 | 50 | 38 | 50 | 52 | 58 | 70 | 52 | 52 | 60 | 62 |
| | IN-A | 44.1 | 43 | 40 | 40 | 42 | 31 | 44 | 31 | 42 | 43 | 47 | 62 | 44 | 44 | 51 | 57 |
| | IN-R | 42.7 | 40 | 39 | 37 | 39 | 37 | 41 | 31 | 39 | 44 | 47 | 58 | 44 | 44 | 50 | 52 |
| | IN-Sketch | 45.5 | 44 | 42 | 42 | 38 | 31 | 42 | 35 | 42 | 46 | 50 | 64 | 49 | 46 | 53 | 55 |
| | ObjNet | 41.9 | 40 | 38 | 38 | 40 | 30 | 43 | 30 | 38 | 38 | 45 | 61 | 39 | 43 | 51 | 54 |
| | IN-Cartoon | 44.4 | 44 | 42 | 41 | 36 | 28 | 41 | 28 | 41 | 40 | 46 | 71 | 46 | 47 | 58 | 54 |
| | IN-Drawing | 46.0 | 45 | 43 | 43 | 34 | 36 | 43 | 33 | 44 | 51 | 49 | 69 | 44 | 48 | 57 | 53 |
| Linear Probing | IN-V2 | 53.1 | 54 | 52 | 52 | 46 | 37 | 50 | 42 | 49 | 52 | 58 | 72 | 53 | 55 | 63 | 63 |
| | IN-A | 53.0 | 53 | 51 | 51 | 46 | 37 | 51 | 42 | 49 | 51 | 59 | 72 | 53 | 55 | 62 | 62 |
| | IN-R | 53.0 | 53 | 51 | 51 | 47 | 38 | 52 | 42 | 49 | 52 | 57 | 72 | 52 | 55 | 62 | 63 |
| | IN-Sketch | 50.9 | 51 | 50 | 49 | 45 | 35 | 48 | 40 | 48 | 50 | 55 | 70 | 50 | 53 | 59 | 61 |
| | ObjNet | 53.4 | 53 | 52 | 51 | 47 | 38 | 52 | 43 | 49 | 52 | 59 | 72 | 51 | 56 | 62 | 63 |
| | IN-Cartoon | 52.6 | 53 | 51 | 51 | 45 | 37 | 49 | 41 | 47 | 50 | 56 | 74 | 52 | 55 | 63 | 64 |
| | IN-Drawing | 54.3 | 55 | 53 | 52 | 46 | 39 | 51 | 42 | 53 | 57 | 58 | 74 | 51 | 58 | 63 | 65 |
| Visual Prompt (Bahng et al., 2022) | IN-V2 | 43.9 | 42 | 41 | 40 | 36 | 28 | 40 | 37 | 39 | 41 | 45 | 66 | 46 | 50 | 53 | 57 |
| | IN-A | 32.1 | 26 | 25 | 22 | 27 | 18 | 30 | 25 | 29 | 29 | 37 | 55 | 35 | 39 | 38 | 45 |
| | IN-R | 35.0 | 34 | 34 | 32 | 26 | 23 | 30 | 26 | 31 | 37 | 34 | 57 | 33 | 41 | 41 | 46 |
| | IN-Sketch | 39.2 | 38 | 37 | 36 | 31 | 24 | 34 | 31 | 35 | 39 | 42 | 61 | 40 | 43 | 45 | 50 |
| | ObjNet | 26.8 | 21 | 20 | 18 | 19 | 15 | 25 | 22 | 23 | 23 | 29 | 51 | 27 | 36 | 33 | 42 |
| | IN-Cartoon | 36.0 | 33 | 32 | 31 | 28 | 20 | 32 | 29 | 30 | 31 | 35 | 63 | 34 | 44 | 48 | 50 |
| | IN-Drawing | 40.7 | 41 | 39 | 39 | 30 | 28 | 35 | 31 | 36 | 45 | 37 | 62 | 38 | 46 | 50 | 51 |
| LoRA (Hu et al., 2021) | IN-V2 | 53.5 | 54 | 52 | 52 | 46 | 37 | 50 | 42 | 49 | 52 | 58 | 73 | 53 | 55 | 63 | 64 |
| | IN-A | 53.5 | 54 | 52 | 52 | 47 | 38 | 52 | 43 | 49 | 52 | 57 | 73 | 51 | 56 | 62 | 63 |
| | IN-R | 52.8 | 54 | 52 | 51 | 46 | 38 | 52 | 42 | 49 | 52 | 54 | 73 | 47 | 56 | 62 | 64 |
| | IN-Sketch | 53.5 | 54 | 52 | 52 | 48 | 37 | 50 | 43 | 49 | 53 | 58 | 73 | 52 | 55 | 62 | 64 |
| | ObjNet | 53.0 | 54 | 52 | 52 | 47 | 38 | 52 | 43 | 49 | 52 | 57 | 73 | 47 | 56 | 62 | 64 |
| | IN-Cartoon | 51.3 | 52 | 50 | 50 | 45 | 37 | 48 | 40 | 46 | 50 | 53 | 72 | 49 | 53 | 62 | 62 |
| | IN-Drawing | 54.6 | 55 | 54 | 53 | 46 | 38 | 51 | 41 | 52 | 58 | 60 | 73 | 54 | 56 | 63 | 64 |
| EWC (Kirkpatrick et al., 2017) | IN-V2 | 55.4 | 55 | 53 | 52 | 50 | 38 | 54 | 43 | 53 | 55 | 62 | 74 | 57 | 57 | 64 | 65 |
| | IN-A | 51.4 | 49 | 46 | 46 | 49 | 34 | 52 | 39 | 49 | 52 | 57 | 71 | 56 | 50 | 58 | 63 |
| | IN-R | 51.8 | 50 | 48 | 48 | 46 | 42 | 51 | 37 | 49 | 55 | 57 | 69 | 54 | 52 | 56 | 62 |
| | IN-Sketch | 54.0 | 53 | 52 | 52 | 47 | 39 | 51 | 43 | 51 | 55 | 59 | 72 | 56 | 55 | 61 | 63 |
| | ObjNet | 51.7 | 51 | 49 | 49 | 48 | 35 | 52 | 39 | 49 | 50 | 57 | 70 | 52 | 52 | 59 | 63 |
| | IN-Cartoon | 49.1 | 49 | 47 | 47 | 42 | 32 | 46 | 35 | 44 | 46 | 54 | 71 | 51 | 51 | 61 | 60 |
| | IN-Drawing | 53.2 | 54 | 52 | 52 | 44 | 37 | 50 | 39 | 51 | 58 | 58 | 71 | 55 | 54 | 62 | 62 |
| LwF (Li & Hoiem, 2017) | IN-V2 | 53.5 | 52 | 51 | 50 | 47 | 38 | 51 | 41 | 51 | 53 | 60 | 72 | 53 | 55 | 63 | 64 |
| | IN-A | 51.7 | 50 | 48 | 48 | 47 | 37 | 51 | 39 | 49 | 51 | 57 | 71 | 54 | 53 | 60 | 62 |
| | IN-R | 51.8 | 50 | 48 | 47 | 45 | 42 | 50 | 40 | 48 | 53 | 56 | 69 | 52 | 54 | 61 | 61 |
| | IN-Sketch | 48.6 | 48 | 46 | 47 | 41 | 33 | 45 | 38 | 46 | 49 | 52 | 68 | 51 | 50 | 58 | 59 |
| | ObjNet | 50.4 | 50 | 48 | 48 | 46 | 36 | 50 | 39 | 46 | 48 | 55 | 70 | 48 | 51 | 60 | 61 |
| | IN-Cartoon | 52.6 | 52 | 51 | 50 | 44 | 36 | 50 | 38 | 49 | 49 | 56 | 77 | 52 | 56 | 64 | 63 |
| | IN-Drawing | 50.8 | 49 | 47 | 47 | 41 | 38 | 50 | 38 | 47 | 54 | 55 | 74 | 48 | 52 | 63 | 60 |
| LP-FT (Kumar et al., 2022) | IN-V2 | 52.3 | 51 | 49 | 49 | 47 | 37 | 51 | 39 | 51 | 53 | 59 | 71 | 52 | 53 | 61 | 63 |
| | IN-A | 48.6 | 47 | 44 | 44 | 46 | 34 | 49 | 36 | 46 | 49 | 52 | 68 | 51 | 48 | 57 | 59 |
| | IN-R | 46.6 | 44 | 42 | 41 | 41 | 39 | 45 | 33 | 43 | 48 | 51 | 63 | 48 | 48 | 55 | 56 |
| | IN-Sketch | 47.9 | 47 | 46 | 46 | 41 | 32 | 45 | 37 | 46 | 48 | 52 | 66 | 51 | 48 | 56 | 57 |
| | ObjNet | 46.6 | 45 | 42 | 42 | 44 | 33 | 47 | 34 | 43 | 44 | 51 | 66 | 44 | 48 | 56 | 58 |
| | IN-Cartoon | 47.2 | 47 | 45 | 44 | 38 | 31 | 44 | 32 | 44 | 44 | 50 | 73 | 48 | 50 | 61 | 57 |
| | IN-Drawing | 48.2 | 47 | 44 | 45 | 37 | 37 | 45 | 34 | 46 | 53 | 52 | 71 | 48 | 50 | 60 | 55 |
| WiSE-FT (Wortsman et al., 2022b) | IN-V2 | 55.0 | 55 | 53 | 53 | 48 | 39 | 52 | 42 | 53 | 55 | 61 | 74 | 56 | 56 | 64 | 65 |
| | IN-A | 54.2 | 53 | 52 | 52 | 49 | 38 | 53 | 41 | 52 | 54 | 60 | 73 | 57 | 55 | 62 | 64 |
| | IN-R | 55.3 | 54 | 52 | 52 | 49 | 43 | 52 | 42 | 53 | 57 | 61 | 72 | 57 | 56 | 63 | 64 |
| | IN-Sketch | 53.6 | 54 | 53 | 53 | 45 | 37 | 49 | 42 | 52 | 55 | 58 | 72 | 56 | 54 | 62 | 63 |
| | ObjNet | 53.5 | 54 | 52 | 52 | 48 | 37 | 52 | 41 | 50 | 51 | 59 | 72 | 53 | 55 | 63 | 64 |
| | IN-Cartoon | 52.9 | 54 | 52 | 51 | 45 | 35 | 49 | 37 | 50 | 50 | 57 | 76 | 55 | 56 | 64 | 63 |
| | IN-Drawing | 55.4 | 56 | 54 | 54 | 46 | 41 | 53 | 41 | 53 | 58 | 60 | 75 | 54 | 56 | 65 | 64 |
| Model Soup PRE-FT-EWC-LwF (Wortsman et al., 2022a) | IN-V2 | 55.1 | 54 | 53 | 52 | 49 | 39 | 53 | 42 | 53 | 55 | 61 | 74 | 56 | 56 | 64 | 65 |
| | IN-A | 53.9 | 53 | 51 | 51 | 49 | 37 | 53 | 41 | 52 | 54 | 59 | 72 | 57 | 54 | 62 | 64 |
| | IN-R | 55.2 | 53 | 52 | 52 | 49 | 44 | 53 | 42 | 52 | 57 | 61 | 72 | 57 | 56 | 63 | 64 |
| | IN-Sketch | 53.6 | 54 | 52 | 52 | 46 | 37 | 49 | 42 | 52 | 54 | 58 | 72 | 56 | 54 | 62 | 63 |
| | ObjNet | 53.5 | 54 | 51 | 51 | 49 | 37 | 53 | 41 | 50 | 51 | 59 | 72 | 53 | 54 | 62 | 64 |
| | IN-Cartoon | 52.7 | 53 | 51 | 51 | 45 | 35 | 49 | 38 | 49 | 50 | 57 | 75 | 54 | 55 | 64 | 63 |
| | IN-Drawing | 55.1 | 55 | 54 | 54 | 46 | 40 | 53 | 41 | 53 | 58 | 59 | 75 | 54 | 56 | 65 | 63 |

Table 47: Accuracy of ImageNet-21K with AugReg pre-trained ViT-S/16 with different fine-tuning methods and downstream datasets on each ImageNet-C corruption For each corruption, accuracy is averaged across 5 levels of severity.

| Dataset | Method | Avg. | Noise | | | Blur | | | | Weather | | | | Digital | | | |
|---|---|---|---|---|---|---|---|---|---|---|---|---|---|---|---|---|---|
| | | | Gauss. | Shot | Impulse | Defocus | Glass | Motion | Zoom | Snow | Frost | Fog | Bright | Contrast | Elastic | Pixel | JPEG |
| Pre-Trained | | 58.0 | 58 | 56 | 56 | 54 | 43 | 59 | 48 | 55 | 55 | 60 | 75 | 56 | 59 | 69 | 68 |
| FT | IN-V2 | 56.8 | 54 | 52 | 53 | 54 | 44 | 58 | 46 | 54 | 55 | 62 | 74 | 58 | 56 | 65 | 67 |
| | IN-A | 52.7 | 49 | 47 | 48 | 51 | 39 | 55 | 43 | 51 | 50 | 57 | 69 | 54 | 51 | 63 | 64 |
| | IN-R | 50.6 | 48 | 46 | 46 | 49 | 45 | 50 | 39 | 46 | 51 | 53 | 66 | 50 | 51 | 58 | 60 |
| | IN-Sketch | 50.7 | 48 | 47 | 47 | 48 | 36 | 51 | 41 | 48 | 50 | 54 | 67 | 52 | 50 | 59 | 60 |
| | ObjNet | 48.5 | 45 | 43 | 43 | 47 | 34 | 52 | 37 | 46 | 45 | 52 | 67 | 49 | 49 | 59 | 61 |
| | IN-Cartoon | 52.2 | 51 | 48 | 49 | 50 | 34 | 56 | 40 | 50 | 45 | 53 | 74 | 53 | 53 | 64 | 63 |
| | IN-Drawing | 52.8 | 46 | 43 | 44 | 50 | 41 | 55 | 42 | 51 | 57 | 57 | 73 | 53 | 54 | 64 | 61 |
| Linear Probing | IN-V2 | 57.7 | 57 | 55 | 56 | 53 | 42 | 59 | 48 | 55 | 54 | 60 | 75 | 56 | 59 | 69 | 67 |
| | IN-A | 57.2 | 56 | 54 | 55 | 53 | 42 | 59 | 47 | 55 | 54 | 60 | 74 | 57 | 58 | 68 | 66 |
| | IN-R | 56.8 | 56 | 54 | 55 | 52 | 42 | 58 | 47 | 54 | 54 | 59 | 74 | 55 | 58 | 68 | 66 |
| | IN-Sketch | 55.1 | 55 | 53 | 53 | 50 | 40 | 56 | 45 | 53 | 53 | 57 | 72 | 53 | 56 | 65 | 64 |
| | ObjNet | 57.9 | 57 | 55 | 55 | 55 | 43 | 60 | 48 | 55 | 54 | 61 | 74 | 58 | 59 | 68 | 67 |
| | IN-Cartoon | 59.3 | 58 | 56 | 57 | 54 | 42 | 61 | 49 | 56 | 55 | 63 | 79 | 59 | 60 | 71 | 70 |
| | IN-Drawing | 60.0 | 59 | 57 | 58 | 55 | 44 | 61 | 48 | 58 | 60 | 64 | 77 | 58 | 61 | 71 | 69 |
| Visual Prompt (Bahng et al., 2022) | IN-V2 | 47.9 | 44 | 42 | 41 | 45 | 34 | 48 | 42 | 43 | 44 | 50 | 68 | 47 | 53 | 59 | 60 |
| | IN-A | 39.6 | 34 | 32 | 30 | 38 | 26 | 41 | 34 | 36 | 35 | 43 | 59 | 40 | 44 | 49 | 54 |
| | IN-R | 38.1 | 33 | 32 | 30 | 36 | 29 | 36 | 31 | 33 | 38 | 39 | 58 | 35 | 44 | 47 | 49 |
| | IN-Sketch | 42.1 | 39 | 38 | 37 | 38 | 28 | 39 | 35 | 40 | 43 | 41 | 63 | 38 | 47 | 51 | 55 |
| | ObjNet | 29.1 | 23 | 22 | 20 | 26 | 17 | 29 | 25 | 25 | 25 | 30 | 50 | 29 | 35 | 37 | 43 |
| | IN-Cartoon | 39.7 | 35 | 33 | 32 | 37 | 24 | 40 | 34 | 35 | 32 | 38 | 64 | 37 | 47 | 56 | 53 |
| | IN-Drawing | 41.0 | 40 | 39 | 38 | 35 | 29 | 37 | 32 | 36 | 44 | 40 | 62 | 31 | 46 | 54 | 52 |
| LoRA (Hu et al., 2021) | IN-V2 | 58.2 | 58 | 56 | 56 | 54 | 43 | 60 | 48 | 56 | 55 | 61 | 76 | 56 | 59 | 69 | 68 |
| | IN-A | 58.9 | 58 | 56 | 56 | 56 | 44 | 61 | 49 | 56 | 55 | 62 | 76 | 59 | 60 | 69 | 68 |
| | IN-R | 58.2 | 58 | 56 | 56 | 54 | 43 | 60 | 48 | 56 | 55 | 60 | 75 | 56 | 59 | 69 | 68 |
| | IN-Sketch | 58.3 | 58 | 56 | 57 | 54 | 42 | 60 | 48 | 56 | 55 | 61 | 75 | 57 | 59 | 69 | 68 |
| | ObjNet | 58.8 | 58 | 56 | 56 | 56 | 43 | 61 | 49 | 56 | 55 | 62 | 76 | 59 | 60 | 69 | 68 |
| | IN-Cartoon | 57.7 | 57 | 55 | 56 | 53 | 42 | 59 | 47 | 55 | 54 | 60 | 75 | 57 | 59 | 69 | 68 |
| | IN-Drawing | 59.0 | 58 | 57 | 57 | 55 | 43 | 60 | 48 | 57 | 59 | 62 | 76 | 58 | 60 | 69 | 68 |
| EWC (Kirkpatrick et al., 2017) | IN-V2 | 59.7 | 58 | 56 | 57 | 56 | 44 | 61 | 49 | 58 | 57 | 64 | 77 | 59 | 60 | 70 | 69 |
| | IN-A | 58.0 | 56 | 54 | 54 | 56 | 42 | 60 | 48 | 56 | 55 | 62 | 74 | 59 | 56 | 68 | 68 |
| | IN-R | 56.4 | 54 | 51 | 51 | 56 | 47 | 57 | 45 | 53 | 57 | 59 | 73 | 56 | 57 | 63 | 66 |
| | IN-Sketch | 58.0 | 57 | 55 | 55 | 55 | 42 | 59 | 49 | 56 | 57 | 62 | 74 | 58 | 58 | 67 | 67 |
| | ObjNet | 56.7 | 56 | 53 | 54 | 55 | 41 | 59 | 47 | 54 | 52 | 60 | 73 | 57 | 55 | 67 | 68 |
| | IN-Cartoon | 55.2 | 53 | 50 | 52 | 52 | 38 | 58 | 44 | 53 | 50 | 59 | 74 | 57 | 56 | 67 | 65 |
| | IN-Drawing | 58.1 | 56 | 55 | 55 | 53 | 43 | 59 | 46 | 56 | 60 | 63 | 75 | 58 | 58 | 67 | 66 |
| LwF (Li & Hoiem, 2017) | IN-V2 | 58.3 | 57 | 55 | 56 | 54 | 44 | 60 | 47 | 57 | 55 | 62 | 75 | 58 | 58 | 68 | 68 |
| | IN-A | 57.8 | 56 | 54 | 55 | 54 | 44 | 59 | 47 | 56 | 55 | 61 | 75 | 57 | 58 | 68 | 67 |
| | IN-R | 57.2 | 56 | 54 | 54 | 53 | 47 | 58 | 46 | 54 | 56 | 60 | 73 | 55 | 58 | 66 | 66 |
| | IN-Sketch | 54.5 | 53 | 51 | 52 | 51 | 38 | 55 | 44 | 53 | 52 | 58 | 71 | 55 | 54 | 64 | 64 |
| | ObjNet | 55.1 | 53 | 50 | 50 | 53 | 40 | 58 | 44 | 53 | 51 | 58 | 73 | 55 | 56 | 66 | 66 |
| | IN-Cartoon | 60.0 | 59 | 57 | 57 | 56 | 44 | 62 | 48 | 57 | 55 | 62 | 81 | 60 | 62 | 72 | 70 |
| | IN-Drawing | 57.8 | 55 | 52 | 53 | 54 | 44 | 60 | 46 | 55 | 59 | 60 | 78 | 55 | 58 | 71 | 68 |
| LP-FT (Kumar et al., 2022) | IN-V2 | 57.6 | 56 | 53 | 54 | 54 | 44 | 59 | 47 | 56 | 55 | 62 | 74 | 58 | 57 | 67 | 67 |
| | IN-A | 56.0 | 53 | 51 | 52 | 53 | 42 | 58 | 46 | 54 | 54 | 60 | 72 | 58 | 55 | 66 | 65 |
| | IN-R | 53.5 | 51 | 49 | 49 | 51 | 46 | 54 | 42 | 50 | 54 | 56 | 69 | 53 | 55 | 62 | 62 |
| | IN-Sketch | 52.3 | 51 | 49 | 50 | 48 | 37 | 53 | 42 | 51 | 51 | 55 | 69 | 53 | 52 | 62 | 62 |
| | ObjNet | 53.4 | 52 | 50 | 49 | 51 | 37 | 56 | 43 | 51 | 50 | 57 | 71 | 53 | 53 | 64 | 64 |
| | IN-Cartoon | 56.0 | 55 | 52 | 53 | 52 | 38 | 59 | 44 | 54 | 50 | 57 | 77 | 57 | 57 | 68 | 66 |
| | IN-Drawing | 55.5 | 50 | 48 | 49 | 52 | 43 | 58 | 43 | 54 | 58 | 60 | 75 | 55 | 56 | 68 | 63 |
| WiSE-FT (Wortsman et al., 2022b) | IN-V2 | 59.3 | 58 | 56 | 57 | 55 | 44 | 61 | 49 | 58 | 57 | 63 | 76 | 59 | 60 | 69 | 69 |
| | IN-A | 58.9 | 57 | 55 | 56 | 55 | 44 | 61 | 49 | 57 | 56 | 63 | 75 | 59 | 59 | 69 | 68 |
| | IN-R | 59.6 | 58 | 56 | 56 | 56 | 49 | 60 | 49 | 57 | 59 | 63 | 75 | 59 | 60 | 68 | 68 |
| | IN-Sketch | 58.0 | 57 | 55 | 56 | 54 | 42 | 58 | 47 | 57 | 57 | 61 | 74 | 59 | 58 | 67 | 67 |
| | ObjNet | 57.6 | 56 | 54 | 55 | 54 | 42 | 60 | 47 | 56 | 54 | 61 | 75 | 57 | 58 | 68 | 67 |
| | IN-Cartoon | 58.4 | 58 | 55 | 56 | 55 | 40 | 61 | 46 | 57 | 53 | 61 | 78 | 58 | 60 | 70 | 68 |
| | IN-Drawing | 60.4 | 59 | 57 | 58 | 56 | 45 | 62 | 49 | 59 | 61 | 64 | 78 | 59 | 61 | 71 | 69 |
| Model Soup PRE-FT-EWC-LwF (Wortsman et al., 2022a) | IN-V2 | 59.3 | 58 | 56 | 57 | 55 | 45 | 61 | 49 | 58 | 57 | 63 | 76 | 59 | 60 | 69 | 69 |
| | IN-A | 58.9 | 57 | 55 | 56 | 56 | 44 | 61 | 49 | 57 | 56 | 63 | 75 | 59 | 59 | 69 | 68 |
| | IN-R | 59.4 | 58 | 56 | 56 | 56 | 49 | 60 | 49 | 57 | 59 | 63 | 75 | 59 | 60 | 68 | 68 |
| | IN-Sketch | 57.9 | 57 | 55 | 56 | 54 | 41 | 58 | 48 | 56 | 56 | 61 | 74 | 59 | 58 | 67 | 67 |
| | ObjNet | 57.6 | 56 | 54 | 54 | 55 | 42 | 60 | 47 | 56 | 54 | 61 | 75 | 57 | 58 | 68 | 67 |
| | IN-Cartoon | 58.6 | 58 | 55 | 56 | 55 | 41 | 61 | 47 | 56 | 53 | 61 | 78 | 59 | 60 | 70 | 69 |
| | IN-Drawing | 60.2 | 59 | 57 | 58 | 56 | 44 | 62 | 49 | 58 | 61 | 63 | 78 | 59 | 60 | 71 | 69 |

Table 48: Accuracy of ImageNet-21K with AugReg pre-trained ViT-S/32 with different fine-tuning methods and downstream datasets on each ImageNet-C corruption. For each corruption, accuracy is averaged across 5 levels of severity.

| Dataset | Method | Avg. | Noise | | | Blur | | | | Weather | | | | Digital | | | |
|---|---|---|---|---|---|---|---|---|---|---|---|---|---|---|---|---|---|
| | | | Gauss. | Shot | Impulse | Defocus | Glass | Motion | Zoom | Snow | Frost | Fog | Bright | Contrast | Elastic | Pixel | JPEG |
| Pre-Trained | | 52.0 | 54 | 53 | 53 | 47 | 40 | 51 | 39 | 43 | 47 | 50 | 69 | 48 | 56 | 65 | 64 |
| FT | IN-V2 | 50.1 | 49 | 47 | 47 | 47 | 40 | 50 | 40 | 43 | 47 | 51 | 66 | 47 | 52 | 62 | 62 |
| | IN-A | 44.0 | 43 | 41 | 41 | 44 | 37 | 46 | 34 | 34 | 39 | 43 | 59 | 42 | 44 | 56 | 56 |
| | IN-R | 43.4 | 42 | 41 | 40 | 42 | 40 | 42 | 31 | 36 | 43 | 44 | 57 | 42 | 45 | 53 | 54 |
| | IN-Sketch | 45.5 | 46 | 45 | 45 | 43 | 34 | 45 | 35 | 39 | 43 | 45 | 60 | 44 | 46 | 56 | 56 |
| | ObjNet | 41.7 | 40 | 39 | 38 | 41 | 33 | 43 | 31 | 32 | 37 | 39 | 58 | 41 | 44 | 55 | 55 |
| | IN-Cartoon | 45.2 | 46 | 44 | 43 | 41 | 30 | 45 | 31 | 38 | 38 | 41 | 68 | 44 | 48 | 63 | 57 |
| | IN-Drawing | 46.9 | 48 | 45 | 45 | 42 | 37 | 45 | 32 | 41 | 50 | 44 | 65 | 43 | 49 | 60 | 55 |
| Linear Probing | IN-V2 | 51.5 | 54 | 52 | 52 | 47 | 40 | 51 | 39 | 42 | 47 | 49 | 68 | 48 | 55 | 65 | 63 |
| | IN-A | 50.4 | 52 | 51 | 51 | 46 | 39 | 50 | 38 | 41 | 46 | 48 | 66 | 48 | 54 | 63 | 61 |
| | IN-R | 51.5 | 52 | 51 | 51 | 48 | 41 | 52 | 40 | 42 | 47 | 51 | 68 | 49 | 55 | 64 | 63 |
| | IN-Sketch | 49.6 | 50 | 49 | 49 | 46 | 39 | 50 | 38 | 41 | 45 | 49 | 65 | 48 | 53 | 61 | 60 |
| | ObjNet | 52.1 | 53 | 52 | 52 | 49 | 41 | 53 | 41 | 42 | 47 | 51 | 68 | 50 | 55 | 64 | 63 |
| | IN-Cartoon | 53.5 | 55 | 54 | 54 | 49 | 41 | 53 | 41 | 43 | 47 | 52 | 73 | 51 | 57 | 68 | 66 |
| | IN-Drawing | 53.6 | 55 | 54 | 54 | 48 | 42 | 52 | 40 | 45 | 52 | 52 | 71 | 49 | 58 | 66 | 65 |
| Visual Prompt (Bahng et al., 2022) | IN-V2 | 42.6 | 43 | 42 | 41 | 38 | 32 | 41 | 35 | 33 | 37 | 39 | 60 | 39 | 49 | 54 | 55 |
| | IN-A | 23.2 | 18 | 17 | 16 | 21 | 17 | 23 | 18 | 16 | 20 | 25 | 38 | 22 | 30 | 32 | 34 |
| | IN-R | 32.3 | 31 | 31 | 29 | 30 | 25 | 29 | 25 | 25 | 32 | 25 | 51 | 23 | 38 | 42 | 46 |
| | IN-Sketch | 35.1 | 36 | 35 | 34 | 31 | 23 | 30 | 25 | 28 | 34 | 30 | 54 | 30 | 40 | 46 | 49 |
| | ObjNet | 24.5 | 21 | 20 | 19 | 22 | 18 | 23 | 20 | 17 | 21 | 19 | 42 | 18 | 33 | 36 | 38 |
| | IN-Cartoon | 36.0 | 37 | 35 | 34 | 32 | 23 | 33 | 27 | 27 | 30 | 28 | 58 | 27 | 44 | 53 | 52 |
| | IN-Drawing | 36.6 | 40 | 39 | 39 | 29 | 28 | 31 | 26 | 28 | 39 | 27 | 55 | 27 | 42 | 49 | 48 |
| LoRA (Hu et al., 2021) | IN-V2 | 52.1 | 54 | 53 | 53 | 47 | 41 | 51 | 39 | 43 | 47 | 50 | 69 | 48 | 56 | 66 | 64 |
| | IN-A | 53.1 | 54 | 53 | 53 | 50 | 42 | 54 | 41 | 43 | 48 | 52 | 70 | 52 | 56 | 66 | 64 |
| | IN-R | 52.8 | 54 | 52 | 53 | 49 | 42 | 54 | 41 | 42 | 48 | 52 | 70 | 50 | 56 | 66 | 64 |
| | IN-Sketch | 52.9 | 54 | 52 | 52 | 50 | 42 | 53 | 41 | 44 | 48 | 52 | 69 | 51 | 56 | 65 | 64 |
| | ObjNet | 53.3 | 54 | 53 | 53 | 50 | 42 | 54 | 42 | 43 | 48 | 53 | 70 | 51 | 57 | 66 | 64 |
| | IN-Cartoon | 51.4 | 53 | 52 | 52 | 47 | 41 | 51 | 39 | 42 | 47 | 48 | 69 | 46 | 55 | 65 | 64 |
| | IN-Drawing | 52.3 | 54 | 53 | 52 | 47 | 41 | 51 | 39 | 45 | 51 | 51 | 69 | 48 | 56 | 64 | 63 |
| EWC (Kirkpatrick et al., 2017) | IN-V2 | 53.6 | 54 | 53 | 53 | 50 | 42 | 54 | 42 | 45 | 49 | 53 | 70 | 50 | 57 | 66 | 65 |
| | IN-A | 50.2 | 49 | 48 | 47 | 50 | 42 | 53 | 40 | 41 | 45 | 49 | 66 | 49 | 51 | 61 | 62 |
| | IN-R | 50.7 | 50 | 49 | 48 | 49 | 45 | 51 | 38 | 43 | 49 | 50 | 66 | 49 | 53 | 58 | 61 |
| | IN-Sketch | 52.6 | 53 | 52 | 52 | 49 | 41 | 52 | 41 | 45 | 50 | 53 | 68 | 51 | 55 | 64 | 63 |
| | ObjNet | 50.7 | 51 | 50 | 50 | 49 | 41 | 53 | 39 | 42 | 45 | 48 | 66 | 48 | 53 | 63 | 63 |
| | IN-Cartoon | 48.8 | 50 | 48 | 48 | 44 | 35 | 49 | 35 | 40 | 43 | 47 | 68 | 47 | 52 | 64 | 61 |
| | IN-Drawing | 52.3 | 53 | 52 | 52 | 46 | 41 | 52 | 37 | 46 | 54 | 52 | 68 | 50 | 55 | 64 | 62 |
| LwF (Li & Hoiem, 2017) | IN-V2 | 51.8 | 53 | 51 | 51 | 47 | 41 | 51 | 40 | 44 | 48 | 51 | 68 | 48 | 55 | 65 | 64 |
| | IN-A | 50.8 | 51 | 50 | 50 | 47 | 40 | 52 | 40 | 41 | 46 | 50 | 67 | 48 | 53 | 64 | 63 |
| | IN-R | 50.9 | 51 | 50 | 49 | 47 | 43 | 50 | 38 | 43 | 49 | 51 | 66 | 48 | 54 | 63 | 62 |
| | IN-Sketch | 48.9 | 51 | 50 | 49 | 45 | 37 | 49 | 39 | 41 | 46 | 47 | 64 | 46 | 51 | 61 | 60 |
| | ObjNet | 48.2 | 48 | 46 | 46 | 47 | 37 | 49 | 37 | 38 | 43 | 47 | 65 | 46 | 50 | 61 | 61 |
| | IN-Cartoon | 53.7 | 55 | 53 | 53 | 49 | 41 | 54 | 39 | 44 | 48 | 50 | 74 | 50 | 58 | 70 | 67 |
| | IN-Drawing | 52.1 | 54 | 51 | 51 | 48 | 41 | 52 | 38 | 44 | 53 | 48 | 71 | 45 | 54 | 67 | 64 |
| LP-FT (Kumar et al., 2022) | IN-V2 | 50.7 | 51 | 49 | 49 | 47 | 40 | 50 | 40 | 43 | 47 | 50 | 67 | 48 | 53 | 63 | 62 |
| | IN-A | 48.2 | 48 | 46 | 46 | 46 | 39 | 49 | 38 | 40 | 44 | 48 | 64 | 47 | 49 | 60 | 59 |
| | IN-R | 47.5 | 46 | 45 | 44 | 45 | 42 | 47 | 35 | 40 | 46 | 47 | 63 | 45 | 50 | 58 | 57 |
| | IN-Sketch | 46.9 | 48 | 47 | 46 | 44 | 36 | 47 | 37 | 40 | 44 | 46 | 61 | 45 | 48 | 58 | 57 |
| | ObjNet | 47.0 | 47 | 45 | 45 | 46 | 36 | 49 | 37 | 37 | 42 | 45 | 63 | 45 | 49 | 60 | 60 |
| | IN-Cartoon | 50.1 | 51 | 49 | 48 | 46 | 35 | 51 | 36 | 41 | 43 | 47 | 72 | 48 | 53 | 68 | 62 |
| | IN-Drawing | 50.3 | 52 | 50 | 50 | 45 | 40 | 49 | 35 | 44 | 52 | 47 | 68 | 45 | 52 | 64 | 60 |
| WiSE-FT (Wortsman et al., 2022b) | IN-V2 | 53.1 | 54 | 52 | 52 | 49 | 42 | 53 | 41 | 45 | 49 | 53 | 70 | 50 | 56 | 66 | 65 |
| | IN-A | 52.8 | 53 | 52 | 52 | 50 | 42 | 54 | 41 | 44 | 49 | 53 | 69 | 50 | 55 | 65 | 64 |
| | IN-R | 53.3 | 53 | 52 | 52 | 50 | 45 | 53 | 40 | 46 | 51 | 54 | 69 | 51 | 56 | 64 | 64 |
| | IN-Sketch | 52.1 | 54 | 53 | 53 | 48 | 40 | 51 | 40 | 45 | 49 | 51 | 68 | 50 | 54 | 64 | 63 |
| | ObjNet | 51.8 | 52 | 51 | 51 | 48 | 40 | 52 | 40 | 43 | 47 | 51 | 68 | 49 | 55 | 65 | 64 |
| | IN-Cartoon | 51.8 | 53 | 52 | 51 | 47 | 37 | 52 | 37 | 43 | 46 | 50 | 72 | 50 | 55 | 68 | 64 |
| | IN-Drawing | 54.3 | 57 | 55 | 55 | 48 | 42 | 54 | 39 | 47 | 54 | 53 | 72 | 51 | 57 | 68 | 65 |
| Model Soup PRE-FT-EWC-LwF (Wortsman et al., 2022a) | IN-V2 | 53.1 | 54 | 52 | 52 | 49 | 42 | 53 | 41 | 45 | 49 | 52 | 70 | 50 | 56 | 66 | 65 |
| | IN-A | 52.5 | 53 | 51 | 51 | 50 | 42 | 54 | 41 | 44 | 48 | 52 | 68 | 50 | 55 | 65 | 64 |
| | IN-R | 53.2 | 53 | 52 | 52 | 50 | 45 | 53 | 40 | 45 | 51 | 54 | 69 | 51 | 56 | 64 | 64 |
| | IN-Sketch | 52.2 | 54 | 53 | 53 | 48 | 40 | 51 | 40 | 45 | 49 | 51 | 68 | 50 | 54 | 64 | 63 |
| | ObjNet | 51.7 | 52 | 51 | 51 | 49 | 40 | 53 | 40 | 43 | 47 | 50 | 68 | 49 | 54 | 65 | 64 |
| | IN-Cartoon | 52.0 | 54 | 52 | 52 | 47 | 38 | 52 | 37 | 43 | 46 | 50 | 72 | 50 | 55 | 68 | 64 |
| | IN-Drawing | 54.2 | 57 | 55 | 55 | 48 | 42 | 53 | 39 | 47 | 54 | 52 | 72 | 50 | 57 | 68 | 65 |

Table 49: Accuracy of ImageNet-21K with AugReg pre-trained ViT-L/16 with different fine-tuning methods and downstream datasets on each ImageNet-C corruption For each corruption, accuracy is averaged across 5 levels of severity.

| Dataset | Method | Avg. | Noise | | | Blur | | | | Weather | | | | Digital | | | |
|---|---|---|---|---|---|---|---|---|---|---|---|---|---|---|---|---|---|
| | | | Gauss. | Shot | Impulse | Defocus | Glass | Motion | Zoom | Snow | Frost | Fog | Bright | Contrast | Elastic | Pixel | JPEG |
| Pre-Trained | | 72.2 | 73 | 73 | 73 | 69 | 62 | 73 | 67 | 72 | 69 | 73 | 82 | 70 | 71 | 79 | 78 |
| FT | IN-V2 | 71.0 | 72 | 71 | 72 | 67 | 62 | 71 | 64 | 70 | 70 | 72 | 81 | 69 | 69 | 79 | 78 |
| | IN-A | 70.4 | 72 | 71 | 71 | 67 | 61 | 71 | 64 | 70 | 69 | 71 | 80 | 68 | 68 | 78 | 77 |
| | IN-R | 65.4 | 68 | 67 | 67 | 61 | 61 | 64 | 55 | 61 | 64 | 65 | 76 | 60 | 64 | 74 | 74 |
| | IN-Sketch | 67.6 | 69 | 68 | 68 | 65 | 58 | 67 | 61 | 67 | 66 | 68 | 78 | 64 | 65 | 75 | 74 |
| | ObjNet | 68.4 | 70 | 69 | 69 | 65 | 60 | 68 | 59 | 68 | 67 | 67 | 79 | 63 | 67 | 78 | 77 |
| | IN-Cartoon | 72.6 | 74 | 72 | 73 | 67 | 59 | 73 | 63 | 73 | 70 | 73 | 89 | 68 | 72 | 83 | 81 |
| | IN-Drawing | 74.7 | 77 | 75 | 76 | 69 | 67 | 74 | 63 | 74 | 77 | 75 | 87 | 67 | 73 | 85 | 81 |
| Linear Probing | IN-V2 | 71.4 | 72 | 72 | 71 | 68 | 61 | 72 | 66 | 72 | 69 | 72 | 81 | 69 | 70 | 78 | 77 |
| | IN-A | 71.4 | 72 | 72 | 71 | 68 | 61 | 72 | 66 | 71 | 69 | 72 | 81 | 70 | 70 | 78 | 77 |
| | IN-R | 69.8 | 70 | 70 | 69 | 67 | 60 | 70 | 64 | 70 | 67 | 71 | 80 | 68 | 69 | 77 | 76 |
| | IN-Sketch | 68.4 | 69 | 68 | 68 | 66 | 59 | 69 | 63 | 68 | 66 | 69 | 78 | 66 | 67 | 75 | 74 |
| | ObjNet | 71.5 | 72 | 71 | 71 | 69 | 62 | 72 | 66 | 71 | 68 | 73 | 81 | 70 | 70 | 78 | 77 |
| | IN-Cartoon | 76.5 | 77 | 76 | 76 | 73 | 65 | 77 | 71 | 76 | 72 | 77 | 88 | 75 | 75 | 85 | 83 |
| | IN-Drawing | 76.0 | 77 | 76 | 76 | 73 | 66 | 76 | 70 | 76 | 75 | 77 | 86 | 74 | 75 | 83 | 81 |
| Visual Prompt (Bahng et al., 2022) | IN-V2 | 57.4 | 52 | 51 | 50 | 53 | 44 | 56 | 54 | 56 | 56 | 62 | 75 | 55 | 61 | 70 | 69 |
| | IN-A | 50.5 | 42 | 41 | 39 | 45 | 36 | 50 | 48 | 50 | 49 | 56 | 71 | 48 | 55 | 65 | 63 |
| | IN-R | 50.3 | 47 | 47 | 45 | 45 | 38 | 47 | 45 | 48 | 51 | 51 | 69 | 45 | 53 | 62 | 62 |
| | IN-Sketch | 48.6 | 46 | 45 | 43 | 42 | 38 | 44 | 43 | 46 | 49 | 48 | 67 | 39 | 52 | 63 | 62 |
| | ObjNet | 47.7 | 38 | 37 | 35 | 43 | 32 | 46 | 44 | 47 | 46 | 53 | 70 | 45 | 53 | 63 | 62 |
| | IN-Cartoon | 55.8 | 50 | 49 | 48 | 51 | 42 | 54 | 52 | 55 | 53 | 57 | 75 | 51 | 60 | 71 | 68 |
| | IN-Drawing | 52.2 | 50 | 49 | 48 | 47 | 43 | 48 | 45 | 49 | 55 | 51 | 70 | 43 | 56 | 66 | 64 |
| LoRA (Hu et al., 2021) | IN-V2 | 72.3 | 73 | 73 | 73 | 69 | 62 | 73 | 67 | 72 | 70 | 73 | 82 | 70 | 71 | 79 | 78 |
| | IN-A | 72.9 | 73 | 73 | 73 | 71 | 63 | 74 | 68 | 72 | 70 | 75 | 82 | 71 | 72 | 79 | 78 |
| | IN-R | 72.6 | 73 | 73 | 73 | 70 | 63 | 73 | 67 | 72 | 69 | 74 | 82 | 70 | 72 | 79 | 78 |
| | IN-Sketch | 72.5 | 74 | 73 | 73 | 70 | 63 | 73 | 67 | 72 | 70 | 74 | 82 | 69 | 72 | 79 | 78 |
| | ObjNet | 72.9 | 73 | 73 | 73 | 71 | 64 | 74 | 68 | 73 | 70 | 75 | 82 | 71 | 72 | 80 | 78 |
| | IN-Cartoon | 72.5 | 73 | 73 | 72 | 70 | 63 | 73 | 67 | 72 | 69 | 74 | 82 | 69 | 71 | 79 | 78 |
| | IN-Drawing | 73.0 | 74 | 73 | 73 | 71 | 64 | 74 | 68 | 73 | 71 | 74 | 82 | 71 | 72 | 79 | 78 |
| EWC (Kirkpatrick et al., 2017) | IN-V2 | 71.0 | 74 | 74 | 74 | 65 | 64 | 70 | 63 | 68 | 69 | 68 | 81 | 66 | 70 | 80 | 78 |
| | IN-A | 72.0 | 73 | 72 | 72 | 68 | 63 | 73 | 66 | 72 | 69 | 73 | 82 | 70 | 70 | 80 | 78 |
| | IN-R | 72.0 | 72 | 71 | 71 | 70 | 65 | 72 | 65 | 72 | 71 | 73 | 82 | 70 | 70 | 78 | 78 |
| | IN-Sketch | 72.4 | 73 | 73 | 72 | 70 | 63 | 72 | 67 | 72 | 71 | 73 | 82 | 70 | 71 | 79 | 78 |
| | ObjNet | 71.2 | 72 | 71 | 70 | 68 | 62 | 72 | 63 | 72 | 70 | 72 | 81 | 70 | 69 | 79 | 78 |
| | IN-Cartoon | 71.8 | 72 | 71 | 71 | 69 | 60 | 73 | 66 | 72 | 68 | 73 | 83 | 70 | 71 | 79 | 78 |
| | IN-Drawing | 73.5 | 75 | 74 | 74 | 70 | 63 | 74 | 66 | 74 | 74 | 75 | 83 | 72 | 72 | 80 | 78 |
| LwF (Li & Hoiem, 2017) | IN-V2 | 71.3 | 74 | 74 | 74 | 66 | 63 | 71 | 65 | 69 | 68 | 71 | 81 | 66 | 71 | 79 | 78 |
| | IN-A | 72.0 | 74 | 73 | 73 | 68 | 63 | 72 | 66 | 72 | 69 | 72 | 82 | 68 | 71 | 79 | 78 |
| | IN-R | 71.4 | 73 | 72 | 72 | 67 | 64 | 71 | 64 | 70 | 70 | 72 | 81 | 67 | 71 | 78 | 78 |
| | IN-Sketch | 70.2 | 71 | 71 | 71 | 67 | 61 | 70 | 64 | 70 | 68 | 70 | 80 | 67 | 69 | 77 | 77 |
| | ObjNet | 71.5 | 73 | 72 | 72 | 68 | 62 | 72 | 65 | 71 | 69 | 72 | 82 | 68 | 70 | 79 | 78 |
| | IN-Cartoon | 75.5 | 79 | 78 | 78 | 70 | 66 | 75 | 65 | 73 | 72 | 71 | 91 | 61 | 78 | 89 | 86 |
| | IN-Drawing | 76.6 | 79 | 77 | 78 | 69 | 71 | 75 | 66 | 76 | 79 | 73 | 90 | 69 | 77 | 88 | 84 |
| LP-FT (Kumar et al., 2022) | IN-V2 | 71.4 | 73 | 73 | 72 | 67 | 62 | 71 | 65 | 72 | 70 | 71 | 81 | 67 | 70 | 79 | 78 |
| | IN-A | 53.7 | 56 | 55 | 55 | 49 | 46 | 53 | 46 | 52 | 50 | 54 | 63 | 51 | 52 | 63 | 62 |
| | IN-R | 49.4 | 51 | 51 | 50 | 46 | 44 | 48 | 39 | 47 | 49 | 49 | 59 | 45 | 48 | 58 | 57 |
| | IN-Sketch | 67.8 | 68 | 68 | 68 | 66 | 58 | 68 | 62 | 67 | 66 | 68 | 77 | 65 | 66 | 75 | 74 |
| | ObjNet | 70.6 | 71 | 70 | 70 | 68 | 61 | 71 | 63 | 70 | 68 | 72 | 80 | 68 | 69 | 78 | 76 |
| | IN-Cartoon | 77.0 | 78 | 77 | 77 | 72 | 64 | 77 | 69 | 77 | 74 | 78 | 90 | 74 | 76 | 86 | 85 |
| | IN-Drawing | 77.5 | 80 | 79 | 79 | 73 | 68 | 77 | 69 | 78 | 78 | 78 | 88 | 70 | 76 | 86 | 83 |
| WiSE-FT (Wortsman et al., 2022b) | IN-V2 | 73.5 | 74 | 74 | 74 | 71 | 64 | 74 | 68 | 74 | 72 | 75 | 83 | 72 | 72 | 80 | 79 |
| | IN-A | 71.5 | 73 | 72 | 72 | 67 | 62 | 72 | 66 | 72 | 69 | 72 | 81 | 69 | 70 | 78 | 77 |
| | IN-R | 73.0 | 74 | 74 | 73 | 70 | 66 | 73 | 65 | 72 | 72 | 74 | 82 | 70 | 72 | 80 | 79 |
| | IN-Sketch | 72.1 | 73 | 73 | 73 | 69 | 62 | 72 | 66 | 72 | 71 | 73 | 82 | 69 | 71 | 79 | 78 |
| | ObjNet | 72.6 | 74 | 73 | 73 | 70 | 63 | 73 | 66 | 73 | 71 | 73 | 82 | 70 | 72 | 80 | 78 |
| | IN-Cartoon | 75.6 | 77 | 76 | 76 | 72 | 63 | 76 | 69 | 76 | 72 | 77 | 87 | 73 | 74 | 83 | 82 |
| | IN-Drawing | 76.8 | 78 | 77 | 78 | 73 | 67 | 77 | 69 | 77 | 77 | 77 | 87 | 73 | 75 | 84 | 82 |
| Model Soup PRE-FT-EWC-LwF (Wortsman et al., 2022a) | IN-V2 | 73.4 | 75 | 75 | 74 | 69 | 64 | 74 | 67 | 73 | 72 | 74 | 83 | 71 | 72 | 80 | 79 |
| | IN-A | 72.7 | 74 | 73 | 73 | 69 | 63 | 73 | 67 | 73 | 70 | 73 | 82 | 71 | 71 | 79 | 78 |
| | IN-R | 73.3 | 74 | 74 | 74 | 70 | 67 | 73 | 66 | 72 | 72 | 74 | 82 | 71 | 72 | 80 | 79 |
| | IN-Sketch | 72.3 | 73 | 73 | 73 | 69 | 63 | 72 | 66 | 73 | 71 | 73 | 82 | 70 | 71 | 79 | 78 |
| | ObjNet | 72.7 | 74 | 73 | 73 | 70 | 63 | 73 | 66 | 73 | 71 | 74 | 82 | 70 | 71 | 79 | 78 |
| | IN-Cartoon | 75.8 | 77 | 76 | 76 | 73 | 64 | 76 | 69 | 76 | 72 | 77 | 88 | 72 | 75 | 84 | 83 |
| | IN-Drawing | 77.4 | 79 | 79 | 79 | 74 | 68 | 78 | 69 | 77 | 77 | 78 | 87 | 73 | 76 | 85 | 83 |

Table 50: Accuracy of ImageNet-1K pre-trained ResNet-50 with different fine-tuning methods and downstream datasets on each ImageNet-C corruption For each corruption, accuracy is averaged across 5 levels of severity.

| Dataset | Method | Avg. | Noise | | | Blur | | | | Weather | | | | Digital | | | |
|---|---|---|---|---|---|---|---|---|---|---|---|---|---|---|---|---|---|
| | | | Gauss. | Shot | Impulse | Defocus | Glass | Motion | Zoom | Snow | Frost | Fog | Bright | Contrast | Elastic | Pixel | JPEG |
| Pre-Trained | | 31.7 | 23 | 21 | 18 | 28 | 23 | 30 | 30 | 24 | 28 | 34 | 59 | 31 | 40 | 42 | 46 |
| FT | IN-V2 | 29.9 | 21 | 20 | 16 | 26 | 22 | 27 | 26 | 25 | 28 | 31 | 56 | 27 | 37 | 41 | 46 |
| | IN-A | 19.9 | 13 | 13 | 10 | 16 | 14 | 17 | 14 | 16 | 18 | 23 | 39 | 20 | 24 | 30 | 31 |
| | IN-R | 25.6 | 21 | 21 | 17 | 20 | 21 | 22 | 20 | 21 | 26 | 26 | 45 | 22 | 30 | 35 | 37 |
| | IN-Sketch | 19.2 | 16 | 15 | 13 | 11 | 12 | 14 | 12 | 17 | 22 | 20 | 37 | 18 | 21 | 28 | 32 |
| | ObjNet | 20.8 | 14 | 14 | 11 | 17 | 12 | 21 | 21 | 15 | 18 | 27 | 42 | 20 | 26 | 27 | 28 |
| | IN-Cartoon | 20.0 | 13 | 12 | 10 | 13 | 11 | 16 | 14 | 14 | 16 | 21 | 50 | 23 | 25 | 34 | 29 |
| | IN-Drawing | 15.1 | 16 | 15 | 11 | 4 | 9 | 9 | 8 | 14 | 24 | 8 | 41 | 6 | 18 | 19 | 22 |
| Linear Probing | IN-V2 | 29.8 | 23 | 22 | 18 | 25 | 23 | 27 | 26 | 23 | 27 | 28 | 56 | 25 | 39 | 41 | 45 |
| | IN-A | 28.7 | 23 | 21 | 18 | 23 | 21 | 25 | 23 | 23 | 28 | 30 | 53 | 26 | 36 | 39 | 42 |
| | IN-R | 25.9 | 19 | 18 | 15 | 19 | 19 | 22 | 22 | 20 | 24 | 23 | 53 | 22 | 36 | 36 | 39 |
| | IN-Sketch | 1.7 | 0 | 0 | 0 | 1 | 1 | 1 | 1 | 3 | 2 | 4 | 2 | 3 | 2 | 3 | 2 |
| | ObjNet | 25.0 | 16 | 15 | 13 | 21 | 16 | 25 | 25 | 18 | 21 | 34 | 49 | 27 | 33 | 30 | 33 |
| | IN-Cartoon | 21.8 | 13 | 12 | 10 | 17 | 15 | 20 | 19 | 15 | 17 | 21 | 49 | 22 | 31 | 32 | 34 |
| | IN-Drawing | 11.4 | 14 | 13 | 9 | 3 | 6 | 5 | 5 | 10 | 19 | 7 | 32 | 7 | 12 | 14 | 16 |
| Visual Prompt (Bahng et al., 2022) | IN-V2 | 21.4 | 18 | 16 | 14 | 13 | 13 | 18 | 20 | 17 | 19 | 17 | 45 | 15 | 29 | 32 | 35 |
| | IN-A | 7.3 | 5 | 5 | 4 | 3 | 3 | 5 | 7 | 6 | 6 | 5 | 22 | 4 | 11 | 12 | 13 |
| | IN-R | 16.6 | 13 | 12 | 10 | 9 | 9 | 13 | 15 | 14 | 15 | 13 | 38 | 12 | 23 | 25 | 27 |
| | IN-Sketch | 17.1 | 13 | 12 | 10 | 9 | 9 | 13 | 16 | 15 | 17 | 15 | 40 | 14 | 22 | 25 | 27 |
| | ObjNet | 12.9 | 8 | 8 | 6 | 7 | 7 | 10 | 14 | 11 | 11 | 12 | 31 | 10 | 19 | 18 | 21 |
| | IN-Cartoon | 17.8 | 12 | 11 | 10 | 11 | 10 | 14 | 16 | 13 | 14 | 16 | 41 | 15 | 24 | 29 | 30 |
| | IN-Drawing | 17.3 | 14 | 13 | 10 | 8 | 9 | 12 | 14 | 16 | 18 | 14 | 40 | 13 | 23 | 26 | 29 |
| EWC (Kirkpatrick et al., 2017) | IN-V2 | 31.5 | 25 | 23 | 19 | 27 | 23 | 29 | 28 | 25 | 29 | 31 | 58 | 26 | 40 | 43 | 47 |
| | IN-A | 22.0 | 18 | 17 | 14 | 16 | 14 | 18 | 16 | 17 | 20 | 24 | 42 | 18 | 27 | 33 | 35 |
| | IN-R | 29.0 | 23 | 22 | 19 | 23 | 23 | 26 | 26 | 22 | 27 | 27 | 54 | 24 | 37 | 38 | 42 |
| | IN-Sketch | 13.3 | 6 | 6 | 3 | 9 | 10 | 10 | 10 | 17 | 15 | 21 | 18 | 17 | 17 | 20 | 19 |
| | ObjNet | 24.9 | 17 | 17 | 13 | 21 | 16 | 25 | 26 | 19 | 20 | 32 | 49 | 23 | 33 | 30 | 34 |
| | IN-Cartoon | 20.7 | 12 | 11 | 9 | 16 | 13 | 19 | 17 | 14 | 16 | 20 | 47 | 22 | 28 | 33 | 33 |
| | IN-Drawing | 12.0 | 16 | 15 | 11 | 3 | 6 | 6 | 5 | 12 | 21 | 6 | 34 | 5 | 12 | 14 | 17 |
| LwF (Li & Hoiem, 2017) | IN-V2 | 31.0 | 22 | 20 | 16 | 27 | 23 | 28 | 27 | 26 | 29 | 33 | 57 | 29 | 39 | 43 | 47 |
| | IN-A | 26.7 | 19 | 18 | 15 | 22 | 19 | 23 | 21 | 22 | 25 | 31 | 50 | 26 | 32 | 37 | 39 |
| | IN-R | 30.3 | 24 | 23 | 20 | 25 | 24 | 27 | 26 | 24 | 29 | 32 | 53 | 28 | 36 | 41 | 43 |
| | IN-Sketch | 21.8 | 17 | 16 | 13 | 14 | 14 | 17 | 15 | 18 | 23 | 24 | 41 | 21 | 25 | 32 | 36 |
| | ObjNet | 25.6 | 17 | 16 | 13 | 22 | 17 | 25 | 25 | 20 | 22 | 32 | 49 | 25 | 31 | 33 | 35 |
| | IN-Cartoon | 29.0 | 19 | 18 | 14 | 22 | 20 | 25 | 24 | 21 | 25 | 31 | 61 | 31 | 37 | 44 | 43 |
| | IN-Drawing | 20.8 | 20 | 18 | 14 | 9 | 13 | 15 | 14 | 19 | 29 | 14 | 52 | 10 | 25 | 28 | 32 |
| LP-FT (Kumar et al., 2022) | IN-V2 | 29.8 | 21 | 20 | 16 | 26 | 22 | 27 | 26 | 25 | 28 | 31 | 56 | 27 | 37 | 41 | 46 |
| | IN-A | 22.6 | 15 | 14 | 12 | 19 | 17 | 20 | 17 | 18 | 21 | 26 | 44 | 22 | 28 | 31 | 34 |
| | IN-R | 27.5 | 23 | 22 | 18 | 22 | 23 | 23 | 22 | 23 | 28 | 29 | 48 | 24 | 32 | 37 | 39 |
| | IN-Sketch | 17.5 | 13 | 12 | 11 | 11 | 12 | 12 | 11 | 17 | 19 | 20 | 32 | 17 | 20 | 26 | 29 |
| | ObjNet | 22.2 | 15 | 15 | 11 | 18 | 14 | 22 | 23 | 17 | 19 | 29 | 44 | 21 | 28 | 28 | 30 |
| | IN-Cartoon | 19.7 | 13 | 12 | 10 | 13 | 10 | 16 | 14 | 14 | 16 | 20 | 49 | 22 | 25 | 32 | 28 |
| | IN-Drawing | 14.5 | 16 | 15 | 12 | 4 | 8 | 8 | 8 | 13 | 24 | 8 | 40 | 6 | 17 | 18 | 21 |
| WiSE-FT (Wortsman et al., 2022b) | IN-V2 | 32.3 | 23 | 21 | 18 | 29 | 24 | 30 | 29 | 26 | 30 | 34 | 59 | 30 | 40 | 44 | 48 |
| | IN-A | 30.7 | 22 | 21 | 18 | 27 | 22 | 27 | 26 | 24 | 28 | 35 | 56 | 31 | 37 | 42 | 45 |
| | IN-R | 33.6 | 27 | 26 | 22 | 29 | 26 | 30 | 29 | 27 | 33 | 36 | 57 | 31 | 39 | 44 | 47 |
| | IN-Sketch | 29.8 | 24 | 22 | 19 | 22 | 21 | 25 | 23 | 24 | 30 | 32 | 55 | 29 | 35 | 42 | 46 |
| | ObjNet | 30.4 | 22 | 21 | 18 | 26 | 20 | 29 | 30 | 23 | 26 | 36 | 60 | 30 | 37 | 39 | 43 |
| | IN-Cartoon | 28.8 | 19 | 17 | 15 | 22 | 18 | 25 | 23 | 22 | 25 | 32 | 60 | 31 | 37 | 43 | 43 |
| | IN-Drawing | 28.8 | 25 | 23 | 19 | 18 | 20 | 23 | 23 | 24 | 32 | 28 | 59 | 23 | 35 | 40 | 42 |
| Model Soup PRE-FT-EWC-LwF (Wortsman et al., 2022a) | IN-V2 | 32.2 | 23 | 22 | 18 | 29 | 24 | 29 | 29 | 26 | 29 | 34 | 59 | 29 | 40 | 44 | 48 |
| | IN-A | 29.2 | 22 | 20 | 17 | 25 | 20 | 26 | 23 | 23 | 26 | 33 | 54 | 28 | 35 | 41 | 43 |
| | IN-R | 32.9 | 27 | 26 | 22 | 28 | 26 | 29 | 28 | 26 | 32 | 34 | 57 | 30 | 39 | 43 | 47 |
| | IN-Sketch | 28.2 | 22 | 21 | 17 | 21 | 20 | 23 | 22 | 24 | 29 | 30 | 52 | 27 | 33 | 40 | 43 |
| | ObjNet | 29.3 | 21 | 20 | 17 | 26 | 19 | 29 | 29 | 23 | 25 | 36 | 54 | 29 | 36 | 37 | 40 |
| | IN-Cartoon | 27.5 | 18 | 17 | 14 | 22 | 18 | 24 | 22 | 20 | 23 | 29 | 58 | 29 | 36 | 42 | 41 |
| | IN-Drawing | 24.2 | 25 | 23 | 18 | 12 | 15 | 17 | 17 | 21 | 31 | 19 | 53 | 15 | 29 | 32 | 36 |

Table 51: Accuracy of ImageNet-1K pre-trained ResNet-50 with different fine-tuning methods and downstream datasets on each ImageNet-C corruption For each corruption, accuracy is averaged across 5 levels of severity.

| Dataset | Method | Avg. | Noise | | | Blur | | | | Weather | | | | Digital | | | |
|---|---|---|---|---|---|---|---|---|---|---|---|---|---|---|---|---|---|
| | | | Gauss. | Shot | Impulse | Defocus | Glass | Motion | Zoom | Snow | Frost | Fog | Bright | Contrast | Elastic | Pixel | JPEG |
| Pre-Trained | | 46.6 | 41 | 39 | 36 | 41 | 27 | 42 | 43 | 40 | 45 | 58 | 71 | 57 | 47 | 50 | 59 |
| FT | IN-V2 | 47.0 | 41 | 40 | 36 | 40 | 28 | 43 | 42 | 42 | 48 | 59 | 72 | 55 | 47 | 51 | 62 |
| | IN-A | 46.3 | 43 | 42 | 40 | 40 | 27 | 41 | 39 | 44 | 47 | 58 | 68 | 56 | 44 | 48 | 58 |
| | IN-R | 43.7 | 40 | 39 | 36 | 34 | 33 | 38 | 36 | 39 | 48 | 52 | 65 | 48 | 45 | 48 | 55 |
| | IN-Sketch | 23.3 | 20 | 19 | 16 | 13 | 11 | 17 | 16 | 26 | 28 | 33 | 41 | 23 | 22 | 29 | 35 |
| | ObjNet | 41.8 | 35 | 34 | 31 | 38 | 24 | 41 | 40 | 35 | 40 | 55 | 65 | 52 | 41 | 43 | 52 |
| | IN-Cartoon | 31.9 | 28 | 26 | 24 | 26 | 15 | 27 | 24 | 24 | 28 | 36 | 62 | 42 | 33 | 41 | 43 |
| | IN-Drawing | 16.3 | 24 | 18 | 20 | 2 | 4 | 5 | 5 | 24 | 37 | 8 | 49 | 7 | 12 | 11 | 17 |
| Linear Probing | IN-V2 | 45.6 | 42 | 40 | 38 | 36 | 28 | 42 | 40 | 40 | 47 | 52 | 70 | 57 | 46 | 49 | 57 |
| | IN-A | 45.4 | 45 | 44 | 42 | 35 | 28 | 39 | 37 | 41 | 48 | 53 | 67 | 54 | 44 | 47 | 55 |
| | IN-R | 40.6 | 35 | 32 | 30 | 33 | 27 | 36 | 37 | 35 | 39 | 47 | 67 | 50 | 47 | 43 | 51 |
| | IN-Sketch | 2.8 | 1 | 1 | 0 | 1 | 1 | 2 | 2 | 5 | 4 | 8 | 3 | 4 | 2 | 3 | 4 |
| | ObjNet | 39.4 | 32 | 30 | 29 | 36 | 23 | 39 | 38 | 33 | 38 | 53 | 64 | 51 | 39 | 39 | 48 |
| | IN-Cartoon | 35.6 | 30 | 25 | 25 | 29 | 22 | 31 | 29 | 29 | 34 | 41 | 65 | 44 | 41 | 44 | 47 |
| | IN-Drawing | 8.8 | 17 | 9 | 15 | 0 | 0 | 0 | 0 | 21 | 32 | 1 | 29 | 1 | 1 | 2 | 3 |
| Visual Prompt (Bahng et al., 2022) | IN-V2 | 36.4 | 31 | 30 | 26 | 28 | 20 | 32 | 36 | 31 | 34 | 44 | 63 | 42 | 40 | 40 | 49 |
| | IN-A | 32.8 | 27 | 26 | 23 | 22 | 16 | 28 | 31 | 29 | 31 | 42 | 59 | 39 | 37 | 38 | 46 |
| | IN-R | 32.8 | 26 | 25 | 22 | 24 | 17 | 29 | 32 | 28 | 31 | 41 | 59 | 40 | 37 | 37 | 45 |
| | IN-Sketch | 33.0 | 27 | 26 | 22 | 24 | 16 | 29 | 33 | 28 | 31 | 41 | 59 | 40 | 36 | 37 | 45 |
| | ObjNet | 33.1 | 27 | 25 | 22 | 24 | 17 | 29 | 34 | 28 | 31 | 42 | 59 | 39 | 37 | 36 | 45 |
| | IN-Cartoon | 34.5 | 29 | 27 | 23 | 26 | 18 | 30 | 34 | 30 | 32 | 43 | 61 | 41 | 39 | 39 | 47 |
| | IN-Drawing | 35.1 | 30 | 29 | 25 | 24 | 18 | 29 | 33 | 31 | 36 | 43 | 61 | 41 | 39 | 40 | 48 |
| EWC (Kirkpatrick et al., 2017) | IN-V2 | 46.0 | 41 | 39 | 36 | 38 | 28 | 43 | 41 | 40 | 46 | 56 | 71 | 54 | 47 | 50 | 59 |
| | IN-A | 46.8 | 44 | 43 | 41 | 38 | 29 | 42 | 40 | 43 | 48 | 58 | 70 | 53 | 46 | 49 | 58 |
| | IN-R | 42.9 | 37 | 35 | 33 | 34 | 28 | 39 | 39 | 38 | 42 | 50 | 68 | 50 | 47 | 47 | 54 |
| | IN-Sketch | 13.2 | 6 | 5 | 3 | 7 | 8 | 10 | 10 | 24 | 22 | 21 | 21 | 12 | 15 | 17 | 18 |
| | ObjNet | 41.3 | 34 | 33 | 30 | 37 | 24 | 40 | 40 | 35 | 38 | 55 | 66 | 53 | 42 | 41 | 51 |
| | IN-Cartoon | 33.1 | 25 | 22 | 21 | 28 | 19 | 30 | 27 | 25 | 30 | 40 | 63 | 42 | 38 | 43 | 46 |
| | IN-Drawing | 7.5 | 10 | 7 | 10 | 0 | 0 | 0 | 0 | 21 | 31 | 0 | 26 | 0 | 1 | 1 | 2 |
| LwF (Li & Hoiem, 2017) | IN-V2 | 47.5 | 42 | 41 | 37 | 41 | 28 | 43 | 42 | 43 | 48 | 59 | 72 | 56 | 47 | 52 | 63 |
| | IN-A | 46.8 | 43 | 42 | 40 | 40 | 28 | 42 | 40 | 44 | 47 | 59 | 69 | 57 | 44 | 49 | 58 |
| | IN-R | 44.9 | 41 | 41 | 37 | 36 | 34 | 39 | 37 | 40 | 48 | 53 | 66 | 50 | 46 | 49 | 56 |
| | IN-Sketch | 21.4 | 18 | 17 | 14 | 13 | 10 | 15 | 15 | 26 | 27 | 32 | 36 | 19 | 20 | 27 | 32 |
| | ObjNet | 43.1 | 37 | 36 | 32 | 39 | 25 | 42 | 41 | 37 | 41 | 56 | 67 | 54 | 42 | 45 | 54 |
| | IN-Cartoon | 34.6 | 30 | 28 | 25 | 28 | 17 | 30 | 27 | 27 | 31 | 39 | 65 | 44 | 36 | 44 | 47 |
| | IN-Drawing | 12.3 | 20 | 13 | 18 | 1 | 2 | 2 | 2 | 25 | 38 | 4 | 42 | 4 | 5 | 5 | 7 |
| LP-FT (Kumar et al., 2022) | IN-V2 | 47.1 | 42 | 40 | 37 | 40 | 28 | 43 | 42 | 42 | 48 | 59 | 71 | 55 | 47 | 51 | 62 |
| | IN-A | 46.4 | 43 | 42 | 39 | 39 | 28 | 42 | 40 | 44 | 48 | 58 | 69 | 56 | 45 | 48 | 58 |
| | IN-R | 44.2 | 40 | 40 | 36 | 35 | 34 | 38 | 37 | 40 | 48 | 52 | 66 | 48 | 46 | 49 | 55 |
| | IN-Sketch | 24.2 | 21 | 19 | 17 | 15 | 13 | 18 | 19 | 27 | 29 | 32 | 41 | 23 | 23 | 30 | 35 |
| | ObjNet | 41.8 | 35 | 34 | 31 | 38 | 24 | 41 | 40 | 35 | 40 | 55 | 66 | 52 | 41 | 43 | 52 |
| | IN-Cartoon | 32.2 | 29 | 27 | 25 | 26 | 15 | 28 | 24 | 24 | 28 | 36 | 62 | 42 | 33 | 41 | 43 |
| | IN-Drawing | 16.7 | 26 | 21 | 22 | 1 | 4 | 6 | 5 | 23 | 37 | 7 | 48 | 5 | 14 | 13 | 19 |
| WiSE-FT (Wortsman et al., 2022b) | IN-V2 | 48.1 | 42 | 40 | 37 | 42 | 28 | 43 | 43 | 42 | 48 | 60 | 73 | 59 | 48 | 53 | 63 |
| | IN-A | 48.9 | 44 | 43 | 40 | 43 | 29 | 43 | 44 | 45 | 48 | 61 | 72 | 60 | 48 | 52 | 61 |
| | IN-R | 49.3 | 45 | 44 | 41 | 43 | 34 | 43 | 43 | 43 | 50 | 59 | 72 | 57 | 50 | 53 | 62 |
| | IN-Sketch | 41.5 | 40 | 40 | 37 | 33 | 26 | 37 | 37 | 36 | 40 | 48 | 66 | 41 | 42 | 47 | 55 |
| | ObjNet | 46.8 | 41 | 40 | 36 | 43 | 27 | 44 | 44 | 40 | 45 | 59 | 71 | 58 | 46 | 50 | 59 |
| | IN-Cartoon | 42.8 | 37 | 35 | 32 | 37 | 22 | 38 | 35 | 37 | 40 | 51 | 72 | 53 | 46 | 50 | 57 |
| | IN-Drawing | 41.8 | 42 | 39 | 38 | 25 | 22 | 34 | 32 | 39 | 48 | 50 | 71 | 48 | 42 | 45 | 53 |
| Model Soup PRE-FT-EWC-LwF (Wortsman et al., 2022a) | IN-V2 | 48.0 | 42 | 40 | 37 | 42 | 28 | 43 | 43 | 43 | 48 | 60 | 73 | 59 | 48 | 52 | 63 |
| | IN-A | 48.9 | 45 | 44 | 41 | 42 | 29 | 44 | 43 | 45 | 49 | 61 | 71 | 60 | 47 | 51 | 61 |
| | IN-R | 48.4 | 44 | 43 | 40 | 41 | 34 | 42 | 42 | 43 | 49 | 58 | 71 | 55 | 50 | 52 | 60 |
| | IN-Sketch | 38.7 | 36 | 35 | 33 | 29 | 22 | 32 | 33 | 37 | 40 | 46 | 63 | 37 | 39 | 45 | 52 |
| | ObjNet | 45.7 | 39 | 38 | 35 | 41 | 26 | 44 | 44 | 39 | 44 | 59 | 70 | 57 | 45 | 48 | 57 |
| | IN-Cartoon | 39.5 | 34 | 31 | 29 | 34 | 21 | 35 | 32 | 32 | 36 | 46 | 69 | 49 | 43 | 48 | 53 |
| | IN-Drawing | 28.9 | 38 | 32 | 34 | 6 | 11 | 14 | 13 | 35 | 46 | 29 | 64 | 24 | 26 | 26 | 35 |

