# OpenReview forum: "ImageNet-RIB Benchmark: Large Pre-Training Datasets Don't Guarantee Robustness after Fine-Tuning"
_ICLR.cc/2025/Conference — Submitted to ICLR 2025_

### Official Review · Reviewer_1GT2 · 2024-11-02

**Soundness:** 2
**Presentation:** 1
**Contribution:** 2
**Rating:** 3
**Confidence:** 5

**Summary:**

Benchmarking Fine-Tuning Robustness: The study assesses the robustness of models by fine-tuning on one OOD dataset and evaluating on others, using diverse fine-tuning strategies (vanilla fine-tuning, linear probing, LoRA, and continual learning methods).

Dataset Distance Metrics: The paper explores how dataset distance metrics can predict robustness changes post-fine-tuning, using Optimal Transport Dataset Distance (OTDD).

Experimental Insights: Findings show that models pre-trained on large, diverse datasets are more robust on OOD datasets. Regularization-based continual learning with Model Soup achieves the best results, and linear probing performs best with LAION-2B pre-training, while pre-training on large datasets like LAION-2B is not always optimal for robustness preservation on downstream tasks.

**Strengths:**

This paper presents a comprehensive set of experiments.

**Weaknesses:**

The clarity of the writing could be improved in several areas:
Figure 3: Could you clarify what each point represents—are they the six downstream datasets? e.g. include a legend explaining what each point represents.
Continual Learning Setup: How exactly is continual learning trained across the pretraining and downstream datasets?
Figure 4: The blue bar for ‘IN-1k’ indicates a model pretrained on IN-1k and evaluated on an OOD downstream dataset. Do the blue bars for “IN-21k,” “IN-21k+augReg,” and “LAION-2B” represent models pretrained on these datasets, then fine-tuned on IN-1k, and evaluated on OOD? How are the red bars trained and evaluated? For example, does the red bar for “LAION-2B” mean pretraining on LAION, then fine-tuning on IN-1k, followed by further fine-tuning on an OOD downstream dataset, and finally evaluating on OOD? Could you elaborate on these details?

Additionally, the evaluation setup of ImageNet-RIB is the same as a single-domain generalization, where training occurs on one domain and evaluation on the remaining N-1. However, related works on single-domain generalization are not discussed. I recommend including a brief discussion of related works on single-domain generalization and explaining how your evaluation protocol differs from these existing ones.

Some observations in Section 4.2 are unsurprising:
Models pre-trained on larger, more diverse datasets perform better on both ImageNet-1K and its variants.
Fine-tuning on a downstream dataset that differs significantly from the pretraining dataset reduces pretraining accuracy. This similarity could be quantified using a distance metric in either image or feature space.


Lastly, the experiment in Section 4.4 is somewhat confusing. This section is crucial, as the authors make a counter-intuitive claim, and it's important to ensure the conclusions are accurate to avoid misleading readers. I will revisit this section once the experimental details in Figure 4 are clarified.

**Questions:**

I recommend that the authors provide more detail on the experimental setup in Sections 4.2, 4.3, and 4.4, this would help ensure a clear understanding of the setup.

---

> ### Author Response · Authors · 2024-11-17
> **Response to Reviewer 1GT2 (1/3)**
>
> We thank the reviewer for their valuable comments and feedback. In our revision, we will aim to carefully follow each of the reviewer’s suggestions. We have made our best effort to answer each of the reviewer’s questions and concerns and would be happy to address any further comments arising in the discussion period.
>
> **Re: Figure 3: Could you clarify what each point represents—are they the six downstream datasets? e.g. include a legend explaining what each point represents.**
>
> Thank you for pointing out. We used different markers for each OOD dataset: ImageNet-V2, ImageNet-A, ImageNet-R, ImageNet-Sketch, and ObjectNet in the revision. The distance 0 denotes the ImageNet-1K validation set. Note that we excluded synthetic datasets since they are made from the ImageNet-1K validation set.
>
> **Re: Continual Learning Setup: How exactly is continual learning trained across the pretraining and downstream datasets?**
>
> EWC and LwF are regularization-based continual learning methods that incorporate a regularization term into the loss function to minimize changes to model parameters. These methods do not necessarily require access to the pre-training dataset. Specifically, EWC uses the Fisher information matrix as a regularization term based on the weights of the pre-trained model  (i.e., $F(W, W_pre$) where $F$, $W$, $W_{pre}$ denotes Fisher information matrix, weights of the current model, and weights of pre-trained models, respectively). In its original formulation, EWC applies a layer-wise importance coefficient for the regularization term to reflect each layer’s relevance to the old task. However, since the pre-training dataset is not accessible (as noted in L269 in the original submission and L323 in the revision), we set the coefficient to 1 for all layers.
>
> LwF, on the other hand, employs the Kullback-Leibler (KL) divergence between the logits of the current model and those of the pre-trained model (i.e., $D_{\text{KL}}(p_{\text{curr}} \parallel p_{\text{pre}})$). To compute the logits, we used a temperature of 2, following Li & Hoiem (2017). For further details on these methods, please refer to the original papers on EWC (Kirkpatrick et al., 2017) and LwF (Li & Hoiem, 2017).
>
> **Re: Figure 4: The blue bar for ‘IN-1k’ indicates a model pretrained on IN-1k and evaluated on an OOD downstream dataset. Do the blue bars for “IN-21k,” “IN-21k+augReg,” and “LAION-2B” represent models pre-trained on these datasets, then fine-tuned on IN-1k, and evaluated on OOD?  How are the red bars trained and evaluated? For example, does the red bar for “LAION-2B” mean pretraining on LAION, then fine-tuning on IN-1k, followed by further fine-tuning on an OOD downstream dataset, and finally evaluating on OOD? Could you elaborate on these details?**
>
> The reviewer’s understanding is correct. As stated in L252-255 and Table 5  (L260-261 and Table 6 in the revision), we used models pre-trained on one dataset, and then fine-tuned them on ImageNet-1K, with the exception of the model directly pre-trained on ImageNet-1K. These models were fine-tuned on each downstream dataset, and mRI was calculated.
>
> The blue bars represent the accuracy of pre-trained models on OOD datasets without fine-tuning (i.e., the average accuracy on OOD datasets shown in Table 6 in the original submission and Table 7 in the revision). The red bars, on the other hand, represent the sum of mRI and accuracy. As mentioned in L406 (L462 in the revision), this corresponds to the average accuracy on OOD datasets. We will clarify this in the revised manuscript.

---

> ### Author Response · Authors · 2024-11-17
> **Response to Reviewer 1GT2 (2/3)**
>
> **Re: The evaluation setup of ImageNet-RIB is the same as a single-domain generalization, where training occurs on one domain and evaluation on the remaining N-1. However, related works on single-domain generalization are not discussed. I recommend including a brief discussion of related works on single-domain generalization and explaining how your evaluation protocol differs from these existing ones.**
>
> We have added a discussion about single-domain generalization to the related work section. Here is the text:
>
> Single-domain generalization [1GT2-1] refers to the task where only one source domain (e.g., MNIST) is available during training, and the model is evaluated on multiple unseen target domains (e.g., SVHN, SYN, MNIST-M, USPS). While the high-level concept is similar to existing robust fine-tuning benchmarks (e.g., Taori et al., 2020), the objectives differ. Robust fine-tuning focuses on maintaining or improving a model’s robustness to OOD datasets during fine-tuning, whereas single-domain generalization aims to achieve generalization to unseen OOD datasets, often through meta-learning-based data augmentation [1GT2-1, 1GT2-4] or adaptive batch normalization [1GT2-2].
>
> Recently, Fan et al. [1GT2-2] applied single-domain generalization to the PACS dataset [1GT2-3], using one domain as the training set and the remaining domains as test sets. This setup resembles our ImageNet-RIB benchmark in that each dataset is used for training while the others are used for testing. However, the goals of the two benchmarks differ: our robust fine-tuning benchmark aims to mitigate robustness degradation during fine-tuning, while single-domain generalization benchmarks focus on improving generalizability from a single source domain.
>
> References:
>
> [1GT2-1] Qiao, Fengchun, Long Zhao, and Xi Peng. "Learning to learn single domain generalization." *Proceedings of the IEEE/CVF conference on computer vision and pattern recognition*. 2020.
>
> [1GT2-2] Fan, Xinjie, et al. "Adversarially adaptive normalization for single domain generalization." *Proceedings of the IEEE/CVF conference on Computer Vision and Pattern Recognition*. 2021.
>
> [1GT2-3] Li, Da, et al. "Deeper, broader and artier domain generalization." *Proceedings of the IEEE international conference on computer vision*. 2017.
>
> [1GT2-4] Chen, Jin, et al. "Meta-causal learning for single domain generalization." *Proceedings of the IEEE/CVF Conference on Computer Vision and Pattern Recognition*. 2023.
>
> **Re: Some observations in Section 4.2 are unsurprising: Models pre-trained on larger, more diverse datasets perform better on both ImageNet-1K and its variants. Fine-tuning on a downstream dataset that differs significantly from the pretraining dataset reduces pretraining accuracy. This similarity could be quantified using a distance metric in either image or feature space.**
>
> We agree with the reviewer’s point that models pre-trained on the larger and more diverse datasets perform better on ImageNet-1K and its variants and it is not surprising as we mentioned in L311 (L348 in the revision). We provided the accuracy of pre-trained models in Table 6 (Table 7 in the revision) as a baseline. Moreover, it might be intuitive that if the downstream dataset is significantly “different” from the pre-training dataset, it leads to more catastrophic forgetting (performance drop on the pre-training dataset).
>
> However, the notion of “difference” is usually determined by humans prior and it is not well established and quantified. Therefore, we measured two different dataset distance metrics on image and feature space and checked whether it is predictable or not. Please refer to Appendix B for more details.
>
>  Additionally, it can be intuitive that fine-tuning on a downstream dataset significantly different from the pre-training dataset leads to more catastrophic forgetting (i.e., a larger performance drop on the pre-training dataset). However, the concept of “difference” is typically informed by human intuition and is not well-defined or consistently quantified. To address this, we measured dataset similarity using two distance metrics in image space and feature space and evaluated whether these metrics can predict performance drops. Further details can be found in Appendix B.

---

> ### Author Response · Authors · 2024-11-17
> **Response to Reviewer 1GT2 (3/3)**
>
> **Re: I recommend that the authors provide more detail on the experimental setup in Sections 4.2, 4.3, and 4.4, this would help ensure a clear understanding of the setup.**
>
> Thank you for the suggestion. We have updated the Section 4 and added Appendix A.3 in the revision. Here are some copies of the updated parts regarding the experimental setup.
>
> Section 4.1 Experimental Details - Pre-Trained Models paragraph
>
> Pre-Trained Models We use several architectures of Vision Transformer (ViT) (Dosovitskiy et al.,
> 2021) and ResNet (He et al., 2016). The models are pre-trained on ImageNet-1K (Russakovsky et al., 2015), or ImageNet-21K (Ridnik et al., 2021) and then fine-tuned on ImageNet-1K. The standard data augmentation and regularization technique for ViT, AugReg (Steiner et al., 2022) can also be used for training on ImageNet-1K or ImageNet-21K. Alternatively, some models are pre-trained on LAION-2B (Schuhmann et al., 2022) or OpenAI CLIP (Radford et al., 2021), followed by fine-tuning on ImageNet-1K. In other words, all pre-trained models are trained on ImageNet-1K to directly leverage its classifier before conducting experiments. For simplicity, we refer to them by the names of the first pre-training datasets (e.g., ImageNet-21K, LAION-2B).
> In the main paper, we focus on ImageNet-1K with AugReg pre-trained ViT-B/16, and experiments
> using other pre-trained models are reported in Appendix D along with ImageNet-1K with SAM (Chen et al., 2022) and ImageNet-21K-P (Ridnik et al., 2021). We employ the timm (Wightman, 2019) and torchvision (maintainers & contributors, 2016) for acquiring model weights and implementation. Please refer to Appendix A.2 for more details.
>
> In Figure 6, we added how to compute each bars.
>
> The average accuracy on OOD datasets before (blue) and after (red) fine-tuning with each method on downstream datasets. The red bar is calculated directly by evaluating pre-trained models on OOD datasets while the blue bar is calculated by adding $mRI$ of each method to the pre-trained models' accuracy. Note that it is identical to the average accuracy on OOD datasets after fine-tuning on each downstream dataset ($mRI + \frac{1}{n}\sum_i^n A^{(i)}\text{pre} = \frac{1}{n}\sum_j \frac{1}{n-1}\sum_{i, i\neq j}^n A^{(i)}_\text{down}$)
>
> Appendix A.3 Training and Hyperparameters
> Each pre-trained model is fine-tuned on the downstream dataset for 10 epochs where the average accuracy on downstream datasets for each pre-trained ViT-B/16 model achieves more than 90% with vanilla fine-tuning. We applied LoRA on query and value projection layers with rank 8 following the original implementation (Hu et al., 2021). We use 2 as a temperature for calculating KL divergence for LwF following Li & Hoiem (2017). For WiSE-FT, we use the interpolation ratio between pre-trained and fine-tuned models as 0.5 following the recommendation by Wortsman et al. (2022b) instead of finding the best hyperparameters evaluated on the benchmark.
>
>
> Again, we truly appreciate the reviewer's comment and feedback. We hope that all concerns are resolved from our response or we are happy to further explain details and willing to update the manuscript.

---

> ### Author Response · Authors · 2024-11-25
>
> We appreciate the reviewer for their insightful review! We’ve made significant additions in direct response to your feedback. We would greatly appreciate any further comments and kindly ask you to consider adjusting your score as the discussion period comes to a close. Thank you!

---

> > ### Author Response · Authors · 2024-12-01
> > **Gentle Reminder to Reviewer 1GT2: Response to Author Response and Revisions**
> >
> > As the response deadline is approaching, we kindly wanted to follow up and ask if you’ve had an opportunity to review our response and the updates we made to the manuscript.
> > The reviewer’s concerns were 1) Difference between ImageNet-RIB and existing benchmarks, and 2) Clarification of benchmark and experiments. We described in the above responses and revision.

---

### Official Review · Reviewer_SbRm · 2024-11-02

**Soundness:** 4
**Presentation:** 3
**Contribution:** 4
**Rating:** 6
**Confidence:** 4

**Summary:**

The paper proposes the ImageNet-RIB Benchmark, which is a new benchmark to evaluate the robustness of models fine-tuned on downstream tasks. The authors did an extensive evaluation of ViT and ResNet models on the benchmark with different robustness and continual learning strategies.

**Strengths:**

The community is going to benefit from having such a benchmark. I found it good, and the authors did an extensive evaluation of different robustness strategies. Overall, the quality is good, and the idea of the paper is clean and easy to follow. The authors did a good job on the evaluation using many experiments.

**Weaknesses:**

There are some technical considerations that need to be added to the material for better reproducibility; for instance, why choose 10 epochs for fine-tuning for downstream tasks? Also, I didn't see any discussion about the LoRa rank selected for the evaluations or a hyperparameter search.

Another hyperparameter is for the WiSE-FT; instead of choosing 0.5, the authors could provide the experiment with the trade-off by choosing 0.5 and generalizing more for OOD with the IMAGENET-RIB and selecting the lambda with D_{down} validation and evaluating it on the OOD. This analysis could be interesting, especially when combined with the insight of having a dataset that is far from the pre-trained and its impact on the ImageNet-RIB.

Figure 2 could have better legends, especially step 3; otherwise, it is hard to understand.

In Table 8, there are some methods with an accuracy of 100.0. Could the authors provide more insight? The task is not so challenging ...

**Questions:**

Could the authors provide more insight on evaluation time for getting one mRI on normal FT, for instance? It would be good to provide such information as the number of total images (sum of all OOD images used for eval) and get the mRI.

Could you provide more insights about the LoRa rank used?

I hope the authors can address the weakness section and the ones added here.

---

> ### Author Response · Authors · 2024-11-17
> **Response to Reviewer SbRm (1/2)**
>
> We are grateful to the reviewer for their detailed review and valuable feedback on our paper. Your comments have provided significant insights that have helped us identify areas for further clarification and improvement. We have carefully considered each of your points and provide detailed responses below, along with the changes we will incorporate into the revised manuscript. We believe these modifications will strengthen our work and better highlight the contributions of our approach to the field.
>
> Moreover, we would like to emphasize that we shared the [anonymous code repository](https://anonymous.4open.science/r/imagenet-rib/) in the first footnote on page 1 for reproducibility.
>
> **Re: Why choose 10 epochs for fine-tuning for downstream tasks?**
>
> We selected 10 epochs for fine-tuning because, as shown in Figure 6 (left) in the revision, the average accuracy on the downstream datasets for each pretrained ViT-B/16 model reaches approximately 90% of its final value within this number of epochs.
>
> **Re: Another hyperparameter is for the WiSE-FT; instead of choosing 0.5, the authors could provide the experiment with the trade-off by choosing 0.5 and generalizing more for OOD with the IMAGENET-RIB and selecting the lambda with D_{down} validation and evaluating it on the OOD. This analysis could be interesting, especially when combined with the insight of having a dataset that is far from the pre-trained and its impact on the ImageNet-RIB.**
>
> Thank you for suggesting a stronger baseline. We conducted a grid search from 0.1 to 0.9 with an increment of 0.1 to find the best-performing ratio between the pre-trained model’s weight and the fine-tuned model’s weight for WiSE-FT (weights = $\alpha$ * PRE + (1-$\alpha$) * FT). Note that this hyperparameter search based on the test result (mRI) is an unfair comparison with other methods.  Table SbRm-1 shows $mRI$ of WiSE with the default ratio and the best ratio. As the reviewer expected, the performance of WiSE-FT is better at different ratios between pre-trained weight and fine-tuned weight. Similarly, hyperparameter search for the model soup that uses network weights from pre-trained model and fine-tuned models with FT, EWC, and LwF can improve the performance as WiSE-FT is a special case of Model Soup when ratios for EWC and LwF are set to 0.
>
> **Table SbRm-1 mRI of ViT-B/16 pre-trained on various datasets using WiSE-FT with default ratio (0.5) and the best ratio (weights = $\alpha$ * PRE + (1-$\alpha$) * FT).**
>
> | Pre-Training Dataset | WiSE-FT ($\alpha$ = 0.5) | WiSE-FT (best $\alpha$) | best $\alpha$|
> | --- | --- | --- |--- |
> | IN-1K + AugReg |  3.6 | 4.7 | 0.4|
> | IN-1K + SAM |  3.6 | 4.1 | 0.3|
> | IN-21K |  2.5 |  2.5 | 0.5|
> | IN-21K-P |  3.0 |  3.0 | 0.5|
> | IN-21K + AugReg |  1.7 | 2.3 | 0.7|
> | OpenAI |  -18.1|  -1.6 | 0.9 |
> | LAION-2B |  -21.6 |  -2.4 | 0.9|
>
> **Re: Figure 2 could have better legends, especially step 3; otherwise, it is hard to understand.**
>
> We appreciate the reviewer’s comment. We have changed the Figure 2 for better understanding.
>
> **Re: In Table 8, there are some methods with an accuracy of 100.0. Could the authors provide more insight? The task is not so challenging ...**
>
> Table 8 presents the performance on the downstream datasets. We did not split the downstream datasets into training and testing/validation sets because these datasets were used exclusively for evaluation when other datasets were treated as downstream datasets. This approach aligns with prior works (e.g., Taori et al., 2020; Hendrycks et al., 2021ab), where the entire downstream datasets were used for testing. We clarified it in the revision.
>
> **Re: Could the authors provide more insight on evaluation time for getting one mRI on normal FT, for instance? It would be good to provide such information as the number of total images (sum of all OOD images used for eval) and get the mRI.**
>
> Across the 8 datasets, there are a total of 4,006,074 images, with the majority coming from ImageNet-C, which includes 15 types of corruption at five levels of severity applied to the ImageNet-1K validation set. This total is approximately 3.1 times larger than the ImageNet-1K training set.
>
> Evaluation using ViT-B/16 on a single NVIDIA A100 GPU took approximately 70 minutes per dataset. Sequential evaluation across all 8 datasets would require 560 minutes in total. However, since the model requires only 2.5GB of VRAM for inference, we were able to run multiple evaluations in parallel on each GPU to reduce the total time.
>
> Below is a table summarizing the number of images per dataset:
>
> **Table SbRm-2**
>
> |Dataset |  Number of Images|
> | --- | --- |
> | ImageNet-V2 | 50,000 |
> | ImageNet-A | 7,500 |
> | ImageNet-R | 30,000 |
> | ObjectNet | 18,574 |
> | ImageNet-Sketch | 50,000 |
> | ImageNet-Cartoon | 50,000 |
> | ImageNet-Drawing | 50,000 |
> | ImageNet-C | 3,750,000 |

---

> ### Author Response · Authors · 2024-11-17
> **Response to Reviewer SbRm (2/2)**
>
> **Re: Also, I didn't see any discussion about the LoRa rank selected for the evaluations or a hyperparameter search. Could you provide more insights about the LoRa rank used?**
>
> We followed the instructions from the official LoRA repository (https://github.com/microsoft/LoRA). Specifically, only the query and value projectors of the network were applied to LoRA with a rank of 8. We have added this detail to Appendix A.3 for clarification.
>
> For more details on the implementation, please refer to our submitted code repository.

---

> ### Comment · Reviewer_SbRm · 2024-11-23
>
> Hello, I would like to thank the authors for the answers. I think that most of my questions were answered.
>
> But I am still not convinced by this: "We selected 10 epochs for fine-tuning because, as shown in Figure 6 (left) in the revision, the average accuracy on the downstream datasets for each pretrained ViT-B/16 model reaches approximately 90% of its final value within this number of epochs."
>
> - For me, this is not a good strategy; I recommend adding more epochs and selecting the best based on validation curves.

---

> > ### Author Response · Authors · 2024-11-25
> >
> > We are glad that the reviewer’s concerns were mostly resolved and we appreciate the reviewer for their response.
> >
> > **Re: For me, this is not a good strategy; I recommend adding more epochs and selecting the best based on validation curves.**
> >
> > Following the reviewer’s recommendation, we split the downstream dataset into training and validation sets and trained longer epochs to find the best-performing model on the validation set. We have used the best model for using robust-fine-tuning methods (LP-FT, WiSE-FT, Model Soup) and evaluating OOD performance. Due to resource constraints, we are only using ViT-B/16 pre-trained on IN-1K + AugReg, IN-21K + AugReg, and LAION-2B. We expect that the result will come out before the author's response deadline.
> > We will share them once it is ready.
> >
> > We appreciate again the reviewer for their insightful comments and we hope that this and the next responses will remove all concerns the reviewer has.

---

> > > ### Comment · Reviewer_SbRm · 2024-11-26
> > >
> > > Hello authors, thanks for the follow-up response to my concerns; I think all my points were addressed, so I decided to increase my confidence score in my review and contribution score, even though I am keeping the same paper score (6: marginally above the acceptance threshold) which is good compared to other reviews.

---

> ### Author Response · Authors · 2024-11-29
> **Results from Splitting the Downstream Dataset and Extending Training for Optimal Model Selection**
>
> We are glad that the reviewer’s concerns were mostly resolved and we appreciate the reviewer for their response. As promised, we have shared results that split the downstream dataset into training and validation sets and choose the best one running more epochs.
>
> **Re: I recommend adding more epochs and selecting the best based on validation curves.**
>
> Following the reviewer’s recommendation, we split the downstream dataset into training and validation sets and extended the training duration (25 epochs) to identify the best-performing model on the validation set. We then used this model to apply robust fine-tuning methods (LP-FT, WiSE-FT, Model Soup) and evaluate out-of-distribution (OOD) performance. Due to resource constraints, we limited our experiments to ViT-B/16 models pre-trained on IN-1K + AugReg, IN-21K + AugReg, and LAION-2B. Table SbRm3 presents the average number of epochs required for each model to achieve its highest validation accuracy on the downstream datasets. Notably, methods such as Visual Prompt, LoRA, and EWC required more training epochs compared to their counterparts.
>
> **Table SbRm3**: The average number of epochs needed to achieve the highest validation accuracy on each downstream dataset. Note that WiSE-FT and Model Soup are post-hoc weight interpolation methods and do not involve training.
>
> | Method | IN-1K + AugReg | IN-21K + AugReg | LAION  |
> | -------- | -------------- | --------------- | ------ |
> | FT       | 4.2           | 7.1           | 12.1 |
> | Linear Probing | 17.8          | 11.5            | 14.1 |
> | Visual Prompt | 22.4         | 22.4          | 20.6 |
> | LoRA     | 20.1         | 16.5            | 15.6 |
> | EWC      | 18.1         | 19.8           | 24.5   |
> | LwF      | 3.8           | 4.2            | 11.5   |
> | LP-FT     | 7.2           | 13.2           | 13.8  |
>
> Table SbRm4 reports the **mRI** values for each method. The performance is not significantly different from the results presented in Table 3 of the manuscript, where the entire downstream dataset was used for fine-tuning with models trained for 10 epochs. If the paper is accepted, we will also include results for all pre-trained models under this revised setting.
>
> **Table SbRm4**: mRI values obtained using the best validation accuracy for each model on the downstream datasets. Parentheses indicate the accuracy difference compared to models fine-tuned for 10 epochs without splitting the training and validation sets.
>
> | Method | IN-1K + AugReg | IN-21K + AugReg | LAION  |
> | ------------------- | -------------- | --------------- | ------ |
> | FT                  | 2.8 (+1.5) | \-5.6 (-0.1)| \-40.4 (-2.3) |
> | Linear Probing            | 1.6 (+0.9) | \-0.7 (-0.4) | **\-2.3 (-0.3)** |
> | Visual Prompt            | \-4.1 (+0.4) | \-9.1 (-0.3) | \-8.9 (-0.7)  |
> | LoRA                | 2.6 (+1.7) | 1.2 (3.3)| \-2.9 (+0.7) |
> | EWC                 | 3.6 (+0.8) | 0.5 (-0.1)| \-14.1 (-1.6) |
> | LwF                 | 4.0 (+0.9) | \-1.4 (-0.4)| \-34.6 (-0.7) |
> | LP-FT                | 3.7 (+1.4) | \-2.4 (0.2)| \-36.7 (0.4) |
> | WiSE-FT         | 4.5 (+0.9) | 1.6 (-0.1)| \-23.0 (-1.4) |
> | MS:PRE-FT-EWC-LwF | **4.8 (+0.9)** | **2.0 (-0.2)**| \-19.2 (-1.3)|
>
> **Re: I am keeping the same paper score (6: marginally above the acceptance threshold)**
> We would like to kindly note that, as the reviewer acknowledges, a rating of 6 denotes a borderline accept in the ICLR rating system, unlike NeurIPS 2024 where a rating of 6 means weak accept. Given that the reviewer mentioned all concerns were resolved and acknowledged the potential of our work to the community, `The community is going to benefit from having such a benchmark. I found it good,`  we are wondering if there might be any additional aspects of the submission that remain unresolved or could be further clarified. If so, we are glad to deal with them.
>
> Again, we truly value your feedback and would appreciate any further suggestions that could strengthen the submission. We would appreciate your consideration for an updated rating if you feel that our responses and the revisions have sufficiently addressed your concerns.

---

### Official Review · Reviewer_t68W · 2024-11-03

**Soundness:** 2
**Presentation:** 3
**Contribution:** 3
**Rating:** 5
**Confidence:** 3

**Summary:**

This paper proposes a benchmark for testing robustness, based on existing diverse image recognition datasets. By training on one dataset and testing on others, the robustness can be evaluated through accuracy before and after training. The results show that current incremental learning and robustness training methods effectively improve robustness, while larger models tend to exhibit greater robustness.

**Strengths:**

1. This paper conducts extensive experiments, exploring the results of various methods that could potentially improve robustness, and also evaluates the effects of models of different sizes.
2. The paper is well-written.
3. The evaluation dataset is more diverse.
4. This paper finds that more data does not necessarily lead to greater robustness.

**Weaknesses:**

1. It seems that the paper did not collect or filter new data; rather, it appears to be more of an evaluation framework than a benchmark.
2. The conclusions drawn are quite obvious, as many similar studies have reached similar findings. It's unclear how this "benchmark" could benefit the community.

**Questions:**

1. Apart from the different datasets, is there any core point that sets this evaluation apart from previous assessments?

---

> ### Author Response · Authors · 2024-11-17
> **Response to Reviewer t68W**
>
> We appreciate the reviewer’s thoughtful comments. In the responses below, we address each of your comments.
>
> **Re: It seems that the´ paper did not collect or filter new data; rather, it appears to be more of an evaluation framework than a benchmark.  It's unclear how this "benchmark" could benefit the community. Apart from the different datasets, is there any core point that sets this evaluation apart from previous assessments?**
>
> We would like to emphasize that ImageNet-RIB differs significantly from the existing benchmark proposed by Taori et al. (2020). As noted in L129-132 and L140-142 (L129-132 and L143-145 in the revision), the Taori et al. benchmark uses ImageNet-1K as the downstream dataset and evaluates robustness changes during fine-tuning by testing on OOD datasets such as ImageNet-V2, ImageNet-A, ImageNet-R, ImageNet-Sketch, and ObjectNet. In contrast, ImageNet-RIB starts with a model pre-trained on a dataset typically larger than ImageNet-1K and then fine-tuned on ImageNet-1K.
>
>  Hence, given a set of ImageNet-based OOD datasets, the pre-trained model is fine-tuned on one OOD dataset and evaluated on the remaining datasets. This process is repeated such that the pre-trained model is fine-tuned on each OOD dataset in turn and evaluated on all others. The mean robustness improvement (mRI) is calculated. As a result, ImageNet-RIB employs a completely different training and evaluation protocol. This design enables a more in-depth analysis of robust fine-tuning. For instance, as shown in Table 4 (Table 5 in the revision), our benchmark allows us to investigate performance changes based on the downstream dataset and OOD dataset (e.g., the relationship between different OOD datasets) and assess how well robustness is maintained regardless of the choice of downstream dataset.
>
> **Re: The conclusions drawn are quite obvious, as many similar studies have reached similar findings.**
>
> We kindly request the reviewer to provide more details about the “similar studies” referenced, so that we can make a more direct comparison. To the best of our knowledge, our conclusions are distinct from existing literature, and Reviewer 1GT2 has even described them as "counter-intuitive."
>
> First, our work is the first to analyze the effect of pre-training dataset size and characteristics on robustness, alongside the concurrent study by Ramanujan et al. [t68W-1], which will appear at NeurIPS this year. Ramanujan et al. demonstrated that both dataset size and label granularity improve robustness to OOD datasets after fine-tuning, using the WILDS benchmark [LcKf-2]. However, their study only tested datasets with up to 150,000 images for label granularity. In contrast, our work reveals that robustness does not always improve as the pretraining dataset size increases. As shown in Figure 4 (FT chart), OOD robustness begins to drop after fine-tuning models pre-trained on ImageNet-21K with AugReg, and a significant robustness degradation occurs with LAION-2B models (and OpenAI CLIP model in the revision).
>
> Second, we identified robust fine-tuning as a one-step continual learning problem and are the first to apply continual learning methods in this domain. Specifically, we show that combining continual learning approaches with post-hoc robust fine-tuning achieves the best performance in the benchmark (L364-373 in the original submission which is L378-387 in the revision).
>
> Finally, we showed that existing dataset distance metrics can predict the performance drop on the pretraining dataset after fine-tuning. By revisiting two metrics—Optimal Transport Dataset Distance (OTDD) and Normalized Compression Distance (NCD)—and applying them to both image and feature spaces, we found a negative correlation between the accuracy on the pre-training dataset and the dataset distance (Please see Table LcKf-1 for exact Pearson’s correlation coefficient in response to Reviewer LcKf).
>
> Alongside the proposal of our benchmark mentioned in the above response, we believe these contributions establish the novelty of our work compared to prior studies.
>
> **References**
>
> [t68W-1] Ramanujan, Vivek, et al. "On the connection between pre-training data diversity and fine-tuning robustness." *Advances in Neural Information Processing Systems* 36 (2024).
>
> [t68W-2] Koh, Pang Wei, et al. "Wilds: A benchmark of in-the-wild distribution shifts." *International conference on machine learning*. PMLR, 2021.

---

> > ### Comment · Reviewer_t68W · 2024-12-03
> > **Thank you for the detailed response.**
> >
> > Thank you for the detailed response. I decided to keep my ratings.

---

> ### Author Response · Authors · 2024-11-25
>
> We appreciate the reviewer for their insightful review! We’ve made significant additions in direct response to your feedback. We would greatly appreciate any further comments and kindly ask you to consider adjusting your score as the discussion period comes to a close. Thank you!

---

> ### Author Response · Authors · 2024-12-01
> **Gentle Reminder to Reviewer t68W: Response to Author Response and Revisions**
>
> As the response deadline is approaching, we kindly wanted to follow up and ask if you’ve had an opportunity to review our response and the updates we made to the manuscript.
>
> The reviewer’s concern was the Contribution of the ImageNet-RIB benchmark. As we mentioned in the previous response, our benchmark enables a more in-depth analysis of robust fine-tuning as it uses multiple downstream datasets, and our conclusion that continual learning is helpful for robust fine-tuning and pre-trained on larger datasets can lead to worse robustness after fine-tuning is novel as the reviewer acknowledged in Strength 4.

---

### Official Review · Reviewer_LcKf · 2024-11-03

**Soundness:** 2
**Presentation:** 2
**Contribution:** 2
**Rating:** 5
**Confidence:** 4

**Summary:**

This paper investigates the robustness and generalization ability of the pre-trained model after fine-tuning. Concretely, how would the model's generalization to out-of-distribution (OOD) samples change after fine-tuning a pre-trained model on a specific downstream dataset? The authors propose a dataset called ImageNet-RIB which combines existing ImageNet-based datasets. Then the authors conduct experiments on the benchmark to observe and compare the model's OOD generalization with different pre-training and fine-tuning settings. The settings include different pre-training datasets and fine-tuning methods. Among the several observations, the author derives a counterintuitive conclusion that large pre-training datasets do not guarantee the model's robustness after fine-tuning.

**Strengths:**

- Investigating the model's robustness and generalization is significant because real-world applications are complex with varying and dynamic data distributions.

- The experiments in this submission are comprehensive and detailed results and code are provided.

**Weaknesses:**

- This paper has very limited novelty in both insights and methodology. The two main contributions are the proposed ImageNet-RIB benchmark and the observation that several existing fine-tuning methods (continue learning and robust fine-tuning) are effective in preserving robustness. Both are very trivial. Compared with the former relevant benchmark [1], ImageNet-RIB follows the same recipe but only adds more existing ImageNet-based datasets. Let alone the fine-tuning methods, as discussed in the related work part, continue learning methods such as EWC and LwF are introduced in Lines 148-150, and robust fine-tuning methods are introduced in Lines 134-137. Therefore, it seems to me more clarification is required on the novelty aspect.

- The presentation is not clear. First, it seems confusing regarding the claim that performance drop on the pre-training dataset can be predicted by the distance between the pre-training and downstream dataset. It is not clear to me which experimental or theoretical part is to support such a claim, i.e., evaluate how a specific statistic correlation can exist and how well when using such correlation for performance drop prediction. Second, the presentation and tables/figures are not well organized. For example, Algorithm 1 is redundant with Figure 2 as the evaluation process is simple and easy to understand. Table 4 should be placed before Table 2 because Table 4 is mentioned in the earlier paragraph. Moreover, Table 4 is redundant with Table 3 and I would suggest putting Table 4 in the appendix. Typo in Lines113-114 and Lines 316-317.

- The experiments are not sound enough to derive the claimed conclusion. For example, the strong claim that large pre-training datasets do not guarantee robustness after fine-tuning is not well supported. The experimental setting only considers three pre-training datasets including ImageNet-1K, ImageNet-21K, and LAION-2B, which is too rough to come to a "conclusion" on pre-training datasets. I would suggest conducting controllable experiments [2] to disentangle the size and diversity of the dataset on LAION-2B, e.g., curating small variants such as LAION-1M, LAION-10M, LAION-100M, and LAION-1B.

- To enhance the significance of the paper, it is necessary to know why the weird observation LAION-2B happens instead of just presenting experimental results and putting forward some hypothesis without any in-depth analysis.

References

[1] Measuring robustness to natural distribution shifts in image classification. NeurIPS 2020

[2] Physics of Language Models.  ICML Tutorial 2024.

**Questions:**

Please see the weakness part.

---

> ### Author Response · Authors · 2024-11-17
> **Response to Reviewer LcKf (1/3)**
>
> We sincerely thank the reviewer for their thorough and constructive feedback. Your insights have been invaluable in highlighting areas that needed further discussion and clarification. Below, we respond to each of your comments and outline the detailed explanations and revisions that will be incorporated into the updated manuscript.
>
> **Re: Compared with the former relevant benchmark [1], ImageNet-RIB follows the same recipe but only adds more existing ImageNet-based datasets.**
>
> We would like to emphasize that ImageNet-RIB differs significantly from the existing benchmark proposed by Taori et al. (2020). As noted in L129-132 and L140-142 (L129-132 and L143-145 in the revision), the Taori et al. benchmark uses ImageNet-1K as the downstream dataset and evaluates robustness changes during fine-tuning by testing on OOD datasets such as ImageNet-V2, ImageNet-A, ImageNet-R, ImageNet-Sketch, and ObjectNet. In contrast, ImageNet-RIB starts with a model pre-trained on a dataset typically larger than ImageNet-1K and then fine-tuned on ImageNet-1K. Hence, given a set of ImageNet-based OOD datasets, the pre-trained model is fine-tuned on one OOD dataset and evaluated on the remaining datasets. This process is repeated such that the pre-trained model is fine-tuned on each OOD dataset in turn and evaluated on all others. The mean robustness improvement (mRI) is calculated. As a result, ImageNet-RIB employs a completely different training and evaluation protocol. This design enables a more in-depth analysis of robust fine-tuning. For instance, as shown in Table 4, our benchmark allows us to investigate performance changes based on the downstream dataset and OOD dataset (e.g., the relationship between different OOD datasets) and assess how well robustness is maintained regardless of the choice of downstream dataset.
>
> **Re: This paper has very limited novelty in both insights and methodology. The two main contributions are the proposed ImageNet-RIB benchmark and the observation that several existing fine-tuning methods (continue learning and robust fine-tuning) are effective in preserving robustness. Both are very trivial.**
>
>  To the best of our knowledge, our conclusions are distinct from existing literature, and Reviewer 1GT2 has even described them as "counter-intuitive."
>
> First, our work is the first to analyze the effect of pre-training dataset size and characteristics on robustness, alongside the concurrent study by Ramanujan et al. [LcKf-1], which will appear at NeurIPS this year. Ramanujan et al. demonstrated that both dataset size and label granularity improve the robustness to OOD datasets after fine-tuning, using the WILDS benchmark [LcKf-2]. However, their study only tested datasets with up to 150,000 images for label granularity. In contrast, our work reveals that robustness does not always improve as the pretraining dataset size increases. As shown in Figure 4 (FT chart), OOD robustness begins to drop after fine-tuning models pre-trained on ImageNet-21K with AugReg, and a significant robustness degradation occurs with LAION-2B models (and OpenAI CLIP model in the revision).
>
> Second, we identified robust fine-tuning as a one-step continual learning problem and are the first to apply continual learning methods in this domain. Specifically, we show that combining continual learning approaches with post-hoc robust fine-tuning achieves the best performance in the benchmark (L364-373 in the original submission which is L378-387 in the revision).
>
> Finally, we showed that existing dataset distance metrics can predict the performance drop on the pretraining dataset after fine-tuning. By revisiting two metrics—Optimal Transport Dataset Distance (OTDD) and Normalized Compression Distance (NCD)—and applying them to both image and feature spaces, we found a negative correlation between the accuracy on the pre-training dataset and the dataset distance (Please see Table LcKf-1 for exact Pearson’s correlation coefficient).
>
> Alongside the proposal of our benchmark mentioned in the above response, we believe these contributions establish the novelty of our work compared to prior studies.
>
> **References**
>
> [LcKf-1] Ramanujan, Vivek, et al. "On the connection between pre-training data diversity and fine-tuning robustness." *Advances in Neural Information Processing Systems* 36 (2024).
>
> [LcKf-2] Koh, Pang Wei, et al. "Wilds: A benchmark of in-the-wild distribution shifts." *International conference on machine learning*. PMLR, 2021.

---

> ### Author Response · Authors · 2024-11-17
> **Response to Reviewer LcKf (2/3)**
>
> **Let alone the fine-tuning methods, as discussed in the related work part, continue learning methods such as EWC and LwF are introduced in Lines 148-150, and robust fine-tuning methods are introduced in Lines 134-137. Therefore, it seems to me more clarification is required on the novelty aspect.**
>
> If our understanding is correct, the reviewer argued that we do not propose any new method instead reuse existing ones. Proposing a new method for robust fine-tuning or continual learning is not our main focus that proposes a new benchmark for robust fine-tuning and breaking our prejudice that training on larger model is always better in all cases. Instead, regarding the methods, we identified robust fine-tuning as a one-step continual learning problem and are the first to apply continual learning methods in this domain. Specifically, we show that combining continual learning approaches with post-hoc robust fine-tuning achieves the best performance in the benchmark  (L364-373 in the original submission which is L378-387 in the revision).
>
> Please inform us if we misunderstood you comment.
>
> **Re: First, it seems confusing regarding the claim that performance drop on the pre-training dataset can be predicted by the distance between the pre-training and downstream dataset. It is not clear to me which experimental or theoretical part is to support such a claim, i.e., evaluate how a specific statistic correlation can exist and how well when using such correlation for performance drop prediction.**
>
> Figure 3 demonstrates the relationship between the accuracy on the pre-training dataset (ImageNet-1K) and the dataset distance between ImageNet-1K and various downstream datasets, measured after fine-tuning using different methods. It is important to note that all models pre-trained on ImageNet-21K and LAION-2B were first fine-tuned on ImageNet-1K before conducting these experiments (see L253-254 and Table 5 in the original submission which are L260-261 and Table 6 in the revision).
>
> As shown in the figure, the accuracy on ImageNet-1K consistently decreases as the pretrained model is fine-tuned on downstream datasets that are farther from ImageNet-1K in terms of dataset distance. To quantify this observation, we computed the Pearson Correlation Coefficient between the dataset distance and the accuracy drop for each fine-tuning method and pre-trained model. The results, presented in Table LcKf-1, show consistent negative correlations across most methods, indicating that larger distances are associated with greater performance degradation on the pre-training dataset.
>
> Prompter demonstrates the strongest negative correlation across all pre-trained models, while vanilla fine-tuning (FT) also shows a strong and consistent correlation (< -0.5) [LcKf-3] between the accuracy on the pre-training dataset and the dataset distance. These results suggest that Optimal Transport Dataset Distance (OTDD), calculated in the feature space of the pre-trained model (i.e., the input to the classification layer), provides meaningful information about the changes in accuracy on the pre-training dataset.
>
> This analysis supports our claim by showing that dataset distance can serve as a predictor of performance drop. While the exact predictive power may vary depending on the fine-tuning method, the consistent negative correlations highlight the utility of dataset distance as a valuable metric for understanding accuracy degradation.
>
> **Table LcKf-1 Pearson correlation coefficient**
>
> | Method | IN1K + AugReg | IN-21K | IN-21K + AugReg | LAION-2B  |
> |-------------------|---------------|--------|-----------------|-----------|
> | FT                | -0.64         | -0.77  | -0.68           | -0.67     |
> | LinearProbing     | -0.22         | -0.36  | 0.10            | -0.19     |
> | Prompter          | -0.91         | -0.92  | -0.86           | -0.74     |
> | LoRA              | -0.63         | -0.25  | -0.63           | -0.31     |
> | EWC               | -0.57         | -0.88  | -0.91           | -0.32     |
> | LwF               | -0.49         | -0.56  | -0.38           | -0.44     |
> | LP-FT             | -0.59         | -0.69  | -0.39           | -0.56     |
> | WiSE-FT           | -0.46         | -0.92  | -0.52           | -0.31     |
> | MS:PRE-FT-ewc-lwf | -0.54         | -0.89  | -0.51           | -0.13     |
>
> [LcKf-3] Witte, Robert S., and John S. Witte. *Statistics*. John Wiley & Sons, 2007.

---

> ### Author Response · Authors · 2024-11-17
> **Response to Reviewer LcKf (3/3)**
>
> **Re: Algorithm 1 is redundant with Figure 2 as the evaluation process is simple and easy to understand.**
>
> Thank you for your suggestion, and we appreciate your positive feedback on Figure 2. We included Algorithm 1 to ensure the protocol was clear for readers who might not fully grasp it from Figure 2 alone. However, based on your feedback, we have moved Algorithm 1 to Appendix in the revised manuscript.
>
> **Re: Table 4 should be placed before Table 2 because Table 4 is mentioned in the earlier paragraph. Moreover, Table 4 is redundant with Table 3 and I would suggest putting Table 4 in the appendix.**
>
> We appreciate the reviewer’s suggestion. Regarding the placement of Table 4 (Table 5 in the revision), we have adhered to this year’s ICLR official recommendation for page 10: "*We recommend that authors only use the longer page limit in order to include larger and more detailed figures.*"
>
> Additionally, Table 3 (Table 4 in the revision) and Table 4 (Table 5 in the revision) serve distinct purposes and are not redundant. Table 3 presents the robustness improvement (RI) for each downstream dataset and the mean robustness improvement (mRI). In contrast, Table 4 provides the accuracy on each OOD dataset after fine-tuning on the downstream dataset. This additional detail allows for in-depth analysis, such as understanding the correlation between specific downstream datasets and OOD datasets (e.g., ImageNet-Sketch and ImageNet-R). For these reasons, we believe Table 4 adds value to the main text and provides complementary insights not captured by Table 3.
>
> **Re: (the claim is not well supported)The experimental setting only considers three pre-training datasets including ImageNet-1K, ImageNet-21K, and LAION-2B, which is too rough to come to a "conclusion" on pre-training datasets. I would suggest conducting controllable experiments [2] to disentangle the size and diversity of the dataset on LAION-2B, e.g., curating small variants such as LAION-1M, LAION-10M, LAION-100M, and LAION-1B.**
>
> Unfortunately, we do not have enough resources to train VIT on these large scale datasets from scratch. Instead, we added entire pre-trained ViT models of which the input size is 224x224 and finally fine-tuned on ImageNet-1K in the revision, i.e., IN-1K + SAM (Sharpness Aware Minimization), IN-21K-P, OpenAI CLIP. Note that IN-21K-P and OpenAI CLIP pre-trained models are fine-tuned on IN-1K same as other models. As shown in Figure 4 in the revision, the robustness is improved after vanilla fine-tuning in IN-1K-based pre-trained models while IN-21K model’s robustness rarely changes. Conversely, the robustness of IN-21K-P and IN-21K + AugReg pre-trained models drops. Especially, after fine-tuning, IN-21K-P pre-trained model has better robustness than IN-21K + AugReg. Finally, OpenAI CLIP model also suffers from a severe robustness drop same as LAION-2B pre-trained model.
>
> We believe that these experiments sufficiently support our claim that larger pre-trained model does not always guarantee to have better robustness. Moreover, the results of models pre-trained on IN-21K and its variants present that different pre-processing, data augmentation, and regularization while training on the same pre-training dataset affects robustness changes in fine-tuning.
>
> **Re: why the weird observation LAION-2B happens instead of just presenting experimental results and putting forward some hypothesis without any in-depth analysis.**
>
> Thank you for pointing out. We are analyzing the cause for the catastrophic robustness drop on the large scale datasets by focusing on how network weight and representation change during fine-tuning. We will share the results once it is done.
>
> **Re: Typo in Lines113-114 and Lines 316-317.**
>
> Thank you for your comment. We changed L113 "the absolute performance" to "**T**he absolute performance"  and L316-317 from "We consider how the accuracy on ImageNet-1K changes after fine-tuning on each downstream dataset, Figure 3." to "We consider how the accuracy on ImageNet-1K changes after fine-tuning on each downstream dataset."

---

> ### Author Response · Authors · 2024-11-25
> **Results with LAION-400M and DataComp-1B pre-trained models**
>
> Following Reviewer LcKf’s recommendation, we used LAION-400M and DataComp-1B pre-trained ViT-B/16 models from the OpenCLIP library. Unfortunately, we could not find a publicly released LAION-100M pre-trained ViT-B/16 model, and we lack the resources to train a ViT from scratch on the LAION datasets. However, we found a LAION-100M pre-trained ViT-B/32 model from Lin et al. [LcKf-4], and we will share the results if fine-tuning is completed during the discussion period.
>
> Unlike the existing models in the Timm library, which are already fine-tuned on ImageNet-1K, we fine-tuned these pre-trained models ourselves on ImageNet-1K. Due to time and resource constraints, we trained the ViT-B/16 models for 25 epochs on ImageNet-1K (ImageNet-1K training accuracy: LAION-400M model: 91.6%; DataComp-1B model: 87.9%).
>
> We fine-tuned the models on each downstream dataset for 2 epochs, as the robustness drop becomes distinctive after 2 epochs of fine-tuning, as shown in Figure 6 of the revision. We will share results after 10 epochs of fine-tuning if they are ready during the discussion period.
>
> As shown in Table LcKf-2, the new LAION-400M and DataComp-1B pre-trained models exhibit much lower accuracy on the ImageNet-1K validation set and in OOD performance, as we had limited time to fine-tune these models on ImageNet-1K. Nonetheless, their mRIs are twice as low as those of the ImageNet-21K + AugReg model. This suggests that models pre-trained on larger and more diverse datasets may be more susceptible to catastrophic forgetting of the robustness to OOD data, even when the pre-trained model’s performance is inferior.
>
> If the paper is accepted, we will use stronger pre-trained models for LAION-100M and DataComp-1B in future experiments.
>
> **Table LcKf-2. Comparison between ViT-B/16 models pre-trained on different datasets before and after fine-tuning for 2 epochs on downstream datasets.**
>
> |                 | IN-1K Accuracy of Pre-Trained Models | Average Accuracy on OOD Datasets before Fine-Tuning | mRI    | Average Accuracy on OOD Datasets after Fine-Tuning |
> | --------------- | --------------------------------- | --------------------------------------------------- | ------ | -------------------------------------------------- |
> | IN-1K + AugReg  | 79.2                              | 46.8                                                | **3.5**    | 50.3                                               |
> | IN-1K + SAM     | 80.2                              | 47.0                                                | 3.1    | 50.1                                               |
> | IN-21K          | 81.8                              | 53.6                                                | 1.0    | 54.6                                               |
> | IN-21K-P        | 84.3                              | 56.8                                                | 1.1    | **57.9**                                               |
> | IN-21K + AugReg | 84.5                              | 60.8                                                | \-3.8  | 56.9                                               |
> | LAION-400M (NEW)     | 73.6                              | 39.8                                                | \-9.0  | 30.8                                               |
> | DataComp-1B (NEW)    | 70.6                              | 39.7                                                | \-7.6  | 32.1                                               |
> | OpenAI          | 85.3                              | 63.4                                                | \-37.8 | 25.6                                               |
> | LAION           | **85.5**                             | **64.2**                                                | \-35.5 | 28.7                                               |
>
>
> We hope that along with the already added pre-training dataset, this will resolve the reviewer’s concern that more datasets are needed.
> Please let us know if any concerns remain.
>
> [Lckf-4] Lin, Yiqi, et al. "Parrot captions teach clip to spot text." *ECCV. 2024*

---

> ### Author Response · Authors · 2024-11-25
>
> We appreciate the reviewer for their insightful review! We’ve made significant additions and conducted enormous experiments in direct response to your feedback. We would greatly appreciate any further comments and kindly ask you to consider adjusting your score as the discussion period comes to a close. Thank you!

---

> ### Author Response · Authors · 2024-11-25
> **Results with LAION-100M, LAION-400M, and DataComp-1B pre-trained models after fine-tuning 10 epochs**
>
> As we promised in the previous response, we shared the results of LAION-400M, DataComp-1B pre-trained ViT-B/16 and LAION-100M pre-trained ViT-B/32. The accuracy of LAION-100M pre-trained ViT-B/32 on the ImageNet-1K training set and validation set is 84.0 and 55.6, respectively.
> As shown in Tables LcKf-3 and LcKf-4, these models have much worse mRI compared to ImageNet-21K with AugReg pre-trained models even though their initial robustness is worse than its counterpart.
>
> We hope that from this response, all concerns are resolved.
>
> **Table LcKf-3. Comparison between ViT-B/16 models pre-trained on different datasets before and after fine-tuning for 10 epochs on downstream datasets.**
>
> | Pre-Trained Models | Average Accuracy on OOD Datasets before Fine-Tuning | mRI | Average Accuracy on OOD Datasets after Fine-Tuning |
> | --- | --- | --- | --- |
> | IN-1K + AugReg | 46.8 | 1.3 | 48.1 |
> | IN-1K + SAM | 47.0 | 2.5 | 49.5 |
> | IN-21K | 53.6 | -0.1 | 53.5 |
> | IN-21K-P | 56.8 | -0.5 | 56.3 |
> | IN-21K + AugReg | 60.8 | -5.5 | 55.3 |
> | LAION-400M (NEW) | 39.8 | -11.2 | 28.6 |
> | DataComp-1B (NEW) | 39.7 | -10.2 | 29.5 |
> | OpenAI | 63.4 | -38.0 | 25.4 |
> | LAION-2B | 64.2 | -38.1 | 26.1 |
>
> **Table LcKf-4. Comparison between ViT-B/32 models pre-trained on different datasets before and after fine-tuning for 10 epochs on downstream datasets.**
>
> | Pre-Trained Models | Average Accuracy on OOD Datasets before Fine-Tuning | mRI | Average Accuracy on OOD Datasets after Fine-Tuning |
> | --- | --- | --- | --- |
> | IN-1K + AugReg | 40.1 | -0.0 | 48.1 |
> | IN-1K + SAM | 37.7 | 1.4 | 39.1 |
> | IN-21K + AugReg | 49.0 | -0.1 | 55.3 |
> | LAION-100M (NEW) | 39.8 | -5.4 | 34.4 |
> | OpenAI | 50.9 | -28.7 | 22.2 |
> | LAION-2B | 52.3 | -31.6 | 21.7 |

---

> > ### Comment · Reviewer_LcKf · 2024-11-26
> > **Thanks for the replies**
> >
> > I sincerely thank the authors for their comprehensive responses and for incorporating many of my suggestions into the revision. My remaining concern is about the strong claim that "large pre-training datasets do not guarantee robustness after fine-tuning." I understand that pre-training from scratch takes much more computing, which can be unaffordable. I also appreciate the new results of LAION-1OOM and LAION-400M. Despite the LAION results approximately supporting the claim, I note such a trend does not hold for IN-1k and IN-21k where IN-21k enjoy better robustness than IN-1k after the fine-tuning. As stated in my initial review, conducting controllable experiments can be a convincing way to support a "counter-intuitive" claim. From my perspective, this submission aims to challenge the popular view of model generalization, which requires more evidence to make it convincing and solid. I think, after the revision, this submission is around the borderline. Therefore, I've decided to increase my rating to 5 and would not be blocking if other reviewers champion the acceptance.

---

> ### Author Response · Authors · 2024-11-26
>
> We appreciate the reviewer's response and are gratified that many of their concerns have already been addressed. We also want to thank the reviewer for increasing the rating. Your feedback is always valuable and has helped improve our paper. We hope that this clarification will provide a better understanding of our work.
>
> **Clarification of Our Claim Regarding the Paper Title: “Large Pre-Training Datasets Do Not Guarantee Robustness After Fine-Tuning”**
>
> Our claim is not that models pre-trained on larger and more diverse datasets always have worse robustness after fine-tuning compared to those pre-trained on smaller datasets. Rather, we argue that such models ***can*** exhibit worse robustness than models pre-trained on smaller datasets after fine-tuning, as robustness degradation becomes significant at certain points (e.g., between IN-21K + AugReg and LAION-2B or DataComp-1B and LAION-2B).
>
> Throughout the paper, we emphasize that larger pre-training datasets ***do not always*** result in better robustness after fine-tuning, even if the initial robustness is higher.
>
> For example:
>
> - “Pre-training on LAION-2B, despite its size and diversity, ***does not always yield the best results*** when fine-tuned on downstream tasks, suggesting that starting with large, rich datasets ***may not always*** be the optimal approach for preserving robustness.” (L100-102)
> - “Starting with rich foundation models ***may*** not always be the best approach.” (L112-114)
>
> From this paper, we caution the machine learning community against the assumption that larger pre-training datasets inherently lead to better robustness, which will hold even after fine-tuning on downstream datasets.
>
> To reduce potential confusion, we will modify the title by adding "always" to:
>
> “ImageNet-RIB Benchmark: Large Pre-Training Datasets Don’t **Always** Guarantee Robustness After Fine-Tuning.”
>
> **Re: Such a trend does not hold for IN-1k and IN-21k where IN-21k enjoy better robustness than IN-1k after the fine-tuning**
>
> We conducted experiments using various versions of ImageNet-1K and ImageNet-21K pre-trained models from the Timm library to include as many pre-trained models as possible. When comparing models with the same pre-training scheme (AugReg), the mRI of IN-1K + AugReg pre-trained models is consistently better than that of IN-21K + AugReg pre-trained models. However, the performance drop for the IN-21K + AugReg pre-trained model is marginal compared to OpenAI and LAION pre-trained models. Consequently, the robustness gap between the IN-1K + AugReg and IN-21K + AugReg pre-trained models is significant, as shown in Table LcKf-5, leading to IN-21K + AugReg models maintaining better robustness overall.
>
> As mentioned in L404-406 (L425-426 in the initial manuscript), we believe that the results from IN-21K + AugReg pre-trained models serve as an early indicator of performance decay in models pre-trained on larger datasets. Thus, there is no conflict with our claim that pre-training on larger datasets can lead to severe catastrophic forgetting of robustness to OOD datasets after fine-tuning.
>
> **Table LcKf-5. Comparison between ViT-B/16 and ViT-B/32 models pre-trained on ImageNet-1K with AugReg and ImageNet-21K with AugReg before and after fine-tuning on downstream datasets.**
>
> | Pre-Training Dataset | Architecture | Average Accuracy on OOD Datasets before Fine-Tuning | mRI   | Average Accuracy on OOD Datasets after Fine-Tuning |
> | -------------------- | ------------ | --------------------------------------------------- | ----- | -------------------------------------------------- |
> | IN-1K + AugReg       | ViT-B/16     | 46.8                                                | 1.3   | 48.1                                               |
> | IN-21K + AugReg      | ViT-B/16     | 60.8                                                | \-5.5 | 55.3                                               |
> | IN-1K + AugReg       | ViT-B/32     | 40.1                                                | 0.0     | 40.1                                               |
> | IN-21K + AugReg      | ViT-B/32     | 49.0                                                  | \-0.1 | 48.9

---

### Author Response · Authors · 2024-11-17
**General Comments regarding the revision**

We truly appreciate all reviewers providing valuable feedback. We have updated the revision with colored as blue accordingly as follows:

- Following Reviewer 1GT2’s comment, we added a new subsection in Section 2 Related Work regarding single-domain generalization.
- Following comments from Reviewer LcKf and SbRm, we modified Step 3 in Figure 2 and moved Algorithm 1 to Appendix to reduce redundancy.
- Following Reviewer 1GT2’s comment, we clarify experimental details in Sections 4.1, 4.4, Figure 6, and Appendix A.3.
- Following Reviewer LcKf’s comment, we calculated Pearson correlation coefficients for Figure 2 and added them in Section 4.2 with a new table (Table 2).
- Following Reviewer LcKf’s comment, we used different pre-training datasets ImageNet-1K with sharpness aware minimization, ImageNet-21K-P, and OpenAI CLIP which are a complete set of the timm library that uses ViT-B/16 with 224x224 sized image and fine-tuned on ImageNet-1K at the end.

Additionally, we added a new analysis to verify that the severe robustness drop of the LAION-2B model after fine-tuning is not due to overfitting to the downstream dataset in Appendix C.  We also corrected the notation for the ImageNet-1K pre-trained ViTs are pre-trained with AugReg as noted in Table 6. We clarify it across the entire paper.

Please let us know if you have further questions or concerns. We will be happy to deal with them.

---

### Meta-Review · Area_Chair_Sk8i · 2024-12-16

**Metareview:**

a) This paper investigates the generalization ability of a pre-trained model after fine-tuning. The authors propose a new dataset/benchmark based on ImageNet. The authors presented a counterintuitive conclusion that large pre-training datasets do not guarantee the model's robustness after fine-tuning.

b) Investigating the robustness of pre-trained models after fine-tuning is an important topic of research sometimes overlooked. The authors proposed a new benchmark (composed of existing datasets) for evaluating such robustness and evaluated several models yielding interesting findings.

c) Reviewers considered most of the contributions of the paper as minor. For instance the new datasets is the use of a different protocol on pre-existing datasets. Most of the results of the paper are expected. The most interesting one that goes against common believe is the fact that pre-training on a large datasets guarantees the model's robustness. However, this is observed in limited experimental setting and no deeper understanding is provided.

d) The main reason for rejection is because the most important/novel observation is not supported by enough empirical results and it could be an effect of the datasets used for the evaluation. Additionally, authors did not investigate what are the possible causes of this phenomena.

**Additional Comments On Reviewer Discussion:**

Rev. LcKf provided a good review of the paper, pointing out the issues related to the fact that the dataset is a collection of previous datasets with a different evaluation protocol and some existing fine-tuning (continual learning and robust fine-tuning) methods work well for maintaining robustness, which is well-known. Authors provided compelling answers and made Rev. to increase their score to 5, considering the pending issue that the only important contribution is not supported by enough results or analysis.

Rev. t68W provided a short review pointing out the same issues with the dataset and minor contributions about robust fine-tuning methods. After rebuttal rev. maintained their score of 5 without further commenting the authors answers.

Rev. SbRm was the most positive about the paper, appreciating the clear writing and the comprehensive benchmark providing a score of 6 and keeping it after authors' rebuttal. After the internal discussion, Rev. agreed on the fact that the paper requires more analysis on their claims.

Rev. 1GT2 was the most negative with a score of 3, pointing out some confusing parts of the paper and some missing related work. After rebuttal, Rev. maintained his score. He mentioned that issues about related work and clarity are not enough for rejection, but found issues with the practical relevance of the proposed benchmark and experiment design that could be biased.

Overall, I agree with revs. that most of the contributions of the paper are not novel/strong enough for this conference. The only contribution that could be of relevant is the fact that in certain cases pre-training large datasets do not guarantee robustness. However, this interesting phenomena is revealed only on some very specific setting and no further understanding or analysis are provided.

---

### Decision · Program_Chairs · 2025-01-22

Reject